# Information theoretic limits of learning a sparse rule

**Clément Luneau**[*]**, Nicolas Macris**
Ecole Polytechnique Fédérale de Lausanne
Suisse

**Jean Barbier**
International Center for Theoretical Physics
Trieste, Italy

## Abstract

We consider generalized linear models in regimes where the number of nonzero components of the signal and accessible data points are *sublinear* with respect to the size of the signal. We prove a variational formula for the asymptotic mutual information per sample when the system size grows to infinity. This result allows us to derive an expression for the minimum mean-square error (MMSE) of the Bayesian estimator when the signal entries have a discrete distribution with finite support. We find that, for such signals and suitable vanishing scalings of the sparsity and sampling rate, the MMSE is nonincreasing piecewise constant. In specific instances the MMSE even displays an *all-or-nothing* phase transition, that is, the MMSE sharply jumps from its maximum value to zero at a critical sampling rate. The all-or-nothing phenomenon has previously been shown to occur in high-dimensional linear regression. Our analysis goes beyond the linear case and applies to learning the weights of a perceptron with general activation function in a teacher-student scenario. In particular, we discuss an all-or-nothing phenomenon for the generalization error with a sublinear set of training examples.

## 1 Introduction

Modern tasks in statistical analysis, signal processing and learning require solving high-dimensional inference problems with a very large number of parameters. This arises in areas as diverse as learning with neural networks [1], high-dimensional regression [2] or compressed sensing [3, 4]. In many situations, there appear barriers to what is possible to estimate or learn when the data becomes too scarce or too noisy. Such barriers can be of *algorithmic* nature, but they can also be *intrinsic* to the very nature of the problem. A celebrated example is the impossibility of reconstructing a noisy signal when the noise is beyond the so-called Shannon capacity of the communication channel [5]. A large amount of interdisciplinary work has shown that these intrinsic barriers can be understood as *static phase transitions* (in the sense of physics) when the system size tends to infinity (see [6, 7, 8]).

When the problem can be formulated as an (optimal) Bayesian inference problem the mathematically rigorous theory of these phase transitions is now quite well developed. Progress initially came from applications of the Guerra-Toninelli interpolation method (developed for the Sherrington-Kirkpatrick spin-glass model [9]) to coding and communication theory [10, 11, 12, 13, 14, 15], and more recently to low-rank matrix and tensor estimation [16, 17, 18, 19, 20, 21, 22, 23, 24], compressive sensing and high-dimensional regression [25, 26, 27, 28], and generalized linear models [29]. In particular, for all these problems it has been possible to reduce the asymptotic mutual information to a low-dimensional variational expression, and deduce from its solution relevant error measures (e.g., minimum mean-square and generalization errors). All these works consider the *traditional regime* of statistical mechanics where the system size goes to infinity while relevant control parameters (such as signal sparsity, sampling rate, or signal-to-noise ratio) are kept fixed.

---

[*]Corresponding author: clement.luneau@epfl.ch

However, there exist *other interesting regimes* for which many of the above mentioned problems also display fundamental intrinsic limits akin to phase transitions. Consider for example the problem of compressive sensing. An interesting regime is one where both the number of nonzero components and of samples scale in a *sublinear* manner as the system size tends to infinity. In this case we would like to identify the phase transition, if there is any, and its nature. This question has first been addressed recently in the framework of compressed sensing for binary Bernoulli signals by [30, 31, 32]. An *all-or-nothing phenomenon* is identified, that is, in an appropriate sparse regime, the minimum mean-square error (MMSE) sharply drops from its maximum possible value (no reconstruction) for "too small" sampling rates to zero (perfect reconstruction) for "large enough" sampling rates. The interest of such regime is not limited to estimation problems. It is also relevant from a learning point of view, e.g., it corresponds to learning scenarios where we have access to a high number of features but only a sublinear number of them – unknown to us – are relevant for the learning task at hand.

Examples abound where the "bet on sparsity principle" [33, 34] is of utmost importance for the interpretability of a high-dimensional model. Let us mention the MNIST handwritten digit database, where each digit can be seen as a $784 = 28 \times 28$-dimensional binary vector representing the pixels whereas the digits effectively live in a space of the order of tens of dimensions [35, 36]. Another example of effective sparsity comes from natural images which are often sparse in a wavelet basis [37]. Then, a fundamental question is *"when is it possible to achieve a low estimation or generalization error with a sublinear amount of samples (sublinear with respect to the total number of features)?"*

In this contribution we address this question for a mathematically simple, but precise and tractable, setting. We consider generalized linear models in the regime of vanishing sparsity and sample rate, or equivalently, of sublinear number of data samples and nonzero signal components. As explained below these models can be used for estimation as well as learning, and we uncover in the sublinear regime intrinsic statistical barriers to these tasks in the form of sharp phase transitions. These statistical barriers are computed exactly and thus provide precise benchmarks to which algorithmic performance can be compared.

Let us outline the mathematical setting (further detailed in Section 2). In a probabilistic setting the *unknown* signal vector $\mathbf{X}^* \in \mathbb{R}^n$ has entries drawn independently at random from a distribution $P_{0,n} \coloneqq \rho_n P_0 + (1 - \rho_n)\delta_0$ with $P_0$ a fixed distribution. The parameter $\rho_n$ controls the sparsity of the signal so that $\mathbf{X}^*$ has $k_n \coloneqq n\rho_n$ nonzero components on average. We observe the data $\mathbf{Y} = \varphi\big(\mathbf{\Phi}\mathbf{X}^*/\sqrt{k_n}\big) \in \mathbb{R}^{m_n}$ obtained by first multiplying the signal with a *known* $m_n \times n$ random matrix $\mathbf{\Phi}$ whose entries are independent standard Gaussian random variables, and then applying $\varphi$ component-wise. The number of data points is controlled by the sampling rate $\alpha_n$, i.e., $m_n \coloneqq \alpha_n n$. We consider the regime $(\rho_n, \alpha_n) \to (0, 0)$ as $n$ goes to infinity with $\alpha_n = \gamma\rho_n|\ln \rho_n|$, for which sharp phase transitions appear when $P_0$ is discrete with finite support. Note that both $m_n$ and $k_n$ scale sublinearly as $n \to +\infty$.

The model can be interpreted as either an estimation problem or a learning problem:

- In the *estimation interpretation*, we assume a purely Bayesian (or optimal) setting. We know the model, the activation function $\varphi$, the prior $P_{0,n}$ as well as the measurement matrix $\mathbf{\Phi}$. Our goal is then to determine what is the lowest reconstruction error that we can achieve, i.e., what is the average minimum mean-square error $k_n^{-1}\mathbb{E}\left\|\mathbf{X}^* - \mathbb{E}[\mathbf{X}^*|\mathbf{Y}, \mathbf{\Phi}]\right\|^2$ when $n$ gets large.

- In the *learning interpretation*, we consider a teacher-student scenario in which a teacher hands out training samples $\{(Y_\mu, (\Phi_{\mu i})_{i=1}^n)\}_{\mu=1}^{m_n}$ to a student. The teacher produces the output label $Y_\mu$ by feeding the input $(\Phi_{\mu i})_{i=1}^n$ to its own one-layer neural network with activation function $\varphi$ and weights $\mathbf{X}^* = (X_i^*)_{i=1}^n$. The student – who is given the model and the prior – has to learn the weights $\mathbf{X}^*$ of the teacher's one-layer neural network by minimizing the empirical training error of the $m_n$ training samples. For example, the binary perceptron corresponds to $\varphi = \mathrm{sign}$ and $Y_\mu \in \{\pm 1\}$. Of particular interest is the generalization error. Given a new – previously unseen – random pattern $\mathbf{\Phi}_{\mathrm{new}} \coloneqq (\Phi_{\mathrm{new},i})_{i=1}^n$ whose true label is $Y_{\mathrm{new}}$ (generated by the teacher's neural network), the optimal generalization error is $\mathbb{E}[(Y_{\mathrm{new}} - \mathbb{E}[\varphi(\mathbf{\Phi}_{\mathrm{new}}^\mathsf{T}\mathbf{X}^*/\sqrt{k_n})|\mathbf{Y}, \mathbf{\Phi}, \mathbf{\Phi}_{\mathrm{new}}])^2]$; the error made when estimating $Y_{\mathrm{new}}$ in a purely Bayesian way.

Let us summarize informally our results. We set $\alpha_n = \gamma\rho_n|\ln \rho_n|$ where $\gamma$ is fixed and $\rho_n$ vanishes as $n$ diverges. We first rigorously determine the mutual information $m_n^{-1}I(\mathbf{X}^*; \mathbf{Y}|\mathbf{\Phi})$ in terms of a low-dimensional variational problem, see Theorem 1 which also provides a precise control of the finite size fluctuations. Remarkably, when $P_0$ is a discrete distribution with finite support, this variational

problem simplifies to a minimization problem over a finite set of values, see Theorem 2. For such signals, using I-MMSE type formulas [38], we can deduce from the solution to this minimization problem the asymptotic MMSE and optimal generalization error, see Theorem 3. Our analysis shows that both errors are nonincreasing piecewise constant functions of $\gamma$. In particular, if the entries of $|\mathbf{X}^*|$ are either 0 or some $a > 0$ then both errors display an all-or-nothing behavior as $n \to +\infty$, with a sharp transition at a threshold $\gamma = \gamma_c$ explicitly computed. These findings are illustrated, and their significance discussed, in Section 3.

In our work the generalized linear model is treated by entirely different methods than the linear model in [30, 31]. Importantly, the sparsity regime treated by our method requires the sparsity $\rho_n$ to go to zero slower than $n^{-1/9}$, while it has to go to zero faster than $n^{-1/2}$ in the results of [31] for the linear case. From this angle, both results complement each other. Our proof technique for Theorem 1 exploits the adaptive interpolation method (see [39, 40]) that is a powerful improvement over the Guerra-Toninelli interpolation and allows to prove replica symmetric formulas for Bayesian inference problems. We adapt the analysis of [29] in a non-trivial way in order to consider the new scaling regime of our problem where $\alpha_n = \gamma\rho_n|\ln\rho_n|$, and $\rho_n \to 0$ as $n$ gets large instead of being fixed. We show that the adaptive interpolation can still be carried through, which requires a more refined control of the error terms compared to [29]. It is interesting, and not a priori obvious, that this can be done since this is *not* the usual statistical mechanics extensive regime. For example, the mutual information has to be normalized by the subextensive quantity $m_n = \mathcal{o}(n)$. Quite remarkably, with this suitable normalization, the asymptotic mutual information, MMSE and generalization error have a similar form to those famously found in ordinary thermodynamic regimes in physics [41, 42, 43, 44].

In Section 2 we present the setting and state our theoretical results on the mutual information and the MMSE in the sublinear regime. We use these results in Section 3 to uncover the all-or-nothing phenomenon for general activation functions. In Section 4 we give an overview of the adaptive interpolation method used to prove Theorem 1. The full proofs of our results are given in the Supplementary Material.

## 2 Problem setting and main results

### 2.1 Generalized linear estimation of low sparsity signals at low sampling rates

Let $n \in \mathbb{N}^*$ and $m_n := \alpha_n n$ with $(\alpha_n)_{n\in\mathbb{N}^*}$ a decreasing sequence of positive sampling rates. Let $P_0$ be a probability distribution with finite second moment $\mathbb{E}_{X\sim P_0}[X^2]$. Let $(X_i^*)_{i=1}^n \overset{\text{iid}}{\sim} P_{0,n}$ be the components of a signal vector $\mathbf{X}^*$ (this is also denoted $\mathbf{X}^* \overset{\text{iid}}{\sim} P_{0,n}$), where

$$P_{0,n} := \rho_n P_0 + (1 - \rho_n)\delta_0 \,. \tag{1}$$

The parameter $\rho_n \in (0, 1)$ controls the sparsity of the signal; the latter being made of $k_n := \rho_n n$ nonzero components in expectation. We will be interested in low sparsity regimes where $k_n = \mathcal{o}(n)$. Let $k_A \in \mathbb{N}$. We consider a measurable function $\varphi : \mathbb{R} \times \mathbb{R}^{k_A} \to \mathbb{R}$ and a probability distribution $P_A$ over $\mathbb{R}^{k_A}$. The $m_n$ data points $\mathbf{Y} := (Y_\mu)_{\mu=1}^{m_n}$ are generated as

$$Y_\mu := \varphi\Big(\frac{1}{\sqrt{k_n}}(\mathbf{\Phi}\mathbf{X}^*)_\mu, \mathbf{A}_\mu\Big) + \sqrt{\Delta}Z_\mu \,, \quad 1 \le \mu \le m_n \,, \tag{2}$$

where $(\mathbf{A}_\mu)_{\mu=1}^{m_n} \overset{\text{iid}}{\sim} P_A$, $(Z_\mu)_{\mu=1}^{m} \overset{\text{iid}}{\sim} \mathcal{N}(0, 1)$ is an additive white Gaussian noise (AWGN), $\Delta > 0$ is the noise variance, and $\mathbf{\Phi}$ is a $m_n \times n$ measurement (or data) matrix with independent entries having zero mean and unit variance. Note that the noise $(Z_\mu)_{\mu=1}^{m}$ can be considered as part of the model, or as a "regularising noise" needed for the analysis but that can be set arbitrarily small. Typically, and as $n$ gets large, $(\mathbf{\Phi}\mathbf{X}^*)_\mu/\sqrt{k_n} = \Theta(1)$. The estimation problem is to recover $\mathbf{X}^*$ from the knowledge of $\mathbf{Y}, \mathbf{\Phi}, \Delta, \varphi, P_{0,n}$ and $P_A$ (the realization of the random stream $(\mathbf{A}_\mu)_{\mu=1}^{m_n}$ itself, if present in the model, is unknown). It will be helpful to think of the measurements as the outputs of a *channel*:

$$Y_\mu \sim P_{\text{out}}\Big( \cdot \Big| \frac{1}{\sqrt{k_n}}(\mathbf{\Phi}\mathbf{X}^*)_\mu \Big), \quad 1 \le \mu \le m_n \,. \tag{3}$$

The transition kernel $P_{\text{out}}$ admits a transition density with respect to Lebesgue's measure given by:

$$P_{\text{out}}(y|x) = \frac{1}{\sqrt{2\pi\Delta}} \int dP_A(\mathbf{a})\, e^{-\frac{1}{2\Delta}(y-\varphi(x,\mathbf{a}))^2} \,. \tag{4}$$

The random stream $(\mathbf{A}_\mu)_{\mu=1}^{m_n}$ represents any source of randomness in the model. For example, the logistic regression $\mathbb{P}(Y_\mu = 1) = f((\mathbf{\Phi}\mathbf{X}^*)_\mu/\sqrt{k_n})$ with $f(x) = (1 + e^{-\lambda x})^{-1}$ is modeled by considering a teacher that draws i.i.d. uniform numbers $A_\mu \sim \mathcal{U}[0,1]$, and then obtains the labels through $Y_\mu = \mathbf{1}_{\{A_\mu \leq f((\mathbf{\Phi}\mathbf{X}^*)_\mu/\sqrt{k_n})\}} - \mathbf{1}_{\{A_\mu \geq f((\mathbf{\Phi}\mathbf{X}^*)_\mu/\sqrt{k_n})\}}$ ($\mathbf{1}_\mathcal{E}$ denotes the indicator function of an event $\mathcal{E}$). In the absence of such a randomness in the model, the activation $\varphi : \mathbb{R} \to \mathbb{R}$ is deterministic, $k_A = 0$ and the integral $\int dP_A(\mathbf{a})$ in (4) simply disappears. Our numerical experiments in Section 3 are for deterministic activations but all of our theoretical results hold for the broader setting.

We have presented the problem from an estimation point of view. In this case, the important quantity to assess the performance of an algorithm estimating $\mathbf{X}^*$ is the mean-square error. Another point of view is the learning one: each row of the matrix $\mathbf{\Phi}$ is the input to a one-layer neural network whose weights $\mathbf{X}^*$ have been sampled independently at random by a teacher. The student is given the input/output pairs $(\mathbf{\Phi}, \mathbf{Y})$ as well as the model used by the teacher. The student's role is then to learn the weights. In this case, more than the mean-square error, the important quantity is the generalization error.

## 2.2 Asymptotic mutual information

The mutual information $I(\mathbf{X}^*; \mathbf{Y}|\mathbf{\Phi})$ between the signal $\mathbf{X}^*$ and the data $\mathbf{Y}$ given the matrix $\mathbf{\Phi}$ is the main quantity of interest in our work. Before stating Theorem 1 on the value of this mutual information, we first introduce two scalar denoising models that play a key role.

The first model is an additive Gaussian channel. Let $X^* \sim P_{0,n}$ be a scalar random variable. We observe $Y^{(r)} := \sqrt{r}X^* + Z$ where $r \geq 0$ plays the role of a signal-to-noise ratio (SNR) and the noise $Z \sim \mathcal{N}(0,1)$ is independent of $X^*$. The mutual information $I_{P_{0,n}}(r) := I(X^*; Y^{(r)})$ between the signal of interest $X^*$ and $Y^{(r)}$ depends on $\rho_n$ through the prior $P_{0,n}$, and it reads:

$$I_{P_{0,n}}(r) = \frac{r\rho_n \mathbb{E}_{X \sim P_0}[X^2]}{2} - \mathbb{E}\ln\int dP_{0,n}(x)e^{rX^*x + \sqrt{r}Zx - \frac{rx^2}{2}} . \tag{5}$$

The second scalar channel is linked to the transition kernel $P_{\text{out}}$ defined by (4). Let $V, W^*$ be two independent standard Gaussian random variables. In this scalar estimation problem we want to infer $W^*$ from the knowledge of $V$ and the observation $\widetilde{Y}^{(q,\rho)} \sim P_{\text{out}}(\cdot | \sqrt{q}\,V + \sqrt{\rho - q}\,W^*)$ where $\rho > 0$ and $q \in [0, \rho]$. The conditional mutual information $I_{P_{\text{out}}}(q, \rho) := I(W^*; \widetilde{Y}^{(q,\rho)}|V)$ is:

$$I_{P_{\text{out}}}(q, \rho) = \mathbb{E}\ln P_{\text{out}}(\widetilde{Y}^{(\rho,\rho)}|\sqrt{\rho}\,V) - \mathbb{E}\ln\int dw\, \frac{e^{-\frac{w^2}{2}}}{\sqrt{2\pi}} P_{\text{out}}(\widetilde{Y}^{(q,\rho)}|\sqrt{q}\,V + \sqrt{\rho - q}\,w) . \tag{6}$$

Both $I_{P_{0,n}}$ and $I_{P_{\text{out}}}$ have nice monotonicity, Lipschitzianity and concavity properties that are important for the proof of Theorem 1 (stated below).

We use the mutual informations (5) and (6) to define the *(replica-symmetric) potential*:

$$i_{\text{RS}}(q, r; \alpha_n, \rho_n) := \frac{1}{\alpha_n}I_{P_{0,n}}\left(\frac{\alpha_n}{\rho_n}r\right) + I_{P_{\text{out}}}(q, \mathbb{E}_{P_0}[X^2]) - \frac{r(\mathbb{E}_{P_0}[X^2] - q)}{2} . \tag{7}$$

Our first result links the extrema of this potential to the mutual information of our original problem.

**Theorem 1** (Mutual information of the GLM at sublinear sparsity and sampling rate)**.** *Suppose that $\Delta > 0$ and that the following hypotheses hold:*

- *(H1) There exists $S > 0$ such that the support of $P_0$ is included in $[-S, S]$.*
- *(H2) $\varphi$ is bounded, and its first and second partial derivatives with respect to its first argument exist, are bounded and continuous. They are denoted $\partial_x \varphi$, $\partial_{xx}\varphi$.*
- *(H3) $(\Phi_{\mu i}) \overset{iid}{\sim} \mathcal{N}(0,1)$.*

*Let $\rho_n = \Theta(n^{-\lambda})$ with $\lambda \in [0, 1/9]$ and $\alpha_n = \gamma\rho_n|\ln \rho_n|$ with $\gamma > 0$. Then for all $n \in \mathbb{N}^*$:*

$$\left|\frac{I(\mathbf{X}^*; \mathbf{Y}|\mathbf{\Phi})}{m_n} - \inf_{q \in [0, \mathbb{E}_{P_0}[X^2]]} \sup_{r \geq 0} i_{\text{RS}}(q, r; \alpha_n, \rho_n)\right| \leq \frac{\sqrt{C}|\ln n|^{1/6}}{n^{\frac{1}{12} - \frac{3\lambda}{4}}} , \tag{8}$$

*where $C$ is a polynomial in $\left(S, \left\|\frac{\varphi}{\sqrt{\Delta}}\right\|_\infty, \left\|\frac{\partial_x \varphi}{\sqrt{\Delta}}\right\|_\infty, \left\|\frac{\partial_{xx}\varphi}{\sqrt{\Delta}}\right\|_\infty, \lambda, \gamma\right)$ with positive coefficients.*

Hence, the asymptotic mutual information is given to leading order by the variational problem $\inf_{q\in[0,\mathbb{E}_{P_0}[X^2]]}\sup_{r\geq 0} i_{\mathrm{RS}}(q,r;\alpha_n,\rho_n)$. Note that this variational problems depends on $n$ and Theorem 1 does not say anything on its value in the asymptotic regime, e.g., does it converge or diverge? Our next theorem answers this question when $P_0$ is a discrete distribution with finite support.

## 2.3 Specialization to discrete priors: all-or-nothing phenomenon and its generalization

**Theorem 2** (Specialization of Theorem 1 to discrete priors with finite support). *Suppose that $\Delta > 0$ and that $P_{0,n} := (1-\rho_n)\delta_0 + \rho_n P_0$ where $P_0$ is a discrete distribution with finite support*

$$\mathrm{supp}(P_0) \subseteq \{-v_K, -v_{K-1}, \dots, -v_1, v_1, v_2, \dots, v_K\} \ ;$$

*where $0 < v_1 < v_2 < \cdots < v_K < v_{K+1} := +\infty$. Further assume that the hypotheses (H2) and (H3) in Theorem 1 hold. Let $\rho_n = \Theta(n^{-\lambda})$ with $\lambda \in (0, 1/9)$ and $\alpha_n = \gamma\rho_n|\ln\rho_n|$ with $\gamma > 0$. Then,*

$$\lim_{n\to+\infty}\frac{I(\mathbf{X}^*;\mathbf{Y}|\boldsymbol{\Phi})}{m_n} = \min_{1\leq k\leq K+1}\left\{I_{P_{\mathrm{out}}}\big(\mathbb{E}[X^2\mathbf{1}_{\{|X|\geq v_k\}}], \mathbb{E}[X^2]\big) + \frac{\mathbb{P}(|X|\geq v_k)}{\gamma}\right\}, \quad (9)$$

*where $X \sim P_0$.*

The proof of Theorem 2 requires computing the limit of $\inf_{q\in[0,\mathbb{E}_{P_0}[X^2]]}\sup_{r\geq 0} i_{\mathrm{RS}}(q,r;\alpha_n,\rho_n)$ and is given in the Supplementary Material.

When doing estimation, one important metric to assess the quality of an estimator $\widehat{\mathbf{X}}(\mathbf{Y},\boldsymbol{\Phi})$ is its mean-square error $\mathbb{E}\|\mathbf{X}^*-\widehat{\mathbf{X}}(\mathbf{Y},\boldsymbol{\Phi})\|^2/k_n$. The latter is always lower bounded by the mean-square error of the Bayesian estimator $\mathbb{E}[\mathbf{X}^*|\mathbf{Y},\boldsymbol{\Phi}]$; the so-called minimum mean-square error (MMSE). Remarkably, once we have Theorem 2, we can obtain the asymptotic MMSE with a little more work. First, we have to introduce a modified inference problem where in addition to the observations $\mathbf{Y}$ we are given $\widetilde{\mathbf{Y}}^{(\tau)} = \sqrt{\alpha_n\tau/\rho_n}\,\mathbf{X}^* + \widetilde{\mathbf{Z}}$. When $\tau$ is close enough to 0, the analysis yielding Theorem 2 can be adapted to obtain the limit

$$\lim_{n\to\infty}\frac{I(\mathbf{X}^*;\mathbf{Y},\widetilde{\mathbf{Y}}^{(\tau)}|\boldsymbol{\Phi})}{m_n}$$
$$= \min_{1\leq k\leq K+1}\left\{I_{P_{\mathrm{out}}}\big(\mathbb{E}[X^2\mathbf{1}_{\{|X|\geq v_k\}}], \mathbb{E}[X^2]\big) + \frac{\mathbb{P}(|X|\geq v_k)}{\gamma} + \frac{\tau\mathbb{E}[X^2\mathbf{1}_{\{|X|<v_k\}}]}{2}\right\}.$$

We can then apply the I-MMSE identity[2][38, 45] to obtain the asymptotic MMSE:

**Theorem 3** (Asymptotic MMSE). *Under the assumptions of Theorem 2, if the minimization problem on the right-hand side of (9) has a unique solution $k^* \in \{1,\dots,K+1\}$ then*

$$\lim_{n\to+\infty}\frac{\mathbb{E}\|\mathbf{X}^* - \mathbb{E}[\mathbf{X}^*|\mathbf{Y},\boldsymbol{\Phi}]\|^2}{k_n} = \mathbb{E}\big[X^2\mathbf{1}_{\{|X|<v_{k^*}\}}\big] \ , \text{ where } X \sim P_0 \ . \quad (10)$$

We prove Theorem 3 in the Supplementary Material. We remark that it is possible with more technical work [29, Appendix C.2] to weaken (H2) in Theorems 2 and 3 to the assumption "There exists $\epsilon > 0$ such that the sequence $\mathbb{E}|\varphi((\boldsymbol{\Phi}\mathbf{X}^*)_1/\sqrt{k_n}, \mathbf{A}_1)|^{2+\epsilon}$ is bounded, and for almost all $\mathbf{a} \sim P_A$ the function $x \mapsto \varphi(x, \mathbf{a})$ is continuous almost everywhere." Hence, Theorems 2 and 3 also apply to the linear activation $\varphi(x) = x$, the perceptron $\varphi(x) = \mathrm{sign}(x)$ and the ReLU $\varphi(x) = \max(0,x)$.

## 3 The all-or-nothing phenomenon

We now highlight interesting consequences of our results regarding the MMSE of the estimation problem as well as the optimal generalization error of the learning problem in the teacher-student scenario. Reeves et al. [31] have proved the existence of an *all-or-nothing phenomenon* for the linear model when $\mathbf{X}^*$ is a 0-1 vector and here we extend their results in two ways: $i$) for the estimation error of a generalized linear model, and $ii$) for the generalization error of a perceptron neural network with general activation function $\varphi$.

We consider signals whose entries are either Bernoulli random variables, i.e., $P_{0,n} := (1 - \rho_n)\delta_0 + \rho_n P_0$ with $P_0 = \delta_1$, or Bernoulli-Rademacher random variables, i.e., $P_{0,n} := (1 - \rho_n)\delta_0 + \rho_n P_0$ with $P_0 = (\delta_1 + \delta_{-1})/2$. In both cases $\mathbb{E}_{P_0}[X^2] = 1$ (we can always assume the latter by rescaling the noise). We place ourselves in the regime of Theorem 3 where $\alpha_n = \gamma \rho_n |\ln \rho_n|$ for some fixed $\gamma > 0$ and $\rho_n \to 0$ in the high-dimensional limit $n \to +\infty$.

**MMSE** In this regime, and for such signals, Theorem 3 states that the minimum mean-square error $\mathrm{MMSE}(\mathbf{X}^* | \mathbf{Y}, \mathbf{\Phi}) := \frac{\mathbb{E}\|\mathbf{X}^* - \mathbb{E}[\mathbf{X}^* | \mathbf{Y}, \mathbf{\Phi}]\|^2}{k_n}$ satisfies:

$$\lim_{n \to +\infty} \mathrm{MMSE}(\mathbf{X}^* | \mathbf{Y}, \mathbf{\Phi}) = \begin{cases} 0 & \text{if } I_{P_{\mathrm{out}}}(0,1) > \gamma^{-1} \text{ ;} \\ 1 & \text{if } I_{P_{\mathrm{out}}}(0,1) < \gamma^{-1} \text{ .} \end{cases} \tag{11}$$

Therefore, we locate an *all-or-nothing phase transition* at the threshold

$$\gamma_c := \frac{1}{I_{P_{\mathrm{out}}}(0,1)} \text{ .} \tag{12}$$

Remember that $\gamma$ controls the amount $m_n$ of training samples. In the high-dimensional limit, perfect reconstruction is possible if $\gamma > \gamma_c$ (the asymptotic MMSE is zero) while it is impossible to do better than a random guess if $\gamma < \gamma_c$ (the asymptotic MMSE is equal to $\lim_{n \to +\infty} \mathbb{E}\|\mathbf{X}^* - \mathbb{E}\mathbf{X}^*\|^2/k_n = 1$; the asymptotic MMSE in the absence of observations). As $I_{P_{\mathrm{out}}}(0,1) := I(W^*; \varphi(W^*, \mathbf{A}) + \sqrt{\Delta}Z)$ where $W^*, Z \overset{\mathrm{iid}}{\sim} \mathcal{N}(0,1) \perp \mathbf{A} \sim P_A$, the threshold $\gamma_c$ is fully determined by the activation function and the amount of noise, and it can be easily evaluated in a number of cases. In Figure 1 we draw $\gamma_c$ for $\varphi(x) = x$, $\varphi(x) = \mathrm{sign}(x)$, $\varphi(x) = \max(0, x)$ and noise variance $\Delta \in [0, 0.5]$. We see that for $\Delta$ small enough the ReLU activation requires less training samples to learn the sparse rule than the linear one; it is the opposite once $\Delta$ becomes large enough. When $\Delta$ diverges both the linear and sign activations have the asymptote $\gamma_c \sim 2\Delta$ while the ReLU activation has another steeper asymptote $\gamma_c \sim a\Delta$, $a \approx 5.87$. The corresponding formulas for $\gamma_c$ are given in Table 1. Note that for the random linear model $\varphi(x) = x$, the threshold $\alpha_c(\rho_n) := \gamma_c \rho_n |\ln \rho_n| = 2\rho_n |\ln \rho_n|/\ln(1 + \Delta^{-1})$ is in agreement with the sample rate $n^*$ for which [31] prove that weak recovery is impossible below it while strong recovery is possible above.

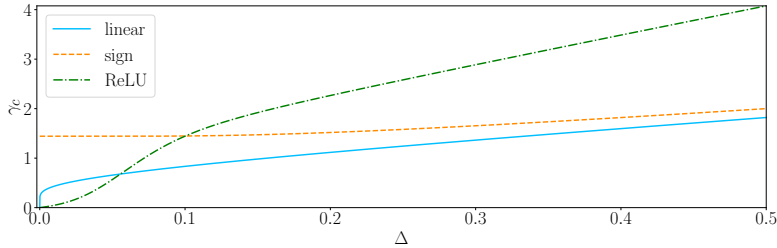

Figure 1: Threshold $\gamma_c$ of the all-or-nothing phase transition for different activation functions as a function of the noise variance $\Delta$.

| Activation $\varphi(x)$ | $\gamma_c(\Delta = 0)$ | $\gamma_c(\Delta)$ for $\Delta > 0$ |
|:---:|:---:|:---|
| $x$ | $0$ | $2/\ln(1 + \Delta^{-1})$ |
| $\mathrm{sign}(x)$ | $1/\ln 2$ | $1/\left(\ln 2 - \mathbb{E}[\ln(1 + e^{-2(1 + \sqrt{\Delta}Z)/\Delta})]\right)$ |
| $\max(0, x)$ | $0$ | $4\Delta/\left(1 - 4\Delta\mathbb{E}[h_\Delta(Z)\ln h_\Delta(Z)]\right)$ <br> with $h_\Delta(Z) := \frac{1}{2} + \sqrt{\frac{\Delta}{1+\Delta}} e^{\frac{Z^2}{2(1+\Delta)}} \int_{-\infty}^{\frac{Z}{\sqrt{1+\Delta}}} \frac{dt}{\sqrt{2\pi}} e^{-\frac{t^2}{2}}$ |

Table 1: Closed-formed formulas of $\gamma_c$ for different activation functions. We use $Z \sim \mathcal{N}(0,1)$.

**Optimal generalization error** When learning in a (matched) teacher-student scenario, the components of $\mathbf{X}^*$ correspond to the unknown weights of the teacher's one-layer neural network. The

student is given the model and training samples $\{(Y_\mu, (\Phi_{\mu,i})_{i=1}^n)\}_{\mu=1}^{m_n}$. Then, the optimal generalization error is the MMSE for predicting the output $Y_{\text{new}} \sim P_{\text{out}}(\,\cdot\,|\mathbf{\Phi}_{\text{new}}^\mathsf{T}\mathbf{X}^*/\sqrt{k_n})$ generated by a new input $\mathbf{\Phi}_{\text{new}} := (\Phi_{\text{new},i}) \stackrel{\text{iid}}{\sim} \mathcal{N}(0,1)$. More precisely, the optimal generalization error is $\text{MMSE}(Y_{\text{new}}|\mathbf{Y},\mathbf{\Phi},\mathbf{\Phi}_{\text{new}}) := \mathbb{E}[(Y_{\text{new}} - \mathbb{E}[Y_{\text{new}}|\mathbf{Y},\mathbf{\Phi},\mathbf{\Phi}_{\text{new}}])^2]$. Based on our proof of Theorem 3 and the optimal generalization error when $\rho_n = \Theta(1)$ (regime of linear sparsity and sampling rate) [29, Theorem 2], we conjecture that, under the assumptions of Theorem 3,

$$\lim_{n \to +\infty} \text{MMSE}(Y_{\text{new}}|\mathbf{Y},\mathbf{\Phi},\mathbf{\Phi}_{\text{new}}) = \Delta + \mathbb{E}\big[\big(\varphi(V,\mathbf{A}) - \mathbb{E}[\varphi(\sqrt{q^*}\,V + \sqrt{\mathbb{E}X^2 - q^*}\,W^*,\mathbf{A})|V]\big)^2\big]$$
(13)

where $V, W^* \sim \mathcal{N}(0,1) \perp \mathbf{A} \sim P_A$ and $q^*$ is such that $\mathbb{E}X^2 - q^* = \mathbb{E}[X^2 \mathbf{1}_{\{|X| < v_{k^*}\}}]$ is the asymptotic MMSE (10). For Bernoulli and Bernoulli-Rademacher signals (the ones considered in this section), it simplifies to

$$\lim_{n \to +\infty} \text{MMSE}(Y_{\text{new}}|\mathbf{Y},\mathbf{\Phi},\mathbf{\Phi}_{\text{new}}) = \begin{cases} \Delta + \mathbb{E}[(\varphi(V,\mathbf{A}) - \mathbb{E}[\varphi(V,\mathbf{A})|V])^2] & \text{if } \gamma > \gamma_c \,; \\ \Delta + \mathbb{V}\text{ar}(\varphi(V,\mathbf{A})) & \text{if } \gamma < \gamma_c \,. \end{cases}$$
(14)

We thus find that the optimal generalization error also displays an all-or-nothing phase transition at $\gamma_c$. More precisely, if $\gamma < \gamma_c$ then the optimal generalization error equals $\Delta + \mathbb{V}\text{ar}(\varphi(V,\mathbf{A}))$ when $n \to +\infty$. This is the same generalization error achieved by the dumb label estimator in the Bayesian sense; the one predicting the new label to be the output value averaged over all possible inputs, weights and noise. If instead $\gamma > \gamma_c$ then it is equal to $\Delta + \mathbb{E}[\mathbb{V}\text{ar}(\varphi(V,\mathbf{A})|V)]$; the irreducible error due to both the noise $\mathbf{Z}$ and the random stream $(\mathbf{A}_\mu)_{\mu=1}^{m_n}$.

Proving (13) entails introducing side observations in the original problem and differentiating with respect to the signal-to-noise ratio of this side channel to exploit the I-MMSE relation, in a similar fashion to what we do in the proof of Theorem 3 (see Supplementary Material). The side observations have the same form than the ones used in [29, Section 5 of SI Appendix] to determine the asymptotic optimal generalization error in the regime of linear sparsity and sampling rate.

**Illustration of the all-or-nothing phenomenon** In Figure 2 we use (11) to draw in solid black lines the asymptotic MMSE in the regime of sublinear sparsity and sampling rate, for both priors Bernoulli and Bernoulli-Rademacher and the activation functions $\varphi(x) = x$, $\varphi(x) = \text{sign}(x)$, $\varphi(x) = \max(0, x)$. For comparison we also draw in dashed colored lines the asymptotic MMSE in regimes of linear sparsity and sampling rate, that is, $\rho_n = \rho$ and $\alpha_n = \gamma\rho|\ln\rho|$ are constant with $n$. In this case, the asymptotic MMSE is given by [29, Theorem 2]

$$\lim_{n \to +\infty} \text{MMSE}(\mathbf{X}^*|\mathbf{Y},\mathbf{\Phi}) = 1 - q^* \,,$$
(15)

whenever $\arg\min_{q \in [0,1]} \sup_{r \geq 0} i_{\text{RS}}(q, r; \gamma\rho|\ln\rho|, \rho)$ is a singleton $\{q^*\}$. To optimize the potential $i_{\text{RS}}(q, r; \gamma\rho|\ln\rho|, \rho)$ we initialize $q \in [0,1]$ at different values and iterate the following fixed point equation (obtained directly by setting the gradient of the potential to zero):

$$r = -2\frac{\partial I_{P_{\text{out}}}}{\partial q}\bigg|_{q,1} \quad, \quad q = -\frac{2}{\rho_n}I'_{P_{0,n}}\left(\frac{\alpha_n}{\rho_n}r\right).$$
(16)

Finally, the fixed point $q^*$ yielding the lowest potential $\sup_{r \geq 0} i_{\text{RS}}(q^*, r; \gamma\rho|\ln\rho|, \rho)$ is used to determine the MMSE thanks to (15). In all configurations the asymptotic MMSE jumps from a value close to 1 to approximately 0 as $\gamma$ increases past $\gamma_c$. As $\rho_n = \rho$ gets closer to 0, this jump becomes sharper with the MMSE approaching 0 or 1 depending on which side of $\gamma_c$ we are. Though this jump becomes sharper, a pure all-or-nothing phase transition only occurs in the regime of sublinear sparsity and sampling rate (solid black lines).

In Figure 3 we use (14) to plot in solid black lines the asymptotic optimal generalization error for the Bernoulli prior and the same activation functions. The dashed colored lines again correspond to regimes of linear sparsity and sampling rate; they are obtained using the formula for the asymptotic optimal generalization error given by [29, Theorem 2]:

$$\lim_{n \to +\infty} \text{MMSE}(Y_{\text{new}}|\mathbf{Y},\mathbf{\Phi},\mathbf{\Phi}_{\text{new}}) = \Delta + \mathbb{E}\big[\big(\varphi(V,\mathbf{A}) - \mathbb{E}[\varphi(\sqrt{q^*}\,V + \sqrt{1 - q^*}\,W^*,\mathbf{A})|V]\big)^2\big].$$
(17)

In all configurations the optimal generalization error jumps from a value close to $\Delta + \mathbb{V}\text{ar}(\varphi(V))$ to approximately $\Delta$ as $\gamma$ increases past $\gamma_c$ (note that the activations are deterministic so there is no

contribution from **A** in the error). The value $\Delta$ is as good as the optimal generalization error can get, i.e., it is equal to the noise variance which is the squared error we would get if we were given the true weights $\mathbf{X}^*$. Again, the jump gets sharper as $\rho_n = \rho$ approaches 0 but a pure all-or-nothing phase transition only occurs in the regime of sublinear sparsity and sampling rate (solid black lines).

The all-or-nothing behavior of the asymptotic MMSE and optimal generalization error is quite striking. Indeed, in the limit of vanishing sparsity and sampling rate either estimation or learning is as good as it can get or as bad as a random guess. This purely dichotomic behavior only occurs in the truly sparse limit, and is shown here to be pretty general in the sense that it occurs for a wide variety of activation functions. An important aspect of our results is to provide a definitive statistical benchmark allowing to measure the quality of algorithms with respect to the minimal amount of sparse data needed to estimate or learn. This benchmark is provided by non-trivial formulas (12) for the threshold $\gamma_c$ given for several examples in Table 1. We note that such precise benchmarks are quite rarely obtained in traditional machine learning approaches.

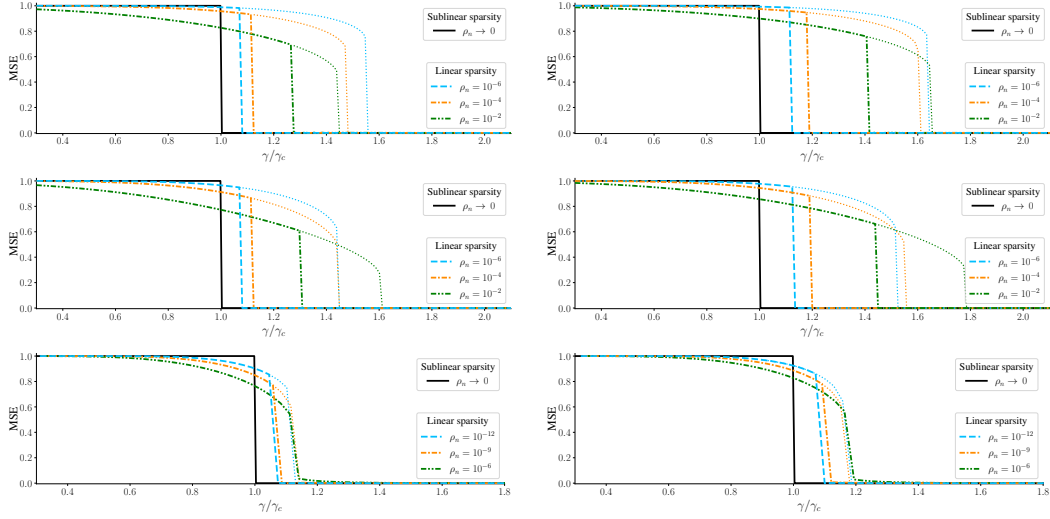

Figure 2: Asymptotic MMSE as a function of $\gamma/\gamma_c$ in the regime of sublinear sparsity and sampling rate ($\rho_n = \Theta(n^{-\lambda})$ with $\lambda \in (0, 1/9)$, solid black line), and in the regime of linear sparsity and sampling rate ($\rho_n$ fixed, dashed colored lines). Dotted lines correspond to algorithmic performance in the regime of linear sparsity and sampling rate (iterating (16) from $q = 10^{-10}$). *Left panels:* Bernoulli prior. *Right panels:* Bernoulli-Rademacher prior. *From top to bottom:* $\varphi(x) = x, \Delta = 0.1$; $\varphi(x) = \text{sign}(x), \Delta = 0$; $\varphi(x) = \max(0, x), \Delta = 0.5$.

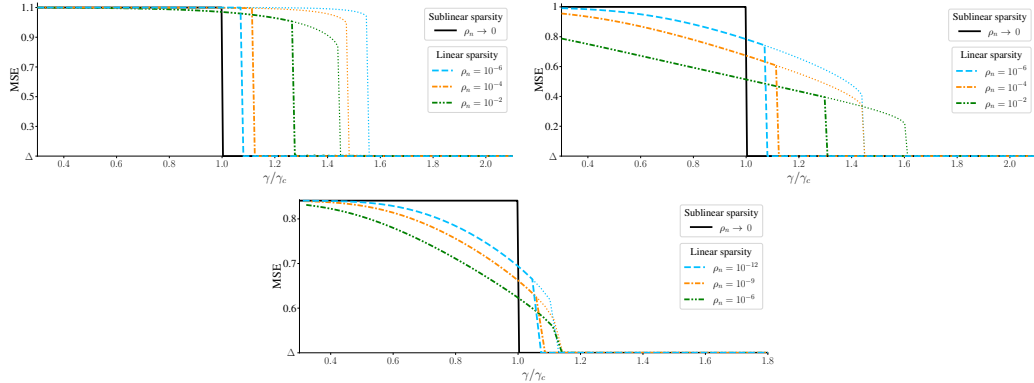

Figure 3: Asymptotic optimal generalization error as a function of $\gamma/\gamma_c$ in the regime of sublinear sparsity and sampling rate ($\rho_n = \Theta(n^{-\lambda})$ with $\lambda \in (0, 1/9)$, solid black line), and in the regime of linear sparsity and sampling rate ($\rho_n$ is fixed, dashed colored lines). Dotted lines correspond to algorithmic performance in the regime of linear sparsity and sampling rate (iterating (16) from $q = 10^{-10}$). *Top left:* random linear model $\varphi(x) = x$, $\Delta = 0.1$. *Top right:* perceptron $\varphi(x) = \text{sign}(x), \Delta = 0$. *Bottom:* ReLU $\varphi(x) = \max(0, x), \Delta = 0.5$.

**Further remarks** Algorithmic aspects are beyond the scope of this paper. However, we make a few remarks about generalized approximate message passing (GAMP) algorithms. In the regime of linear sparsity and sampling rate, the state evolution equations precisely tracking the asymptotic performance of the algorithm are linked to the fixed point equation (16) [46]. The fixed point $q^{\text{alg}}$ reached by initializing (16) arbitrarily close to $q = 0$ can be used in (15) and (17) – instead of $q^*$– to obtain both the mean-square and generalization errors of GAMP algorithms. These errors are represented with dotted colored lines in Figures 2 and 3. We observe an algorithmic-to-statistical gap, that is, the dotted lines corresponding to the algorithmic performance do not drop to zero around $\gamma_c$ but at a higher *algorithmic threshold*. In this work we don't study the performance of GAMP algorithms in the regime of sublinear sparsity and sampling rate. However, reference [32] rigorously shows that in this regime the all-or-nothing behavior also occurs at an algorithmic level for GAMP algorithms. It would be highly desirable to extend their results to other activations and derive the corresponding thresholds.

## 4   Overview of the proof of Theorem 1

The interested reader will find the proof of Theorem 1 in the supplementary material. In this section we give an outline of the proof and its main ideas. The proof is based on the adaptive interpolation method [39, 40] whose main difference with the canonical interpolation method [47, 48] is the increased flexibility given to the path followed by the interpolation between its two extremes. The method has been developed separately for symmetric rank-one tensor problems where the spike has i.i.d. components [39, 40], and for one-layer GLMs whose input signal has again i.i.d. components [29]. The sparse regime of the problem studied in this contribution differs of the usual scaling for which such techniques have been developed. They have been used in a regime where the number of measurements and sparsity are linear in $n$ as in [29]. Working in the sparse regime requires writing more refined concentration bounds and proving that the key steps of the adaptive interpolation can still be carried through.

**1. Interpolating estimation problem** To simplify the presentation we assume that $\Delta = 1$ and $\mathbb{E}_{X \sim P_0}[X^2] = 1$. The proof starts by introducing an interpolating inference problem that depends on a parameter $t \in [0,1]$ and two continuous interpolation functions $R_1, R_2 : [0,1] \to \mathbb{R}_+$ with $R_1(0) = R_2(0) = 0$. Let $\mathbf{X}^* \overset{\text{iid}}{\sim} P_{0,n}$, $\boldsymbol{\Phi} := (\Phi_{\mu i}) \overset{\text{iid}}{\sim} \mathcal{N}(0,1)$, $\mathbf{V} := (V_\mu)_{\mu=1}^{m_n} \overset{\text{iid}}{\sim} \mathcal{N}(0,1)$ and $\mathbf{W}^* := (W_\mu^*)_{\mu=1}^{m_n} \overset{\text{iid}}{\sim} \mathcal{N}(0,1)$. We define for all $t \in [0,1]$ an "interpolating pre-activation":

$$S_\mu^{(t)} := \sqrt{(1-t)/k_n}\, (\boldsymbol{\Phi}\mathbf{X}^*)_\mu + \sqrt{R_2(t)}\, V_\mu + \sqrt{t - R_2(t)}\, W_\mu^* \,.$$

The inference problem at a fixed $t$ is to recover both unknowns $\mathbf{X}^*, \mathbf{W}^*$ from the knowledge of $\mathbf{V}$, $\boldsymbol{\Phi}$ and the data

$$\begin{cases} Y_\mu^{(t)} & \sim \quad P_{\text{out}}(\cdot \,|\, S_\mu^{(t)}) \quad, \ 1 \le \mu \le m_n\,; \\ \widetilde{Y}_i^{(t)} & = \ \sqrt{R_1(t)}\, X_i^* + \widetilde{Z}_i \,, \ 1 \le i \le n \ \, ; \end{cases}$$

where $Z_\mu, \widetilde{Z}_i \overset{\text{iid}}{\sim} \mathcal{N}(0,1)$. The corresponding *interpolating mutual information* is:

$$i_n(t) := m_n^{-1} I\big((\mathbf{X}^*, \mathbf{W}^*)\,;\, (\mathbf{Y}^{(t)}, \widetilde{\mathbf{Y}}^{(t)}) \big| \boldsymbol{\Phi}, \mathbf{V}\big) \,.$$

**2. Fundamental sum-rule** Note that at $t = 0$ we recover the original problem of interest and $i_n(0) = I(\mathbf{X}^*; \mathbf{Y}|\boldsymbol{\Phi})/m_n$. At the other extreme $t = 1$, the mutual information can be written in terms of the simple mutual informations $I_{P_{0,n}}$ and $I_{P_{\text{out}}}$, that is, $i_n(1) = I_{P_{0,n}}(R_1(1))/\alpha_n + I_{P_{\text{out}}}(R_2(1), 1)$. We link the mutual information at both extremes by computing the derivative $i_n'(\cdot)$ of $i_n(\cdot)$ and then using the fundamental identity $i_n(0) = i_n(1) - \int_0^1 i_n'(t)dt$. It yields the sum-rule:

$$\frac{I(\mathbf{X}^*; \mathbf{Y}|\boldsymbol{\Phi})}{m_n} = \frac{1}{\alpha_n} I_{P_{0,n}}(R_1(1)) + I_{P_{\text{out}}}(R_2(1), 1) - \frac{\rho_n}{2\alpha_n} \int_0^1 R_1'(t)\big(1 - R_2'(t)\big)dt + \mathcal{R}_n \,.$$

The last term $\mathcal{R}_n$ is a remainder whose absolute value we want to control in order to get Theorem 1.

**3. Controlling the remainder** This is done by plugging two different choices of interpolation functions $(R_1, R_2)$ in the sum-rule. One choice yields an upper bound on the difference in the left-hand side of (8), while another yields a lower bound. Each choice of interpolation functions $(R_1, R_2)$ is defined implicitly as the solution to a first-order ordinary differential equation. Remarkably, under these two choices, the remainder $\mathcal{R}_n$ can be controlled using precise concentration results.

## Broader Impact

We believe that it is difficult to clearly foresee societal consequence of the present, purely theoretical, work. The results presented inscribe themselves in the larger theme of providing guidelines for better and parsimonious use of data when possible, for example when learning a sparse rule. On the long run, such guidelines must be taken into account for building engineering systems that are more efficient in terms of computational and energetic cost.

## Acknowledgments and Disclosure of Funding

The work of C. L. is supported by the Swiss National Foundation for Science grant number 200021E 17554.

## Footnotes

[2]The derivative of $I(\mathbf{X}^*;\mathbf{Y},\widetilde{\mathbf{Y}}^{(\tau)}|\boldsymbol{\Phi})/m_n$ with respect to $\tau$ at $\tau = 0$ is equal to half the MMSE of the original problem.

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
