[Supplementary Material]

# Supplementary material:
# Information theoretic limits of learning a sparse rule

**Clément Luneau**[*]**, Nicolas Macris**
Ecole Polytechnique Fédérale de Lausanne
Suisse

**Jean Barbier**
International Center for Theoretical Physics
Trieste, Italy

## Contents

[*]Corresponding author: clement.luneau@epfl.ch

# 1 Problem setting and results

For the reader's convenience, we repeat the problem setting and Theorems 1, 2 and 3 whose proofs are given in this Supplementary Material.

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

*where $C$ is a polynomial in $\big(S, \big\|\frac{\varphi}{\sqrt{\Delta}}\big\|_\infty, \big\|\frac{\partial_x\varphi}{\sqrt{\Delta}}\big\|_\infty, \big\|\frac{\partial_{xx}\varphi}{\sqrt{\Delta}}\big\|_\infty, \lambda, \gamma\big)$ with positive coefficients.*

We prove Theorem 1 in Section 2.

**Theorem 2** (Specialization of Theorem 1 to discrete priors with finite support)**.** *Suppose that $\Delta > 0$ and that $P_{0,n} := (1-\rho_n)\delta_0 + \rho_n P_0$ where $P_0$ is a discrete distribution with finite support*

$$\text{supp}(P_0) \subseteq \{-v_K, -v_{K-1}, \ldots, -v_1, v_1, v_2, \ldots, v_K\}\ ;$$

*where $0 < v_1 < v_2 < \cdots < v_K < v_{K+1} := +\infty$. Further assume that the hypotheses (H2),(H3) in Theorem 1 hold. Let $\rho_n = \Theta(n^{-\lambda})$ with $\lambda \in (0, 1/9)$ and $\alpha_n = \gamma\rho_n|\ln\rho_n|$ with $\gamma > 0$. Then,*

$$\lim_{n \to +\infty} \frac{I(\mathbf{X}^*; \mathbf{Y}|\mathbf{\Phi})}{m_n} = \min_{1 \leq k \leq K+1} \left\{ I_{P_{\text{out}}}\big(\mathbb{E}[X^2\mathbf{1}_{\{|X| \geq v_k\}}], \mathbb{E}[X^2]\big) + \frac{\mathbb{P}(|X| \geq v_k)}{\gamma} \right\}\ , \quad (9)$$

*where $X \sim P_0$.*

We prove Theorem 2 for the special case of a Bernoulli prior $P_{0,n}$ in Section 3. The proof of this special case contains all the main ideas needed to establish Theorem 2 while being technically simpler. We give the proof for a general discrete prior with finite support in Appendix F.

**Theorem 3** (Asymptotic MMSE)**.** *Suppose that $\Delta > 0$ and that $P_{0,n} := (1-\rho_n)\delta_0 + \rho_n P_0$ where $P_0$ is a discrete distribution with finite support*

$$\text{supp}(P_0) \subseteq \{-v_K, -v_{K-1}, \ldots, -v_1, v_1, v_2, \ldots, v_K\}\ ;$$

*where $0 < v_1 < v_2 < \cdots < v_K < v_{K+1} := +\infty$. Further assume that the hypotheses (H2),(H3) in Theorem 1 hold. Let $\rho_n = \Theta(n^{-\lambda})$ with $\lambda \in (0, 1/9)$ and $\alpha_n = \gamma\rho_n|\ln\rho_n|$ with $\gamma > 0$. If the minimization problem on the right-hand side of (9) has a unique solution $k^* \in \{1, \ldots, K+1\}$ then*

$$\lim_{n \to +\infty} \frac{\mathbb{E}\|\mathbf{X}^* - \mathbb{E}[\mathbf{X}^*|\mathbf{Y}, \mathbf{\Phi}]\|^2}{k_n} = \mathbb{E}\big[X^2\mathbf{1}_{\{|X| < v_{k^*}\}}\big]\ , \text{ where } X \sim P_0\ . \quad (10)$$

We prove Theorem 3 in Section 4.

## 2 Proof of Theorem 1 with the adaptive interpolation method

Note that it is the same to observe (2) or their rescaled versions $\frac{1}{\sqrt{\Delta}}\varphi\big(\frac{1}{\sqrt{k_n}}(\boldsymbol{\Phi X}^*)_\mu, \mathbf{A}_\mu\big) + Z_\mu$. Therefore, up to a rescaling of $\varphi$ by $1/\sqrt{\Delta}$, we will suppose that $\Delta = 1$ all along the proof of Theorem 1. For a similar reason, we can suppose that $\mathbb{E}_{X \sim P_0}[X^2] = 1$.

### 2.1 Interpolating estimation problem

We fix a sequence $(s_n)_{n\in\mathbb{N}^*} \in (0, 1/2]$ and define $\mathcal{B}_n := [s_n, 2s_n]^2$. Let $r_{\max} := -2\frac{\partial I_{P_{\text{out}}}}{\partial q}\big|_{q=1,\rho=1}$ a positive real number. For all $\epsilon = (\epsilon_1, \epsilon_2) \in \mathcal{B}_n$, we define the *interpolation functions*

$$R_1(\cdot, \epsilon) : t \in [0, 1] \mapsto \epsilon_1 + \int_0^t r_\epsilon(v)dv \quad \text{and} \quad R_2(\cdot, \epsilon) : t \in [0, 1] \mapsto \epsilon_2 + \int_0^t q_\epsilon(v)dv \,,$$

where $q_\epsilon : [0, 1] \to [0, 1]$ and $r_\epsilon : [0, 1] \to [0, \frac{\alpha_n}{\rho_n}r_{\max}]$ are two continuous functions. We say that the families of functions $(q_\epsilon)_{\epsilon\in\mathcal{B}_n}$ and $(r_\epsilon)_{\epsilon\in\mathcal{B}_n}$ are *regular* if $\forall t \in [0, 1] : \epsilon \mapsto \big(R_1(t, \epsilon), R_2(t, \epsilon)\big)$ is a $\mathcal{C}^1$ diffeomorphism from $\mathcal{B}_n$ onto its image whose Jacobian determinant is greater than, or equal, to one. This property will reveal important later in our proof. Let $\mathbf{X}^* \overset{\text{iid}}{\sim} P_{0,n}$, $\boldsymbol{\Phi} := (\Phi_{\mu i}) \overset{\text{iid}}{\sim} \mathcal{N}(0, 1)$, $\mathbf{V} := (V_\mu)_{\mu=1}^{m_n} \overset{\text{iid}}{\sim} \mathcal{N}(0, 1)$ and $\mathbf{W}^* := (W_\mu^*)_{\mu=1}^{m_n} \overset{\text{iid}}{\sim} \mathcal{N}(0, 1)$. We define:

$$S_\mu^{(t,\epsilon)} = S_\mu^{(t,\epsilon)}(\mathbf{X}^*, W_\mu^*) := \sqrt{\frac{1-t}{k_n}}\,(\boldsymbol{\Phi X}^*)_\mu + \sqrt{R_2(t, \epsilon)}\,V_\mu + \sqrt{t + 2s_n - R_2(t, \epsilon)}\,W_\mu^* \,. \tag{11}$$

Consider the following observations coming from two types of channels:

$$\begin{cases} Y_\mu^{(t,\epsilon)} & \sim \quad P_{\text{out}}(\,\cdot\,|\,S_\mu^{(t,\epsilon)}) \quad , \; 1 \le \mu \le m_n \,; \\ \widetilde{Y}_i^{(t,\epsilon)} & = \sqrt{R_1(t, \epsilon)}\,X_i^* + \widetilde{Z}_i \,, \; 1 \le i \le n \quad ; \end{cases} \tag{12}$$

where $(\widetilde{Z}_i)_{i=1}^n \overset{\text{iid}}{\sim} \mathcal{N}(0, 1)$. The inference problem (at time $t$) is to recover both unknowns $\mathbf{X}^*, \mathbf{W}^*$ from the knowledge of $\mathbf{V}$, $\boldsymbol{\Phi}$ and the observations $\mathbf{Y}^{(t,\epsilon)} := (Y_\mu^{(t,\epsilon)})_{\mu=1}^{m_n}$, $\widetilde{\mathbf{Y}}^{(t,\epsilon)} := (\widetilde{Y}_i^{(t,\epsilon)})_{i=1}^n$. The joint posterior density of $(\mathbf{X}^*, \mathbf{W}^*)$ given $(\mathbf{Y}^{(t,\epsilon)}, \widetilde{\mathbf{Y}}^{(t,\epsilon)}, \boldsymbol{\Phi}, \mathbf{V})$ reads:

$$dP(\mathbf{x}, \mathbf{w}|\mathbf{Y}^{(t,\epsilon)}, \widetilde{\mathbf{Y}}^{(t,\epsilon)}, \boldsymbol{\Phi}, \mathbf{V})$$

$$:= \frac{1}{\mathcal{Z}_{t,\epsilon}} \prod_{i=1}^n dP_{0,n}(x_i)\, e^{-\frac{1}{2}\big(\sqrt{R_1(t,\epsilon)}\,x_i - \widetilde{Y}_i^{(t,\epsilon)}\big)^2} \prod_{\mu=1}^{m_n} \frac{dw_\mu}{\sqrt{2\pi}} e^{-\frac{w_\mu^2}{2}} P_{\text{out}}(Y_\mu^{(t,\epsilon)}|s_\mu^{(t,\epsilon)}) \,, \tag{13}$$

where $s_\mu^{(t,\epsilon)} := S_\mu^{(t,\epsilon)}(\mathbf{x}, w_\mu)$ and $\mathcal{Z}_{t,\epsilon} \equiv \mathcal{Z}_{t,\epsilon}(\mathbf{Y}^{(t,\epsilon)}, \widetilde{\mathbf{Y}}^{(t,\epsilon)}, \boldsymbol{\Phi}, \mathbf{V})$ is the normalization. The *interpolating mutual information* is:

$$i_{n,\epsilon}(t) := \frac{1}{m_n}I\big((\mathbf{X}^*, \mathbf{W}^*); (\mathbf{Y}^{(t,\epsilon)}, \widetilde{\mathbf{Y}}^{(t,\epsilon)})\big|\boldsymbol{\Phi}, \mathbf{V}\big) \,. \tag{14}$$

The *perturbation* $\epsilon$ only induces a small change in mutual information. In particular, at $t = 0$:

**Lemma 1.** *Suppose that (H1), (H2), (H3) hold, that $\Delta = \mathbb{E}_{P_0}[X^2] = 1$ and that there exist real positive numbers $M_\alpha, M_{\rho/\alpha}$ such that $\forall n \in \mathbb{N}^*$: $\alpha_n \le M_\alpha$ and $\rho_n/\alpha_n \le M_{\rho/\alpha}$. For all $\epsilon \in \mathcal{B}_n$:*

$$\left| i_{n,\epsilon}(0) - \frac{I(\mathbf{X}^*; \mathbf{Y}|\boldsymbol{\Phi})}{m_n} \right| \le \sqrt{C}\,\frac{s_n}{\sqrt{\rho_n}} \,,$$

*where $C$ is a polynomial in $\big(S, \|\varphi\|_\infty, \|\partial_x\varphi\|_\infty, \|\partial_{xx}\varphi\|_\infty, M_\alpha, M_{\rho/\alpha}\big)$ with positive coefficients.*

We prove Lemma 1 in Appendix B.2. By the chain rule for mutual information and the Lipschitzianity of $I_{P_{0,n}}, I_{P_{\text{out}}}$ (see Lemmas 6 and 7 in Appendix A), at $t = 1$ we have for all $\epsilon \in \mathcal{B}_n$:

$$i_{n,\epsilon}(1) = \frac{I(\mathbf{X}^*; \widetilde{\mathbf{Y}}^{(1,\epsilon)}|\boldsymbol{\Phi}) + I(\mathbf{W}^*; \mathbf{Y}^{(1,\epsilon)}|\boldsymbol{\Phi}, \mathbf{V})}{m_n} = \frac{I_{P_{0,n}}(R_1(1, \epsilon))}{\alpha_n} + I_{P_{\text{out}}}(R_2(1, \epsilon), 1 + 2s_n)$$

$$= \frac{1}{\alpha_n}I_{P_{0,n}}\left(\int_0^1 r_\epsilon(t)dt\right) + I_{P_{\text{out}}}\left(\int_0^1 q_\epsilon(t)dt, 1\right) + \mathcal{O}(s_n) \,, \tag{15}$$

assuming there exists $M_{\rho/\alpha} > 0$ such that $\forall n \in \mathbb{N}^* : \rho_n/\alpha_n \le M_{\rho/\alpha}$. $\mathcal{O}(s_n)$ is a quantity whose absolute value is bounded by $Cs_n$ where $C$ is a polynomial in $\big(S, \|\varphi\|_\infty, \|\partial_x\varphi\|_\infty, \|\partial_{xx}\varphi\|_\infty, M_{\rho/\alpha}\big)$ with positive coefficients.

## 2.2 Fundamental sum rule

We want to *compare* the original model of interest (model at $t = 0$) to the purely scalar one ($t = 1$). To do so, we use $i_{n,\epsilon}(0) = i_{n,\epsilon}(1) - \int_0^1 i'_{n,\epsilon}(t)dt$ where $i'_{n,\epsilon}(\cdot)$ is the derivative of $i_{n,\epsilon}(\cdot)$. Once combined with Lemma 1 and (15), it yields (note that $\mathcal{O}(s_n) = \mathcal{O}(s_n/\sqrt{\rho_n})$ since $0 < \rho_n < 1$):

$$\frac{I(\mathbf{X}^*; \mathbf{Y} | \boldsymbol{\Phi})}{m_n} = \mathcal{O}\left(\frac{s_n}{\sqrt{\rho_n}}\right) + \frac{1}{\alpha_n} I_{P_{0,n}}\left(\int_0^1 r_\epsilon(t)dt\right) + I_{P_{\text{out}}}\left(\int_0^1 q_\epsilon(t)dt, 1\right)$$
$$- \int_0^1 i'_{n,\epsilon}(t)dt \,. \tag{16}$$

From now on let $(\mathbf{x}, \mathbf{w}) \in \mathbb{R}^n \times \mathbb{R}^{m_n}$ be a pair of random vectors sampled from the joint posterior distribution (13). The angular brackets $\langle - \rangle_{n,t,\epsilon}$ denote an expectations w.r.t. the distribution (13), i.e., $\langle g(\mathbf{x}, \mathbf{w}) \rangle_{n,t,\epsilon} := \int g(\mathbf{x}, \mathbf{w}) dP(\mathbf{x}, \mathbf{w} | \mathbf{Y}^{(t,\epsilon)}, \widetilde{\mathbf{Y}}^{(t,\epsilon)}, \boldsymbol{\Phi}, \mathbf{V})$ for every integrable function $g$. We define the scalar overlap $Q := \frac{1}{k_n} \sum_{i=1}^n X_i^* x_i$. The computation of $i'_{n,\epsilon}$ is found in Appendix B.1.

**Proposition 1.** *Suppose that (H1), (H2), (H3) hold and that $\Delta = \mathbb{E}_{X \sim P_0}[X^2] = 1$. Further assume that there exist real positive numbers $M_\alpha, M_{\rho/\alpha}$ such that $\forall n \in \mathbb{N}^*$: $\alpha_n \le M_\alpha$ and $\rho_n/\alpha_n \le M_{\rho/\alpha}$. Define $u_y(x) := \ln P_{\text{out}}(y|x)$ and $u'_y(\cdot)$ its derivative w.r.t. $x$. For all $(t, \epsilon) \in [0, 1] \times \mathcal{B}_n$:*

$$i'_{n,\epsilon}(t) = \mathcal{O}\left(\frac{1}{\rho_n \sqrt{n}}\right) + \frac{\rho_n}{2\alpha_n} r_\epsilon(t)(1 - q_\epsilon(t))$$
$$+ \frac{1}{2} \mathbb{E}\left\langle (Q - q_\epsilon(t)) \left( \frac{1}{m_n} \sum_{\mu=1}^{m_n} u'_{Y_\mu^{(t,\epsilon)}}(S_\mu^{(t,\epsilon)}) u'_{Y_\mu^{(t,\epsilon)}}(s_\mu^{(t,\epsilon)}) - \frac{\rho_n}{\alpha_n} r_\epsilon(t) \right) \right\rangle_{n,t,\epsilon}, \quad (17)$$

*where $\left| \mathcal{O}\left(\frac{1}{\rho_n \sqrt{n}}\right) \right| \le \frac{\sqrt{C}}{\rho_n \sqrt{n}}$, with $C$ a polynomial in $\left( S, \|\varphi\|_\infty, \|\partial_x \varphi\|_\infty, \|\partial_{xx}\varphi\|_\infty, M_\alpha, M_{\rho/\alpha} \right)$ with positive coefficients, uniformly in $(t, \epsilon)$.*

The next key result states that the overlap concentrates on its expectation. This behavior is called *replica symmetric* in statistical physics. Similar results have been obtained in the spin glass literature [1, 2]. In this work we use a formulation taylored to Bayesian inference problems as developed in the context of LDPC codes, random linear estimation [3] and Nishimori symmetric spin glasses [4, 5, 6].

**Proposition 2** (Overlap concentration). *Suppose that (H1), (H2), (H3) hold, that $\Delta = \mathbb{E}_{P_0}[X^2] = 1$ and that the family of functions $(r_\epsilon)_{\epsilon \in \mathcal{B}_n}, (q_\epsilon)_{\epsilon \in \mathcal{B}_n}$ are regular. Further assume that there exist real positive numbers $M_\alpha, M_{\rho/\alpha}, m_{\rho/\alpha}$ such that $\forall n \in \mathbb{N}^*$: $\alpha_n \le M_\alpha$ and $\frac{m_{\rho/\alpha}}{n} < \frac{\rho_n}{\alpha_n} \le M_{\rho/\alpha}$. Let $M_n := \left( s_n^2 \rho_n^2 \left( \frac{\rho_n n}{\alpha_n m_{\rho/\alpha}} \right)^{1/3} - s_n^2 \rho_n^2 \right)^{-1} > 0$. We have for all $t \in [0, 1]$:*

$$\int_{\mathcal{B}_n} \frac{d\epsilon}{s_n^2} \int_0^1 dt \, \mathbb{E}\left\langle \left( Q - \mathbb{E}\langle Q \rangle_{n,t,\epsilon} \right)^2 \right\rangle_{n,t,\epsilon} \le C M_n \,, \tag{18}$$

*where $C$ is a polynomial in $\left( S, \|\varphi\|_\infty, \|\partial_x \varphi\|_\infty, \|\partial_{xx}\varphi\|_\infty, M_\alpha, M_{\rho/\alpha}, m_{\rho/\alpha} \right)$ with positive coefficients.*

We prove Proposition 2 in Appendix D. We can now prove the fundamental sum rule.

**Proposition 3** (Fundamental sum rule). *Suppose that $\forall (t, \epsilon) \in [0, 1] \times \mathcal{B}_n : q_\epsilon(t) = \mathbb{E}\langle Q \rangle_{n,t,\epsilon}$. Under the assumptions of Proposition 2, we have:*

$$\frac{I(\mathbf{X}^*; \mathbf{Y} | \boldsymbol{\Phi})}{m_n} = \mathcal{O}\left(\sqrt{M_n}\right) + \mathcal{O}\left(\frac{s_n}{\sqrt{\rho_n}}\right)$$
$$+ \int_{\mathcal{B}_n} \frac{d\epsilon}{s_n^2} \left\{ \frac{1}{\alpha_n} I_{P_{0,n}}\left(\int_0^1 r_\epsilon(t)dt\right) + I_{P_{\text{out}}}\left(\int_0^1 q_\epsilon(t)dt, 1\right) - \frac{\rho_n}{2\alpha_n} \int_0^1 r_\epsilon(t)(1 - q_\epsilon(t))dt \right\} \,.$$

*The constant factors in $\mathcal{O}\left(\sqrt{M_n}\right)$ and $\mathcal{O}\left(s_n/\sqrt{\rho_n}\right)$ are $\sqrt{C_1}$ and $\sqrt{C_2}$ where $C_1, C_2$ are polynomials in $\left( S, \|\varphi\|_\infty, \|\partial_x \varphi\|_\infty, \|\partial_{xx}\varphi\|_\infty, M_\alpha, M_{\rho/\alpha}, m_{\rho/\alpha} \right)$ with positive coefficients.*

*Proof.* Let $\mathbb{E}_{\epsilon,t} := \int_{\mathcal{B}_n} \frac{d\epsilon}{s_n^2} \int_0^1 dt$. By Cauchy-Schwarz inequality:

$$\left| \int_{\mathcal{B}_n} \frac{d\epsilon}{s_n^2} \int_0^1 dt\, \mathbb{E} \left\langle (Q - q_\epsilon(t)) \left( \frac{1}{m_n} \sum_{\mu=1}^{m_n} u'_{Y_\mu^{(t,\epsilon)}}(S_\mu^{(t,\epsilon)}) u'_{Y_\mu^{(t,\epsilon)}}(s_\mu^{(t,\epsilon)}) - \frac{\rho_n}{\alpha_n} r_\epsilon(t) \right) \right\rangle_{n,t,\epsilon} \right|^2$$

$$\leq \int_{\mathcal{B}_n} \frac{d\epsilon}{s_n^2} \int_0^1 dt\, \mathbb{E} \left\langle \left( \frac{1}{m_n} \sum_{\mu=1}^{m_n} u'_{Y_\mu^{(t,\epsilon)}}(S_\mu^{(t,\epsilon)}) u'_{Y_\mu^{(t,\epsilon)}}(s_\mu^{(t,\epsilon)}) - \frac{\rho_n}{\alpha_n} r_\epsilon(t) \right)^2 \right\rangle_{n,t,\epsilon}$$

$$\cdot \int_{\mathcal{B}_n} \frac{d\epsilon}{s_n^2} \int_0^1 dt\, \mathbb{E} \left\langle (Q - q_\epsilon(t))^2 \right\rangle_{n,t,\epsilon} .$$

The first factor on the right-hand side of this inequality is bounded by a constant that depends polynomially on $\|\varphi\|_\infty, \|\partial_x \varphi\|_\infty{}^2$. Since $\forall (t,\epsilon) \in [0,1] \times \mathcal{B}_n : q_\epsilon(t) = \mathbb{E}\langle Q \rangle_{n,t,\epsilon}$, the second term is in $\mathcal{O}(M_n)$ (see Proposition 2). Therefore, by Proposition 1:

$$\mathbb{E}_{\epsilon,t}\, i'_{n,\epsilon}(t) = \mathcal{O}(\sqrt{M_n}) + \mathcal{O}\left( \frac{1}{\rho_n \sqrt{n}} \right) + \mathbb{E}_{\epsilon,t}\, \frac{\rho_n}{2\alpha_n} r_\epsilon(t)(1 - q_\epsilon(t)) . \tag{19}$$

Note that $1/\rho_n \sqrt{n} = \mathcal{O}(\sqrt{M_n})$. Integrating (16) over $\epsilon \in \mathcal{B}_n$ and making use of (19) give the result. $\square$

## 2.3 Matching bounds

To prove Theorem 1, we will lower and upper bound $I(\mathbf{X}^*; \mathbf{Y}|\Phi)/m_n$ by the same quantity, up to a small error. To do so we will plug two different choices of interpolation functions $R_1(\cdot, \epsilon), R_2(\cdot, \epsilon)$ in the sum-rule of Proposition 3. In both cases, the interpolation functions will be the solutions of a second-order ordinary differential equation (ODE). We now describe these ODEs.

Fix $t \in [0,1]$ and $R = (R_1, R_2) \in [0, +\infty) \times [0, t + 2s_n]$. Consider the observations:

$$\begin{cases} Y_\mu^{(t,R_2)} & \sim P_{\text{out}}(\cdot \,|\, S_\mu^{(t,R_2)}) , \quad 1 \leq \mu \leq m_n ; \\ \widetilde{Y}_i^{(t,R_1)} & = \sqrt{R_1}\, X_i^* + \widetilde{Z}_i , \quad 1 \leq i \leq n ; \end{cases} \tag{20}$$

where $S_\mu^{(t,R_2)} = S_\mu^{(t,R_2)}(\mathbf{X}^*, \mathbf{W}^*) := \sqrt{(1-t)/k_n} (\Phi \mathbf{X}^*)_\mu + \sqrt{R_2}\, V_\mu + \sqrt{t + 2s_n - R_2}\, W_\mu^*$. The joint posterior density of $(\mathbf{X}^*, \mathbf{W}^*)$ given $(\mathbf{Y}^{(t,R_2)}, \widetilde{\mathbf{Y}}^{(t,R_1)}, \Phi, \mathbf{V})$ is:

$$dP(\mathbf{x}, \mathbf{w}|\mathbf{Y}^{(t,R_2)}, \widetilde{\mathbf{Y}}^{(t,R_1)}, \Phi, \mathbf{V})$$
$$\propto \prod_{i=1}^n dP_{0,n}(x_i)\, e^{-\frac{1}{2} \left( \sqrt{R_1} x_i - \widetilde{Y}_i^{(t,R_1)} \right)^2} \prod_{\mu=1}^{m_n} \frac{dw_\mu}{\sqrt{2\pi}} e^{-\frac{w_\mu^2}{2}} P_{\text{out}}(Y_\mu^{(t,R_2)}|S_\mu^{(t,R_2)}(\mathbf{x}, w_\mu)) .$$

The angular brackets $\langle - \rangle_{n,t,R}$ denotes the expectation w.r.t. this posterior. Let $r \in [0, r_{\max}]$, $F_2^{(n)}(t,R) := \mathbb{E}\langle Q \rangle_{n,t,R}$ and $F_1^{(n)}(t,R) := -2\frac{\alpha_n}{\rho_n} \frac{\partial I_{P_{\text{out}}}}{\partial q}\big|_{q=\mathbb{E}\langle Q \rangle_{n,t,R}, \rho=1}$. We will consider the two following second-order ODEs with initial value $\epsilon \in [s_n, 2s_n]^2$:

$$y'(t) = \left( \frac{\alpha_n}{\rho_n} r, F_2^{(n)}(t, y(t)) \right) , \quad y(0) = \epsilon ; \tag{21}$$

$$y'(t) = \left( F_1^{(n)}(t, y(t)), F_2^{(n)}(t, y(t)) \right) , \quad y(0) = \epsilon . \tag{22}$$

The next proposition sums up useful properties on the solutions of these two ODEs, i.e., our two kinds of interpolation functions. The proof is given in Appendix E.

**Proposition 4.** *Suppose that (H1), (H2), (H3) hold and that $\Delta = \mathbb{E}_{X \sim P_0}[X^2] = 1$. For all $\epsilon \in \mathcal{B}_n$, there exists a unique global solution $R(\cdot, \epsilon) : [0,1] \to [0, +\infty)^2$ to (22). This solution is continuously differentiable and its derivative $R'(\cdot, \epsilon)$ satisfies $R'([0,1], \epsilon) \subseteq [0, \alpha_n r_{\max}/\rho_n] \times [0,1]$. Besides, for all $t \in [0,1]$, $R(t, \cdot)$ is a $\mathcal{C}^1$-diffeomorphism from $\mathcal{B}_n$ onto its image whose Jacobian determinant is greater than, or equal to, one. Finally, the same statement holds if we consider (21) instead.*

$(|Y_\mu^{(t,\epsilon)}| + \|\varphi\|_\infty) \Delta^{-1} \|\partial_x \varphi\|_\infty = (\sqrt{\Delta}|Z_\mu| + 2\|\varphi\|_\infty) \Delta^{-1} \|\partial_x \varphi\|_\infty$ (see the inequality (93) in Appendix A). The noise $Z_\mu$ is averaged over thanks to the expectation.

**Proposition 5** (Upper bound). *Suppose that (H1), (H2), (H3) hold, that $\Delta = \mathbb{E}_{P_0}[X^2] = 1$ and that $\forall n \in \mathbb{N}^*$: $\alpha_n \leq M_\alpha$, $\frac{m_{\rho/\alpha}}{n} < \frac{\rho_n}{\alpha_n} \leq M_{\rho/\alpha}$ for positive numbers $M_\alpha, M_{\rho/\alpha}, m_{\rho/\alpha}$. Then:*

$$\forall n \in \mathbb{N}^* : \quad \frac{I(\mathbf{X}^*; \mathbf{Y}|\mathbf{\Phi})}{m_n} \leq \inf_{r \in [0, r_{\max}]} \sup_{q \in [0,1]} i_{\mathrm{RS}}(q, r; \alpha_n, \rho_n) + \mathcal{O}(\sqrt{M_n}) + \mathcal{O}\left(\frac{s_n}{\sqrt{\rho_n}}\right). \quad (23)$$

*Proof.* Fix $r \in r_{\max}$. For all $\epsilon \in \mathcal{B}_n$, $(R_1(\cdot, \epsilon), R_2(\cdot, \epsilon))$ is the unique solution to the ODE (21) (see Proposition 4). Let $q_\epsilon(t) := R_2'(t, \epsilon) = \mathbb{E}\langle Q\rangle_{n,t,\epsilon}$, $r_\epsilon(t) := R_1'(t, \epsilon) = \frac{\alpha_n r}{\rho_n}$. By Proposition 4, the families of functions $(q_\epsilon)_{\epsilon \in \mathcal{B}_n}$, $(r_\epsilon)_{\epsilon \in \mathcal{B}_n}$ are *regular*. We can now apply Proposition 3 to get:

$$\frac{I(\mathbf{X}^*; \mathbf{Y}|\mathbf{\Phi})}{m_n} = \int_{\mathcal{B}_n} \frac{d\epsilon}{s_n^2} i_{\mathrm{RS}}\left(\int_0^1 q_\epsilon(t)\, dt, r; \alpha_n, \rho_n\right) + \mathcal{O}(\sqrt{M_n}) + \mathcal{O}\left(\frac{s_n}{\sqrt{\rho_n}}\right)$$

$$\leq \sup_{q \in [0,1]} i_{\mathrm{RS}}(q, r; \alpha_n, \rho_n) + \mathcal{O}(\sqrt{M_n}) + \mathcal{O}\left(\frac{s_n}{\sqrt{\rho_n}}\right). \quad (24)$$

The inequality (24) holds for all $r \in [0, r_{\max}]$ and the constant factors in the quantities $\mathcal{O}(\sqrt{M_n})$, $\mathcal{O}(s_n/\sqrt{\rho_n})$ are uniform in $r$. Hence the inequality (23) with the infimum over $r$. $\qquad\square$

**Proposition 6** (Lower bound). *Under the same hypotheses than Proposition 5, we have:*

$$\forall n \in \mathbb{N}^* : \quad \frac{I(\mathbf{X}^*; \mathbf{Y}|\mathbf{\Phi})}{m_n} \geq \inf_{r \in [0, r_{\max}]} \sup_{q \in [0,1]} i_{\mathrm{RS}}(q, r; \alpha_n, \rho_n) + \mathcal{O}(\sqrt{M_n}) + \mathcal{O}\left(\frac{s_n}{\sqrt{\rho_n}}\right). \quad (25)$$

*Proof.* For all $\epsilon \in \mathcal{B}_n$, $(R_1(\cdot, \epsilon), R_2(\cdot, \epsilon))$ is the unique solution to the ODE (22) (see Proposition 4). We define $q_\epsilon(t) := R_2'(t, \epsilon) = \mathbb{E}\langle Q\rangle_{n,t,\epsilon}$, $r_\epsilon(t) := R_1'(t, \epsilon) = -\frac{2\alpha_n}{\rho_n} \frac{\partial I_{P_{\mathrm{out}}}}{\partial q}\big|_{q = q_\epsilon(t), \rho = 1}$. By Proposition 4, the families of functions $(q_\epsilon)_{\epsilon \in \mathcal{B}_n}$, $(r_\epsilon)_{\epsilon \in \mathcal{B}_n}$ are *regular*. Note that $\forall \epsilon \in \mathcal{B}_n$:

$$\frac{1}{\alpha_n} I_{P_{0,n}}\left(\int_0^1 r_\epsilon(t)\, dt\right) + I_{P_{\mathrm{out}}}\left(\int_0^1 q_\epsilon(t)\, dt, 1\right) - \frac{\rho_n}{2\alpha_n} \int_0^1 r_\epsilon(t)(1 - q_\epsilon(t))\, dt$$

$$\geq \int_0^1 \left\{ \frac{1}{\alpha_n} I_{P_{0,n}}(r_\epsilon(t)) + I_{P_{\mathrm{out}}}(q_\epsilon(t), 1) - \frac{\rho_n}{2\alpha_n} r_\epsilon(t)(1 - q_\epsilon(t)) \right\} dt$$

$$= \int_0^1 \left\{ \sup_{q \in [0,1]} \frac{1}{\alpha_n} I_{P_{0,n}}(r_\epsilon(t)) + I_{P_{\mathrm{out}}}(q, 1) - \frac{\rho_n}{2\alpha_n} r_\epsilon(t)(1 - q) \right\} dt$$

$$= \int_0^1 \sup_{q \in [0,1]} i_{\mathrm{RS}}\left(q, \frac{\rho_n}{\alpha_n} r_\epsilon(t); \alpha_n, \rho_n\right) dt \quad (26)$$

$$\geq \inf_{r \in [0, r_{\max}]} \sup_{q \in [0,1]} i_{\mathrm{RS}}(q, r; \alpha_n, \rho_n). \quad (27)$$

The first inequality is an application of Jensen's inequality to the concave functions $I_{P_{0,n}}, I_{P_{\mathrm{out}}}(\cdot, 1)$ (see Lemmas 6 and 7). The subsequent equality is because the global maximum of the concave function $h : q \in [0,1] \mapsto I_{P_{\mathrm{out}}}(q, 1) - \frac{\rho_n}{2\alpha_n} r_\epsilon(t)(1 - q)$ is reached at $q_\epsilon(t)$ since $h'(q_\epsilon(t)) = 0$. The equality (26) follows from the definition (7) of $i_{\mathrm{RS}}$. Finally, the inequality (27) is because $r_\epsilon(t) \in \left[0, \frac{\alpha_n}{\rho_n} r_{\max}\right]$ and we simply lowerbound the integrand in (26) by a quantity independent of $t \in [0,1]$. We now apply Proposition 3 and make use of (27) to obtain the inequality (25). $\qquad\square$

To prove Theorem 1, it remains to combine Propositions 5 and 6 with the identity

$$\inf_{r \in [0, r_{\max}]} \sup_{q \in [0,1]} i_{\mathrm{RS}}(q, r; \alpha_n, \rho_n) = \inf_{r \geq 0} \sup_{q \in [0,1]} i_{\mathrm{RS}}(q, r; \alpha_n, \rho_n) = \inf_{q \in [0,1]} \sup_{r \geq 0} i_{\mathrm{RS}}(q, r; \alpha_n, \rho_n), \quad (28)$$

and the choice $\rho_n = \Theta(n^{-\lambda})$, $\alpha_n = \gamma \rho_n |\ln \rho_n|$ and $s_n = \Theta(n^{-\beta})$ with $\lambda \in [0, 1/9)$, $\gamma > 0$ and $\beta \in (\lambda/2, 1/6 - \lambda)$. Optimizing over $\beta$ to maximize the convergence rate of

$$\mathcal{O}(\sqrt{M_n}) + \mathcal{O}\left(\frac{s_n}{\sqrt{\rho_n}}\right) = \mathcal{O}\left(\max\left\{\frac{1}{n^{\beta - \lambda/2}}, \frac{|\ln n|^{1/6}}{n^{1/6 - \lambda - \beta}}\right\}\right)$$

yields Theorem 1. The identity (28) has been proved in [7, Proposition 7 and Corollary 7 in SI].

# 3 Proof of Theorem 2 for a Bernoulli prior

In this section, we assume that $P_{0,n} := (1 - \rho_n)\delta_0 + \rho_n\delta_1$ and we prove Theorem 2 for this specific case. The proof contains all the main ideas needed to establish Theorem 2 while being technically simpler. The interested reader can find the proof of Theorem 2 for a general discrete prior with finite support in Appendix F.

For $\rho_n, \alpha_n > 0$ we denote the variational problem appearing in Theorem 1 by

$$I(\rho_n, \alpha_n) := \inf_{q \in [0,1]} \sup_{r \geq 0} i_{\mathrm{RS}}(q, r; \alpha_n, \rho_n) \,, \tag{29}$$

where the potential $i_{\mathrm{RS}}$ is defined in (7). Let $X^* \sim P_{0,n}$, $Z \sim \mathcal{N}(0,1)$ be independent random variables. We define for all $r \geq 0$:

$$\psi_{P_{0,n}}(r) := \mathbb{E}\left[ \ln \left( 1 - \rho_n + \rho_n e^{-\frac{r}{2} + rX^* + \sqrt{r}Z} \right) \right] . \tag{30}$$

Note that $I_{P_{0,n}}(r) := I(X^*; \sqrt{r}\,X^* + Z) = \frac{r\rho_n}{2} - \psi_{P_{0,n}}(r)$ so

$$I(\rho_n, \alpha_n) = \inf_{q \in [0,1]} I_{P_{\mathrm{out}}}(q, 1) + \sup_{r \geq 0} \left\{ \frac{rq}{2} - \frac{1}{\alpha_n}\psi_{P_{0,n}}\left( \frac{\alpha_n}{\rho_n}r \right) \right\} . \tag{31}$$

The latter expression for $I(\rho_n, \alpha_n)$ is easier to work with. We point out that $\psi_{P_{0,n}}$ is twice differentiable, nondecreasing, strictly convex and $\frac{\rho_n}{2}$-Lipschitz on $[0, +\infty)$ (see Lemma 6) while $I_{P_{\mathrm{out}}}(\cdot, 1)$ is nonincreasing and concave on $[0, 1]$ (see [7, Appendix B.2, Proposition 18]).

Our goal is now to compute the limit of $I(\rho_n, \alpha_n)$ when $\alpha_n := \gamma\rho_n|\ln \rho_n|$ for a fix $\gamma > 0$ and $\rho_n \to 0$. Once we know this limit, we directly obtain Theorem 2 thanks to Theorem 1. We first show that – for $q$ in a growing interval – the point at which the supremum over $r$ is achieved is located in an interval shrinking on $r^* := 2/\gamma$.

**Lemma 2.** *Let $P_{0,n} := (1 - \rho_n)\delta_0 + \rho_n\delta_1$ and $\alpha_n := \gamma\rho_n|\ln \rho_n|$ for a fix $\gamma > 0$. Define $g_{\rho_n} : r \in (0, +\infty) \mapsto \frac{2}{\rho_n}\psi'_{P_{0,n}}\left( \frac{\alpha_n}{\rho_n}r \right)$ and $\forall \rho_n \in (0, e^{-1})$ :*

$$a_{\rho_n} := g_{\rho_n}\left( \frac{2(1 - |\ln \rho_n|^{-\frac{1}{4}})}{\gamma} \right) \quad , \quad b_{\rho_n} := g_{\rho_n}\left( \frac{2(1 + |\ln \rho_n|^{-\frac{1}{4}})}{\gamma} \right) . \tag{32}$$

*We have $[a_{\rho_n}, b_{\rho_n}] \subset (\rho_n, 1)$ and $\lim_{\rho_n \to 0} a_{\rho_n} = 0$, $\lim_{\rho_n \to 0} b_{\rho_n} = 1$. Besides, for every $q \in (\rho_n, 1)$ there exists a unique $r_n^*(q) \in (0, +\infty)$ such that*

$$\frac{r_n^*(q)q}{2} - \frac{1}{\alpha_n}\psi_{P_{0,n}}\left( \frac{\alpha_n}{\rho_n}r_n^*(q) \right) = \sup_{r \geq 0} \frac{rq}{2} - \frac{1}{\alpha_n}\psi_{P_{0,n}}\left( \frac{\alpha_n}{\rho_n}r \right) , \tag{33}$$

*and*

$$\forall q \in [a_{\rho_n}, b_{\rho_n}] : \frac{2(1 - |\ln \rho_n|^{-\frac{1}{4}})}{\gamma} \leq r_n^*(q) \leq \frac{2(1 + |\ln \rho_n|^{-\frac{1}{4}})}{\gamma} , \tag{34}$$

$$\forall q \in [b_{\rho_n}, 1) : r_n^*(q) \geq \frac{2(1 + |\ln \rho_n|^{-\frac{1}{4}})}{\gamma} . \tag{35}$$

*Proof.* For every $q \in (0, 1)$ we define $f_{\rho_n, q} : r \in [0, +\infty) \mapsto \frac{rq}{2} - \frac{1}{\alpha_n}\psi_{P_{0,n}}\left( \frac{\alpha_n}{\rho_n}r \right)$ whose supremum over $r$ we want to compute. The derivative of $f_{\rho_n, q}$ with respect to $r$ reads

$$f'_{\rho_n, q}(r) = \frac{q}{2} - \frac{1}{\rho_n}\psi'_{P_{0,n}}\left( \frac{\alpha_n}{\rho_n}r \right) . \tag{36}$$

The derivative $\psi'_{P_{0,n}}$ is continuously increasing and thus one-to-one from $(0, +\infty)$ onto $(\rho_n^2/2, \rho_n/2)$. Therefore, if $q \in (0, \rho_n]$ then $f'_{\rho_n, q} \leq 0$ and the supremum of $f_{\rho_n, q}$ is achieved at $r = 0$. On the contrary, if $q \in (\rho_n, 1)$ then there exists a unique solution $r_n^*(q) \in (0, +\infty)$ to the critical point equation $f'_{\rho_n, q}(r) = 0$. As $f_{\rho_n, q}$ is concave (given that $\psi_{P_{0,n}}$ is convex), this solution $r_n^*(q)$ is the global maximum of $f_{\rho_n, q}$. We now transform the critical point equation:

$$f_{\rho_n, q}(r) = 0 \Leftrightarrow \frac{2}{\rho_n}\psi'_{P_{0,n}}\left( \frac{\alpha_n}{\rho_n}r \right) = q \Leftrightarrow g_{\rho_n}(r) = q \,, \tag{37}$$

where $g_{\rho_n} : r \mapsto \frac{2}{\rho_n}\psi'_{P_{0,n}}\left(\frac{\alpha_n}{\rho_n}r\right)$ is increasing and one-to-one from $(0,+\infty)$ to $(\rho_n, 1)$. For all $\rho_n \in (0, e^{-1}) : |\ln \rho_n|^{-\frac{1}{4}} \in (0,1)$. By Lemma 3 (directly following the proof) applied with $\epsilon = |\ln \rho_n|^{-\frac{1}{4}}$, we have:

$$\rho_n < a_{\rho_n} := g_{\rho_n}\left(\frac{2(1-|\ln\rho_n|^{-\frac{1}{4}})}{\gamma}\right) \leq \frac{\exp\left(-\frac{|\ln\rho_n|^{\frac{1}{2}}}{16(1-|\ln\rho_n|^{-1/4})}\right)}{2} + \frac{\exp\left(-\frac{|\ln\rho_n|^{\frac{3}{4}}}{2}\right)}{1-\rho_n} ; \quad (38)$$

$$1 > b_{\rho_n} := g_{\rho_n}\left(\frac{2(1+|\ln\rho_n|^{-\frac{1}{4}})}{\gamma}\right) \geq \frac{1 - 0.5\exp\left(-\frac{|\ln\rho_n|^{1/2}}{16}\right)}{1 + \exp\left(-\frac{|\ln\rho_n|^{3/4}}{2}\right)} . \quad (39)$$

It directly follows from (38) that $\lim_{\rho_n \to 0} a_{\rho_n} = 0$ and from (39) that $\lim_{\rho_n \to 0} b_{\rho_n} = 1$. As $g_{\rho_n}$ is increasing, if $q = g_{\rho_n}(r_n^*(q)) \in [a_{\rho_n}, b_{\rho_n}]$ then

$$\frac{2(1-|\ln\rho_n|^{-\frac{1}{4}})}{\gamma} \leq r_n^*(q) \leq \frac{2(1+|\ln\rho_n|^{-\frac{1}{4}})}{\gamma}$$

while if $q = g_{\rho_n}(r_n^*(q)) \in [b_{\rho_n}, 1)$ then $r_n^*(q) \geq \frac{2(1+|\ln\rho_n|^{-\frac{1}{4}})}{\gamma}$. $\qquad \square$

**Lemma 3.** *Let* $\alpha_n := \gamma\rho_n|\ln\rho_n|$ *for a fix* $\gamma > 0$ *and define* $g_{\rho_n} : r \mapsto \frac{2}{\rho_n}\psi'_{P_{0,n}}\left(\frac{\alpha_n}{\rho_n}r\right)$. *For all* $(\rho_n, \epsilon) \in (0,1)^2$ *we have:*

$$g_{\rho_n}\left(\frac{2(1-\epsilon)}{\gamma}\right) \leq \frac{\exp\left(-\frac{\epsilon^2}{16}\frac{|\ln\rho_n|}{1-\epsilon}\right)}{2} + \frac{\exp\left(-\frac{\epsilon}{2}|\ln\rho_n|\right)}{1-\rho_n} ; \quad (40)$$

$$g_{\rho_n}\left(\frac{2(1+\epsilon)}{\gamma}\right) \geq \frac{1 - 0.5\exp\left(-\frac{\epsilon^2}{16}|\ln\rho_n|\right)}{1 + \exp\left(-\frac{\epsilon}{2}|\ln\rho_n|\right)} . \quad (41)$$

*Proof.* The derivative of $\psi_{P_{0,n}}$ reads $\psi'_{P_{0,n}}(r) = \frac{\rho_n}{2}\mathbb{E}\left[\left(1 + \frac{1-\rho_n}{\rho_n}e^{-\frac{r}{2}-\sqrt{r}Z}\right)^{-1}\right]$. Therefore:

$$g_{\rho_n}(r) = \mathbb{E}\left[\frac{1}{1 + (1-\rho_n)\exp\left\{|\ln\rho_n|\left(1 - \gamma r/2 - \sqrt{\frac{\gamma r}{|\ln\rho_n|}}Z\right)\right\}}\right] \in (0,1) . \quad (42)$$

Hence for all $\epsilon \in (0,1)$ we have:

$$g_{\rho_n}\left(\frac{2(1\pm\epsilon)}{\gamma}\right) = \mathbb{E}\left[\frac{1}{1 + (1-\rho_n)\exp\left\{|\ln\rho_n|\left(\mp\epsilon - \sqrt{\frac{2(1\pm\epsilon)}{|\ln\rho_n|}}Z\right)\right\}}\right] . \quad (43)$$

By the dominated convergence theorem $\lim_{\rho_n \to 0} g_{\rho_n}\left(2(1+\epsilon)/\gamma\right) = 1$ and $\lim_{\rho_n \to 0} g_{\rho_n}\left(2(1-\epsilon)/\gamma\right) = 0$. We first lower bound $g_{\rho_n}\left(2(1+\epsilon)/\gamma\right)$. Note that $\forall z \geq -\frac{\epsilon}{2}\sqrt{\frac{|\ln\rho_n|}{2(1+\epsilon)}} : -\epsilon - \sqrt{\frac{2(1+\epsilon)}{|\ln\rho_n|}}z \leq -\frac{\epsilon}{2}$. Hence:

$$g_{\rho_n}\left(\frac{2(1+\epsilon)}{\gamma}\right) = \int_{-\infty}^{+\infty} \frac{dz}{\sqrt{2\pi}} \frac{e^{-\frac{z^2}{2}}}{1 + (1-\rho_n)\exp\left\{|\ln\rho_n|\left(-\epsilon - \sqrt{\frac{2(1+\epsilon)}{|\ln\rho_n|}}z\right)\right\}}$$

$$\geq \int_{-\frac{\epsilon}{2}\sqrt{\frac{|\ln\rho_n|}{2(1+\epsilon)}}}^{+\infty} \frac{dz}{\sqrt{2\pi}} \frac{e^{-\frac{z^2}{2}}}{1 + (1-\rho_n)\exp\left(-\frac{\epsilon}{2}|\ln\rho_n|\right)}$$

$$= \frac{1 - F\left(-\frac{\epsilon}{2}\sqrt{\frac{|\ln\rho_n|}{2(1+\epsilon)}}\right)}{1 + (1-\rho_n)\exp\left(-\frac{\epsilon}{2}|\ln\rho_n|\right)} \geq \frac{1 - F\left(-\frac{\epsilon}{2}\sqrt{\frac{|\ln\rho_n|}{2}}\right)}{1 + \exp\left(-\frac{\epsilon}{2}|\ln\rho_n|\right)} , \quad (44)$$

where $F(x) := \int_{-\infty}^{x} \frac{dz}{\sqrt{2\pi}}e^{-\frac{z^2}{2}}$ is the cumulative distribution function of the standard normal distribution. Making use of the upper bound $F(-x) \leq \frac{e^{-x^2/2}}{2}$ for $x > 0$ yields

$$g_{\rho_n}\left(\frac{2(1+\epsilon)}{\gamma}\right) \geq \frac{1 - 0.5\exp\left(-\frac{\epsilon^2}{16}|\ln\rho_n|\right)}{1 + \exp\left(-\frac{\epsilon}{2}|\ln\rho_n|\right)} . \quad (45)$$

Next we prove the upper bound on $g_{\rho_n}\big(2(1-\epsilon)/\gamma\big)$. We denote the indicator function of an event $\mathcal{E}$ by $\mathbf{1}_{\mathcal{E}}$. We have:

$$g_{\rho_n}\left(\frac{2(1-\epsilon)}{\gamma}\right) = \mathbb{E}\left[\frac{1}{1+(1-\rho_n)\exp\left\{|\ln\rho_n|\left(\epsilon-\sqrt{\frac{2(1-\epsilon)}{|\ln\rho_n|}}Z\right)\right\}}\right] \tag{46}$$

$$\leq \mathbb{E}\left[\mathbf{1}_{\left\{Z\geq\frac{\epsilon}{2}\sqrt{\frac{|\ln\rho_n|}{2(1-\epsilon)}}\right\}} + \frac{\mathbf{1}_{\left\{Z<\frac{\epsilon}{2}\sqrt{\frac{|\ln\rho_n|}{2(1-\epsilon)}}\right\}}}{1+(1-\rho_n)\exp\left(\frac{\epsilon}{2}|\ln\rho_n|\right)}\right]$$

$$= F\left(-\frac{\epsilon}{2}\sqrt{\frac{|\ln\rho_n|}{2(1-\epsilon)}}\right) + \frac{1-F\left(-\frac{\epsilon}{2}\sqrt{\frac{|\ln\rho_n|}{2(1-\epsilon)}}\right)}{1+(1-\rho_n)\exp\left(\frac{\epsilon}{2}|\ln\rho_n|\right)}$$

$$\leq F\left(-\frac{\epsilon}{2}\sqrt{\frac{|\ln\rho_n|}{2(1-\epsilon)}}\right) + \frac{\exp\left(-\frac{\epsilon}{2}|\ln\rho_n|\right)}{1-\rho_n}$$

$$\leq \frac{\exp\left(-\frac{\epsilon^2}{16}\frac{|\ln\rho_n|}{1-\epsilon}\right)}{2} + \frac{\exp\left(-\frac{\epsilon}{2}|\ln\rho_n|\right)}{1-\rho_n} . \tag{47}$$

The last inequality follows from the same upper bound on $F(-x)$ that we used to obtain (45). $\qquad\square$

Lemma 2 essentially states that the global maximum of $r \mapsto \frac{rq}{2} - \frac{1}{\alpha_n}\psi_{P_{0,n}}\left(\frac{\alpha_n}{\rho_n}r\right)$ is located in a tight interval around $2/\gamma$ when $q \in [a_{\rho_n}, b_{\rho_n}]$. The next step is to use this knowledge to tightly bound the maximum value $\sup_{r\geq 0}\frac{rq}{2} - \frac{1}{\alpha_n}\psi_{P_{0,n}}\left(\frac{\alpha_n}{\rho_n}r\right)$ for all $q \in [a_{\rho_n}, b_{\rho_n}]$. The following lemma gives a bound on $\frac{1}{\alpha_n}\psi_{P_{0,n}}\left(\frac{\alpha_n}{\rho_n}r\right)$ for $0 \leq r \leq 2(1+\epsilon)/\gamma$.

**Lemma 4.** *Let* $P_{0,n} := (1-\rho_n)\delta_0 + \rho_n\delta_1$ *and* $\alpha_n := \gamma\rho_n|\ln\rho_n|$ *for a fix* $\gamma > 0$. *For every* $\epsilon \in (0,1)$ *and* $r \in [0, 2(1+\epsilon)/\gamma]$ *we have*

$$0 \leq \frac{1}{\alpha_n}\psi_{P_{0,n}}\left(\frac{\alpha_n}{\rho_n}r\right) \leq \frac{\epsilon}{\gamma} + \frac{\ln 2}{\gamma|\ln\rho_n|} + \frac{1}{\gamma}\sqrt{\frac{2}{\pi|\ln\rho_n|}} . \tag{48}$$

*Proof.* The function $\psi_{P_{0,n}}$ is nondecreasing on $[0,+\infty)$ so $\forall r \in [0, 2(1+\epsilon)/\gamma]$ :

$$0 \leq \frac{1}{\alpha_n}\psi_{P_{0,n}}\left(\frac{\alpha_n}{\rho_n}r\right) \leq \frac{1}{\alpha_n}\psi_{P_{0,n}}\left(\frac{\alpha_n}{\rho_n}\frac{2(1+\epsilon)}{\gamma}\right) = \frac{\psi_{P_{0,n}}\big(2(1+\epsilon)|\ln\rho_n|\big)}{\gamma\rho_n|\ln\rho_n|} . \tag{49}$$

The upper bound on the right-hand side of (49) reads (remember the definition 30 of $\psi_{P_{0,n}}$):

$$\frac{\psi_{P_{0,n}}\big(2(1+\epsilon)|\ln\rho_n|\big)}{\gamma\rho_n|\ln\rho_n|} = \frac{1-\rho_n}{\gamma\rho_n|\ln\rho_n|}\mathbb{E}\left[\ln\left(1-\rho_n+\rho_n e^{-(1+\epsilon)|\ln\rho_n|+\sqrt{2(1+\epsilon)|\ln\rho_n|}Z}\right)\right]$$

$$+ \frac{1}{\gamma|\ln\rho_n|}\mathbb{E}\left[\ln\left(1-\rho_n+\rho_n e^{(1+\epsilon)|\ln\rho_n|+\sqrt{2(1+\epsilon)|\ln\rho_n|}Z}\right)\right]$$

$$= \frac{1-\rho_n}{\gamma\rho_n|\ln\rho_n|}\mathbb{E}\left[\ln\left(1-\rho_n+\rho_n e^{-(1+\epsilon)|\ln\rho_n|+\sqrt{2(1+\epsilon)|\ln\rho_n|}Z}\right)\right]$$

$$+ \frac{1}{\gamma|\ln\rho_n|}\mathbb{E}\left[\ln\left(1-\rho_n+e^{\epsilon|\ln\rho_n|+\sqrt{2(1+\epsilon)|\ln\rho_n|}Z}\right)\right] . \tag{50}$$

To control the first term on the right-hand side of (50) we use that $\ln(1+x) \leq x$:

$$\frac{1-\rho_n}{\gamma\rho_n|\ln\rho_n|}\mathbb{E}\left[\ln\left(1-\rho_n+\rho_n e^{-(1+\epsilon)|\ln\rho_n|+\sqrt{2(1+\epsilon)|\ln\rho_n|}Z}\right)\right]$$

$$\leq \frac{\mathbb{E}\left[e^{-(1+\epsilon)|\ln\rho_n|+\sqrt{2(1+\epsilon)|\ln\rho_n|}Z}-1\right]}{\gamma|\ln\rho_n|}$$

$$= \frac{e^{-(1+\epsilon)|\ln\rho_n|}\mathbb{E}\left[e^{\sqrt{2(1+\epsilon)|\ln\rho_n|}Z}\right]-1}{\gamma|\ln\rho_n|} = 0 . \tag{51}$$

To control the second term on the right-hand side of (50), we use that:

$$\forall z \leq 0 : \ln\left(1 - \rho_n + e^{\epsilon|\ln\rho_n| + \sqrt{2(1+\epsilon)|\ln\rho_n|}z}\right) \leq \ln(1 + e^{\epsilon|\ln\rho_n|}) \leq \ln(2e^{\epsilon|\ln\rho_n|}) \; ;$$

$$\forall z \geq 0 : \ln\left(1 - \rho_n + e^{\epsilon|\ln\rho_n| + \sqrt{2(1+\epsilon)|\ln\rho_n|}z}\right) \leq \ln(2e^{\epsilon|\ln\rho_n| + \sqrt{2(1+\epsilon)|\ln\rho_n|}z}) \; .$$

It directly follows that:

$$\frac{1}{\gamma|\ln\rho_n|}\mathbb{E}\left[\ln\left(1 - \rho_n + e^{\epsilon|\ln\rho_n| + \sqrt{2(1+\epsilon)|\ln\rho_n|}Z}\right)\right] \leq \frac{\epsilon}{\gamma} + \frac{\ln 2}{\gamma|\ln\rho_n|} + \frac{1}{\gamma}\sqrt{\frac{1+\epsilon}{\pi|\ln\rho_n|}} \; .$$

The latter combined with (50) and (51) ends the proof. $\qquad\square$

We can now compute the limit of $I(\rho_n, \alpha_n)$ when $\rho_n \to 0$ and $\alpha_n := \gamma\rho_n|\ln\rho_n|$.

**Proposition 7.** *Let* $P_{0,n} := (1 - \rho_n)\delta_0 + \rho_n\delta_1$ *and* $\alpha_n := \gamma\rho_n|\ln\rho_n|$ *for a fix* $\gamma > 0$. *Then the quantity* $I(\rho_n, \alpha_n) := \inf\limits_{q\in[0,1]}\sup\limits_{r\geq 0} i_{\mathrm{RS}}(q, r; \alpha_n, \rho_n)$ *converges when* $\rho_n \to 0^+$ *and*

$$\lim_{\rho_n \to 0^+} I(\rho_n, \alpha_n) = \min\left\{I_{P_{\mathrm{out}}}(0, 1), \frac{1}{\gamma}\right\} \; .$$

*Proof.* Let $a_{\rho_n}$, $b_{\rho_n}$ the quantities defined in Lemma 2. By Lemmas 2 and 4 (applied with $\epsilon = |\ln\rho_n|^{-\frac{1}{4}}$ for $\rho_n$ small enough), we have $\forall q \in [a_{\rho_n}, b_{\rho_n}]$:

$$\frac{(1 - |\ln\rho_n|^{-\frac{1}{4}})q}{\gamma} - \frac{1}{\gamma}\left(\frac{1}{|\ln\rho_n|^{\frac{1}{4}}} + \frac{\ln 2}{|\ln\rho_n|} + \sqrt{\frac{2}{\pi|\ln\rho_n|}}\right)$$

$$\leq \frac{r_n^*(q)q}{2} - \frac{1}{\alpha_n}\psi_{P_{0,n}}\left(\frac{\alpha_n}{\rho_n}r_n^*(q)\right) \leq \frac{(1 + |\ln\rho_n|^{-\frac{1}{4}})q}{\gamma} \; . \quad (52)$$

Therefore, $\forall q \in [a_{\rho_n}, b_{\rho_n}]$:

$$I_{P_{\mathrm{out}}}(q, 1) + \frac{q}{\gamma} - \frac{1}{\gamma}\left(\frac{2}{|\ln\rho_n|^{\frac{1}{4}}} + \frac{\ln 2}{|\ln\rho_n|} + \sqrt{\frac{2}{\pi|\ln\rho_n|}}\right)$$

$$\leq \sup_{r\geq 0} i_{\mathrm{RS}}(q, r; \alpha_n, \rho_n) \leq I_{P_{\mathrm{out}}}(q, 1) + \frac{q}{\gamma} + \frac{1}{\gamma|\ln\rho_n|^{\frac{1}{4}}} \; .$$

It directly follows that:

$$-\frac{1}{\gamma}\left(\frac{2}{|\ln\rho_n|^{\frac{1}{4}}} + \frac{\ln 2}{|\ln\rho_n|} + \sqrt{\frac{2}{\pi|\ln\rho_n|}}\right) + \left\{\inf_{q\in[a_{\rho_n}, b_{\rho_n}]} I_{P_{\mathrm{out}}}(q, 1) + \frac{q}{\gamma}\right\}$$

$$\leq \inf_{q\in[a_{\rho_n}, b_{\rho_n}]}\sup_{r\geq 0} i_{\mathrm{RS}}(q, r; \alpha_n, \rho_n) \leq \frac{1}{\gamma|\ln\rho_n|^{\frac{1}{4}}} + \left\{\inf_{q\in[a_{\rho_n}, b_{\rho_n}]} I_{P_{\mathrm{out}}}(q, 1) + \frac{q}{\gamma}\right\} \; . \quad (53)$$

Note that $q \mapsto I_{P_{\mathrm{out}}}(q, 1) + \frac{q}{\gamma}$ is concave on $[0, 1]$ so

$$\inf_{q\in[a_{\rho_n}, b_{\rho_n}]} I_{P_{\mathrm{out}}}(q, 1) + \frac{q}{\gamma} = \min\left\{I_{P_{\mathrm{out}}}(a_{\rho_n}, 1) + \frac{a_{\rho_n}}{\gamma}, I_{P_{\mathrm{out}}}(b_{\rho_n}, 1) + \frac{b_{\rho_n}}{\gamma}\right\}$$

$$\xrightarrow[\rho_n\to 0]{} \min\left\{I_{P_{\mathrm{out}}}(0, 1), \frac{1}{\gamma}\right\} \; . \quad (54)$$

Combining the bounds (53) on $\inf_{q\in[a_{\rho_n}, b_{\rho_n}]}\sup_{r\geq 0} i_{\mathrm{RS}}(q, r; \alpha_n, \rho_n)$ with the limit (54) yields:

$$\lim_{\rho_n\to 0}\inf_{q\in[a_{\rho_n}, b_{\rho_n}]}\sup_{r\geq 0} i_{\mathrm{RS}}(q, r; \alpha_n, \rho_n) = \min\left\{I_{P_{\mathrm{out}}}(0, 1), \frac{1}{\gamma}\right\} \; . \quad (55)$$

**Upper bound on the limit superior of** $I(\rho_n, \alpha_n)$    The upper bound on the limit superior of $I(\rho_n, \alpha_n) := \inf_{q \in [0,1]} \sup_{r \geq 0} i_{\text{RS}}(q, r; \alpha_n, \rho_n)$ directly follows from the limit (55) and the upper bound $I(\rho_n, \alpha_n) \leq \inf_{q \in [a_{\rho_n}, b_{\rho_n}]} \sup_{r \geq 0} i_{\text{RS}}(q, r; \alpha_n, \rho_n)$:

$$\limsup_{\rho_n \to 0^+} I(\rho_n, \alpha_n) \leq \min\left\{ I_{P_{\text{out}}}(0, 1), \frac{1}{\gamma} \right\}. \tag{56}$$

**Matching lower bound on the limit inferior of** $I(\rho_n, \alpha_n)$    We first rewrite $I(\rho_n, \alpha_n)$ by splitting the segment $[0, 1] = [0, a_{\rho_n}] \cup [a_{\rho_n}, b_{\rho_n}] \cup [b_{\rho_n}, 1]$:

$$I(\rho_n, \alpha_n) = \min\left\{ \inf_{q \in [0, a_{\rho_n}]} \sup_{r \geq 0} i_{\text{RS}}(q, r; \alpha_n, \rho_n) \,; \inf_{q \in [a_{\rho_n}, b_{\rho_n}]} \sup_{r \geq 0} i_{\text{RS}}(q, r; \alpha_n, \rho_n) \,; \right.$$
$$\left. \inf_{q \in [b_{\rho_n}, 1]} \sup_{r \geq 0} i_{\text{RS}}(q, r; \alpha_n, \rho_n) \right\}. \tag{57}$$

For all $q \in [0, a_{\rho_n}]$ we have:

$$\sup_{r \geq 0} i_{\text{RS}}(q, r; \alpha_n, \rho_n) = I_{P_{\text{out}}}(q, 1) + \sup_{r \geq 0}\left\{ \frac{rq}{2} - \frac{1}{\alpha_n} \psi_{P_{0,n}}\left( \frac{\alpha_n}{\rho_n} r \right) \right\}$$
$$\geq I_{P_{\text{out}}}(q, 1) + \lim_{r \to 0^+}\left\{ \frac{rq}{2} - \frac{1}{\alpha_n} \psi_{P_{0,n}}\left( \frac{\alpha_n}{\rho_n} r \right) \right\} = I_{P_{\text{out}}}(q, 1).$$

As $q \mapsto I_{P_{\text{out}}}(q, 1)$ is decreasing it follows that:

$$\inf_{q \in [0, a_{\rho_n}]} \sup_{r \geq 0} i_{\text{RS}}(q, r; \alpha_n, \rho_n) \geq \inf_{q \in [0, a_{\rho_n}]} I_{P_{\text{out}}}(q, 1) = I_{P_{\text{out}}}(a_{\rho_n}, 1). \tag{58}$$

For all $q \in [b_{\rho_n}, 1)$ we have:

$$\sup_{r \geq 0} i_{\text{RS}}(q, r; \alpha_n, \rho_n) = I_{P_{\text{out}}}(q, 1) + \sup_{r \geq 0}\left\{ \frac{rq}{2} - \frac{1}{\alpha_n} \psi_{P_{0,n}}\left( \frac{\alpha_n}{\rho_n} r \right) \right\}$$
$$\geq \frac{q(1 + |\ln \rho_n|^{-\frac{1}{4}})}{\gamma} - \frac{1}{\alpha_n} \psi_{P_{0,n}}\left( \frac{\alpha_n}{\rho_n} \frac{2(1 + |\ln \rho_n|^{-\frac{1}{4}})}{\gamma} \right)$$
$$\geq \frac{b_{\rho_n}}{\gamma} - \frac{1}{\alpha_n} \psi_{P_{0,n}}\left( \frac{\alpha_n}{\rho_n} \frac{2(1 + |\ln \rho_n|^{-\frac{1}{4}})}{\gamma} \right)$$
$$\geq \frac{b_{\rho_n}}{\gamma} - \frac{1}{\gamma}\left( \frac{1}{|\ln \rho_n|^{\frac{1}{4}}} + \frac{\ln 2}{|\ln \rho_n|} + \sqrt{\frac{2}{\pi |\ln \rho_n|}} \right). \tag{59}$$

The first inequality follows from the trivial lower bounds $I_{P_{\text{out}}}(q, 1) \geq 0$ and

$$\sup_{r \geq 0} \frac{rq}{2} - \frac{1}{\alpha_n} \psi_{P_{0,n}}\left( \frac{\alpha_n}{\rho_n} r \right) \geq \frac{\widetilde{r} q}{2} - \frac{1}{\alpha_n} \psi_{P_{0,n}}\left( \frac{\alpha_n}{\rho_n} \widetilde{r} \right) \quad \text{where} \quad \widetilde{r} := \frac{2(1 + |\ln \rho_n|^{-\frac{1}{4}})}{\gamma}.$$

The last inequality follows from Lemma 4 applied with $\epsilon = |\ln \rho_n|^{-\frac{1}{4}}$:

$$\frac{1}{\alpha_n} \psi_{P_{0,n}}\left( \frac{\alpha_n}{\rho_n} \frac{2(1 + |\ln \rho_n|^{-\frac{1}{4}})}{\gamma} \right) \leq \frac{1}{\gamma}\left( \frac{1}{|\ln \rho_n|^{\frac{1}{4}}} + \frac{\ln 2}{|\ln \rho_n|} + \sqrt{\frac{2}{\pi |\ln \rho_n|}} \right).$$

Note that the final lower bound (59) does not depend on $q \in [b_{\rho_n}, 1)$ so the same inequality holds for the infimum of $\sup_{r \geq 0} i_{\text{RS}}(q, r; \alpha_n, \rho_n)$ over $q \in [b_{\rho_n}, 1]$. Combining (57), (58) and (59) yields:

$$I(\rho_n, \alpha_n) \geq \min\left\{ I_{P_{\text{out}}}(a_{\rho_n}, 1); \inf_{q \in [a_{\rho_n}, b_{\rho_n}]} \sup_{r \geq 0} i_{\text{RS}}(q, r; \alpha_n, \rho_n) \,; \right.$$
$$\left. \frac{b_{\rho_n}}{\gamma} - \frac{1}{\gamma}\left( \frac{1}{|\ln \rho_n|^{\frac{1}{4}}} + \frac{\ln 2}{|\ln \rho_n|} + \sqrt{\frac{2}{\pi |\ln \rho_n|}} \right) \right\}.$$

Hence we have (remember the limit (55) and that $a_{\rho_n} \to 0$ and $b_{\rho_n} \to 1$ when $\rho_n$ vanishes):

$$\liminf_{\rho_n \to 0^+} I(\rho_n, \alpha_n) \geq \min \left\{ I_{P_{\text{out}}}(0,1) \, ; \min \left\{ I_{P_{\text{out}}}(0,1), \frac{1}{\gamma} \right\} ; \frac{1}{\gamma} \right\} = \min \left\{ I_{P_{\text{out}}}(0,1), \frac{1}{\gamma} \right\} .$$

(60)

We see thanks to (56) and (60) that the superior and inferior limits of $I(\rho_n, \alpha_n)$ match each other and $\lim_{\rho_n \to 0^+} I(\rho_n, \alpha_n) = \min \left\{ I_{P_{\text{out}}}(0,1), \frac{1}{\gamma} \right\}$. $\qquad \square$

Finally, we obtain Theorem 2 for the specific choice $P_{0,n} := (1 - \rho_n)\delta_0 + \rho_n \delta_1$ by combining Theorem 1 and Proposition 7 together:

$$\lim_{n \to +\infty} \frac{I(\mathbf{X}^*; \mathbf{Y} | \mathbf{\Phi})}{m_n} = \min \left\{ I_{P_{\text{out}}}(0,1) \, ; \, \frac{1}{\gamma} \right\} .$$

(61)

# 4 Asymptotic minimum mean-square error: proof of Theorem 3

Let $\widehat{\mathbf{X}} = \widehat{\mathbf{X}}(\mathbf{Y}, \mathbf{\Phi})$ be an estimator of $\mathbf{X}^*$ that is a function of the observations $\mathbf{Y}$ and the measurement matrix $\mathbf{\Phi}$. Then the mean-square error of this estimator is $\mathbb{E}\|\mathbf{X}^* - \widehat{\mathbf{X}}\|^2 / k_n \in [0, \mathbb{E}_{X \sim P_0} X^2]$ where the normalization factor $k_n := n\rho_n$ is the expected sparsity of $\mathbf{X}^*$. It is well-known that the Bayes estimator $\mathbb{E}[\mathbf{X}^* | \mathbf{Y}, \mathbf{\Phi}]$ achieves the minimum mean-square error (MMSE) among all estimators of the form $\widehat{\mathbf{X}}(\mathbf{Y}, \mathbf{\Phi})$. We denote the mean-square error of the Bayes estimator by

$$\text{MMSE}(\mathbf{X}^* | \mathbf{Y}, \mathbf{\Phi}) := \frac{\mathbb{E}\|\mathbf{X}^* - \mathbb{E}[\mathbf{X}^* | \mathbf{Y}, \mathbf{\Phi}]\|^2}{k_n} .$$

(62)

The MMSE is therefore a tight lower bound on the error that we achieve when estimating $\mathbf{X}^*$ from the observations $\mathbf{Y}$ and the known measurement matrix $\mathbf{\Phi}$. For this reason a result on the MMSE is easier to interpret than a result on the normalized mutual information $I(\mathbf{X}^*; \mathbf{Y} | \mathbf{\Phi}) / m_n$. In this section, we prove Theorem 3, that is, a formula for the asymptotic MMSE when $n$ diverges to infinity while $\rho_n = \Theta(n^{-\lambda})$ with $\lambda \in (0, 1/9)$ and $\alpha_n = \gamma \rho_n |\ln \rho_n|$ with $\gamma > 0$. The proof of this theorem is given at the end of this section. The proof relies on the I-MMSE relation [8] that links the MMSE to the derivative of the mutual information with respect to the signal-to-noise ratio of some well-chosen observation channel. For this reason, we first have to determine the asymptotic mutual information of a modified inference problem in which, in addition to the observations (2), we have access to the side information $\widetilde{\mathbf{Y}}^{(\tau)} = \sqrt{\alpha_n \tau / \rho_n} \, \mathbf{X}^* + \widetilde{\mathbf{Z}}$ with $\tau > 0$ and $\widetilde{\mathbf{Z}}$ an additive white Gaussian noise. Indeed, the parameter $\tau$ is akin to a signal-to-noise ratio and the derivative of the mutual information $I(\mathbf{X}^*; \mathbf{Y}, \widetilde{\mathbf{Y}}^{(\tau)} | \mathbf{\Phi}) / m_n$ with respect to $\tau$ yields half the MMSE [8]:

$$\frac{\partial}{\partial \tau} \left( \frac{I(\mathbf{X}^*; \mathbf{Y}, \widetilde{\mathbf{Y}}^{(\tau)} | \mathbf{\Phi})}{m_n} \right) = \frac{\text{MMSE}(\mathbf{X}^* | \mathbf{Y}, \widetilde{\mathbf{Y}}^{(\tau)}, \mathbf{\Phi})}{2} \xrightarrow[\tau \to 0^+]{} \frac{\text{MMSE}(\mathbf{X}^* | \mathbf{Y}, \mathbf{\Phi})}{2} .$$

## 4.1 Generalized linear estimation with side information

Let $(X_i^*)_{i=1}^n \overset{\text{iid}}{\sim} P_{0,n}$ be the components of the signal vector $\mathbf{X}^*$. We now have access to the observations:

$$\begin{cases} Y_\mu & \sim P_{\text{out}}\left( \cdot \, \left| \, \frac{(\mathbf{\Phi}\mathbf{X}^*)_\mu}{\sqrt{k_n}} \right. \right) , & 1 \leq \mu \leq m_n ; \\ \widetilde{Y}_i^{(\tau)} & = \sqrt{\frac{\alpha_n}{\rho_n} \tau} \, X_i^* + \widetilde{Z}_i & , & 1 \leq i \leq n ; \end{cases}$$

(63)

where $\tau \geq 0$. Remember that the transition kernel $P_{\text{out}}$ is defined in (4) using the activation function $\varphi$ and the probability distribution $P_A$. The side information induces only a small change in the *(replica-symmetric) potential* whose extremization gives the asymptotic normalized mutual information. More precisely, the potential now reads:

$$i_{\text{RS}}(q, r, \tau; \alpha_n, \rho_n) := \frac{1}{\alpha_n} I_{P_{0,n}}\left( \frac{\alpha_n}{\rho_n}(r + \tau) \right) + I_{P_{\text{out}}}\left( q, \mathbb{E}X^2 \right) - \frac{r(\mathbb{E}X^2 - q)}{2} ,$$

(64)

where $X \sim P_0$. We then have the following generalization of Theorem 1.

**Theorem 4** (Mutual information of the GLM with side information at sublinear sparsity and sampling rate). *Suppose that $\Delta > 0$ and that the following hypotheses hold:*

- *(H1) There exists $S > 0$ such that the support of $P_0$ is included in $[-S, S]$.*
- *(H2) $\varphi$ is bounded, and its first and second partial derivatives with respect to its first argument exist, are bounded and continuous. They are denoted $\partial_x \varphi$, $\partial_{xx} \varphi$.*
- *(H3) $(\Phi_{\mu i}) \overset{iid}{\sim} \mathcal{N}(0, 1)$.*

*Let $\rho_n = \Theta(n^{-\lambda})$ with $\lambda \in [0, 1/9)$ and $\alpha_n = \gamma \rho_n |\ln \rho_n|$ with $\gamma > 0$. Then for all $n \in \mathbb{N}^*$:*

$$\left| \frac{I(\mathbf{X}^*; \mathbf{Y}, \widetilde{\mathbf{Y}}^{(\tau)} | \mathbf{\Phi})}{m_n} - \inf_{q \in [0, \mathbb{E}_{P_0}[X^2]]} \sup_{r \geq 0} i_{\mathrm{RS}}(q, r, \tau; \alpha_n, \rho_n) \right| \leq \frac{\sqrt{C} |\ln n|^{1/6}}{n^{\frac{1}{12} - \frac{3\lambda}{4}}} , \qquad (65)$$

*where $C$ is a polynomial in $\left( \tau, S, \left\| \frac{\varphi}{\sqrt{\Delta}} \right\|_\infty, \left\| \frac{\partial_x \varphi}{\sqrt{\Delta}} \right\|_\infty, \left\| \frac{\partial_{xx} \varphi}{\sqrt{\Delta}} \right\|_\infty, \lambda, \gamma \right)$ with positive coefficients.*

*Proof.* The proof is similar to the proof of Theorem 1 except for a small change in the adaptive interpolation method due to the side information. More precisely, at $t \in [0, 1]$ we have access to the observations

$$\begin{cases} Y_\mu^{(t,\epsilon)} & \sim \dfrac{P_{\mathrm{out}}(\cdot \mid S_\mu^{(t,\epsilon)})}{} \quad , \ 1 \leq \mu \leq m_n ; \\ \widetilde{Y}_i^{(t,\epsilon,\tau)} & = \sqrt{\dfrac{\alpha_n}{\rho_n} \tau + R_1(t, \epsilon)} \, X_i^* + \widetilde{Z}_i , \ 1 \leq i \leq n \ ; \end{cases} \qquad (66)$$

where $X_i^* \overset{iid}{\sim} P_{0,n}$, $\widetilde{Z}_i \overset{iid}{\sim} \mathcal{N}(0, 1)$ and

$$S_\mu^{(t,\epsilon)} := \sqrt{\frac{1-t}{k_n}} \sum_{i=1}^n \Phi_{\mu i} X_i^* + \sqrt{R_2(t, \epsilon)} \, V_\mu + \sqrt{t + 2s_n - R_2(t, \epsilon)} \, W_\mu^*$$

with $\Phi_{\mu i}, V_\mu, W_\mu^* \overset{iid}{\sim} \mathcal{N}(0, 1)$. The proof then goes by looking to the interpolating mutual information $I((\mathbf{X}^*, \mathbf{W}^*); (\mathbf{Y}^{(t,\epsilon)}, \widetilde{\mathbf{Y}}^{(t,\epsilon,\tau)}) | \mathbf{\Phi}) / m_n$, and follows exactly the same lines than the proof of Theorem 1. In particular, the interpolation functions $(R_1, R_2)$ are chosen a posteriori as the solutions to the same second-order ordinary differential equations than for Theorem 1. $\qquad \square$

Let $X^* \sim P_{0,n} \perp Z \sim \mathcal{N}(0, 1)$. We define for all $r \geq 0$:

$$\psi_{P_{0,n}}(r) := \mathbb{E}\left[ \ln \int dP_{0,n}(x) e^{-\frac{r}{2} x^2 + r X^* x + \sqrt{r} x Z} \right] .$$

Note that $I_{P_{0,n}}(r) := I(X^*; \sqrt{r} \, X^* + Z) = \frac{r \rho_n \mathbb{E}[X^2]}{2} - \psi_{P_{0,n}}(r)$ where $X \sim P_0$. For $\rho_n, \alpha_n > 0$ and $\tau \geq 0$, we denote the variational problem appearing in Theorem 1 by

$$I(\rho_n, \alpha_n, \tau) := \inf_{q \in [0, \mathbb{E}X^2]} \sup_{r \geq 0} i_{\mathrm{RS}}(q, r, \tau; \alpha_n, \rho_n)$$

$$= \inf_{q \in [0, \mathbb{E}X^2]} I_{P_{\mathrm{out}}}(q, \mathbb{E}X^2) + \frac{\tau \mathbb{E}X^2}{2} + \sup_{r \geq 0} \left\{ \frac{rq}{2} - \frac{1}{\alpha_n} \psi_{P_{0,n}}\left( \frac{\alpha_n}{\rho_n}(r + \tau) \right) \right\}$$

$$= \inf_{q \in [0, \mathbb{E}X^2]} I_{P_{\mathrm{out}}}(q, \mathbb{E}X^2) + \frac{\tau(\mathbb{E}X^2 - q)}{2} + \sup_{r \geq \tau} \left\{ \frac{rq}{2} - \frac{1}{\alpha_n} \psi_{P_{0,n}}\left( \frac{\alpha_n}{\rho_n} r \right) \right\}, \quad (67)$$

where $X \sim P_0$. Similarly to what is done in Appendix F, we can compute the limit of $I(\rho_n, \alpha_n, \tau)$ for a discrete distribution with finite support $P_0$.

**Proposition 8.** *Let $P_{0,n} := (1 - \rho_n)\delta_0 + \rho_n P_0$ where $P_0$ is a discrete distribution with finite support $\mathrm{supp}(P_0) \subseteq \{-v_K, -v_{K-1}, \ldots, -v_1, v_1, v_2, \ldots, v_K\}$ where $0 < v_1 < \cdots < v_K < v_{K+1} = +\infty$. Let $\alpha_n := \gamma \rho_n |\ln \rho_n|$ for a fix $\gamma > 0$. For every $\tau \in [0, 2/\gamma v_K^2)$, $I(\rho_n, \alpha_n, \tau)$ defined in (67) converges when $\rho_n \to 0^+$ and (in what follows $X \sim P_0$):*

$$\lim_{\rho_n \to 0^+} I(\rho_n, \alpha_n, \tau)$$

$$= \min_{1 \leq k \leq K+1} \left\{ I_{P_{\mathrm{out}}}\left( \mathbb{E}[X^2 \mathbf{1}_{\{|X| \geq v_k\}}], \mathbb{E}[X^2] \right) + \frac{\mathbb{P}(|X| \geq v_k)}{\gamma} + \frac{\tau \mathbb{E}[X^2 \mathbf{1}_{\{|X| < v_k\}}]}{2} \right\}. \quad (68)$$

*Proof.* Fix $\tau \in [0, 2/\gamma v_K^2)$. Define $\widetilde{I}_{P_{\text{out}}}(q, \mathbb{E}X^2) = I_{P_{\text{out}}}(q, \mathbb{E}X^2) + \frac{\tau(\mathbb{E}X^2 - q)}{2}$. From (67) we have

$$I(\rho_n, \alpha_n, \tau) = \inf_{q \in [0, \mathbb{E}X^2]} \widetilde{I}_{P_{\text{out}}}(q, \mathbb{E}X^2) + \sup_{r \geq \tau} \left\{ \frac{rq}{2} - \frac{1}{\alpha_n} \psi_{P_{0,n}}\left(\frac{\alpha_n}{\rho_n} r\right) \right\}. \tag{69}$$

Note that $\widetilde{I}_{P_{\text{out}}}(\cdot, \mathbb{E}X^2)$ is concave nonincreasing on $[0, \mathbb{E}X^2]$ – exactly as $I_{P_{\text{out}}}(\cdot, \mathbb{E}X^2)$ –, and that the variational problem (69) has a form similar to the quantity $I(\rho_n, \alpha_n)$ whose limit is given by Proposition 15 in Appendix F. The only difference that we have to take into account in the analysis is that the supremum is over $r \in [\tau, +\infty)$ instead of $r \in [0, +\infty)$.

Remember the definition (196) of $a_{\rho_n}^{(K)}$. By Lemma 15, for every $q \in (\rho_n \mathbb{E}[X]^2, \mathbb{E}[X^2])$ there exists a unique $r_n^*(q) \in (0, +\infty)$ such that

$$\frac{r_n^*(q)q}{2} - \frac{1}{\alpha_n} \psi_{P_{0,n}}\left(\frac{\alpha_n}{\rho_n} r_n^*(q)\right) = \sup_{r \geq 0} \frac{rq}{2} - \frac{1}{\alpha_n} \psi_{P_{0,n}}\left(\frac{\alpha_n}{\rho_n} r\right), \tag{70}$$

and $\forall q \in [a_{\rho_n}^{(K)}, \mathbb{E}X^2] : r_n^*(q) \geq 2(1 - |\ln \rho_n|^{-\frac{1}{4}})/\gamma v_K^2$. By assumption $\tau < 2/\gamma v_K^2$ so, for $\rho_n$ small enough, $\forall q \in [a_{\rho_n}^{(K)}, \mathbb{E}X^2] : r_n^*(q) > \tau$. It follows that $\forall q \in [a_{\rho_n}^{(K)}, \mathbb{E}X^2] : r_n^*(q)$ satisfies

$$\frac{r_n^*(q)q}{2} - \frac{1}{\alpha_n} \psi_{P_{0,n}}\left(\frac{\alpha_n}{\rho_n} r_n^*(q)\right) = \sup_{r \geq \tau} \frac{rq}{2} - \frac{1}{\alpha_n} \psi_{P_{0,n}}\left(\frac{\alpha_n}{\rho_n} r\right). \tag{71}$$

Thanks to the identity (71) the same analysis leading to Propositions 14 and 15 can be repeated, replacing $I_{P_{\text{out}}}(\cdot, \mathbb{E}X^2)$ by $\widetilde{I}_{P_{\text{out}}}(\cdot, \mathbb{E}X^2)$ (this makes no difference as we only need for $\widetilde{I}_{P_{\text{out}}}(\cdot, \mathbb{E}X^2)$ to be concave nonincreasing), in order to obtain the limit:

$$\lim_{\rho_n \to 0^+} \inf_{q \in [a_{\rho_n}^{(K)}, \mathbb{E}X^2]} \sup_{r \geq \tau} \widetilde{I}_{P_{\text{out}}}(q, \mathbb{E}X^2) + \frac{rq}{2} - \frac{1}{\alpha_n} \psi_{P_{0,n}}\left(\frac{\alpha_n}{\rho_n} r\right)$$

$$= \min_{1 \leq k \leq K+1} \left\{ \widetilde{I}_{P_{\text{out}}}\left(\mathbb{E}[X^2 \mathbf{1}_{\{|X| \geq v_k\}}], \mathbb{E}X^2\right) + \frac{\mathbb{P}(|X| \geq v_k)}{\gamma} \right\}. \tag{72}$$

Note that the limit (72) is for the infimum over $q \in [a_{\rho_n}^{(K)}, \mathbb{E}X^2]$, not the infimum over $q \in [0, \mathbb{E}X^2]$. This is because, for $q \in (\rho_n \mathbb{E}X^2, a_{\rho_n}^{(K)})$, $r_n^*(q)$ does not necessarily satisfy (71). However, the limit (72) directly implies the following upper bound on the limit superior:

$$\limsup_{\rho_n \to 0^+} I(\rho_n, \alpha_n, \tau) \leq \min_{1 \leq k \leq K+1} \left\{ \widetilde{I}_{P_{\text{out}}}\left(\mathbb{E}[X^2 \mathbf{1}_{\{|X| \geq v_k\}}], \mathbb{E}X^2\right) + \frac{\mathbb{P}(|X| \geq v_k)}{\gamma} \right\}. \tag{73}$$

In order to lower bound the limit inferior, we have to lower bound the infimum over $q \in [0, a_{\rho_n}^{(K)}]$ of $\widetilde{I}_{P_{\text{out}}}(q, \mathbb{E}X^2) + \sup_{r \geq \tau} \left\{ \frac{rq}{2} - \frac{1}{\alpha_n} \psi_{P_{0,n}}\left(\alpha_n r / \rho_n\right) \right\}$. Because $\widetilde{I}_{P_{\text{out}}}(\cdot, \mathbb{E}X^2)$ is nonincreasing and $q \mapsto \sup_{r \geq \tau} \left\{ \frac{rq}{2} - \frac{1}{\alpha_n} \psi_{P_{0,n}}\left(\alpha_n r / \rho_n\right) \right\}$ is nondecreasing (it is the supremum of nondecreasing functions), we have:

$$\inf_{q \in [0, a_{\rho_n}^{(K)}]} \widetilde{I}_{P_{\text{out}}}(q, \mathbb{E}X^2) + \sup_{r \geq \tau} \left\{ \frac{rq}{2} - \frac{1}{\alpha_n} \psi_{P_{0,n}}\left(\frac{\alpha_n}{\rho_n} r\right) \right\}$$

$$\geq \widetilde{I}_{P_{\text{out}}}(a_{\rho_n}^{(K)}, \mathbb{E}X^2) + \sup_{r \geq \tau} \left\{ -\frac{1}{\alpha_n} \psi_{P_{0,n}}\left(\frac{\alpha_n}{\rho_n} r\right) \right\}$$

$$\geq \widetilde{I}_{P_{\text{out}}}(a_{\rho_n}^{(K)}, \mathbb{E}X^2) - \frac{1}{\alpha_n} \psi_{P_{0,n}}\left(\frac{\alpha_n}{\rho_n} \tau\right). \tag{74}$$

The last inequality follows from $\psi_{P_{0,n}}$ being nondecreasing (see Lemma 6). We can use the computations in the proof of Lemma 16 to write $\frac{1}{\alpha_n} \psi_{P_{0,n}}\left(\frac{\alpha_n \tau}{\rho_n}\right)$ more explicitly:

$$\frac{1}{\alpha_n} \psi_{P_{0,n}}\left(\frac{\alpha_n \tau}{\rho_n}\right) = \frac{B_{\rho_n}}{\gamma} + \frac{\tau \mathbb{E}X^2}{2} - \frac{1}{\gamma} + \frac{1}{\gamma} \sum_{j=1}^K p_j^+ \mathbb{E}\left[\frac{\ln \widetilde{h}\left(Z, \gamma\tau |\ln \rho_n|, v_j; \rho_n, \mathbf{v}, \mathbf{p}^+, \mathbf{p}^-\right)}{|\ln \rho_n|}\right]$$

$$+ \frac{1}{\gamma} \sum_{j=1}^K p_j^- \mathbb{E}\left[\frac{\ln \widetilde{h}\left(Z, \gamma\tau |\ln \rho_n|, v_j; \rho_n, \mathbf{v}, \mathbf{p}^-, \mathbf{p}^+\right)}{|\ln \rho_n|}\right], \tag{75}$$

where

$$B_{\rho_n} = \frac{1-\rho_n}{\rho_n |\ln \rho_n|} \mathbb{E} \ln \left(1 - \rho_n + \rho_n \sum_{i=1}^{K} e^{-\frac{\gamma\tau}{2v_k^2}|\ln\rho_n|} \left(p_i^+ e^{\sqrt{\gamma\tau|\ln\rho_n|v_i^2}Z} + p_i^- e^{-\sqrt{\gamma\tau|\ln\rho_n|v_i^2}Z}\right)\right)$$

and $\forall z \in \mathbb{R}$ :

$$\widetilde{h}\big(z, \gamma\tau|\ln\rho_n|, v_j; \rho_n, \mathbf{v}, \mathbf{p}^{\pm}, \mathbf{p}^{\mp}\big) = (1-\rho_n)e^{|\ln\rho_n|\left(1-\frac{\gamma\tau v_j^2}{2} - \sqrt{\frac{\gamma\tau v_j^2}{|\ln\rho_n|}}z\right)}$$
$$+ \sum_{i=1}^{K} e^{-|\ln\rho_n|\left(\frac{\gamma\tau(v_i-v_j)^2}{2} - \sqrt{\frac{\gamma\tau}{|\ln\rho_n|}}(v_i-v_j)z\right)} \left(p_i^{\pm} + p_i^{\mp} e^{-2|\ln\rho_n|v_i\left(\gamma\tau v_j + z\sqrt{\frac{\gamma\tau}{|\ln\rho_n|}}\right)}\right). \quad (76)$$

We can show, exactly as it is done for $A_{\rho_n}$ in the proof of Lemma 16, that $|B_{\rho_n}| \leq {}^1/|\ln\rho_n|$. As $\tau < {}^2/\gamma v_K^2$ we have $\forall j \in \{1, \ldots, K\} : 1 - \gamma\tau v_j^2/2 > 0$, and from (76) we then easily deduce that $\forall j \in \{1, \ldots, K\}, \forall z \in \mathbb{R}$ :

$$\lim_{\rho_n \to 0^+} \frac{\ln \widetilde{h}\big(z, \gamma\tau|\ln\rho_n|, v_j; \rho_n, \mathbf{v}, \mathbf{p}^{\pm}, \mathbf{p}^{\mp}\big)}{|\ln\rho_n|} = 1 - \frac{\gamma\tau v_j^2}{2} . \quad (77)$$

By the dominated convergence theorem, making use of the pointwise limits (77), we have:

$$\sum_{j=1}^{K} p_j^+ \mathbb{E}\left[\frac{\ln \widetilde{h}\big(Z, \gamma\tau|\ln\rho_n|, v_j; \rho_n, \mathbf{v}, \mathbf{p}^+, \mathbf{p}^-\big)}{|\ln\rho_n|}\right] + p_j^- \mathbb{E}\left[\frac{\ln \widetilde{h}\big(Z, \gamma\tau|\ln\rho_n|, v_j; \rho_n, \mathbf{v}, \mathbf{p}^-, \mathbf{p}^+\big)}{|\ln\rho_n|}\right]$$
$$\xrightarrow[\rho_n \to 0^+]{} \sum_{j=1}^{K} (p_j^+ + p_j^-)\left(1 - \frac{\gamma\tau v_j^2}{2}\right) = 1 - \frac{\gamma\tau\mathbb{E}X^2}{2} . \quad (78)$$

Combining the identity (75), $\lim_{\rho_n \to 0^+} B_{\rho_n} = 0$ and the limit (78) yields:

$$\lim_{\rho_n \to 0} \frac{1}{\alpha_n} \psi_{P_{0,n}}\left(\frac{\alpha_n}{\rho_n}\tau\right) = \frac{\tau\mathbb{E}X^2}{2} - \frac{1}{\gamma} + \frac{1}{\gamma}\left(1 - \frac{\gamma\tau\mathbb{E}X^2}{2}\right) = 0 . \quad (79)$$

The lower bound (74) together with the limits (79) and $\lim_{\rho_n \to 0^+} a_{\rho_n}^{(K)} = 0$ (see Lemma 15) implies:

$$\liminf_{\rho_n \to 0^+} \inf_{q \in [0, a_{\rho_n}^{(K)}]} \widetilde{I}_{P_{\mathrm{out}}}(q, \mathbb{E}X^2) + \sup_{r \geq \tau}\left\{\frac{rq}{2} - \frac{1}{\alpha_n}\psi_{P_{0,n}}\left(\frac{\alpha_n}{\rho_n}r\right)\right\} \geq \widetilde{I}_{P_{\mathrm{out}}}(0, \mathbb{E}X^2) . \quad (80)$$

Finally, we combine the latter inequality with the limit (72) to obtain

$$\liminf_{\rho_n \to 0^+} I(\rho_n, \alpha_n, \tau) \geq \min_{1 \leq k \leq K+1}\left\{\widetilde{I}_{P_{\mathrm{out}}}\big(\mathbb{E}[X^2\mathbf{1}_{\{|X| \geq v_k\}}], \mathbb{E}X^2\big) + \frac{\mathbb{P}(|X| \geq v_k)}{\gamma}\right\} . \quad (81)$$

The upper bound (73) on the limit superior matches the lower bound (81) on the limit inferior. Hence,

$$\lim_{\rho_n \to 0^+} I(\rho_n, \alpha_n, \tau) = \min_{1 \leq k \leq K+1}\left\{\widetilde{I}_{P_{\mathrm{out}}}\big(\mathbb{E}[X^2\mathbf{1}_{\{|X| \geq v_k\}}], \mathbb{E}X^2\big) + \frac{\mathbb{P}(|X| \geq v_k)}{\gamma}\right\}$$
$$= \min_{1 \leq k \leq K+1}\left\{I_{P_{\mathrm{out}}}\big(\mathbb{E}[X^2\mathbf{1}_{\{|X| \geq v_k\}}], \mathbb{E}X^2\big) + \frac{\tau\mathbb{E}[X^2\mathbf{1}_{\{|X| < v_k\}}]}{2} + \frac{\mathbb{P}(|X| \geq v_k)}{\gamma}\right\};$$

where the last equality follows simply from the definition of $\widetilde{I}_{P_{\mathrm{out}}}$. $\qquad\square$

The next theorem is a direct corollary of Theorem 4 and Proposition 8.

**Theorem 5.** *Suppose that $\Delta > 0$ and that $P_{0,n} \coloneqq (1-\rho_n)\delta_0 + \rho_n P_0$ where $P_0$ is a discrete distribution with finite support $\mathrm{supp}(P_0) \subseteq \{-v_K, -v_{K-1}, \ldots, -v_2, -v_1, v_1, v_2, \ldots, v_{K-1}, v_K\}$ where $0 < v_1 < v_2 < \cdots < v_K < v_{K+1} = +\infty$. Further assume that the following hypotheses hold:*

   *(H2) $\varphi$ is bounded, and its first and second partial derivatives with respect to its first argument exist, are bounded and continuous. They are denoted $\partial_x\varphi$, $\partial_{xx}\varphi$.*

*(H3)* $(\Phi_{\mu i}) \overset{iid}{\sim} \mathcal{N}(0,1)$.

*Let $\rho_n = \Theta(n^{-\lambda})$ with $\lambda \in (0, 1/9)$ and $\alpha_n = \gamma \rho_n |\ln \rho_n|$ with $\gamma > 0$. Then $\forall \tau \in [0, 2/\gamma v_K^2)$:*

$$\lim_{n \to +\infty} \frac{I(\mathbf{X}^*; \mathbf{Y}, \widetilde{\mathbf{Y}}^{(\tau)} | \boldsymbol{\Phi})}{m_n}$$

$$= \min_{1 \le k \le K+1} \left\{ I_{P_{\text{out}}} \left( \mathbb{E}[X^2 \mathbf{1}_{\{|X| \ge v_k\}}], \mathbb{E} X^2 \right) + \frac{\tau \mathbb{E}[X^2 \mathbf{1}_{\{|X| < v_k\}}]}{2} + \frac{\mathbb{P}(|X| \ge v_k)}{\gamma} \right\}.$$

## 4.2 Proof of Theorem 3

For all $n \in \mathbb{N}^*$ and $\tau \in [0, +\infty)$ we define $i_n(\tau) := I(\mathbf{X}^*; \mathbf{Y}, \widetilde{\mathbf{Y}}^{(\tau)} | \boldsymbol{\Phi})/m_n$ the normalized conditional mutual information between $\mathbf{X}^*$ and the observations $\mathbf{Y}, \widetilde{\mathbf{Y}}^{(\tau)}$ – defined in (63) – given $\boldsymbol{\Phi}$. We place ourselves in the regime of Theorem 3, that is, $\rho_n = \Theta(n^{-\lambda})$ with $\lambda \in [0, 1/9)$ and $\alpha_n = \gamma \rho_n |\ln \rho_n|$ with $\gamma > 0$. By Theorem 5 if the side-information is low enough, namely $\tau < 2/\gamma v_K^2$, then $\lim_{n \to +\infty} i_n(\tau) = i(\tau)$ where

$$i(\tau) := \min_{1 \le k \le K+1} \left\{ I_{P_{\text{out}}} \left( \mathbb{E}[X^2 \mathbf{1}_{\{|X| \ge v_k\}}], \mathbb{E} X^2 \right) + \frac{\tau \mathbb{E}[X^2 \mathbf{1}_{\{|X| < v_k\}}]}{2} + \frac{\mathbb{P}(|X| \ge v_k)}{\gamma} \right\}. \quad (82)$$

We first establish a few properties of the function $i_n$. The posterior density of $\mathbf{X}^*$ given the observations $(\mathbf{Y}, \widetilde{\mathbf{Y}}^{(\tau)})$ defined in (63) reads:

$$dP(\mathbf{x} | \mathbf{Y}, \widetilde{\mathbf{Y}}^{(\tau)}) = \frac{1}{\mathcal{Z}(\mathbf{Y}, \widetilde{\mathbf{Y}}^{(\tau)})} \prod_{i=1}^{n} dP_{0,n}(x_i) e^{-\frac{1}{2} \left( \widetilde{Y}_i^{(\tau)} - \sqrt{\frac{\alpha_n \tau}{\rho_n}} x_i \right)^2} \prod_{\mu=1}^{m_n} P_{\text{out}} \left( Y_\mu \Big| \frac{(\boldsymbol{\Phi} \mathbf{x})_\mu}{\sqrt{k_n}} \right), \quad (83)$$

where $\mathcal{Z}(\mathbf{Y}, \widetilde{\mathbf{Y}}^{(\tau)})$ is a normalization factor. In what follows $\mathbf{x}$ denotes a $n$-dimensional random vector distributed with respect to the posterior distribution (83). We will use the brackets $\langle - \rangle_{n,\tau}$ to denote an expectation with respect to $\mathbf{x}$. By definition of the mutual information we have:

$$i_n(\tau) = -\frac{1}{m_n} \mathbb{E} \ln \mathcal{Z}(\mathbf{Y}, \widetilde{\mathbf{Y}}^{(\tau)}) + \frac{1}{m_n} \mathbb{E} \left[ \ln \prod_{i=1}^{n} e^{-\frac{1}{2} \left( \widetilde{Y}_i^{(\tau)} - \sqrt{\frac{\alpha_n \tau}{\rho_n}} X_i^* \right)^2} \prod_{\mu=1}^{m_n} P_{\text{out}} \left( Y_\mu \Big| \frac{(\boldsymbol{\Phi} \mathbf{X}^*)_\mu}{\sqrt{k_n}} \right) \right]$$

$$= -\frac{1}{m_n} \mathbb{E} \ln \mathcal{Z}(\mathbf{Y}, \widetilde{\mathbf{Y}}^{(\tau)}) - \frac{1}{2\alpha_n} + \mathbb{E} \left[ \ln P_{\text{out}} \left( Y_1 \Big| \frac{(\boldsymbol{\Phi} \mathbf{X}^*)_1}{\sqrt{k_n}} \right) \right]. \quad (84)$$

Derivation under the expectation sign, justified by the dominated convergence theorem, yields the first derivative:

$$i_n'(\tau) = \frac{1}{m_n} \sum_{i=1}^{n} \mathbb{E} \left[ \left\langle \left( \widetilde{Y}_i^{(\tau)} - \sqrt{\frac{\alpha_n \tau}{\rho_n \tau}} x_i \right) \frac{1}{2} \sqrt{\frac{\alpha_n}{\rho_n \tau}} (X_i^* - x_i) \right\rangle_{n,\tau} \right]$$

$$= \frac{1}{m_n} \sum_{i=1}^{n} \mathbb{E} \left[ \left\langle \left( \widetilde{Y}_i^{(\tau)} - \sqrt{\frac{\alpha_n \tau}{\rho_n \tau}} X_i^* \right) \frac{1}{2} \sqrt{\frac{\alpha_n}{\rho_n \tau}} (x_i - X_i^*) \right\rangle_{n,\tau} \right]$$

$$= \frac{1}{2 m_n} \sqrt{\frac{\alpha_n}{\rho_n \tau}} \mathbb{E} \left[ \widetilde{Z}_i (\langle x_i \rangle_{n,\tau} - X_i^*) \right]$$

$$= \frac{1}{2 m_n} \sqrt{\frac{\alpha_n}{\rho_n \tau}} \sum_{i=1}^{n} \mathbb{E} \left[ \widetilde{Z}_i \langle x_i \rangle_{n,\tau} \right]$$

$$= \frac{1}{2 m_n} \frac{\alpha_n}{\rho_n} \sum_{i=1}^{n} \mathbb{E} \left[ \langle x_i^2 \rangle_{n,\tau} - \langle x_i \rangle_{n,\tau}^2 \right]$$

$$= \frac{\mathbb{E} \| \mathbf{X}^* - \mathbb{E}[\mathbf{X}^* | \mathbf{Y}, \mathbf{Y}^{(\tau)}, \boldsymbol{\Phi}] \|^2}{2 k_n}. \quad (85)$$

The second equality above follows from Nishimori identity. The fifth equality is obtained thanks to a Gaussian integration by parts with respect to $\widetilde{Z}_i$. The final identity (85) is the I-MMSE relation

previously mentioned. Further differentiating with respect to $\tau$ and integrating by parts with respect to the Gaussian random variables $\widetilde{Z}_i$ give

$$i_n''(\tau) = -\frac{1}{2k_n}\sum_{i=1}^n \mathbb{E}\big[\langle (x_i - \langle x_i\rangle_{n,\tau})^2\rangle_{n,\tau}^2\big] . \tag{86}$$

The identity (86) shows that $i_n$ is concave as its second derivative is nonpositive. By Griffiths' lemma it follows that whenever the pointwise limit (82) is differentiable at $\tau \in (0, {}^2/_{\gamma v_K^2})$ we have:

$$\lim_{n\to+\infty} i_n'(\tau) = i'(\tau) .$$

The final step is to determine $i'(\tau)$. Suppose that the minimization problem

$$\min_{1\le k\le K+1}\left\{ I_{P_{\mathrm{out}}}\big(\mathbb{E}[X^2\mathbf{1}_{\{|X|\ge v_k\}}], \mathbb{E}[X^2]\big) + \frac{\mathbb{P}(|X| \ge v_k)}{\gamma}\right\} \tag{87}$$

has a unique solution $k^* \in \{1,\ldots,K+1\}$. Then, there exists $\epsilon \in [0, {}^2/_{\gamma v_K^2})$ such that $\forall \tau \in [0,\epsilon) : k^*$ is the unique solution to the minimization problem

$$\min_{1\le k\le K+1}\left\{ I_{P_{\mathrm{out}}}\big(\mathbb{E}[X^2\mathbf{1}_{\{|X|\ge v_k\}}], \mathbb{E}[X^2]\big) + \frac{\tau\mathbb{E}[X^2\mathbf{1}_{\{|X|<v_k\}}]}{2} + \frac{\mathbb{P}(|X| \ge v_k)}{\gamma}\right\} .$$

Therefore, $\forall \tau \in [0,\epsilon)$ :

$$i(\tau) = I_{P_{\mathrm{out}}}\big(\mathbb{E}[X^2\mathbf{1}_{\{|X|\ge v_{k^*}\}}], \mathbb{E}[X^2]\big) + \frac{\tau\mathbb{E}[X^2\mathbf{1}_{\{|X|<v_{k^*}\}}]}{2} + \frac{\mathbb{P}(|X| \ge v_{k^*})}{\gamma} ,$$

$$i'(\tau) = \frac{\mathbb{E}[X^2\mathbf{1}_{\{|X|<v_{k^*}\}}]}{2} .$$

We conclude that whenever the minimization problem (87) has a unique solution $k^*$ we have

$$\lim_{n\to+\infty}\frac{\mathbb{E}\|\mathbf{X}^* - \mathbb{E}[\mathbf{X}^*|\mathbf{Y}, \mathbf{\Phi}]\|^2}{k_n} = \lim_{n\to+\infty} 2i_n'(0) = 2i'(0) = \mathbb{E}[X^2\mathbf{1}_{\{|X|<v_{k^*}\}}] .$$

## 4.3 All-or-nothing phenomenon and its generalization

We now look at the asymptotic MMSE as a function of the number of measurements, i.e., as a function of the parameter $\gamma$ that controls the number of measurements $m_n = \gamma \cdot n\rho_n|\log \rho_n|$. Let $X \sim P_0$ and assume that $\mathrm{supp}|X| = K$. We place ourselves under the assumptions of Theorem 3. The functions $k \mapsto I_{P_{\mathrm{out}}}\big(\mathbb{E}[X^2\mathbf{1}_{\{|X|\ge v_k\}}], \mathbb{E}[X^2]\big)$ and $k \mapsto \mathbb{P}(|X| \ge v_k)$ are nondecreasing and increasing on $\{1, 2, \ldots, K+1\}$, respectively. Hence, the minimization problem on the right-hand side of (9) has a unique solution denoted $k^*(\gamma)$ for all but $K$ or less values of $\gamma \in (0, +\infty)$, and $\gamma_1 < \gamma_2 \Rightarrow k^*(\gamma_1) \ge k^*(\gamma_2)$ (assuming $k^*(\gamma_1), k^*(\gamma_2)$ are well-defined). By Theorem 3, it implies that the asymptotic MMSE as a function of $\gamma$ is nonincreasing and piecewise constant; its image is included in $\{\mathbb{E}X^2, \mathbb{E}[X^2\mathbf{1}_{\{|X|\le v_{K-1}\}}], \ldots, \mathbb{E}[X^2\mathbf{1}_{\{|X|\le v_1\}}], 0\}$. The asymptotic MMSE has at most $K$ discontinuities. As $\gamma$ increases past a discontinuity, the asymptotic MMSE jumps from $\mathbb{E}[X^2\mathbf{1}_{\{|X|<v_{k_1^*}\}}]$ for some $k_1^* \in \{2, \ldots, K+1\}$ down to a lower value $\mathbb{E}[X^2\mathbf{1}_{\{|X|<v_{k_2^*}\}}]$ where $k_2^* \in \{1, \ldots, k_1^* - 1\}$.

Therefore, when $K = 1$, the asymptotic MMSE has one discontinuity at $\gamma_c := 1/I_{P_{\mathrm{out}}}(0, \mathbb{E}X^2)$ where it jumps down from $\mathbb{E}X^2$ to $0$: this is the all-or-nothing phenomenon previously observed in [9, 10, 11] for a linear activation function $\varphi(x) = x$ and a deterministic distribution $P_0$. Theorem 3 generalizes this all-or-nothing phenomenon to activation functions satisfying mild conditions and any discrete distribution $P_0$ whose support is included in $\{-v, v\}$ for some $v > 0$.

When $K > 1$, the phenomenology is more complex. The asymptotic MMSE exhibits intermerdiate plateaus in between the plateaus "MMSE $= \mathbb{E}X^2$" (no reconstruction at all) for low values of $\gamma$ and "MMSE $= 0$" (perfect reconstruction) for large values of $\gamma$. For illustration purposes we now define the following three discrete distributions with support size $K \ge 1$:

- $P_{\mathrm{unif}}^{(K)}$ is the uniform distribution on $\{\sqrt{a}, 2\sqrt{a}, \ldots, K\sqrt{a}\}$ with $a := {}^6/_{(K+1)(2K+1)}$ so that $\mathbb{E}X^2 = 1$ for $X \sim P_0$.

- $P_{\text{linear}}^{(K)}$ is the distribution on $\{\sqrt{b}, 2\sqrt{b}, \ldots, K\sqrt{b}\}$ with $b := \sum_{j=1}^{K} 1/Kj^2$ and $P_{\text{linear}}^{(K)}(i\sqrt{b}) = 1/Ki^2b$ so that $\mathbb{E}X^2 = 1$ and $\mathbb{E}[X^2\mathbf{1}_{\{|X|<k\sqrt{b}\}}] = k-1/K$ for $X \sim P_0$, i.e., the quantity $\mathbb{E}[X^2\mathbf{1}_{\{|X|<v_k\}}]$ increases linearly with $k$.

- $P_{\text{binom}}^{(K,p)}$ is the binomial distribution on $\{\sqrt{c}, 2\sqrt{c}, \ldots, K\sqrt{c}\}$ with

$$c = 1/(K-1)(K-2)p^2 + 3(K-1)p + 1$$

and $P_{\text{binom}}^{(K,p)}(i\sqrt{c}) = \binom{K-1}{i-1}p^{i-1}(1-p)^{K-i}$ so that $\mathbb{E}X^2 = 1$.

In Figure 1 we plot the asymptotic MMSE (using Theorem 3) as a function of the noise variance $\Delta$ and the parameter $\gamma$ for three different activation functions and $P_0 \in \{P_{\text{unif}}^{(5)}, P_{\text{linear}}^{(5)}, P_{\text{binom}}^{(5,0.2)}\}$.

Figure 1: Minimum mean-square error in the asymptotic regime of Theorem 3 for $\Delta \in [0, 4]$ and $\gamma \in (0, 10.5]$. *From left to right:* the activation function is linear $\varphi(x) = x$, the ReLU $\varphi(x) = \max(0, x)$ and the sign function $\varphi(x) = \text{sign}(x)$. *Top to bottom:* the prior distribution $P_0$ of the nonzero elements of $\mathbf{X}^*$ is $P_{\text{unif}}^{(5)}$, $P_{\text{linear}}^{(5)}$ and $P_{\text{binom}}^{(5,0.2)}$.

## Acknowledgments and Disclosure of Funding

The work of C. L. is supported by the Swiss National Foundation for Science grant number 200021E 17554.

## Footnotes

[2] Remember that $r_\epsilon$ takes its values in $[0, \frac{\alpha_n}{\rho_n} r_{\max}]$. Besides, under (H2), $u'_{Y_\mu^{(t,\epsilon)}}$ is upper bounded by

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

# A    Properties of the mutual informations of the scalar channels

This appendix gives important properties on the mutual informations of the scalar channels defined in Section 1. We first recall the important Nishimori identity that we will use in this appendix and others as well.

**Lemma 5** (Nishimori identity). *Let $(\mathbf{X}, \mathbf{Y}) \in \mathbb{R}^{n_1} \times \mathbb{R}^{n_2}$ be a pair of jointly distributed random vectors. Let $k \geq 1$. Let $\mathbf{X}^{(1)}, \ldots, \mathbf{X}^{(k)}$ be $k$ independent samples drawn from the conditional distribution $P(\mathbf{X} = \cdot | \mathbf{Y})$, independently of every other random variables. The angular brackets $\langle - \rangle$ denote the expectation operator with respect to $P(\mathbf{X} = \cdot | \mathbf{Y})$, while $\mathbb{E}$ denotes the expectation with respect to $(\mathbf{X}, \mathbf{Y})$. Then, for every integrable function $g$ the two following quantities are equal:*

$$\mathbb{E}\langle g(\mathbf{Y}, \mathbf{X}^{(1)}, \ldots, \mathbf{X}^{(k-1)}, \mathbf{X}^{(k)})\rangle := \mathbb{E}\int g(\mathbf{Y}, \mathbf{x}^{(1)}, \ldots, \mathbf{x}^{(k-1)}, \mathbf{x}^{(k)}) \prod_{i=1}^{k} dP(\mathbf{x}^{(i)}|\mathbf{Y}) \ ;$$

$$\mathbb{E}\langle g(\mathbf{Y}, \mathbf{X}^{(1)}, \ldots, \mathbf{X}^{(k-1)}, \mathbf{X})\rangle := \mathbb{E}\int g(\mathbf{Y}, \mathbf{x}^{(1)}, \ldots, \mathbf{x}^{(k-1)}, \mathbf{X}) \prod_{i=1}^{k-1} dP(\mathbf{x}^{(i)}|\mathbf{Y}) \ .$$

*Proof.* This is a simple consequence of Bayes' formula. It is equivalent to sample the pair $(\mathbf{X}, \mathbf{Y})$ according to its joint distribution, or to first sample $\mathbf{Y}$ according to its marginal distribution and to then sample $\mathbf{X}$ conditionally to $\mathbf{Y}$ from its conditional distribution $P(\mathbf{X} = \cdot | \mathbf{Y})$. Hence the $(k+1)$-tuple $(\mathbf{Y}, \mathbf{X}^{(1)}, \ldots, \mathbf{X}^{(k)})$ is equal in law to $(\mathbf{Y}, \mathbf{X}^{(1)}, \ldots, \mathbf{X}^{(k-1)}, \mathbf{X})$. $\qquad\square$

**Lemma 6.** *Let $X \sim P_X$ be a real random variable with finite second moment. Let $Z \sim \mathcal{N}(0,1)$ be independent of $X$. Define $I_{P_X}(r) := I(X; Y^{(r)})$ the mutual information between $X$ and $Y^{(r)} := \sqrt{r}X + Z$, and*

$$\psi_{P_X}(r) := \mathbb{E}\ln\int dP_X(x)e^{\sqrt{r}xY^{(r)} - \frac{rx^2}{2}} \ .$$

*Then, $I_{P_X}$ (resp. $\psi_{P_X}$) is twice continuously differentiable, nondecreasing, Lipschitz with Lipschitz constant $\mathbb{E}[X^2]/2$, and concave (resp. convex) on $[0, +\infty)$. Besides, if $P_X$ is not deterministic then $I_{P_X}$ (resp. $\psi_{P_X}$) is strictly concave (resp. strictly convex).*

*Proof.* The properties of the mutual information $I_{P_X}$ are well-known and proved in [8, 12]. Note that $\forall r \geq 0 : I_{P_X}(r) = r\mathbb{E}[X^2]/2 - \psi_{P_X}(r)$. The properties of $\psi_{P_X}$ follow directly from those of $I_{P_X}$ and the latter identity. $\qquad\square$

**Lemma 7.** *Let $\Delta \in (0, +\infty)$. Let $\varphi : \mathbb{R} \times \mathbb{R}^{k_A} \to \mathbb{R}$ be a bounded measurable function. Further assume that the first and second partial derivatives of $\varphi$ with respect to its first argument, denoted $\partial_x \varphi$ and $\partial_{xx}\varphi$, exist and are bounded.*
*Let $W^*, V, Z \sim \mathcal{N}(0,1)$ and $\mathbf{A} \sim P_A$ – $P_A$ is a probability distribution over $\mathbb{R}^{k_A}$ – be independent random variables. Define $I_{P_{\text{out}}}(q, \rho) := I(W^*; \widetilde{Y}^{(q,\rho)}|V)$ the conditional mutual information between $W^*$ and $\widetilde{Y}^{(q,\rho)} := \varphi(\sqrt{\rho - q}\,W^* + \sqrt{q}\,V, \mathbf{A}) + \sqrt{\Delta}\,Z$ given $V$. Then:*

- *$\forall \rho \in (0, +\infty)$ the function $q \mapsto I_{P_{\text{out}}}(q, \rho)$ is continuously twice differentiable, concave and nonincreasing on $[0, \rho]$;*

- *For all $\rho \in (0, +\infty)$, the function $q \mapsto I_{P_{\text{out}}}(q, \rho)$ is Lipschitz on $[0, \rho]$ with Lipschitz constant $C_1\big(\big\|\frac{\varphi}{\sqrt{\Delta}}\big\|_\infty, \big\|\frac{\partial_x \varphi}{\sqrt{\Delta}}\big\|_\infty\big)$ where:*

$$C_1(a, b) := (4a^2 + 1)b^2 \ .$$

- *For all $q \in [0, +\infty)$, the function $\rho \mapsto I_{P_{\text{out}}}(q, \rho)$ is Lipschitz on $[q, +\infty)$ with Lipschitz constant $C_2\big(\big\|\frac{\varphi}{\sqrt{\Delta}}\big\|_\infty, \big\|\frac{\partial_x \varphi}{\sqrt{\Delta}}\big\|_\infty, \big\|\frac{\partial_{xx}\varphi}{\sqrt{\Delta}}\big\|_\infty\big)$ where:*

$$C_2(a, b, c) := b^2(128a^4 + 12a^2 + 27) + c\big(16a^3 + 4\sqrt{2/\pi}\big) \ .$$

*Proof.* Let $P_{\text{out}}(y|x) = \int \frac{dP_A(\mathbf{a})}{\sqrt{2\pi\Delta}}e^{-\frac{1}{2\Delta}(y - \varphi(x, \mathbf{a}))^2}$. The posterior density of $W^*$ given $(V, \widetilde{Y}^{(q,\rho)})$ is

$$dP(w|V, \widetilde{Y}^{(q,\rho)}) := \frac{1}{\mathcal{Z}_{q,\rho}(V, \widetilde{Y}^{(q,\rho)})} \frac{dw}{\sqrt{2\pi}} e^{-\frac{w^2}{2}} P_{\text{out}}(\widetilde{Y}^{(q,\rho)}|\sqrt{\rho - q}\,w + \sqrt{q}\,V) \ , \qquad (88)$$

where $\mathcal{Z}(q, \rho) := \int \frac{dw}{\sqrt{2\pi}}e^{-\frac{w^2}{2}}P_{\text{out}}(\widetilde{Y}^{(q,\rho)}|\sqrt{\rho - q}\,w + \sqrt{q}\,V)$ is the normalization factor. Then:

$$I_{P_{\text{out}}}(q, \rho) = \mathbb{E}\big[\ln P_{\text{out}}(\widetilde{Y}^{(q,\rho)}|\sqrt{\rho - q}\,W^* + \sqrt{q}\,V)\big] - \mathbb{E}\ln\mathcal{Z}(q, \rho)$$

$$= \mathbb{E}\ln\mathcal{Z}(\rho, \rho) - \mathbb{E}\ln\mathcal{Z}(q, \rho) \ . \qquad (89)$$

It is shown in [7, Appendix B.2, Proposition 18] that, for all $\rho \in (0, +\infty)$, $q \mapsto \mathbb{E}\ln\mathcal{Z}(q,\rho)$ is continuously twice differentiable, convex and nondecreasing on $[0,\rho]$, i.e., $q \mapsto I_{P_{\text{out}}}(q,\rho)$ is continuously twice differentiable, concave and nonincreasing on $[0,\rho]$.

We prove the second point of the lemma by upper bounding the partial derivative of $I_{P_{\text{out}}}$ with respect to $q$. The Lipschitzianity will then follow directly from the mean-value theorem. We denote an expectation with respect to the posterior distribution (88) using the angular brackets $\langle - \rangle_{q,\rho}$, i.e., $\langle g(w) \rangle_{q,\rho} := \int g(w) dP(w|V, \widetilde{Y}^{(q,\rho)})$. Let $u_y(x) := \ln P_{\text{out}}(y|x)$. We know from [7, Appendix B.2, Proposition 18] that $\forall \rho \in (0, +\infty), \forall q \in [0,\rho]$:

$$\left.\frac{\partial I_{P_{\text{out}}}}{\partial q}\right|_{q,\rho} = -\left.\frac{\partial \mathbb{E}\ln\mathcal{Z}}{\partial q}\right|_{q,\rho} = -\frac{1}{2}\mathbb{E}\left[\left\langle u'_{\widetilde{Y}(q,\rho)}\left(\sqrt{\rho-q}\,w + \sqrt{q}\,V\right)\right\rangle^2_{q,\rho}\right]. \tag{90}$$

By Jensen's inequality and Nishimory identity, it directly follows from (90):

$$\left|\left.\frac{\partial I_{P_{\text{out}}}}{\partial q}\right|_{q,\rho}\right| \leq \frac{1}{2}\mathbb{E}\left[\left\langle u'_{\widetilde{Y}(q,\rho)}\left(\sqrt{\rho-q}\,w + \sqrt{q}\,V\right)^2\right\rangle_{q,\rho}\right] = \frac{1}{2}\mathbb{E}\left[u'_{\widetilde{Y}(q,\rho)}\left(\sqrt{\rho-q}\,W^* + \sqrt{q}\,V\right)^2\right]. \tag{91}$$

Remember that $\partial_x \varphi$, $\partial_{xx}\varphi$ denote the first and second partial derivatives of $\varphi$ with respect to its first coordinate. The infinity norms $\|\varphi\|_\infty$ and $\|\partial_x\varphi\|_\infty$ are finite by assumptions. Note that $\forall x \in \mathbb{R}$:

$$u'_y(x) = \frac{\int \frac{y - \varphi(x,\mathbf{a})}{\Delta}\partial_x\varphi(x,\mathbf{a})\frac{dP_A(\mathbf{a})}{\sqrt{2\pi\Delta}}e^{-\frac{1}{2\Delta}(y-\varphi(x,\mathbf{a}))^2}}{\int \frac{dP_A(\mathbf{a})}{\sqrt{2\pi\Delta}}e^{-\frac{1}{2\Delta}(y-\varphi(x,\mathbf{a}))^2}} \; ; \tag{92}$$

$$|u'_y(x)| \leq \frac{|y| + \|\varphi\|_\infty}{\Delta}\|\partial_x\varphi\|_\infty \tag{93}$$

Then $|u'_{\widetilde{Y}(q,\rho)}(x)| \leq \frac{2\|\varphi\|_\infty + \sqrt{\Delta}|Z|}{\Delta}\|\partial_x\varphi\|_\infty$. This upper bound combined with (91) yields:

$$\left|\left.\frac{\partial I_{P_{\text{out}}}}{\partial q}\right|_{q,\rho}\right| \leq \frac{4\|\varphi\|^2_\infty + \Delta}{\Delta^2}\|\partial_x\varphi\|^2_\infty , \tag{94}$$

which implies the second point of the lemma thanks to the mean-value theorem.

To prove the third, and last, point of the lemma we will now upper bound the partial derivative of $I_{P_{\text{out}}}$ with respect to $\rho$. Note that

$$\mathbb{E}\ln\mathcal{Z}(q,\rho) = \mathbb{E}\left[\int dy\, e^{u_y(\sqrt{\rho-q}\,W^* + \sqrt{q}\,V)}\ln\int\frac{dw}{\sqrt{2\pi}}e^{u_y(\sqrt{\rho-q}\,w + \sqrt{q}\,V) - \frac{w^2}{2}}\right].$$

Therefore:

$$\begin{aligned}
\left.\frac{\partial \mathbb{E}\ln\mathcal{Z}}{\partial\rho}\right|_{q,\rho} &= \mathbb{E}\left[\frac{W^*}{2\sqrt{\rho-q}}\int dy\,\left(u'_y(x)e^{u_y(x)}\right)\Big|_{x=\sqrt{\rho-q}\,W^* + \sqrt{q}\,V}\ln\int\frac{dw}{\sqrt{2\pi}}e^{u_y(\sqrt{\rho-q}\,w + \sqrt{q}\,V) - \frac{w^2}{2}}\right] \\
&\quad + \mathbb{E}\left[\left\langle\frac{w}{2\sqrt{\rho-q}}u'_{\widetilde{Y}(q,\rho)}\left(\sqrt{\rho-q}\,w + \sqrt{q}\,V\right)\right\rangle_{q,\rho}\right] \\
&= \mathbb{E}\left[\frac{W^*}{2\sqrt{\rho-q}}\int dy\,\left(u'_y(x)e^{u_y(x)}\right)\Big|_{x=\sqrt{\rho-q}\,W^* + \sqrt{q}\,V}\ln\int\frac{dw}{\sqrt{2\pi}}e^{u_y(\sqrt{\rho-q}\,w + \sqrt{q}\,V) - \frac{w^2}{2}}\right] \\
&\quad + \mathbb{E}\left[\frac{W^*}{2\sqrt{\rho-q}}u'_{\widetilde{Y}(q,\rho)}\left(\sqrt{\rho-q}\,W^* + \sqrt{q}\,V\right)\right] \\
&= \frac{1}{2}\mathbb{E}\left[\left(u''_{\widetilde{Y}(q,\rho)}(x) + u'_{\widetilde{Y}(q,\rho)}(x)^2\right)\Big|_{x=\sqrt{\rho-q}\,W^* + \sqrt{q}\,V}\ln\mathcal{Z}(q,\rho)\right] \\
&\quad + \frac{1}{2}\mathbb{E}\left[u''_{\widetilde{Y}(q,\rho)}\left(\sqrt{\rho-q}\,W^* + \sqrt{q}\,V\right)\right] \\
&= \frac{1}{2}\mathbb{E}\left[\left(u''_{\widetilde{Y}(q,\rho)}(x) + u'_{\widetilde{Y}(q,\rho)}(x)^2\right)\Big|_{x=\sqrt{\rho-q}\,W^* + \sqrt{q}\,V}\left(\ln\mathcal{Z}(q,\rho) + 1\right)\right] \\
&\quad - \frac{1}{2}\mathbb{E}\left[u'_{\widetilde{Y}(q,\rho)}\left(\sqrt{\rho-q}\,W^* + \sqrt{q}\,V\right)^2\right]. \tag{95}
\end{aligned}$$

The second equality follows from Nishimori identity and the third one from integrating by parts with respect to $W^*$. We now define $\forall \rho \in [0, +\infty) : h(\rho) := \mathbb{E}\ln \mathcal{Z}(\rho, \rho) = \mathbb{E}[\int dy \, e^{u_y(\sqrt{\rho}\,V)} u_y(\sqrt{\rho}\,V)]$. We have:

$$h'(\rho) = \mathbb{E}\left[\frac{V}{2\sqrt{\rho}} \int dy \, e^{u_y(\sqrt{\rho}\,V)} \left(u_y(\sqrt{\rho}\,V) + 1\right) u_y'(\sqrt{\rho}\,V)\right]$$

$$= \frac{1}{2}\mathbb{E}\left[\int dy \, e^{u_y(\sqrt{\rho}\,V)} \left(u_y''(\sqrt{\rho}\,V) + u_y'(\sqrt{\rho}\,V)^2\right)\left(u_y(\sqrt{\rho}\,V) + 1\right)\right]$$

$$\qquad + \frac{1}{2}\mathbb{E}\left[\int dy \, e^{u_y(\sqrt{\rho}\,V)} u_y'(\sqrt{\rho}\,V)^2\right]$$

$$= \frac{1}{2}\mathbb{E}\left[\left(u_{\widetilde{Y}(\rho,\rho)}''(x) + u_{\widetilde{Y}(\rho,\rho)}'(x)^2\right)\Big|_{x=\sqrt{\rho}\,V}\left(\ln \mathcal{Z}(\rho,\rho) + 1\right)\right] + \frac{1}{2}\mathbb{E}\left[u_{\widetilde{Y}(\rho,\rho)}'(\sqrt{\rho}\,V)^2\right]. \qquad (96)$$

Combining (89), (95) and (96) yields

$$\frac{\partial I_{P_{\mathrm{out}}}}{\partial \rho}\bigg|_{q,\rho} = \frac{1}{2}\mathbb{E}\left[\left(u_{\widetilde{Y}(\rho,\rho)}''(x) + u_{\widetilde{Y}(\rho,\rho)}'(x)^2\right)\Big|_{x=\sqrt{\rho}\,V}\left(\ln \mathcal{Z}(\rho,\rho) + 1\right)\right]$$

$$\qquad - \frac{1}{2}\mathbb{E}\left[\left(u_{\widetilde{Y}(q,\rho)}''(x) + u_{\widetilde{Y}(q,\rho)}'(x)^2\right)\Big|_{x=\sqrt{\rho-q}\,W^* + \sqrt{q}\,V}\left(\ln \mathcal{Z}(q,\rho) + 1\right)\right]$$

$$\qquad + \frac{1}{2}\mathbb{E}\left[u_{\widetilde{Y}(q,\rho)}'(\sqrt{\rho}\,V)^2\right] + \frac{1}{2}\mathbb{E}\left[u_{\widetilde{Y}(\rho,\rho)}'(\sqrt{\rho-q}\,W^* + \sqrt{q}\,V)^2\right]. \qquad (97)$$

The last two summands on the right-hand side of (97) are upper bounded by $\frac{4\|\varphi\|_\infty^2 + \Delta}{\Delta^2}\|\partial_x\varphi\|_\infty^2$ (see the proof of the second point of the lemma). The first two summands on the right-hand side of (97) involve the function $(x,y) \mapsto u_y''(x) + u_y'(x)^2$. We have:

$$u_y''(x) + u_y'(x)^2 = \frac{\int \frac{(y-\varphi(x,\mathbf{a}))^2\partial_x\varphi(x,\mathbf{a})^2 - \Delta\partial_x\varphi(x,\mathbf{a})^2 + \Delta\partial_{xx}\varphi(x,\mathbf{a})(y-\varphi(x,\mathbf{a}))}{\Delta^2}\frac{dP_A(\mathbf{a})}{\sqrt{2\pi\Delta}}e^{-\frac{1}{2\Delta}(y-\varphi(x,\mathbf{a}))^2}}{\int \frac{dP_A(\mathbf{a})}{\sqrt{2\pi\Delta}}e^{-\frac{1}{2\Delta}(y-\varphi(x,\mathbf{a}))^2}}.$$

$$(98)$$

Then, by a direct computation, we obtain:

$$\int_{-\infty}^{+\infty}(u_y''(x) + u_y'(x)^2)e^{u_y(x)}dy$$

$$= \int dP_A(\mathbf{a})\int_{-\infty}^{+\infty}\frac{\left((y-\varphi(x,\mathbf{a}))^2 - \Delta\right)\partial_x\varphi(x,\mathbf{a})^2 + \Delta\partial_{xx}\varphi(x,\mathbf{a})(y-\varphi(x,\mathbf{a}))}{\Delta^2}\frac{e^{-\frac{(y-\varphi(x,\mathbf{a}))^2}{2\Delta}}}{\sqrt{2\pi\Delta}}dy$$

$$= \int dP_A(\mathbf{a})\int_{-\infty}^{+\infty}\frac{(\widetilde{y}^2 - 1)\partial_x\varphi(x,\mathbf{a})^2 + \sqrt{\Delta}\partial_{xx}\varphi(x,\mathbf{a})\widetilde{y}}{\Delta}\frac{e^{-\frac{\widetilde{y}^2}{2}}d\widetilde{y}}{\sqrt{2\pi}}$$

$$= 0. \qquad (99)$$

Therefore:

$$\mathbb{E}\left[\left(u_{\widetilde{Y}(q,\rho)}''(x) + u_{\widetilde{Y}(q,\rho)}'(x)^2\right)\Big|_{x=\sqrt{\rho-q}\,W^* + \sqrt{q}\,V}\right]$$

$$= \mathbb{E}\left[\left(\int_{-\infty}^{+\infty}(u_y''(x) + u_y'(x)^2)e^{u_y(x)}dy\right)\Big|_{x=\sqrt{\rho-q}\,W^* + \sqrt{q}\,V}\right] = 0.$$

This directly implies:

$$\mathbb{E}\left[\left(u_{\widetilde{Y}(q,\rho)}''(x) + u_{\widetilde{Y}(q,\rho)}'(x)^2\right)\Big|_{x=\sqrt{\rho-q}\,W^* + \sqrt{q}\,V}\left(\ln \mathcal{Z}(q,\rho) + 1\right)\right]$$

$$= \mathbb{E}\left[\left(u_{\widetilde{Y}(q,\rho)}''(x) + u_{\widetilde{Y}(q,\rho)}'(x)^2\right)\Big|_{x=\sqrt{\rho-q}\,W^* + \sqrt{q}\,V}\left(\ln \mathcal{Z}(q,\rho) + \frac{\ln(2\pi\Delta)}{2}\right)\right]. \qquad (100)$$

We use the formula (98) for $u_y''(x) + u_y'(x)^2$ to get the upper bound:

$$\left|u_{\widetilde{Y}(q,\rho)}''(x) + u_{\widetilde{Y}(q,\rho)}'(x)^2\right| \leq \frac{\left((2\|\varphi\|_\infty + \sqrt{\Delta}|Z|)^2 + \Delta\right)\|\partial_x\varphi\|_\infty^2 + \Delta\|\partial_{xx}\varphi\|_\infty(2\|\varphi\|_\infty + \sqrt{\Delta}|Z|)}{\Delta^2}.$$

$$(101)$$

Trivially, $P_{\mathrm{out}}(y|x) \leq 1/\sqrt{2\pi\Delta}$. This implies

$$\ln \mathcal{Z}(q,\rho) = \ln \int \frac{dw}{\sqrt{2\pi}}e^{-\frac{w^2}{2}}P_{\mathrm{out}}\left(\widetilde{Y}^{(q,\rho)}|\sqrt{\rho-q}\,w + \sqrt{q}\,V\right) \leq -\frac{\ln(2\pi\Delta)}{2},$$

while, by Jensen's inequality, we have

$$\ln \mathcal{Z}(q, \rho) = \ln \int \frac{dw}{\sqrt{2\pi}} e^{-\frac{w^2}{2}} dP_A(\mathbf{a}) \frac{1}{\sqrt{2\pi\Delta}} e^{-\frac{1}{2\Delta}(\widetilde{Y}^{(q,\rho)} - \varphi(x,\mathbf{a}))^2}$$

$$\geq \int \frac{dw}{\sqrt{2\pi}} e^{-\frac{w^2}{2}} dP_A(\mathbf{a}) \left( -\frac{\ln(2\pi\Delta)}{2} - \frac{(\widetilde{Y}^{(q,\rho)} - \varphi(x,\mathbf{a}))^2}{2\Delta} \right)$$

$$\geq -\frac{\ln(2\pi\Delta)}{2} - \frac{(2\|\varphi\|_\infty + \sqrt{\Delta}|Z|)^2}{2\Delta} .$$

Hence

$$\left| \ln \mathcal{Z}(q, \rho) + \frac{\ln(2\pi\Delta)}{2} \right| \leq \frac{(2\|\varphi\|_\infty + \sqrt{\Delta}|Z|)^2}{2\Delta} . \tag{102}$$

Combining (100), (101), (102) yields the following upper bound of the second term on the right-hand side of (97):

$$\left| \frac{1}{2} \mathbb{E}\left[ \left( u''_{\widetilde{Y}(q,\rho)}(x) + u'_{\widetilde{Y}(q,\rho)}(x)^2 \right) \Big|_{x = \sqrt{\rho-q}\, W^* + \sqrt{q}\, V} (\ln \mathcal{Z}(q, \rho) + 1) \right] \right|$$

$$\leq C\left( \left\| \frac{\varphi}{\sqrt{\Delta}} \right\|_\infty, \left\| \frac{\partial_x \varphi}{\sqrt{\Delta}} \right\|_\infty, \left\| \frac{\partial_{xx} \varphi}{\sqrt{\Delta}} \right\|_\infty \right) , \tag{103}$$

where $C(a, b, c) := b^2 (64a^4 + 6a^2 + 13.5) + c(8a^3 + 2\sqrt{\frac{2}{\pi}})$. This upper bound holds for all $q \in [0, \rho]$. In particular, it holds for the first term on the right-hand side of (97) where $q = \rho$. We now have an upper bound for each summand on the right-hand side of (97) and we can combine them to get:

$$\frac{\partial I_{P_{\text{out}}}}{\partial \rho}\Big|_{q,\rho} \leq 2C\left( \left\| \frac{\varphi}{\sqrt{\Delta}} \right\|_\infty, \left\| \frac{\partial_x \varphi}{\sqrt{\Delta}} \right\|_\infty, \left\| \frac{\partial_{xx} \varphi}{\sqrt{\Delta}} \right\|_\infty \right) + 2\left( 4\left\| \frac{\varphi}{\sqrt{\Delta}} \right\|_\infty^2 + 1 \right) \left\| \frac{\partial_x \varphi}{\sqrt{\Delta}} \right\|_\infty^2 .$$

We can conclude the proof of the third point of the lemma using this last upper bound and the mean-value theorem. $\square$

# B    Properties of the interpolating mutual information

We recall that $u_y(x) := \ln P_{\text{out}}(y|x)$, and that $u'_y(\cdot)$ and $u''_y(\cdot)$ are the first and second derivatives of $u_y(\cdot)$. We denote $P'_{\text{out}}(y|x)$ and $P''_{\text{out}}(y|x)$ the first and second derivatives of $x \mapsto P_{\text{out}}(y|x)$. Finally, the scalar overlap is $Q := \frac{1}{k_n} \sum_{i=1}^n X_i^* x_i$.

## B.1    Derivative of the interpolating mutual information

**Proposition 1 (extended).** *Suppose that $\Delta > 0$ and that all of (H1), (H2) and (H3) hold. Further assume that $\mathbb{E}_{X \sim P_0}[X^2] = 1$. The derivative of the interpolating mutual information (14) with respect to $t$ satisfies for all $(t, \epsilon) \in [0, 1] \times \mathcal{B}_n$:*

$$i'_{n,\epsilon}(t) = \mathcal{O}\left( \frac{1}{\sqrt{n\rho_n}} \right) + \mathcal{O}\left( \sqrt{\frac{\alpha_n}{\rho_n} \mathbb{V}\text{ar} \frac{\ln \mathcal{Z}_{t,\epsilon}}{m_n}} \right) + \frac{\rho_n}{2\alpha_n} r_\epsilon(t)(1 - q_\epsilon(t))$$

$$+ \frac{1}{2} \mathbb{E}\left\langle (Q - q_\epsilon(t)) \left( \frac{1}{m_n} \sum_{\mu=1}^{m_n} u'_{Y_\mu^{(t,\epsilon)}}(S_\mu^{(t,\epsilon)}) u'_{Y_\mu^{(t,\epsilon)}}(s_\mu^{(t,\epsilon)}) - \frac{\rho_n}{\alpha_n} r_\epsilon(t) \right) \right\rangle_{n,t,\epsilon} , \tag{104}$$

*where*

$$\left| \mathcal{O}\left( \frac{1}{\sqrt{n\rho_n}} \right) \right| \leq \frac{S^2 C}{\sqrt{n\rho_n}} \quad and \quad \left| \mathcal{O}\left( \sqrt{\frac{\alpha_n}{\rho_n} \mathbb{V}\text{ar} \frac{\ln \mathcal{Z}_{t,\epsilon}}{m_n}} \right) \right| \leq S^2 \sqrt{D \frac{\alpha_n}{\rho_n} \mathbb{V}\text{ar} \frac{\ln \mathcal{Z}_{t,\epsilon}}{m_n}} ;$$

*with ($\partial_x \varphi$ and $\partial_{xx} \varphi$ denote the first and second partial derivatives of $\varphi$ with respect to its first argument):*

$$C := \left\| \frac{\partial_x \varphi}{\sqrt{\Delta}} \right\|_\infty^2 \left( 64 \left\| \frac{\varphi}{\sqrt{\Delta}} \right\|_\infty^4 + 2 \left\| \frac{\varphi}{\sqrt{\Delta}} \right\|_\infty^2 + 12.5 \right) + \left\| \frac{\partial_{xx} \varphi}{\sqrt{\Delta}} \right\|_\infty \left( 8 \left\| \frac{\varphi}{\sqrt{\Delta}} \right\|_\infty^3 + 2\sqrt{\frac{2}{\pi}} \right) ;$$

$$D := \left\| \frac{\partial_x \varphi}{\sqrt{\Delta}} \right\|_\infty^4 + \frac{1}{2} \left\| \frac{\partial_{xx} \varphi}{\sqrt{\Delta}} \right\|_\infty^2 .$$

*In addition, if both sequences $(\alpha_n)_n$ and $(\rho_n/\alpha_n)_n$ are bounded, i.e., if there exist real positive numbers $M_\alpha, M_{\rho/\alpha}$ such that $\forall n \in \mathbb{N}^*: \alpha_n \le M_\alpha, \rho_n/\alpha_n \le M_{\rho/\alpha}$ then for all $(t, \epsilon) \in [0,1] \times \mathcal{B}_n$:*

$$i'_{n,\epsilon}(t) = \mathcal{O}\left(\frac{1}{\sqrt{n}\,\rho_n}\right) + \frac{\rho_n}{2\alpha_n} r_\epsilon(t)(1 - q_\epsilon(t))$$

$$+ \frac{1}{2}\mathbb{E}\left\langle \left(Q - q_\epsilon(t)\right)\left(\frac{1}{m_n}\sum_{\mu=1}^{m_n} u'_{Y_\mu^{(t,\epsilon)}}(S_\mu^{(t,\epsilon)}) u'_{Y_\mu^{(t,\epsilon)}}(s_\mu^{(t,\epsilon)}) - \frac{\rho_n}{\alpha_n} r_\epsilon(t)\right)\right\rangle_{n,t,\epsilon} , \quad (105)$$

*where*

$$\left|\mathcal{O}\left(\frac{1}{\sqrt{n}\,\rho_n}\right)\right| \le \frac{S^2 C + S^2\sqrt{D\big(\widetilde{C}_1 + M_{\rho/\alpha}\widetilde{C}_2 + M_\alpha \widetilde{C}_3\big)}}{\sqrt{n}\,\rho_n} .$$

*Here $\widetilde{C}_1, \widetilde{C}_2, \widetilde{C}_3$ are the polynomials in $\big(S, \big\|\frac{\varphi}{\sqrt{\Delta}}\big\|_\infty, \big\|\frac{\partial_x\varphi}{\sqrt{\Delta}}\big\|_\infty, \big\|\frac{\partial_{xx}\varphi}{\sqrt{\Delta}}\big\|_\infty\big)$ defined in Proposition 9.*

*Proof.* We recall that $\mathcal{Z}_{t,\epsilon}$ is the normalization to the joint posterior density of $(\mathbf{X}^*, \mathbf{W}^*)$ given $(\mathbf{Y}^{(t,\epsilon)}, \widetilde{\mathbf{Y}}^{(t,\epsilon)}, \mathbf{\Phi}, \mathbf{V})$. We define the average interpolating free entropy $f_{n,\epsilon}(t) := \mathbb{E}\ln\mathcal{Z}_{t,\epsilon}/m_n$. Note that $i_{n,\epsilon}(t) := I((\mathbf{X}^*, \mathbf{W}^*); (\mathbf{Y}^{(t,\epsilon)}, \widetilde{\mathbf{Y}}^{(t,\epsilon)})|\mathbf{\Phi}, \mathbf{V})/m_n$ satisfies:

$$i_{n,\epsilon}(t) = -\frac{\mathbb{E}\ln\mathcal{Z}_{t,\epsilon}}{m_n} + \frac{1}{m_n}\mathbb{E}\big[\ln\big(e^{-\frac{\|\tilde{\mathbf{z}}\|^2}{2}} P_{\text{out}}(Y_\mu^{(t,\epsilon)}|S_\mu^{(t,\epsilon)})\big)\big]$$

$$= -f_{n,\epsilon}(t) - \frac{1}{2\alpha_n} + \mathbb{E}\big[\ln P_{\text{out}}(Y_1^{(t,\epsilon)}|S_1^{(t,\epsilon)})\big]$$

Given $\mathbf{X}^*$, $S_1^{(t,\epsilon)} \sim \mathcal{N}(0, V^{(t)})$ where $\rho^{(t)} := \frac{1-t}{k_n}\|\mathbf{X}^*\|^2 + t + 2s_n$. Then:

$$\mathbb{E}\ln P_{\text{out}}(Y_1^{(t,\epsilon)}|S_1^{(t,\epsilon)}) = \mathbb{E}\big[\mathbb{E}[\ln P_{\text{out}}(Y_1^{(t,\epsilon)}|S_1^{(t,\epsilon)})|\mathbf{X}^*]\big] = \mathbb{E}[h(\rho^{(t)})] ,$$

where $h : \rho \in [0, +\infty) \mapsto \mathbb{E}_{V\sim\mathcal{N}(0,1)}\int u_y(\sqrt{\rho}\,V)e^{u_y(\sqrt{\rho}\,V)}dy$. All in all, we have:

$$i_{n,\epsilon}(t) = \mathbb{E}[h(\rho^{(t)})] - f_{n,\epsilon}(t) - \frac{1}{2\alpha_n} . \quad (106)$$

We directly obtain for the derivative of $i_{n,\epsilon}(\cdot)$:

$$i'_{n,\epsilon}(t) = -\mathbb{E}\left[h'(\rho^{(t)})\left(\frac{\|\mathbf{X}^*\|^2}{k_n} - 1\right)\right] - f'_{n,\epsilon}(t) , \quad (107)$$

where $h', f'_{n,\epsilon}$ are the derivatives of $h, f_{n,\epsilon}$. In Lemma 7 of Appendix A, we compute $h'$ and show:

$$\forall \rho \in [0, +\infty) : |h'(\rho)| \le C := C\left(\left\|\frac{\varphi}{\sqrt{\Delta}}\right\|_\infty, \left\|\frac{\partial_x\varphi}{\sqrt{\Delta}}\right\|_\infty, \left\|\frac{\partial_{xx}\varphi}{\sqrt{\Delta}}\right\|_\infty\right)$$

with $C(a, b, c) := b^2(64a^4 + 2a^2 + 12.5) + c\big(8a^3 + 2\sqrt{\frac{2}{\pi}}\big)$. The first term on the right-hand side of (107) thus satisfies:

$$\left|\mathbb{E}\left[h'(\rho^{(t)})\left(\frac{\|\mathbf{X}^*\|^2}{k_n} - 1\right)\right]\right| \le C\sqrt{\mathbb{V}\text{ar}\left(\frac{\|\mathbf{X}^*\|^2}{k_n}\right)} = \frac{C}{k_n}\sqrt{n\mathbb{V}\text{ar}((X_1^*)^2)} = \frac{CS^2}{\sqrt{n}\rho_n} . \quad (108)$$

We now turn to the computation of $f'_{n,\epsilon}$.

**Derivative of the average interpolating free entropy**    Note that

$$f_{n,\epsilon}(t) = \frac{1}{m_n}\mathbb{E}\left[\int \frac{d\mathbf{y}\,d\widetilde{\mathbf{y}}}{\sqrt{2\pi}^n} e^{-\mathcal{H}_{t,\epsilon}(\mathbf{X}^*, \mathbf{W}^*; \mathbf{y}, \widetilde{\mathbf{y}}, \mathbf{\Phi}, \mathbf{V})}\ln\int dP_{0,n}(\mathbf{x})\mathcal{D}\mathbf{w}\, e^{-\mathcal{H}_{t,\epsilon}(\mathbf{x}, \mathbf{w}; \mathbf{y}, \widetilde{\mathbf{y}}, \mathbf{\Phi}, \mathbf{V})}\right] \quad (109)$$

where the expectation is over $\mathbf{X}^*, \mathbf{\Phi}, \mathbf{V}, \mathbf{W}^*$, $\mathcal{D}\mathbf{w} := \frac{d\mathbf{w}\,e^{-\frac{\|\mathbf{w}\|^2}{2}}}{\sqrt{2\pi}^{m_n}}$ and the Hamiltonian $\mathcal{H}_{t,\epsilon}$ is:

$$\mathcal{H}_{t,\epsilon}(\mathbf{x}, \mathbf{w}; \mathbf{y}, \widetilde{\mathbf{y}}, \mathbf{\Phi}, \mathbf{V}) := -\sum_{\mu=1}^{m_n}\ln P_{\text{out}}(y_\mu|s_\mu^{(t,\epsilon)}) + \frac{1}{2}\sum_{i=1}^{n}\big(\widetilde{y}_i - \sqrt{R_1(t,\epsilon)}\,x_i\big)^2 . \quad (110)$$

We will need its derivative $\mathcal{H}'_{t,\epsilon}$ with respect to $t$:

$$\mathcal{H}'_{t,\epsilon}(\mathbf{x}, \mathbf{w}; \mathbf{y}, \widetilde{\mathbf{y}}, \mathbf{\Phi}, \mathbf{V}) := -\sum_{\mu=1}^{m_n}\frac{\partial s_\mu^{(t,\epsilon)}}{\partial t} u'_{y_\mu}(s_\mu^{(t,\epsilon)}) - \frac{r_\epsilon(t)}{2\sqrt{R_1(t,\epsilon)}}\sum_{i=1}^{n} x_i(\widetilde{y}_i - \sqrt{R_1(t,\epsilon)}x_i) . \quad (111)$$

The derivative of $f_{n,\epsilon}$ can be obtained by differentiating (109) under the expectation:

$$f'_{n,\epsilon}(t) = -\frac{1}{m_n}\mathbb{E}\big[\mathcal{H}'_{t,\epsilon}(\mathbf{X}^*,\mathbf{W}^*;\mathbf{Y}^{(t,\epsilon)},\widetilde{\mathbf{Y}}^{(t,\epsilon)},\mathbf{\Phi},\mathbf{V})\ln\mathcal{Z}_{t,\epsilon}\big]$$
$$-\frac{1}{m_n}\mathbb{E}\big\langle\mathcal{H}'_{t,\epsilon}(\mathbf{x},\mathbf{w};\mathbf{Y}^{(t,\epsilon)},\widetilde{\mathbf{Y}}^{(t,\epsilon)},\mathbf{\Phi},\mathbf{V})\big\rangle_{n,t,\epsilon}$$
$$=-\frac{1}{m_n}\mathbb{E}\big[\mathcal{H}'_{t,\epsilon}(\mathbf{X}^*,\mathbf{W}^*;\mathbf{Y}^{(t,\epsilon)},\widetilde{\mathbf{Y}}^{(t,\epsilon)},\mathbf{\Phi},\mathbf{V})\ln\mathcal{Z}_{t,\epsilon}\big]$$
$$-\frac{1}{m_n}\mathbb{E}[\mathcal{H}'_{t,\epsilon}(\mathbf{X}^*,\mathbf{W}^*;\mathbf{Y}^{(t,\epsilon)},\widetilde{\mathbf{Y}}^{(t,\epsilon)},\mathbf{\Phi},\mathbf{V})]\,. \quad (112)$$

The last equality follows from the Nishimory identity

$$\mathbb{E}\,\langle\mathcal{H}'_{t,\epsilon}(\mathbf{x},\mathbf{w};\mathbf{Y}^{(t,\epsilon)},\widetilde{\mathbf{Y}}^{(t,\epsilon)},\mathbf{\Phi},\mathbf{V})\rangle_{n,t,\epsilon}=\mathbb{E}[\mathcal{H}'_{t,\epsilon}(\mathbf{X}^*,\mathbf{W}^*;\mathbf{Y}^{(t,\epsilon)},\widetilde{\mathbf{Y}}^{(t,\epsilon)},\mathbf{\Phi},\mathbf{V})]\,.$$

Evaluating (111) at $(\mathbf{x},\mathbf{w};\mathbf{y},\widetilde{\mathbf{y}},\mathbf{\Phi},\mathbf{V})=(\mathbf{X}^*,\mathbf{W}^*;\mathbf{Y}^{(t,\epsilon)},\widetilde{\mathbf{Y}}^{(t,\epsilon)},\mathbf{\Phi},\mathbf{V})$ yields:

$$\mathcal{H}'_{t,\epsilon}(\mathbf{X}^*,\mathbf{W}^*;\mathbf{Y}^{(t,\epsilon)},\widetilde{\mathbf{Y}}^{(t,\epsilon)},\mathbf{\Phi},\mathbf{V})=-\sum_{\mu=1}^{m_n}\frac{\partial S_\mu^{(t,\epsilon)}}{\partial t}u'_{Y_\mu^{(t,\epsilon)}}(S_\mu^{(t,\epsilon)})-\frac{r_\epsilon(t)}{2\sqrt{R_1(t,\epsilon)}}\sum_{i=1}^n X_i^*\widetilde{Z}_i\,. \quad (113)$$

The expectation of (113) is zero:

$$\mathbb{E}\,\mathcal{H}'_{t,\epsilon}(\mathbf{X}^*,\mathbf{W}^*;\mathbf{Y}^{(t,\epsilon)},\widetilde{\mathbf{Y}}^{(t,\epsilon)},\mathbf{\Phi},\mathbf{V})=-\sum_{\mu=1}^{m_n}\mathbb{E}\bigg[\frac{\partial S_\mu^{(t,\epsilon)}}{\partial t}u'_{Y_\mu^{(t,\epsilon)}}(S_\mu^{(t,\epsilon)})\bigg]$$
$$=-\sum_{\mu=1}^{m_n}\mathbb{E}\bigg[\frac{\partial S_\mu^{(t,\epsilon)}}{\partial t}\mathbb{E}\Big[u'_{Y_\mu^{(t,\epsilon)}}(S_\mu^{(t,\epsilon)})\Big|\mathbf{X}^*,\mathbf{W}^*,\mathbf{V},\mathbf{\Phi}\Big]\bigg]$$
$$=-\sum_{\mu=1}^{m_n}\mathbb{E}\bigg[\frac{\partial S_\mu^{(t,\epsilon)}}{\partial t}\int u'_y(S_\mu^{(t,\epsilon)})P_{\text{out}}(y\,|\,S_\mu^{(t,\epsilon)})dy\bigg]$$
$$=-\sum_{\mu=1}^{m_n}\mathbb{E}\bigg[\frac{\partial S_\mu^{(t,\epsilon)}}{\partial t}\int P'_{\text{out}}(y\,|\,S_\mu^{(t,\epsilon)})dy\bigg]$$
$$=0\,.$$

The last equality is because for all $x$:

$$\int P'_{\text{out}}(y\,|\,x)dy=\int dP_A(\mathbf{a})\partial_x\varphi(x,\mathbf{a})\int\frac{y-\varphi(x,\mathbf{a})}{\Delta}\frac{e^{-\frac{(y-\varphi(x,\mathbf{a}))^2}{2\Delta}}}{\sqrt{2\pi\Delta}}dy=0\,.$$

The expectation of (113) being zero, the identity (112) reads:

$$f'_{n,\epsilon}(t)=\frac{1}{m_n}\sum_{\mu=1}^{m_n}\mathbb{E}\bigg[\frac{\partial S_\mu^{(t,\epsilon)}}{\partial t}u'_{Y_\mu^{(t,\epsilon)}}(S_\mu^{(t,\epsilon)})\ln\mathcal{Z}_{t,\epsilon}\bigg]+\frac{1}{m_n}\frac{r_\epsilon(t)}{2\sqrt{R_1(t,\epsilon)}}\sum_{i=1}^n\mathbb{E}\big[X_i^*\widetilde{Z}_i\ln\mathcal{Z}_{t,\epsilon}\big]\,. \quad (114)$$

First, we compute the first kind of expectation on the right-hand side of (114). $\forall\mu\in\{1,\dots,m_n\}$:

$$\mathbb{E}\bigg[\frac{\partial S_\mu^{(t,\epsilon)}}{\partial t}u'_{Y_\mu^{(t,\epsilon)}}(S_\mu^{(t,\epsilon)})\ln\mathcal{Z}_{t,\epsilon}\bigg]$$
$$=\frac{1}{2}\mathbb{E}\bigg[\bigg(-\frac{(\mathbf{\Phi}\mathbf{X}^*)_\mu}{\sqrt{k_n(1-t)}}+\frac{q_\epsilon(t)V_\mu}{\sqrt{R_2(t,\epsilon)}}+\frac{(1-q_\epsilon(t))W_\mu^*}{\sqrt{t+2s_n-R_2(t,\epsilon)}}\bigg)u'_{Y_\mu^{(t,\epsilon)}}(S_\mu^{(t,\epsilon)})\ln\mathcal{Z}_{t,\epsilon}\bigg]\,. \quad (115)$$

An integration by parts w.r.t. the independent standard Gaussians $(\Phi_{\mu i})_{i=1}^n$ yields:

$$\mathbb{E}\bigg[\frac{(\mathbf{\Phi}\mathbf{X}^*)_\mu}{\sqrt{k_n(1-t)}}u'_{Y_\mu^{(t,\epsilon)}}(S_\mu^{(t,\epsilon)})\ln\mathcal{Z}_{t,\epsilon}\bigg]$$
$$=\sum_{i=1}^n\mathbb{E}\bigg[\frac{\Phi_{\mu i}X_i^*}{\sqrt{k_n(1-t)}}\int d\mathbf{y}d\widetilde{\mathbf{y}}\,u'_{y_\mu}(S_\mu^{(t,\epsilon)})e^{-\mathcal{H}_{t,\epsilon}(\mathbf{X}^*,\mathbf{W}^*;\mathbf{y},\widetilde{\mathbf{y}},\mathbf{\Phi},\mathbf{V})}\ln\int dP_{0,n}(\mathbf{x})\mathcal{D}\mathbf{w}\,e^{-\mathcal{H}_{t,\epsilon}(\mathbf{x},\mathbf{w};\mathbf{y},\widetilde{\mathbf{y}},\mathbf{\Phi},\mathbf{V})}\bigg]$$
$$=\sum_{i=1}^n\mathbb{E}\bigg[\frac{(X_i^*)^2}{k_n}\big(u''_{Y_\mu^{(t,\epsilon)}}(S_\mu^{(t,\epsilon)})+u'_{Y_\mu^{(t,\epsilon)}}(S_\mu^{(t,\epsilon)})^2\big)\ln\mathcal{Z}_{t,\epsilon}+\frac{X_i^*u'_{Y_\mu^{(t,\epsilon)}}(S_\mu^{(t,\epsilon)})}{k_n}\big\langle x_iu'_{Y_\mu^{(t,\epsilon)}}(s_\mu^{(t,\epsilon)})\big\rangle_{n,t,\epsilon}\bigg]$$
$$=\mathbb{E}\bigg[\frac{\|\mathbf{X}^*\|^2}{k_n}\frac{P''_{\text{out}}(Y_\mu^{(t,\epsilon)}|S_\mu^{(t,\epsilon)})}{P_{\text{out}}(Y_\mu^{(t,\epsilon)}|S_\mu^{(t,\epsilon)})}\ln\mathcal{Z}_{t,\epsilon}\bigg]+\mathbb{E}\bigg\langle Q\,u'_{Y_\mu^{(t,\epsilon)}}(S_\mu^{(t,\epsilon)})u'_{Y_\mu^{(t,\epsilon)}}(s_\mu^{(t,\epsilon)})\bigg\rangle_{n,t,\epsilon}\,, \quad (116)$$

where, in the last equality, we used the identity $u_y''(x) + u_y'(x)^2 = \frac{P_{\text{out}}''(y|x)}{P_{\text{out}}(y|x)}$. Another Gaussian integration by parts, this time with respect to $V_\mu \sim \mathcal{N}(0,1)$, gives:

$$
\mathbb{E}\left[\frac{q_\epsilon(t)V_\mu}{\sqrt{R_2(t,\epsilon)}} u_{Y_\mu^{(t,\epsilon)}}'(S_\mu^{(t,\epsilon)}) \ln \mathcal{Z}_{t,\epsilon}\right]
$$

$$
= \mathbb{E}\left[\frac{q_\epsilon(t)V_\mu}{\sqrt{R_2(t,\epsilon)}} \int d\mathbf{y}\, d\widetilde{\mathbf{y}}\, u_{y_\mu}'(S_\mu^{(t,\epsilon)}) e^{-\mathcal{H}_{t,\epsilon}(\mathbf{X}^*,\mathbf{W}^*;\mathbf{y},\widetilde{\mathbf{y}},\mathbf{\Phi},\mathbf{V})} \ln \int dP_{0,n}(\mathbf{x})\mathcal{D}\mathbf{w}\, e^{-\mathcal{H}_{t,\epsilon}(\mathbf{x},\mathbf{w};\mathbf{y},\widetilde{\mathbf{y}},\mathbf{\Phi},\mathbf{V})}\right]
$$

$$
= \mathbb{E}\left[q_\epsilon(t)\big(u_{Y_\mu^{(t,\epsilon)}}''(S_\mu^{(t,\epsilon)}) + u_{Y_\mu^{(t,\epsilon)}}'(S_\mu^{(t,\epsilon)})^2\big) \ln \mathcal{Z}_{t,\epsilon} + q_\epsilon(t)u_{Y_\mu^{(t,\epsilon)}}'(S_\mu^{(t,\epsilon)})\langle u_{Y_\mu^{(t,\epsilon)}}'(s_\mu^{(t,\epsilon)})\rangle_{n,t,\epsilon}\right]
$$

$$
= \mathbb{E}\left[q_\epsilon(t)\frac{P_{\text{out}}''(Y_\mu^{(t,\epsilon)}|S_\mu^{(t,\epsilon)})}{P_{\text{out}}(Y_\mu^{(t,\epsilon)}|S_\mu^{(t,\epsilon)})} \ln \mathcal{Z}_{t,\epsilon}\right] + \mathbb{E}\left\langle q_\epsilon(t)u_{Y_\mu^{(t,\epsilon)}}'(S_\mu^{(t,\epsilon)})u_{Y_\mu^{(t,\epsilon)}}'(s_\mu^{(t,\epsilon)})\right\rangle_{n,t,\epsilon}, \tag{117}
$$

Finally, a Gaussian integration by part w.r.t. $W_\mu^* \sim \mathcal{N}(0,1)$ gives:

$$
\mathbb{E}\left[\frac{(1-q_\epsilon(t))W_\mu^*}{\sqrt{t+2s_n-R_2(t,\epsilon)}} u_{Y_\mu^{(t,\epsilon)}}'(S_\mu^{(t,\epsilon)}) \ln \mathcal{Z}_{t,\epsilon}\right] = \mathbb{E}\left[(1-q_\epsilon(t))\frac{P_{\text{out}}''(Y_\mu^{(t,\epsilon)}|S_\mu^{(t,\epsilon)})}{P_{\text{out}}(Y_\mu^{(t,\epsilon)}|S_\mu^{(t,\epsilon)})} \ln \mathcal{Z}_{t,\epsilon}\right]. \tag{118}
$$

Plugging (116), (117) and (118) back in (115), we obtain:

$$
\mathbb{E}\left[\frac{\partial S_\mu^{(t,\epsilon)}}{\partial t} u_{Y_\mu^{(t,\epsilon)}}'(S_\mu^{(t,\epsilon)}) \ln \mathcal{Z}_{t,\epsilon}\right] = -\frac{1}{2}\mathbb{E}\left[\frac{P_{\text{out}}''(Y_\mu^{(t,\epsilon)}|S_\mu^{(t,\epsilon)})}{P_{\text{out}}(Y_\mu^{(t,\epsilon)}|S_\mu^{(t,\epsilon)})}\left(\frac{\|\mathbf{X}^*\|^2}{k_n}-1\right)\ln \mathcal{Z}_{t,\epsilon}\right]
$$

$$
-\frac{1}{2}\mathbb{E}\left\langle (Q-q_\epsilon(t))u_{Y_\mu^{(t,\epsilon)}}'(S_\mu^{(t,\epsilon)})u_{Y_\mu^{(t,\epsilon)}}'(s_\mu^{(t,\epsilon)})\right\rangle_{n,t,\epsilon}. \tag{119}
$$

It remains to compute the first kind of expectation on the right-hand side of (114), i.e.,

$$
\mathbb{E}\big[X_i^*\widetilde{Z}_i \ln \mathcal{Z}_{t,\epsilon}\big] = \mathbb{E}\left[X_i^*\widetilde{Z}_i \ln \int dP_{0,n}(\mathbf{x})\mathcal{D}\mathbf{w}\, P_{\text{out}}(Y_\mu^{(t,\epsilon)}|s_\mu^{(t,\epsilon)})e^{-\sum_{i=1}^n \frac{(\sqrt{R_1(t,\epsilon)}(X_i^*-x_i)+\widetilde{Z}_i)^2}{2}}\right]
$$

$$
= -\mathbb{E}\big[X_i^*\big\langle\sqrt{R_1(t,\epsilon)}(X_i^*-x_i)+\widetilde{Z}_i\big\rangle_{n,t,\epsilon}\big]
$$

$$
= -\sqrt{R_1(t,\epsilon)}\,\mathbb{E}\big\langle(\rho_n - X_i^*x_i)\big\rangle_{n,t,\epsilon}. \tag{120}
$$

The second equality follows from a Gaussian integration by parts w.r.t. $\widetilde{Z}_i \sim \mathcal{N}(0,1)$. Plugging the two simplified expectations (119) and (120) back in (114) yields:

$$
f_{n,\epsilon}'(t) = -\frac{\rho_n}{2\alpha_n}r_\epsilon(t)(1-q_\epsilon(t)) - \frac{1}{2}\mathbb{E}\left[\sum_{\mu=1}^{m_n}\frac{P_{\text{out}}''(Y_\mu^{(t,\epsilon)}|S_\mu^{(t,\epsilon)})}{P_{\text{out}}(Y_\mu^{(t,\epsilon)}|S_\mu^{(t,\epsilon)})}\left(\frac{\|\mathbf{X}^*\|^2}{k_n}-1\right)\frac{\ln \mathcal{Z}_{t,\epsilon}}{m_n}\right]
$$

$$
-\frac{1}{2}\mathbb{E}\left\langle (Q-q_\epsilon(t))\left(\frac{1}{m_n}\sum_{\mu=1}^{m_n}u_{Y_\mu^{(t,\epsilon)}}'(S_\mu^{(t,\epsilon)})u_{Y_\mu^{(t,\epsilon)}}'(s_\mu^{(t,\epsilon)}) - \frac{\rho_n}{\alpha_n}r_\epsilon(t)\right)\right\rangle_{n,t,\epsilon}. \tag{121}
$$

The last step to end the proof of the proposition is to upper bound

$$
A_n^{(t,\epsilon)} := \mathbb{E}\left[\sum_{\mu=1}^{m_n}\frac{P_{\text{out}}''(Y_\mu^{(t,\epsilon)}|S_\mu^{(t,\epsilon)})}{P_{\text{out}}(Y_\mu^{(t,\epsilon)}|S_\mu^{(t,\epsilon)})}\left(\frac{\|\mathbf{X}^*\|^2}{k_n}-1\right)\frac{\ln \mathcal{Z}_{t,\epsilon}}{m_n}\right] \tag{122}
$$

which appears on the right-hand side of (121).

**Upper bouding the quantity** (122)  Remember that $u_y''(x)+u_y'(x)^2 = \frac{P_{\text{out}}''(y|x)}{P_{\text{out}}(y|x)}$ and $P_{\text{out}}(y|x)=e^{u_y(x)}$. Therefore, $\forall x$:

$$
\int_{-\infty}^{+\infty} P_{\text{out}}''(y|x)dy = \int_{-\infty}^{+\infty}(u_y''(x)+u_y'(x)^2)e^{u_y(x)}dy = 0,
$$

where the second equality follows from the direct computation (99) in Lemma 7 of Appendix A. Consequently, using the tower property of the conditionnal expectation, for all $\mu \in \{1,\ldots,m\}$:

$$
\mathbb{E}\left[\sum_{\mu=1}^{m_n}\frac{P_{\text{out}}''(Y_\mu^{(t,\epsilon)}|S_\mu^{(t,\epsilon)})}{P_{\text{out}}(Y_\mu^{(t,\epsilon)}|S_\mu^{(t,\epsilon)})}\left(\frac{\|\mathbf{X}^*\|^2}{k_n}-1\right)\right] = \mathbb{E}\left[\left(\frac{\|\mathbf{X}^*\|^2}{k_n}-1\right)\sum_{\mu=1}^{m_n}\mathbb{E}\left[\frac{P_{\text{out}}''(Y_\mu^{(t,\epsilon)}|S_\mu^{(t,\epsilon)})}{P_{\text{out}}(Y_\mu^{(t,\epsilon)}|S_\mu^{(t,\epsilon)})}\Big|\mathbf{X}^*,\mathbf{S}^{(t,\epsilon)}\right]\right]
$$

$$
= \mathbb{E}\left[\left(\frac{\|\mathbf{X}^*\|^2}{k_n}-1\right)\sum_{\mu=1}^{m_n}\int_{-\infty}^{+\infty}P_{\text{out}}''(y|S_\mu^{(t,\epsilon)})dy\right] = 0. \tag{123}
$$

Making use of (123) and Cauchy-Schwarz inequality, we have:

$$|A_n^{(t,\epsilon)}| = \left| \mathbb{E}\left[ \sum_{\mu=1}^{m_n} \frac{P''_{\text{out}}(Y_\mu^{(t,\epsilon)}|S_\mu^{(t,\epsilon)})}{P_{\text{out}}(Y_\mu^{(t,\epsilon)}|S_\mu^{(t,\epsilon)})} \left( \frac{\|\mathbf{X}^*\|^2}{k_n} - 1 \right) \left( \frac{\ln \mathcal{Z}_{t,\epsilon}}{m_n} - f_{n,\epsilon}(t) \right) \right] \right|$$

$$\leq \mathbb{E}\left[ \left( \sum_{\mu=1}^{m_n} \frac{P''_{\text{out}}(Y_\mu^{(t,\epsilon)}|S_\mu^{(t,\epsilon)})}{P_{\text{out}}(Y_\mu^{(t,\epsilon)}|S_\mu^{(t,\epsilon)})} \right)^2 \left( \frac{\|\mathbf{X}^*\|^2}{k_n} - 1 \right)^2 \right]^{\frac{1}{2}} \sqrt{\mathbb{V}\text{ar}\, \frac{\ln \mathcal{Z}_{t,\epsilon}}{m_n}} \,. \quad (124)$$

Using again the tower property of the conditional expectation gives:

$$\mathbb{E}\left[ \left( \sum_{\mu=1}^{m_n} \frac{P''_{\text{out}}(Y_\mu^{(t,\epsilon)}|S_\mu^{(t,\epsilon)})}{P_{\text{out}}(Y_\mu^{(t,\epsilon)}|S_\mu^{(t,\epsilon)})} \right)^2 \left( \frac{\|\mathbf{X}^*\|^2}{k_n} - 1 \right)^2 \right]$$

$$= \mathbb{E}\left[ \left( \frac{\|\mathbf{X}^*\|^2}{k_n} - 1 \right)^2 \mathbb{E}\left[ \left( \sum_{\mu=1}^{m_n} \frac{P''_{\text{out}}(Y_\mu^{(t,\epsilon)}|S_\mu^{(t,\epsilon)})}{P_{\text{out}}(Y_\mu^{(t,\epsilon)}|S_\mu^{(t,\epsilon)})} \right)^2 \,\Big|\, \mathbf{X}^*, \mathbf{S}^{(t,\epsilon)} \right] \right] \,. \quad (125)$$

Note that conditionally on $\mathbf{S}^{(t,\epsilon)}$ the random variables $\big(P''_{\text{out}}(Y_\mu^{(t,\epsilon)}|S_\mu^{(t,\epsilon)})/P_{\text{out}}(Y_\mu^{(t,\epsilon)}|S_\mu^{(t,\epsilon)})\big)_{\mu=1}^{m_n}$ are i.i.d. and centered. Therefore:

$$\mathbb{E}\left[ \left( \sum_{\mu=1}^{m_n} \frac{P''_{\text{out}}(Y_\mu^{(t,\epsilon)}|S_\mu^{(t,\epsilon)})}{P_{\text{out}}(Y_\mu^{(t,\epsilon)}|S_\mu^{(t,\epsilon)})} \right)^2 \,\Big|\, \mathbf{X}^*, \mathbf{S}^{(t,\epsilon)} \right] = \mathbb{E}\left[ \left( \sum_{\mu=1}^{m_n} \frac{P''_{\text{out}}(Y_\mu^{(t,\epsilon)}|S_\mu^{(t,\epsilon)})}{P_{\text{out}}(Y_\mu^{(t,\epsilon)}|S_\mu^{(t,\epsilon)})} \right)^2 \,\Big|\, \mathbf{S}^{(t,\epsilon)} \right]$$

$$= m_n \mathbb{E}\left[ \left( \frac{P''_{\text{out}}(Y_1^{(t,\epsilon)}|S_1^{(t,\epsilon)})}{P_{\text{out}}(Y_1^{(t,\epsilon)}|S_1^{(t,\epsilon)})} \right)^2 \,\Big|\, \mathbf{S}^{(t,\epsilon)} \right]$$

$$= m_n \mathbb{E}\left[ \int_{-\infty}^{+\infty} \frac{P''_{\text{out}}(y|S_1^{(t,\epsilon)})^2}{P_{\text{out}}(y|S_1^{(t,\epsilon)})} dy \right] \,. \quad (126)$$

We now use the formula (98) for $u''_y(x) + u'_y(x)^2 = {}^{P''_{\text{out}}(y|x)}/{}_{P_{\text{out}}(y|x)}$ (obtained in Lemma 7 of Appendix A) with Jensen's equality to show that for all $x$:

$$\left( \frac{P''_{\text{out}}(y|x)}{P_{\text{out}}(y|x)} \right)^2 \leq \frac{\int \left( \frac{(y-\varphi(x,\mathbf{a}))^2 \partial_x\varphi(x,\mathbf{a})^2 - \Delta\partial_x\varphi(x,\mathbf{a})^2 + \Delta\partial_{xx}\varphi(x,\mathbf{a})(y-\varphi(x,\mathbf{a}))}{\Delta^2} \right)^2 \frac{dP_A(\mathbf{a})}{\sqrt{2\pi\Delta}} e^{-\frac{(y-\varphi(x,\mathbf{a}))^2}{2\Delta}}}{\int \frac{dP_A(\mathbf{a})}{\sqrt{2\pi\Delta}} e^{-\frac{(y-\varphi(x,\mathbf{a}))^2}{2\Delta}}}$$

$$= \frac{\int \left( \frac{(y-\varphi(x,\mathbf{a}))^2 \partial_x\varphi(x,\mathbf{a})^2 - \Delta\partial_x\varphi(x,\mathbf{a})^2 + \Delta\partial_{xx}\varphi(x,\mathbf{a})(y-\varphi(x,\mathbf{a}))}{\Delta^2} \right)^2 \frac{dP_A(\mathbf{a})}{\sqrt{2\pi\Delta}} e^{-\frac{(y-\varphi(x,\mathbf{a}))^2}{2\Delta}}}{P_{\text{out}}(y|x)} \,.$$

It follows that for all $x$:

$$\int_{-\infty}^{+\infty} \frac{P''_{\text{out}}(y|x)^2}{P_{\text{out}}(y|x)} dy = \int dP_A(\mathbf{a}) \int_{-\infty}^{+\infty} \left( \frac{(u^2-1)\partial_x\varphi(x,\mathbf{a})^2 + \sqrt{\Delta}\partial_{xx}\varphi(x,\mathbf{a})u}{\Delta} \right)^2 \frac{du}{\sqrt{2\pi}} e^{-\frac{u^2}{2}}$$

$$\leq 4 \left\| \frac{\partial_x\varphi}{\sqrt{\Delta}} \right\|_\infty^4 + 2 \left\| \frac{\partial_{xx}\varphi}{\sqrt{\Delta}} \right\|_\infty^2 \,.$$

Let $D := \left\| \frac{\partial_x\varphi}{\sqrt{\Delta}} \right\|_\infty^4 + \frac{1}{2} \left\| \frac{\partial_{xx}\varphi}{\sqrt{\Delta}} \right\|_\infty^2$. Combining this last upper bound with (126) and (125) yields:

$$\mathbb{E}\left[ \left( \sum_{\mu=1}^{m_n} \frac{P''_{\text{out}}(Y_\mu^{(t,\epsilon)}|S_\mu^{(t,\epsilon)})}{P_{\text{out}}(Y_\mu^{(t,\epsilon)}|S_\mu^{(t,\epsilon)})} \right)^2 \left( \frac{\|\mathbf{X}^*\|^2}{k_n} - 1 \right)^2 \right] \leq 4D\, m_n \mathbb{V}\text{ar}\left( \frac{\|\mathbf{X}^*\|^2}{k_n} \right) = \frac{4D\alpha_n S^4}{\rho_n} \quad (127)$$

Going back to (124), we have $\forall (t,\epsilon) \in [0,1] \times \mathcal{B}_n$:

$$|A_n^{(t,\epsilon)}| \leq 2S^2 \sqrt{D \frac{\alpha_n}{\rho_n} \mathbb{V}\text{ar}\, \frac{\ln \mathcal{Z}_{t,\epsilon}}{m_n}} \,. \quad (128)$$

**Putting everything together: proofs of** (104) **and** (105)  Combining (107) and (121) yields the following formula for the derivative of $i_{n,\epsilon}$ (remember the definition (122) of $A_n^{(t,\epsilon)}$):

$$i'_{n,\epsilon}(t) = \frac{A_n^{(t,\epsilon)}}{2} - \mathbb{E}\left[ h'(\rho^{(t)}) \left( \frac{\|\mathbf{X}^*\|^2}{k_n} - 1 \right) \right] + \frac{\rho_n}{2\alpha_n} r_\epsilon(t)(1 - q_\epsilon(t))$$

$$+ \frac{1}{2} \mathbb{E}\left\langle \left( Q - q_\epsilon(t) \right) \left( \frac{1}{m_n} \sum_{\mu=1}^{m_n} u'_{Y_\mu^{(t,\epsilon)}}(S_\mu^{(t,\epsilon)}) u'_{Y_\mu^{(t,\epsilon)}}(s_\mu^{(t,\epsilon)}) - \frac{\rho_n}{\alpha_n} r_\epsilon(t) \right) \right\rangle_{n,t,\epsilon} \,. \quad (129)$$

Combining the identity (129) with the upper bounds (108) and (128) yields (104).

It remains to prove the identity (105) that holds under the additional assumption that $\forall n : \alpha_n \leq M_\alpha$, $\rho_n/\alpha_n \leq M_{\rho/\alpha}$. Combining (128) with the upper bound (144) on the variance of $\mathbb{Var}(\ln \mathcal{Z}_{t,\epsilon}/m_n)$ (see Proposition 9 of Appendix C) gives:

$$\left|\frac{A_n^{(t,\epsilon)}}{2}\right| \leq \frac{S^2 \sqrt{D(\widetilde{C}_1 + M_{\rho/\alpha}\widetilde{C}_2 + M_\alpha \widetilde{C}_3)}}{\sqrt{n}\rho_n} .$$

The constants $\widetilde{C}_1, \widetilde{C}_2, \widetilde{C}_3$ are defined in Proposition 9 while $D$ has been defined earlier in the proof. Besides, as $\rho_n \leq 1$, we have $\frac{1}{\sqrt{n\rho_n}} \leq \frac{1}{\sqrt{n}\rho_n}$ and we can loosen the upper bound (108): $\left|\mathbb{E}\left[h'(\rho^{(t)})\left(\frac{\|\mathbf{X}^*\|^2}{k_n} - 1\right)\right]\right| \leq \frac{CS^2}{\sqrt{n}\rho_n}$. Then, the term $A_n^{(t,\epsilon)}/2 - \mathbb{E}\left[h'(\rho^{(t)})(\|\mathbf{X}^*\|^2/k_n - 1)\right]$ on the right-hand side of (129) is in $\mathcal{O}(1/\sqrt{n}\rho_n)$ and this proves the identity (105). $\qquad\square$

## B.2   Proof of Lemma 1

*Proof.* At $t = 0$ the functions $r_\epsilon$ and $q_\epsilon$ do not play any role in the observations (12) since $R_1(t,\epsilon) = \epsilon_1$ and $R_2(t,\epsilon) = \epsilon_2$. While in the main text we restricted $\epsilon$ to be in $\mathcal{B}_n := [s_n, 2s_n]^2$, we can define observations $(\mathbf{Y}^{(0,\epsilon)}, \widetilde{\mathbf{Y}}^{(0,\epsilon)})$ using (12) for $t = 0$ and $\epsilon \in [0, 2s_n]^2$. We then extend the *interpolating mutual information* at $t = 0$ to all $\epsilon \in [0, 2s_n]^2$:

$$i_{n,\epsilon}(0) := \frac{1}{m_n} I\big((\mathbf{X}^*, \mathbf{W}^*); (\mathbf{Y}^{(0,\epsilon)}, \widetilde{\mathbf{Y}}^{(0,\epsilon)})\big|\mathbf{\Phi}, \mathbf{V}\big) .$$

Note that the variation we want to control in this lemma satisfies:

$$\left|i_{n,\epsilon}(0) - \frac{I(\mathbf{X}^*; \mathbf{Y}|\mathbf{\Phi})}{m_n}\right| \leq \left|i_{n,\epsilon}(0) - i_{n,\epsilon=(0,0)}(0)\right| + \left|i_{n,\epsilon=(0,0)}(0) - \frac{I(\mathbf{X}^*; \mathbf{Y}|\mathbf{\Phi})}{m_n}\right| . \tag{130}$$

We will upper bound the two terms on the right-hand side of (130) separately.
**1.** By the I-MMSE relation (see [8]), we have for all $\epsilon \in [0, 2s_n]^2$:

$$\left|\frac{\partial i_{n,\epsilon}(0)}{\partial \epsilon_1}\right| = \frac{1}{2\alpha_n}\mathbb{E}\big[(X_1^* - \langle x_1 \rangle_{n,0,\epsilon})^2\big] \leq \frac{\mathbb{E}[(X_1^*)^2]}{2\alpha_n} = \frac{\rho_n}{2\alpha_n} . \tag{131}$$

To upper bound the absolute value of the partial derivative with respect to $\epsilon_2$, we use that $\epsilon \in [0, 2s_n]^2$:

$$\frac{\partial i_{n,\epsilon}(0)}{\partial \epsilon_2} = -\frac{1}{2}\mathbb{E}\big[u'_{Y_1^{(0,\epsilon)}}(S_1^{(0,\epsilon)})\langle u'_{Y_1^{(0,\epsilon)}}(s_1^{(0,\epsilon)})\rangle_{n,0,\epsilon}\big] .$$

This identity is obtained in a similar fashion to the computation of the derivative of $i_{n,\epsilon}(\cdot)$ in Appendix B.1 (see (117) and (118) in particular). Under the hypothesis (H2), we obtain in the proof of Lemma 7 the upper bound (93) on $|u'_y(x)|$ for all $x \in \mathbb{R}$. Making use of this upper bound yields $\forall x \in \mathbb{R} : \left|u'_{Y_1^{(0,\epsilon)}}(x)\right| \leq (2\|\varphi\|_\infty + |Z_1|)\|\partial_x\varphi\|_\infty$. Therefore:

$$\left|\frac{\partial i_{n,\epsilon}(0)}{\partial \epsilon_2}\right| \leq \frac{1}{2}\mathbb{E}\big[(2\|\varphi\|_\infty + |Z_1|)^2\|\partial_x\varphi\|_\infty^2\big] \leq (4\|\varphi\|_\infty^2 + 1)\|\partial_x\varphi\|_\infty^2 . \tag{132}$$

By the mean value theorem, and the upper bounds (131) and (132), we have:

$$\begin{aligned}
\left|i_{n,\epsilon}(0) - i_{n,\epsilon=(0,0)}(0)\right| &\leq \frac{\rho_n}{2\alpha_n}|\epsilon_1| + (4\|\varphi\|_\infty^2 + 1)\|\partial_x\varphi\|_\infty^2|\epsilon_2| \\
&\leq \left(\frac{\rho_n}{2\alpha_n} + (4\|\varphi\|_\infty^2 + 1)\|\partial_x\varphi\|_\infty^2\right)2s_n \\
&\leq \left(M_{\rho/\alpha} + 2(4\|\varphi\|_\infty^2 + 1)\|\partial_x\varphi\|_\infty^2\right)s_n .
\end{aligned} \tag{133}$$

**2.** It remains to upper bound the second term on the right-hand side of (130). Define the following observations where $\mathbf{X}^* \overset{\text{iid}}{\sim} P_{0,n}$, $\mathbf{\Phi} := (\Phi_{\mu i}) \overset{\text{iid}}{\sim} \mathcal{N}(0,1)$, $\mathbf{W}^* := (W_\mu^*)_{\mu=1}^{m_n} \overset{\text{iid}}{\sim} \mathcal{N}(0,1)$ and $\eta \in [0, +\infty)$:

$$Y_\mu^{(\eta)} \sim P_{\text{out}}\left(\cdot \left| \frac{(\mathbf{\Phi}\mathbf{X}^*)_\mu}{\sqrt{k_n}} + \sqrt{\eta}\,W_\mu^*\right.\right) + Z_\mu , \quad 1 \leq \mu \leq m_n . \tag{134}$$

The joint posterior density of $(\mathbf{X}^*, \mathbf{W}^*)$ given $(\mathbf{Y}^{(\eta)}, \mathbf{\Phi})$ reads:

$$dP(\mathbf{x}, \mathbf{w}|\mathbf{Y}^{(\eta)}, \mathbf{\Phi}) := \frac{1}{\mathcal{Z}_\eta} dP_{0,n}(\mathbf{x}) \prod_{\mu=1}^{m_n} \frac{dw_\mu}{\sqrt{2\pi}} e^{-\frac{w_\mu^2}{2}} P_{\text{out}}\left(Y_\mu^{(\eta)}\left|\frac{(\mathbf{\Phi}\mathbf{x})_\mu}{\sqrt{k_n}} + \sqrt{\eta}\,w_\mu\right.\right) , \tag{135}$$

where $\mathcal{Z}_\eta$ is the normalization factor. Define the average free entropy $f_n(\eta) := \mathbb{E}\ln\mathcal{Z}_\rho/m_n$. The mutual information $i_n(\eta) := \frac{1}{m_n}I\big((\mathbf{X}^*,\mathbf{W}^*);\mathbf{Y}^{(\eta)}\big|\mathbf{\Phi}\big)$ satisfies:

$$i_n(\rho) = \mathbb{E}\bigg[h\bigg(\frac{\|\mathbf{X}^*\|^2}{k_n}+\eta\bigg)\bigg] - f_n(\rho) - \frac{1}{2\alpha_n} \; . \tag{136}$$

where $h : \rho \in [0,+\infty) \mapsto \mathbb{E}_{V\sim\mathcal{N}(0,1)}\int u_y(\sqrt{\rho}\,V)e^{u_y(\sqrt{\rho}\,V)}dy$. The identity (136) can be obtained exactly as the identity (106) in Appendix B.1. Under the assumptions of the lemma, all the hypotheses of domination are reunited to make sure that $\eta \mapsto i_n(\eta)$ is continuous on $[0,2s_n]$ and differentiable on $(0,2s_n)$. Therefore, by the mean-value theorem, there exists $\eta^* \in (0,2s_n)$ such that:

$$\bigg|i_{n,\epsilon=(0,0)}(0) - \frac{I(\mathbf{X}^*;\mathbf{Y}|\mathbf{\Phi})}{m_n}\bigg| = \big|i_n(2s_n)-i_n(0)\big| = |i_n'(\eta^*)|2s_n \; . \tag{137}$$

Again, in a similar fashion to the computation of the derivative of $i_{n,\epsilon}(\cdot)$ in Appendix B.1, we can show that $\forall\eta \in [0,+\infty)$:

$$i_n'(\rho) = \mathbb{E}\bigg[h'\bigg(\frac{\|\mathbf{X}^*\|^2}{k_n}+\eta\bigg)\bigg] - f_n'(\rho) \; ; \tag{138}$$

$$f_n'(\rho) = \frac{1}{2}\mathbb{E}\bigg[\sum_{\mu=1}^{m_n}\frac{P_{\text{out}}''\big(Y_\mu^{(\rho)}\big|\frac{(\mathbf{\Phi}\mathbf{X}^*)_\mu}{\sqrt{k_n}}+\sqrt{\eta}\,W_\mu^*\big)}{P_{\text{out}}\big(Y_\mu^{(\rho)}\big|\frac{(\mathbf{\Phi}\mathbf{X}^*)_\mu}{\sqrt{k_n}}+\sqrt{\eta}\,W_\mu^*\big)}\frac{\ln\mathcal{Z}_\rho}{m_n}\bigg] \; . \tag{139}$$

In Lemma 7 of Appendix A, we compute $h'$ and show:

$$\forall\rho\in[0,+\infty) : |h'(\rho)| \le C := C\bigg(\bigg\|\frac{\varphi}{\sqrt{\Delta}}\bigg\|_\infty, \bigg\|\frac{\partial_x\varphi}{\sqrt{\Delta}}\bigg\|_\infty, \bigg\|\frac{\partial_{xx}\varphi}{\sqrt{\Delta}}\bigg\|_\infty\bigg)$$

with $C(a,b,c) := b^2(64a^4+2a^2+12.5)+c\big(8a^3+2\sqrt{\frac{2}{\pi}}\,\big)$. The first term on the right-hand side of (138) thus satisfies:

$$\bigg|\mathbb{E}\bigg[h'\bigg(\frac{\|\mathbf{X}^*\|^2}{k_n}+\eta\bigg)\bigg]\bigg| \le C \; . \tag{140}$$

The second term, i.e., $f_n'(\rho)$ is similar to the quantity $A_n^{(t,\epsilon)}$ defined in (122). We upper bound $A_n^{(t,\epsilon)}$ in the last part of the proof in Appendix B.1. We can follow the same steps than for upper bounding $A_n^{(t,\epsilon)}$ and obtain:

$$|f_n'(\eta)| \le \sqrt{Dm_n\mathbb{V}\text{ar}\frac{\ln\mathcal{Z}_\eta}{m_n}} \; . \tag{141}$$

Note that $\mathcal{Z}_{\eta=2s_n} = \mathcal{Z}_{t=0,\epsilon=(0,0)}$. By Proposition 9 in Appendix C we have $\mathbb{V}\text{ar}\frac{\ln\mathcal{Z}_{\eta=2s_n}}{m_n} \le \frac{\widetilde{C}}{n\alpha_n\rho_n}$ where $\widetilde{C}$ is a polynomial in $\big(S,\|\varphi\|_\infty,\|\partial_x\varphi\|_\infty,\|\partial_{xx}\varphi\|_\infty,M_\alpha,M_{\rho/\alpha}\big)$ with positive coefficients. In fact, this upper bound holds for all $\eta \in [0,2s_n]$, i.e.,

$$\forall\eta\in[0,2s_n] : \mathbb{V}\text{ar}\bigg(\frac{\ln\mathcal{Z}_\eta}{m_n}\bigg) \le \frac{\widetilde{C}}{n\alpha_n\rho_n} \; .$$

The proof of this uniform bound on $\mathbb{V}\text{ar}\big(\ln\mathcal{Z}_\eta/m_n\big)$ is the same as the one of Proposition 9, only that it is simpler because there is no second channel similar to $\widetilde{\mathbf{Y}}^{(t,\epsilon)}$. We now combine (137), (138), (140), (141) to finally obtain:

$$\bigg|i_{n,\epsilon=(0,0)}(0) - \frac{I(\mathbf{X}^*;\mathbf{Y}|\mathbf{\Phi})}{m_n}\bigg| \le \bigg(C+\sqrt{\frac{D\widetilde{C}}{\rho_n}}\bigg)2s_n \; . \tag{142}$$

**3.** We now plug (133) and (142) back in (130) and use that $\rho_n \in (0,1]$ to end the proof of the lemma:

$$\bigg|i_{n,\epsilon}(0) - \frac{I(\mathbf{X}^*;\mathbf{Y}|\mathbf{\Phi})}{m_n}\bigg| \le \big(M_{\rho/\alpha}+2(4\|\varphi\|_\infty^2+1)\|\partial_x\varphi\|_\infty^2+2C+\sqrt{D\widetilde{C}}\,\big)\frac{s_n}{\sqrt{\rho_n}} \; .$$

$\square$

## C   Concentration of the free entropy

In this appendix we show that the log-partition function per data point, or *free entropy*, of the interpolating model studied in Section 2.1 concentrates around its expectation.

**Proposition 9** (Free entropy concentration). *Suppose that $\Delta > 0$ and that all of (H1), (H2) and (H3) hold. Further assume that $\mathbb{E}_{X \sim P_0}[X^2] = 1$. We have for all $(t, \epsilon) \in [0, 1] \times \mathcal{B}_n$:*

$$\mathbb{V}\text{ar}\left(\frac{\ln \mathcal{Z}_{t,\epsilon}}{m_n}\right) \leq \frac{1}{n\alpha_n\rho_n}\left(\widetilde{C}_1 + \frac{\rho_n}{\alpha_n}\widetilde{C}_2 + \alpha_n\widetilde{C}_3\right), \tag{143}$$

*where ($\partial_x \varphi$ and $\partial_{xx}\varphi$ denote the first and second partial derivatives of $\varphi$ with respect to its first argument):*

$$\widetilde{C}_1 := 1.5 + 4\left\|\frac{\varphi}{\sqrt{\Delta}}\right\|_\infty^2 + 8S^2\left(4\left\|\frac{\varphi}{\sqrt{\Delta}}\right\|_\infty^2 + 1\right)\left\|\frac{\partial_x\varphi}{\sqrt{\Delta}}\right\|_\infty^2$$
$$+ \left(2\left\|\frac{\varphi}{\sqrt{\Delta}}\right\|_\infty + \sqrt{\frac{2}{\pi}}\right)^2\left(\left\|\frac{\varphi}{\sqrt{\Delta}}\right\|_\infty^2 + (16 + 4S^2)\left\|\frac{\partial_x\varphi}{\sqrt{\Delta}}\right\|_\infty^2\right);$$

$$\widetilde{C}_2 := 1.5 + 12S^2;$$

$$\widetilde{C}_3 := 8S^2\left(3\left\|\frac{\partial_x\varphi}{\sqrt{\Delta}}\right\|_\infty^2 + \left\|\frac{\varphi}{\sqrt{\Delta}}\right\|_\infty\left\|\frac{\partial_{xx}\varphi}{\sqrt{\Delta}}\right\|_\infty + 12\left\|\frac{\partial_x\varphi}{\sqrt{\Delta}}\right\|_\infty^2\left\|\frac{\varphi}{\sqrt{\Delta}}\right\|_\infty^2 + 2\sqrt{\frac{2}{\pi}}\left\|\frac{\varphi}{\sqrt{\Delta}}\right\|_\infty\left\|\frac{\partial_x\varphi}{\sqrt{\Delta}}\right\|_\infty^2\right)^2.$$

*In addition, if both sequences $(\alpha_n)_n$ and $(\rho_n/\alpha_n)_n$ are bounded, i.e., if there exist real positive numbers $M_\alpha, M_{\rho/\alpha}$ such that $\forall n \in \mathbb{N}^* : \alpha_n \leq M_\alpha, \rho_n/\alpha_n \leq M_{\rho/\alpha}$ then for all $(t, \epsilon) \in [0, 1] \times \mathcal{B}_n$:*

$$\mathbb{V}\text{ar}\left(\frac{\ln \mathcal{Z}_{t,\epsilon}}{m_n}\right) \leq \frac{C}{n\alpha_n\rho_n}, \tag{144}$$

*where $C := \widetilde{C}_1 + M_{\rho/\alpha}\widetilde{C}_2 + M_\alpha\widetilde{C}_3$.*

To lighten notations, we define $k_1 := \sqrt{R_2(t,\epsilon)}$, $k_2 := \sqrt{t + 2s_n - R_2(t,\epsilon)}$. Let $\mathbf{X}^* \overset{\text{iid}}{\sim} P_{0,n}$, $\boldsymbol{\Phi} := (\Phi_{\mu i}) \overset{\text{iid}}{\sim} \mathcal{N}(0,1)$, $\mathbf{V} := (V_\mu)_{\mu=1}^{m_n} \overset{\text{iid}}{\sim} \mathcal{N}(0,1)$ and $\mathbf{W}^* := (W_\mu^*)_{\mu=1}^{m_n} \overset{\text{iid}}{\sim} \mathcal{N}(0,1)$. Remember that

$$S_\mu^{(t,\epsilon)} := \sqrt{\frac{1-t}{k_n}}(\boldsymbol{\Phi}\mathbf{X}^*)_\mu + k_1 V_\mu + k_2 W_\mu^*, \tag{145}$$

and that, in the interpolation problem, we observe:

$$\begin{cases} Y_\mu^{(t,\epsilon)} & \sim \quad \varphi\big(S_\mu^{(t,\epsilon)}, \mathbf{A}_\mu\big) + \sqrt{\Delta}Z_\mu, \quad 1 \leq \mu \leq m_n; \\ \widetilde{Y}_i^{(t,\epsilon)} & = \quad \sqrt{R_1(t,\epsilon)}X_i^* + \widetilde{Z}_i \qquad\qquad, \quad 1 \leq i \leq n; \end{cases} \tag{146}$$

where $(Z_\mu)_{\mu=1}^{m_n}, (\widetilde{Z}_i)_{i=1}^n \overset{\text{iid}}{\sim} \mathcal{N}(0,1)$ and $(\mathbf{A}_\mu)_{\mu=1}^{m_n} \overset{\text{iid}}{\sim} P_A$. $\mathcal{Z}_{t,\epsilon}$ is the normalization to the joint posterior density of $(\mathbf{X}^*, \mathbf{W}^*)$ given $(\mathbf{Y}^{(t,\epsilon)}, \widetilde{\mathbf{Y}}^{(t,\epsilon)}, \boldsymbol{\Phi}, \mathbf{V})$, i.e.,

$$\mathcal{Z}_{t,\epsilon} := \int dP_{0,n}(\mathbf{x})\mathcal{D}\mathbf{w}\, e^{-\frac{\|\sqrt{R_1(t,\epsilon)}\mathbf{x} - \widetilde{\mathbf{Y}}^{(t,\epsilon)}\|^2}{2}} P_{\text{out}}\big(Y_\mu^{(t,\epsilon)}|s_\mu^{(t,\epsilon)}\big),$$

where $\mathcal{D}\mathbf{w} := \frac{d\mathbf{w}e^{-\frac{\|\mathbf{w}\|^2}{2}}}{\sqrt{2\pi}^{m_n}}$ and $s_\mu^{(t,\epsilon)} := \sqrt{\frac{1-t}{k_n}}(\boldsymbol{\Phi}\mathbf{x})_\mu + k_1 V_\mu + k_2 w_\mu$. We define:

$$\Gamma_\mu^{(t,\epsilon)} := \frac{\varphi\big(S_\mu^{(t,\epsilon)}, \mathbf{A}_\mu\big) - \varphi\big(s_\mu^{(t,\epsilon)}, \mathbf{a}_\mu\big)}{\Delta}.$$

By definition, $P_{\text{out}}(Y_\mu^{(t,\epsilon)}|s_\mu^{(t,\epsilon)}) = \int dP_A(\mathbf{a}_\mu)\frac{1}{\sqrt{2\pi\Delta}}e^{-\frac{1}{2}(\Gamma_\mu^{(t,\epsilon)} + Z_\mu)^2}$. Therefore, the interpolating free entropy satisfies:

$$\frac{\ln \mathcal{Z}_{t,\epsilon}}{m_n} = \frac{1}{2}\ln(2\pi\Delta) - \frac{1}{2m_n}\sum_{\mu=1}^{m_n}Z_\mu^2 - \frac{1}{2m_n}\sum_{i=1}^n \widetilde{Z}_i^2 + \frac{\ln \widehat{\mathcal{Z}}_{t,\epsilon}}{m_n} \tag{147}$$

where

$$\widehat{\mathcal{Z}}_{t,\epsilon} := \int dP_{0,n}(\mathbf{x})\mathcal{D}\mathbf{w}dP_A(\mathbf{a}_\mu)\, e^{-\widehat{\mathcal{H}}_{t,\epsilon}(\mathbf{x},\mathbf{w},\mathbf{a})}; \tag{148}$$

$$\widehat{\mathcal{H}}_{t,\epsilon}(\mathbf{x},\mathbf{w},\mathbf{a}) := \frac{1}{2}\sum_{\mu=1}^{m_n}(\Gamma_\mu^{(t,\epsilon)})^2 + 2Z_\mu\Gamma_\mu^{(t,\epsilon)}$$

$$+ \frac{1}{2}\sum_{i=1}^n R_1(t,\epsilon)(X_i^* - x_i)^2 + 2Z_i'\sqrt{R_1(t,\epsilon)}(X_i^* - x_i). \tag{149}$$

From (147), it follows directly that:

$$\mathbb{Var}\left(\frac{\ln \mathcal{Z}_{t,\epsilon}}{m_n}\right) \leq 3\mathbb{Var}\left(\frac{1}{2m_n}\sum_{\mu=1}^{m_n} Z_\mu^2\right) + 3\mathbb{Var}\left(\frac{1}{2m_n}\sum_{i=1}^{n}\widetilde{Z}_i^2\right) + 3\mathbb{Var}\left(\frac{\ln \widehat{\mathcal{Z}}_{t,\epsilon}}{m_n}\right)$$

$$= \frac{3}{2\alpha_n n} + \frac{3}{2\alpha_n^2 n} + 3\mathbb{Var}\left(\frac{\ln \widehat{\mathcal{Z}}_{t,\epsilon}}{m_n}\right) \tag{150}$$

In order to prove Proposition 9, it remains to show that $\ln \widehat{\mathcal{Z}}_{t,\epsilon}/m_n$ concentrates. We recall here the classical variance bounds that we will use. We refer to [13, Chapter 3] for detailed proofs of these statements.

**Proposition 10** (Gaussian Poincaré inequality). *Let $\mathbf{U} = (U_1, \ldots, U_N)$ be a vector of $N$ independent standard normal random variables. Let $g : \mathbb{R}^N \to \mathbb{R}$ be a $\mathcal{C}^1$ function. Then*

$$\mathbb{Var}(g(\mathbf{U})) \leq \mathbb{E}\big[\|\nabla g(\mathbf{U})\|^2\big]. \tag{151}$$

**Proposition 11** (Bounded difference). *Let $\mathcal{U} \subset \mathbb{R}$. Let $g : \mathcal{U}^N \to \mathbb{R}$ a function that satisfies the bounded difference property, i.e., there exists some constants $c_1, \ldots, c_N \geq 0$ such that*

$$\sup_{\substack{(u_1,\ldots,u_N)\in\mathcal{U}^N \\ u_i'\in\mathcal{U}}} |g(u_1,\ldots,u_i,\ldots,u_N) - g(u_1,\ldots,u_i',\ldots,u_N)| \leq c_i \quad \text{for all} \quad 1 \leq i \leq N.$$

*Let $\mathbf{U} = (U_1, \ldots, U_N)$ be a vector of $N$ independent random variables that take values in $\mathcal{U}$. Then*

$$\mathbb{Var}(g(\mathbf{U})) \leq \frac{1}{4}\sum_{i=1}^{N} c_i^2. \tag{152}$$

**Proposition 12** (Efron-Stein inequality). *Let $\mathcal{U} \subset \mathbb{R}$, and a function $g : \mathcal{U}^N \to \mathbb{R}$. Let $\mathbf{U} = (U_1, \ldots, U_N)$ be a vector of $N$ independent random variables with law $P_U$ that take values in $\mathcal{U}$. Let $\mathbf{U}^{(i)}$ a vector which differs from $\mathbf{U}$ only by its $i$-th component, which is replaced by $U_i'$ drawn from $P_U$ independently of $\mathbf{U}$. Then*

$$\mathbb{Var}(g(\mathbf{U})) \leq \frac{1}{2}\sum_{i=1}^{N}\mathbb{E}_{\mathbf{U}}\mathbb{E}_{U_i'}\big[(g(\mathbf{U}) - g(\mathbf{U}^{(i)}))^2\big]. \tag{153}$$

We first show the concentration w.r.t. all Gaussian variables $\mathbf{\Phi}, \mathbf{V}, \mathbf{Z}, \mathbf{Z}', \mathbf{W}^*$, then the concentration w.r.t. $\mathbf{A}$ and finally the one w.r.t. $\mathbf{X}^*$. The order in which we prove the concentrations does matter.

We will denote $\partial_x \varphi$ and $\partial_{xx} \varphi$ the first and second partial derivatives of $\varphi$ with respect to its first argument. Note that $|R_1| \leq 2s_n + \frac{\alpha_n}{\rho_n} r_{\max}$ and, by the inequality (94) in Lemma 7 of Appendix A, $r_{\max} := 2\big|\frac{\partial I_{P_{\text{out}}}}{\partial q}\big|_{1,1}\big| \leq 2C_1(\|\frac{\varphi}{\sqrt{\Delta}}\|_\infty, \|\frac{\partial_x \varphi}{\sqrt{\Delta}}\|_\infty)$ with $C_1(a,b) := (4a^2 + 1)b^2$. Then, the quantity

$$K_n := 2\left(s_n + \frac{\alpha_n}{\rho_n}C_1\left(\left\|\frac{\varphi}{\sqrt{\Delta}}\right\|_\infty, \left\|\frac{\partial_x \varphi}{\sqrt{\Delta}}\right\|_\infty\right)\right)$$

upper bounds $|R_1|$. Besides, $|R_2|$ is upper bounded by 2.

### Concentration with respect to the Gaussian random variables

**Lemma 8.** *Let $\mathbb{E}_{\mathbf{Z},\widetilde{\mathbf{Z}}}$ be the expectation w.r.t. $(\mathbf{Z},\widetilde{\mathbf{Z}})$ only. Under the assumptions of Theorem 1, we have for all $(t,\epsilon) \in [0,1] \times \mathcal{B}_n$:*

$$\mathbb{E}\left[\left(\frac{\ln \hat{\mathcal{Z}}_{t,\epsilon}}{m_n} - \frac{1}{m_n}\mathbb{E}_{\mathbf{Z},\mathbf{Z}'}\ln \hat{\mathcal{Z}}_{t,\epsilon}\right)^2\right] \leq \frac{C_2}{n\alpha_n\rho_n} + \frac{C_3}{n\alpha_n^2}, \tag{154}$$

*where $C_2 := 4\|\frac{\varphi}{\sqrt{\Delta}}\|_\infty^2 + 8S^2 C_1\big(\|\frac{\varphi}{\sqrt{\Delta}}\|_\infty, \|\frac{\partial_x \varphi}{\sqrt{\Delta}}\|_\infty\big)$ and $C_3 = 4S^2$.*

*Proof.* In this proof we see $g := \ln \hat{\mathcal{Z}}_{t,\epsilon}/m_n$ as a function of $\mathbf{Z}$ and $\widetilde{\mathbf{Z}}$, and we work conditionally on all other random variables. We have $\|\nabla g\|^2 = \|\nabla_{\mathbf{Z}} g\|^2 + \|\nabla_{\widetilde{\mathbf{Z}}} g\|^2$. Each partial derivative has the form $\partial_u g = m_n^{-1}\langle \partial_u \widehat{\mathcal{H}}_{t,\epsilon}\rangle_{t,\epsilon}$. We find:

$$\|\nabla_{\mathbf{Z}} g\|^2 = m_n^{-2}\sum_{\mu=1}^{m_n}\langle \Gamma_\mu^{(t,\epsilon)}\rangle_{t,\epsilon}^2 \leq 4m_n^{-1}\left\|\frac{\varphi}{\sqrt{\Delta}}\right\|_\infty^2,$$

$$\|\nabla_{\widetilde{\mathbf{Z}}} g\|^2 = m_n^{-2}R_1(t,\epsilon)\sum_{i=1}^{n}(X_i^* - \langle x_i\rangle_{t,\epsilon})^2 \leq 4K_n S^2 m_n^{-2} n.$$

So $\|\nabla g\|^2 \le 4m_n^{-1}\big(\big\|\frac{\varphi}{\sqrt{\Delta}}\big\|_\infty^2 + \frac{K_n S^2}{\alpha_n}\big)$. Applying Proposition 10 yields:

$$
\begin{aligned}
\mathbb{E}_{\mathbf{z},\tilde{\mathbf{z}}}\Big[\Big(\frac{\ln \hat{\mathcal{Z}}_{t,\epsilon}}{m_n} - \frac{\mathbb{E}_{\mathbf{z},\tilde{\mathbf{z}}}\ln \hat{\mathcal{Z}}_{t,\epsilon}}{m_n}\Big)^2\Big] &\le \frac{4}{n\alpha_n}\Big(\Big\|\frac{\varphi}{\sqrt{\Delta}}\Big\|_\infty^2 + \frac{K_n S^2}{\alpha_n}\Big)\\
&= \frac{4}{n\alpha_n}\Big(\Big\|\frac{\varphi}{\sqrt{\Delta}}\Big\|_\infty^2 + \frac{2S^2 s_n}{\alpha_n} + \frac{2S^2}{\rho_n}C_1\Big(\Big\|\frac{\varphi}{\sqrt{\Delta}}\Big\|_\infty, \Big\|\frac{\partial_x\varphi}{\sqrt{\Delta}}\Big\|_\infty\Big)\Big)\\
&\le \frac{4}{n\alpha_n\rho_n}\Big(\Big\|\frac{\varphi}{\sqrt{\Delta}}\Big\|_\infty^2 + 2S^2 C_1\Big(\Big\|\frac{\varphi}{\sqrt{\Delta}}\Big\|_\infty, \Big\|\frac{\partial_x\varphi}{\sqrt{\Delta}}\Big\|_\infty\Big)\Big) + \frac{4S^2}{n\alpha_n^2}\ .
\end{aligned}
$$

The last inequality follows from $\rho_n \le 1$ and $2s_n \le 1$. Taking the expectation on both sides of this last inequality gives the lemma. $\qquad\square$

**Lemma 9.** *Let $\mathbb{E}_G$ denotes the expectation w.r.t. $(\mathbf{Z},\widetilde{\mathbf{Z}},\mathbf{V},\mathbf{W}^*,\mathbf{\Phi})$ only. Under the assumptions of Theorem 1, we have for all $(t,\epsilon) \in [0,1]\times\mathcal{B}_n$:*

$$
\mathbb{E}\Big[\Big(\frac{\mathbb{E}_{\mathbf{z},\tilde{\mathbf{z}}}\ln\widehat{\mathcal{Z}}_{t,\epsilon}}{m_n} - \frac{\mathbb{E}_G\ln\widehat{\mathcal{Z}}_{t,\epsilon}}{m_n}\Big)^2\Big] \le \frac{C_4}{n\alpha_n\rho_n}\ . \tag{155}
$$

*where $C_4 := \big(4\big\|\frac{\varphi}{\sqrt{\Delta}}\big\|_\infty + 2\sqrt{\frac{2}{\pi}}\big)^2 (4 + S^2)\big\|\frac{\partial_x\varphi}{\sqrt{\Delta}}\big\|_\infty^2$.*

*Proof.* In this proof we see $g = \mathbb{E}_{\mathbf{z},\tilde{\mathbf{z}}}\ln\widehat{\mathcal{Z}}_{t,\epsilon}/m_n$ as a function of $\mathbf{V},\mathbf{W}^*,\mathbf{\Phi}$ and we work conditionally on $\mathbf{A},\mathbf{X}^*$. Once again each partial derivative has the form $\partial_u g = m_n^{-1}\langle\partial_u\widehat{\mathcal{H}}_{t,\epsilon}\rangle_{t,\epsilon}$. We first compute the partial derivatives of $g$ w.r.t. $\{V_\mu\}_{\mu=1}^{m_n}$:

$$
\begin{aligned}
\Big|\frac{\partial g}{\partial V_\mu}\Big| &= m_n^{-1}\Big|\mathbb{E}_{\mathbf{z},\tilde{\mathbf{z}}}\Big\langle(\Gamma_\mu^{(t,\epsilon)} + Z_\mu)\frac{\partial\Gamma_\mu^{(t,\epsilon)}}{\partial V_\mu}\Big\rangle_{t,\epsilon}\Big| \le m_n^{-1}\mathbb{E}_{\mathbf{z},\tilde{\mathbf{z}}}\Big[\Big(\big(2\big\|\frac{\varphi}{\sqrt{\Delta}}\big\|_\infty + |Z_\mu|\big)2\sqrt{2}\big\|\frac{\partial_x\varphi}{\sqrt{\Delta}}\big\|_\infty\Big]\\
&= m_n^{-1}\Big(4\big\|\frac{\varphi}{\sqrt{\Delta}}\big\|_\infty + 2\sqrt{\frac{2}{\pi}}\Big)\sqrt{2}\big\|\frac{\partial_x\varphi}{\sqrt{\Delta}}\big\|_\infty\ .
\end{aligned}
$$

The same inequality holds for $|\frac{\partial g}{\partial W_\mu^*}|$. To compute the derivative w.r.t. $\Phi_{\mu i}$, we first remark that:

$$
\frac{\partial\Gamma_\mu^{(t,\epsilon)}}{\partial\Phi_{\mu i}} = \sqrt{\frac{1-t}{\Delta k_n}}\Big\{X_i^*\,\partial_x\varphi\Big(\sqrt{\frac{1-t}{k_n}}(\mathbf{\Phi X}^*)_\mu + k_1 V_\mu + k_2 W_\mu^*, \mathbf{A}_\mu\Big)\\
- x_i\,\partial_x\varphi\Big(\sqrt{\frac{1-t}{k_n}}(\mathbf{\Phi x})_\mu + k_1 V_\mu + k_2 w_\mu, \mathbf{a}_\mu\Big)\Big\}\ .
$$

Therefore:

$$
\begin{aligned}
\Big|\frac{\partial g}{\partial\Phi_{\mu i}}\Big| &= m_n^{-1}\Big|\mathbb{E}_{\mathbf{z},\tilde{\mathbf{z}}}\Big\langle(\Gamma_\mu^{(t,\epsilon)} + Z_\mu)\frac{\partial\Gamma_\mu^{(t,\epsilon)}}{\partial\Phi_{\mu i}}\Big\rangle_{t,\epsilon}\Big| \le \frac{1}{m_n\sqrt{k_n}}\mathbb{E}_{\mathbf{z},\tilde{\mathbf{z}}}\Big[\Big(2\big\|\frac{\varphi}{\sqrt{\Delta}}\big\|_\infty + |Z_\mu|\big)2S\big\|\frac{\partial_x\varphi}{\sqrt{\Delta}}\big\|_\infty\Big]\\
&= \frac{1}{m_n\sqrt{k_n}}\Big(4\big\|\frac{\varphi}{\sqrt{\Delta}}\big\|_\infty + 2\sqrt{\frac{2}{\pi}}\Big)S\big\|\frac{\partial_x\varphi}{\sqrt{\Delta}}\big\|_\infty\ .
\end{aligned}
$$

Putting together these inequalities on the partial derivatives of $g$, we find:

$$
\begin{aligned}
\|\nabla g\|^2 &= \sum_{\mu=1}^{m_n}\Big|\frac{\partial g}{\partial V_\mu}\Big|^2 + \sum_{\mu=1}^{m_n}\Big|\frac{\partial g}{\partial W_\mu^*}\Big|^2 + \sum_{\mu=1}^{m_n}\sum_{i=1}^{n}\Big|\frac{\partial g}{\partial\Phi_{\mu i}}\Big|^2\\
&\le \frac{4}{m_n}\Big(4\big\|\frac{\varphi}{\sqrt{\Delta}}\big\|_\infty + 2\sqrt{\frac{2}{\pi}}\Big)^2\big\|\frac{\partial_x\varphi}{\sqrt{\Delta}}\big\|_\infty^2 + \frac{1}{m_n\rho_n}\Big(4\big\|\frac{\varphi}{\sqrt{\Delta}}\big\|_\infty + 2\sqrt{\frac{2}{\pi}}\Big)^2 S^2\big\|\frac{\partial_x\varphi}{\sqrt{\Delta}}\big\|_\infty^2\\
&\le \frac{1}{m_n\rho_n}\Big(4\big\|\frac{\varphi}{\sqrt{\Delta}}\big\|_\infty + 2\sqrt{\frac{2}{\pi}}\Big)^2 (4 + S^2)\big\|\frac{\partial_x\varphi}{\sqrt{\Delta}}\big\|_\infty^2
\end{aligned}
$$

In the last inequality we used that $\rho_n \le 1$. To end the proof of the lemma it remains to apply Proposition 10 as we did in Lemma 8. $\qquad\square$

**Concentration with respect to the random stream** We now apply the variance bound of Proposition 11 to show that $\mathbb{E}_G\ln\widehat{\mathcal{Z}}_{t,\epsilon}/m_n$ concentrates w.r.t. $\mathbf{A}$.

**Lemma 10.** *Let $\mathbb{E}_{\mathbf{A}}$ denotes the expectation w.r.t. $\mathbf{A}$ only. Under the assumptions of Theorem 1, we have for all $(t, \epsilon) \in [0, 1] \times \mathcal{B}_n$:*

$$\mathbb{E}\left[\left(\frac{\mathbb{E}_G \ln \widehat{\mathcal{Z}}_{t,\epsilon}}{m_n} - \frac{\mathbb{E}_{G,\mathbf{A}} \ln \widehat{\mathcal{Z}}_{t,\epsilon}}{m_n}\right)^2\right] \leq \frac{C_5}{n\alpha_n} . \tag{156}$$

*where $C_5 := \left(2\left\|\frac{\varphi}{\sqrt{\Delta}}\right\|_{\infty} + \sqrt{\frac{2}{\pi}}\right)^2 \left\|\frac{\varphi}{\sqrt{\Delta}}\right\|_{\infty}^2$.*

*Proof.* We see $g = \mathbb{E}_G \ln \widehat{\mathcal{Z}}_{t,\epsilon}/m_n$ as a function of $\mathbf{A}$ only. Let $\nu \in \{1, \ldots, m_n\}$. We want to estimate the difference $g(\mathbf{A}) - g(\mathbf{A}^{(\nu)})$ corresponding to two configurations $\mathbf{A}$ and $\mathbf{A}^{(\nu)}$ such that $A_\mu^{(\nu)} = A_\mu$ for $\mu \neq \nu$ and $A_\nu^{(\nu)} \sim P_A$ independently of everything else. We will denote $\widehat{\mathcal{H}}_{t,\epsilon}^{(\nu)}$ and $\Gamma_\mu^{(t,\epsilon)(\nu)}$ the quantities $\widehat{\mathcal{H}}_{t,\epsilon}$ and $\Gamma_\mu^{(t,\epsilon)}$ when $\mathbf{A}$ is replaced by $\mathbf{A}^{(\nu)}$. By Jensen's inequality, we have:

$$\frac{1}{m_n}\mathbb{E}_G\langle\widehat{\mathcal{H}}_{t,\epsilon}^{(\nu)} - \widehat{\mathcal{H}}_{t,\epsilon}\rangle_{t,\epsilon}^{(\nu)} \leq g(\mathbf{A}) - g(\mathbf{A}^{(\nu)}) \leq \frac{1}{m_n}\mathbb{E}_G\langle\widehat{\mathcal{H}}_{t,\epsilon}^{(\nu)} - \widehat{\mathcal{H}}_{t,\epsilon}\rangle_{t,\epsilon} \tag{157}$$

where the angular brackets $\langle - \rangle_{t,\epsilon}$ and $\langle - \rangle_{t,\epsilon}^{(\nu)}$ denote expectation with respect to the distributions $\propto dP_{0,n}(\mathbf{x})\mathcal{D}\mathbf{w}dP_A(\mathbf{a}_\mu)\,e^{-\widehat{\mathcal{H}}_{t,\epsilon}(\mathbf{x},\mathbf{w},\mathbf{a})}$ and $\propto dP_{0,n}(\mathbf{x})\mathcal{D}\mathbf{w}dP_A(\mathbf{a}_\mu)\,e^{-\widehat{\mathcal{H}}_{t,\epsilon}^{(\nu)}(\mathbf{x},\mathbf{w},\mathbf{a})}$, respectively. From the definition (149) of $\widehat{\mathcal{H}}_{t,\epsilon}$,

$$\widehat{\mathcal{H}}_{t,\epsilon}^{(\nu)} - \widehat{\mathcal{H}}_{t,\epsilon} = \frac{1}{2}\left(\left(\Gamma_\nu^{(t,\epsilon)(\nu)}\right)^2 - \left(\Gamma_\nu^{(t,\epsilon)}\right)^2 + 2Z_\nu\left(\Gamma_\nu^{(t,\epsilon)(\nu)} - \Gamma_\nu^{(t,\epsilon)}\right)\right) .$$

Note that:

$$\left|\left(\Gamma_\nu^{(t,\epsilon)(\nu)}\right)^2 - \left(\Gamma_\nu^{(t,\epsilon)}\right)^2 + 2Z_\nu\left(\Gamma_\nu^{(t,\epsilon)(\nu)} - \Gamma_\nu^{(t,\epsilon)}\right)\right| \leq 8\left\|\frac{\varphi}{\sqrt{\Delta}}\right\|_{\infty}^2 + 4|Z_\nu|\left\|\frac{\varphi}{\sqrt{\Delta}}\right\|_{\infty} .$$

We thus conclude that $g$ satisfies the bounded difference property:

$$\forall \nu \in \{1, \ldots, m_n\} : \left|g(\mathbf{A}) - g(\mathbf{A}^{(\nu)})\right| \leq \frac{2}{m_n}\left(2\left\|\frac{\varphi}{\sqrt{\Delta}}\right\|_{\infty} + \sqrt{\frac{2}{\pi}}\right)\left\|\frac{\varphi}{\sqrt{\Delta}}\right\|_{\infty} . \tag{158}$$

To end the proof of Lemma 10, we just need to apply Proposition 11. $\qquad\square$

**Concentration with respect to the signal**  Let $\mathbb{E}_{\sim\mathbf{X}^*} \equiv \mathbb{E}_{\mathbf{A},G}$ denote the expectation w.r.t. all quenched variables except $\mathbf{X}^*$. It remains to bound the variance of $\mathbb{E}_{\sim\mathbf{X}^*} \ln \widehat{\mathcal{Z}}_{t,\epsilon}/m_n$ (which only depends on $\mathbf{X}^*$).

**Lemma 11.** *Under the assumptions of Theorem 1, we have for all $(t, \epsilon) \in [0, 1] \times \mathcal{B}_n$:*

$$\mathbb{E}\left[\left(\frac{\mathbb{E}[\ln \widehat{\mathcal{Z}}_{t,\epsilon}|\mathbf{X}^*]}{m_n} - \frac{\mathbb{E}\ln \widehat{\mathcal{Z}}_{t,\epsilon}}{m_n}\right)^2\right] \leq \frac{C_6}{n\rho_n} + \frac{C_7\rho_n}{n\alpha_n^2}$$

*where $C_7 := 8S^2$ and*

$$C_6 := 8S^2\left(3\left\|\frac{\partial_x\varphi}{\sqrt{\Delta}}\right\|_{\infty}^2 + \left\|\frac{\varphi}{\sqrt{\Delta}}\right\|_{\infty}\left\|\frac{\partial_{xx}\varphi}{\sqrt{\Delta}}\right\|_{\infty} + 12\left\|\frac{\partial_x\varphi}{\sqrt{\Delta}}\right\|_{\infty}^2\left\|\frac{\varphi}{\sqrt{\Delta}}\right\|_{\infty}^2 + 2\sqrt{\frac{2}{\pi}}\left\|\frac{\varphi}{\sqrt{\Delta}}\right\|_{\infty}\left\|\frac{\partial_x\varphi}{\sqrt{\Delta}}\right\|_{\infty}^2\right)^2 .$$

*Proof.* $g = \mathbb{E}[\ln \widehat{\mathcal{Z}}_{t,\epsilon}|\mathbf{X}^*]/m_n$ is a function of $\mathbf{X}^*$. For $j \in \{1, \ldots, n\}$, we have:

$$\frac{\partial g}{\partial X_j^*} = -\frac{1}{m_n}\mathbb{E}\left[\left\langle\frac{\partial\widehat{\mathcal{H}}_{t,\epsilon}}{\partial X_j^*}\right\rangle_{n,t,\epsilon}\bigg|\mathbf{X}^*\right]$$

$$= -\frac{1}{m_n}\sqrt{\frac{1-t}{\Delta k_n}}\sum_{\mu=1}^{m_n}\mathbb{E}\left[\Phi_{\mu j}\partial_x\varphi(S_\mu^{(t,\epsilon)}, \mathbf{A}_\mu)\left(\langle\Gamma_\mu^{(t,\epsilon)}\rangle_{n,t,\epsilon} + Z_\mu\right)\bigg|\mathbf{X}^*\right]$$

$$+ \frac{1}{m_n}\mathbb{E}\left[\langle R_1(t,\epsilon)(X_j^* - x_j) + \sqrt{R_1(t,\epsilon)}\widetilde{Z}_j\rangle_{n,t,\epsilon}\bigg|\mathbf{X}^*\right]$$

$$= -\frac{1}{m_n}\sqrt{\frac{1-t}{\Delta k_n}}\sum_{\mu=1}^{m_n}\mathbb{E}\left[\Phi_{\mu j}\partial_x\varphi(S_\mu^{(t,\epsilon)}, \mathbf{A}_\mu)\langle\Gamma_\mu^{(t,\epsilon)}\rangle_{n,t,\epsilon}\bigg|\mathbf{X}^*\right]$$

$$+ \frac{R_1(t,\epsilon)}{m_n}\mathbb{E}\left[(X_j^* - \langle x_j\rangle_{n,t,\epsilon})\big|\mathbf{X}^*\right] \tag{159}$$

To get the last equality we use $\mathbb{E}[\Phi_{\mu j}\partial_x\varphi(S_\mu^{(t,\epsilon)},\mathbf{A}_\mu)Z_\mu|\mathbf{X}^*]=\mathbb{E}[\Phi_{\mu j}\partial_x\varphi(S_\mu^{(t,\epsilon)},\mathbf{A}_\mu)|\mathbf{X}^*]\mathbb{E}[Z_\mu]=0$ and $\mathbb{E}[\sqrt{R_1(t,\epsilon)}\widetilde{Z}_j|\mathbf{X}^*]=0$. An integration by parts with respect to $\Phi_{\mu j}$ yields:

$$\mathbb{E}\Big[\Phi_{\mu j}\partial_x\varphi(S_\mu^{(t,\epsilon)},\mathbf{A}_\mu)\langle\Gamma_\mu^{(t,\epsilon)}\rangle_{n,t,\epsilon}\Big|\mathbf{X}^*\Big]$$

$$=\sqrt{\frac{1-t}{k_n\Delta}}\mathbb{E}\Big[X_j^*(\partial_x\varphi^2+\varphi\,\partial_{xx}\varphi)(S_\mu^{(t,\epsilon)},\mathbf{A}_\mu)\Big|\mathbf{X}^*\Big]$$

$$-\sqrt{\frac{1-t}{\Delta k_n}}\mathbb{E}\Big[X_j^*\partial_{xx}\varphi(S_\mu^{(t,\epsilon)},\mathbf{A}_\mu)\langle\varphi(s_\mu^{(t,\epsilon)},\mathbf{a}_\mu)\rangle_{n,t,\epsilon}\Big|\mathbf{X}^*\Big]$$

$$-\sqrt{\frac{1-t}{\Delta k_n}}\mathbb{E}\Big[\partial_x\varphi(S_\mu^{(t,\epsilon)},\mathbf{A}_\mu)\langle x_j\partial_x\varphi(s_\mu^{(t,\epsilon)},\mathbf{a}_\mu)\rangle_{n,t,\epsilon}\Big|\mathbf{X}^*\Big]$$

$$+\sqrt{\frac{1-t}{\Delta k_n}}\mathbb{E}\Big[\partial_x\varphi(S_\mu^{(t,\epsilon)},\mathbf{A}_\mu)\big\langle\varphi(s_\mu^{(t,\epsilon)},\mathbf{a}_\mu)$$

$$\big(X_j^*\partial_x\varphi(S_\mu^{(t,\epsilon)},\mathbf{A}_\mu)-x_j\partial_x\varphi(s_\mu^{(t,\epsilon)},\mathbf{a}_\mu)\big)(\Gamma_\mu^{(t,\epsilon)}+Z_\mu)\big\rangle_{n,t,\epsilon}\Big|\mathbf{X}^*\Big]$$

$$-\sqrt{\frac{1-t}{\Delta k_n}}\mathbb{E}\Big[\partial_x\varphi(S_\mu^{(t,\epsilon)},\mathbf{A}_\mu)\langle\varphi(s_\mu^{(t,\epsilon)},\mathbf{a}_\mu)\rangle_{n,t,\epsilon}$$

$$\big\langle\big(X_j^*\partial_x\varphi(S_\mu^{(t,\epsilon)},\mathbf{A}_\mu)-x_j\partial_x\varphi(s_\mu^{(t,\epsilon)},\mathbf{a}_\mu)\big)(\Gamma_\mu^{(t,\epsilon)}+Z_\mu)\big\rangle_{n,t,\epsilon}\Big|\mathbf{X}^*\Big]$$

It directly follows that: $\big|\mathbb{E}\big[\Phi_{\mu j}\partial_x\varphi(S_\mu^{(t,\epsilon)},\mathbf{A}_\mu)\langle\Gamma_\mu^{(t,\epsilon)}\rangle_{n,t,\epsilon}|\mathbf{X}^*\big]\big|\leq\sqrt{\frac{\Delta}{k_n}}\widetilde{C}_6$ where:

$$\widetilde{C}_6:=2S\Big(\Big\|\frac{\partial_x\varphi}{\sqrt{\Delta}}\Big\|_\infty^2+\Big\|\frac{\varphi}{\sqrt{\Delta}}\Big\|_\infty\Big\|\frac{\partial_{xx}\varphi}{\sqrt{\Delta}}\Big\|_\infty+4\Big\|\frac{\partial_x\varphi}{\sqrt{\Delta}}\Big\|_\infty^2\Big\|\frac{\varphi}{\sqrt{\Delta}}\Big\|_\infty^2+2\sqrt{\frac{2}{\pi}}\Big\|\frac{\varphi}{\sqrt{\Delta}}\Big\|_\infty\Big\|\frac{\partial_x\varphi}{\sqrt{\Delta}}\Big\|_\infty^2\Big)\,.$$

Making use of this upper bound, we obtain for all $j\in\{1,\ldots,n\}$:

$$\Big|\frac{\partial g}{\partial X_j^*}\Big|\leq\frac{\widetilde{C}_6}{k_n}+\frac{2SK_n}{m_n}=\frac{\widetilde{C}_6}{k_n}+\frac{2S}{m_n}\Big(2s_n+2\frac{\alpha_n}{\rho_n}C_1\Big(\Big\|\frac{\varphi}{\sqrt{\Delta}}\Big\|_\infty,\Big\|\frac{\partial_x\varphi}{\sqrt{\Delta}}\Big\|_\infty\Big)\Big)$$

$$=\frac{1}{n\rho_n}\Big(\widetilde{C}_6+4SC_1\Big(\Big\|\frac{\varphi}{\sqrt{\Delta}}\Big\|_\infty,\Big\|\frac{\partial_x\varphi}{\sqrt{\Delta}}\Big\|_\infty\Big)\Big)+\frac{2S}{n\alpha_n}\,. \tag{160}$$

For a fixed $j\in\{1,\ldots,n\}$, let $\mathbf{X}^{(j)}$ be a vector such that $X_i^{(j)}=X_i^*$ for $i\neq j$ and $X_j^{(j)}\sim P_{0,n}$ independently of everything else. By the mean-value theorem and thanks to (160), we have:

$$\mathbb{E}_{\mathbf{X}^*}\mathbb{E}_{X_j^{(j)}}\big[\big(g(\mathbf{X}^*)-g(\mathbf{X}^{*(j)})\big)^2\big]$$

$$\leq\Big(\frac{1}{n\rho_n}\Big(\widetilde{C}_6+4SC_1\Big(\Big\|\frac{\varphi}{\sqrt{\Delta}}\Big\|_\infty,\Big\|\frac{\partial_x\varphi}{\sqrt{\Delta}}\Big\|_\infty\Big)\Big)+\frac{2S}{n\alpha_n}\Big)^2\mathbb{E}\big[\big(X_j^*-X_j^{(j)}\big)^2\big]$$

$$\leq\frac{4}{n^2\rho_n}\Big(\widetilde{C}_6+4SC_1\Big(\Big\|\frac{\varphi}{\sqrt{\Delta}}\Big\|_\infty,\Big\|\frac{\partial_x\varphi}{\sqrt{\Delta}}\Big\|_\infty\Big)\Big)^2+\frac{16S^2\rho_n}{n^2\alpha_n^2}\,.$$

We used $\mathbb{E}\big[\big(X_j^*-X_j^{(j)}\big)^2\big]=2\rho_n\mathbb{E}_{X\sim P_0}[X^2]-2\rho_n^2\mathbb{E}_{X\sim P_0}[X]^2\leq 2\rho_n\mathbb{E}_{X\sim P_0}[X^2]=2\rho_n$ and Jensen's inequality $(a+b)^2\leq 2a^2+2b^2$ to get the last inequality. To end the proof it now suffices to apply Proposition 12. $\square$

**Proof of Proposition 9:** Combining Lemmas 8, 9, 10 and 11 yields:

$$\mathbb{V}\mathrm{ar}\Big(\frac{\ln\widehat{\mathcal{Z}}_{t,\epsilon}}{m_n}\Big)\leq\frac{C_2+C_4}{n\alpha_n\rho_n}+\frac{C_3+C_7\rho_n}{n\alpha_n^2}+\frac{C_5}{n\alpha_n}+\frac{C_6}{n\rho_n}\,. \tag{161}$$

Plugging (161) back in (150) gives:

$$\mathbb{V}\mathrm{ar}\Big(\frac{\ln\mathcal{Z}_{t,\epsilon}}{m_n}\Big)\leq\frac{C_2+C_4}{n\alpha_n\rho_n}+\frac{C_3+C_7\rho_n+1.5}{n\alpha_n^2}+\frac{C_5+1.5}{n\alpha_n}+\frac{C_6}{n\rho_n}$$

$$\leq\frac{C_2+C_4+C_5+1.5}{n\alpha_n\rho_n}+\frac{C_3+C_7+1.5}{n\alpha_n^2}+\frac{C_6}{n\rho_n}$$

$$=\frac{1}{n\alpha_n\rho_n}\Big(C_2+C_4+C_5+1.5+\frac{\rho_n}{\alpha_n}(C_3+C_7+1.5)+\alpha_n C_6\Big)\,. \tag{162}$$

The second inequality follows from $\rho_n\leq 1$. It ends the proof of Proposition 9.

# D Concentration of the overlap

In this appendix we prove Proposition 2. Define the average free entropy $f_{n,\epsilon}(t) := \frac{1}{m_n} \mathbb{E} \ln \mathcal{Z}_{t,\epsilon}$. In this section we think of it as a function of $R_1 = R_1(t,\epsilon)$ and $R_2 = R_2(t,\epsilon)$, i.e., $(R_1, R_2) \mapsto f_{n,\epsilon}(t)$. Similarly, we also view the free entropy for a realization of the quenched variables as a function

$$(R_1, R_2) \mapsto F_{n,\epsilon}(t) \equiv \frac{1}{m_n} \ln \mathcal{Z}_{t,\epsilon}(\mathbf{Y}_t, \mathbf{Y}'_t, \mathbf{\Phi}, \mathbf{V}) .$$

In this appendix, to lighten the notations, we drop the indices of the angular brackets $\langle - \rangle_{n,t,\epsilon}$ and simply write $\langle - \rangle$. We denote with $\cdot$ the scalar product between two vectors. We define:

$$\mathcal{L} := \frac{1}{k_n} \left( \frac{\|\mathbf{x}\|^2}{2} - \mathbf{x} \cdot \mathbf{X}^* - \frac{\mathbf{x} \cdot \widetilde{\mathbf{Z}}}{2\sqrt{R_1}} \right) .$$

The fluctuations of the overlap $Q := \frac{1}{k_n} \mathbf{X}^* \cdot \mathbf{x}$ and those of $\mathcal{L}$ are related through the inequality:

$$\frac{1}{4} \mathbb{E} \langle (Q - \mathbb{E}\langle Q \rangle)^2 \rangle \leq \mathbb{E} \langle (\mathcal{L} - \mathbb{E}\langle \mathcal{L} \rangle)^2 \rangle . \tag{163}$$

The proof of (163) is based on integrations by parts with respect to $\widetilde{Z}$ and a repeated use of the Nishimori identity (see Lemma 5). Proposition 2 is then a direct consequence of the following:

**Proposition 13** (Concentration of $\mathcal{L}$ on $\mathbb{E}\langle \mathcal{L} \rangle$)**.** *Suppose that $\Delta > 0$, that all of (H1), (H2), (H3) hold, that $\mathbb{E}_{X \sim P_0}[X^2] = 1$ and that the family of functions $(r_\epsilon)_{\epsilon \in \mathcal{B}_n}, (q_\epsilon)_{\epsilon \in \mathcal{B}_n}$ are regular. Further assume that there exist real positive numbers $M_\alpha, M_{\rho/\alpha}, m_{\rho/\alpha}$ such that $\forall n \in \mathbb{N}^*$:*

$$\alpha_n \leq M_\alpha \quad and \quad \frac{m_{\rho/\alpha}}{n} < \frac{\rho_n}{\alpha_n} \leq M_{\rho/\alpha} .$$

*Let $(s_n)_{n \in \mathbb{N}^*}$ be a sequence of real numbers in $(0, {}^1\!/_2]$. Define $\mathcal{B}_n := [s_n, 2s_n]^2$. We have $\forall t \in [0,1]$:*

$$\int_{\mathcal{B}_n} d\epsilon \, \mathbb{E} \langle (\mathcal{L} - \mathbb{E}\langle \mathcal{L} \rangle_{n,t,\epsilon})^2 \rangle_{n,t,\epsilon} \leq \frac{C}{\rho_n^2 \left( \frac{\rho_n n}{\alpha_n m_{\rho/\alpha}} \right)^{\frac{1}{3}} - \rho_n^2} , \tag{164}$$

*where $C$ is a polynomial in $\left( S, \left\| \frac{\varphi}{\sqrt{\Delta}} \right\|_\infty, \left\| \frac{\partial_x \varphi}{\sqrt{\Delta}} \right\|_\infty, \left\| \frac{\partial_{xx} \varphi}{\sqrt{\Delta}} \right\|_\infty, M_\alpha, M_{\rho/\alpha}, m_{\rho/\alpha} \right)$ with positive coefficients.*

Because $\mathbb{E}\langle (\mathcal{L} - \mathbb{E}\langle \mathcal{L} \rangle)^2 \rangle = \mathbb{E}\langle (\mathcal{L} - \langle \mathcal{L} \rangle)^2 \rangle + \mathbb{E}[(\langle \mathcal{L} \rangle - \mathbb{E}\langle \mathcal{L} \rangle)^2]$, Proposition 2 follows directly from the next two lemmas.

**Lemma 12** (Concentration of $\mathcal{L}$ on $\langle \mathcal{L} \rangle$)**.** *Under the assumptions of Proposition 13, $\forall t \in [0,1]$:*

$$\int_{\mathcal{B}_n} d\epsilon \, \mathbb{E} \langle (\mathcal{L} - \langle \mathcal{L} \rangle_{n,t,\epsilon})^2 \rangle_{n,t,\epsilon} \leq \frac{1}{n \rho_n} .$$

The second lemma states that $\mathcal{L}$ concentrates w.r.t. the realizations of quenched disorder variables. It is a consequence of the concentration of the free entropy (see Proposition 9 in Appendix C).

**Lemma 13** (Concentration of $\langle \mathcal{L} \rangle$ on $\mathbb{E}\langle \mathcal{L} \rangle$)**.** *Under the assumptions of Proposition 2, $\forall t \in [0,1]$:*

$$\int_{\mathcal{B}_n} d\epsilon \, \mathbb{E}[(\langle \mathcal{L} \rangle_{n,t,\epsilon} - \mathbb{E}\langle \mathcal{L} \rangle_{n,t,\epsilon})^2] \leq \frac{C}{\rho_n^2 \left( \frac{\rho_n n}{\alpha_n m_{\rho/\alpha}} \right)^{\frac{1}{3}} - \rho_n^2} , \tag{165}$$

*where $C$ is a polynomial in $\left( S, \left\| \frac{\varphi}{\sqrt{\Delta}} \right\|_\infty, \left\| \frac{\partial_x \varphi}{\sqrt{\Delta}} \right\|_\infty, \left\| \frac{\partial_{xx} \varphi}{\sqrt{\Delta}} \right\|_\infty, M_\alpha, M_{\rho/\alpha}, m_{\rho/\alpha} \right)$ with positive coefficients.*

We now turn to the proof of Lemmas 12 and 13. The main ingredient will be a set of formulas for the first two partial derivatives of the free entropy w.r.t. $R_1 = R_1(t,\epsilon)$. For any given realisation of the quenched disorder:

$$\frac{dF_{n,\epsilon}(t)}{dR_1} = -\frac{\rho_n}{\alpha_n} \langle \mathcal{L} \rangle - \frac{1}{2m_n} \left( \|\mathbf{X}^*\|^2 + \frac{\mathbf{X}^* \cdot \widetilde{\mathbf{Z}}}{\sqrt{R_1}} \right) , \tag{166}$$

$$\frac{1}{m_n} \frac{d^2 F_{n,\epsilon}(t)}{dR_1^2} = \left( \frac{\rho_n}{\alpha_n} \right)^2 (\langle \mathcal{L}^2 \rangle - \langle \mathcal{L} \rangle^2) + \frac{1}{4m_n^2 R_1^{3/2}} \widetilde{\mathbf{Z}} \cdot (\mathbf{X}^* - \langle \mathbf{x} \rangle) . \tag{167}$$

Averaging (166) yields:

$$\frac{df_{n,\epsilon}(t)}{dR_1} = -\frac{\rho_n}{\alpha_n} \left( \mathbb{E}\langle \mathcal{L} \rangle + \frac{1}{2} \right) = \frac{\rho_n}{2\alpha_n} \left( \frac{\mathbb{E}\|\langle \mathbf{x} \rangle\|^2}{k_n} - 1 \right) . \tag{168}$$

To obtain the second equality we simplified $\mathbb{E}\langle \mathcal{L} \rangle$ by using an integration by parts w.r.t. the standard Gaussian random vector $\widetilde{\mathbf{Z}}$ and $\mathbb{E}\langle \mathbf{x} \cdot \mathbf{X}^* \rangle = \mathbb{E}\|\langle \mathbf{x} \rangle\|^2$ (by Nishimori identity, see Lemma 5). Averaging (167) and integrating by parts w.r.t. the standard Gaussian random vector $\widetilde{\mathbf{Z}}$ gives:

$$\frac{1}{m_n} \frac{d^2 f_{n,\epsilon}(t)}{dR_1^2} = \left( \frac{\rho_n}{\alpha_n} \right)^2 \mathbb{E}[\langle \mathcal{L}^2 \rangle - \langle \mathcal{L} \rangle^2] - \frac{1}{4m_n^2 R_1} \mathbb{E}[\langle \|\mathbf{x}\|^2 \rangle - \|\langle \mathbf{x} \rangle\|^2] . \tag{169}$$

**Proof of Lemma 12**  From (169) we have:

$$\mathbb{E}\langle(\mathcal{L}-\langle\mathcal{L}\rangle)^2\rangle = \left(\frac{\alpha_n}{\rho_n}\right)^2 \frac{1}{m_n}\frac{d^2 f_{n,\epsilon}(t)}{dR_1^2} + \left(\frac{\alpha_n}{\rho_n}\right)^2 \frac{1}{4m_n^2 R_1}\mathbb{E}\big[\langle\|\mathbf{x}\|^2\rangle - \|\langle\mathbf{x}\rangle\|^2\big]$$

$$\leq \frac{\alpha_n}{\rho_n^2 n}\frac{d^2 f_{n,\epsilon}(t)}{dR_1^2} + \frac{1}{4\epsilon_1 n\rho_n} , \tag{170}$$

where we used $\mathbb{E}\langle\|\mathbf{x}\|^2\rangle = \mathbb{E}\|\mathbf{X}^*\|^2 = n\rho_n$ by the Nishimori identity and $R_1 \geq \epsilon_1$. Recall $\mathcal{B}_n := [s_n, 2s_n]^2$. By assumption the families of functions $(q_\epsilon)_{\epsilon\in\mathcal{B}_n}$ and $(r_\epsilon)_{\epsilon\in\mathcal{B}_n}$ are regular. Therefore, $R^t : (\epsilon_1, \epsilon_2) \mapsto (R_1(t,\epsilon), R_2(t,\epsilon))$ is a $\mathcal{C}^1$-diffeomorphism whose Jacobian determinant $|J_{R^t}|$ satisfies $\forall \epsilon \in \mathcal{B}_n : |J_{R^t}(\epsilon)| \geq 1$. Integrating (170) over $\epsilon \in \mathcal{B}_n$ yields:

$$\int_{\mathcal{B}_n} d\epsilon\, \mathbb{E}\langle(\mathcal{L}-\langle\mathcal{L}\rangle)^2\rangle \leq \frac{\alpha_n}{\rho_n^2 n}\int_{R^t(\mathcal{B}_n)} \frac{dR_1 dR_2}{|J_{R^t}((R^t)^{-1}(R_1,R_2))|}\frac{d^2 f_{n,\epsilon}(t)}{dR_1^2} + \frac{1}{4n\rho_n}\int_{\mathcal{B}_n}\frac{d\epsilon_1}{\epsilon_1}d\epsilon_2$$

$$\leq \frac{\alpha_n}{\rho_n^2 n}\int_{R^t(\mathcal{B}_n)} dR_1 dR_2\,\frac{d^2 f_{n,\epsilon}(t)}{dR_1^2} + \frac{s_n}{4n\rho_n}\ln 2 . \tag{171}$$

Note that $R^t(\mathcal{B}_n) \subset \big[s_n, 2s_n + \frac{\alpha_n}{\rho_n}r_{\max}\big] \times [s_n, 2s_n + 1]$ (by definition of the interpolation functions). Thus:

$$\int_{\mathcal{B}_n} d\epsilon\, \mathbb{E}\langle(\mathcal{L}-\langle\mathcal{L}\rangle)^2\rangle \leq \frac{\alpha_n}{\rho_n^2 n}\int_{s_n}^{2s_n+1} dR_2 \left[\frac{df_{n,\epsilon}(t)}{dR_1}\right]_{R_1=s_n}^{2s_n+\frac{\alpha_n}{\rho_n}r_{\max}} + \frac{s_n}{4n\rho_n}\ln 2$$

$$\leq \frac{1+s_n}{2\rho_n n} + \frac{s_n}{4n\rho_n}\ln 2 \leq \frac{1}{n\rho_n} . \tag{172}$$

The last inequality follows from $s_n \leq 1/2$ and $(\ln 2)/2 < 1$. To obtain the second inequality we bounded the partial derivative of the free entropy using (168) and $\mathbb{E}\|\langle\mathbf{x}\rangle\|^2 \leq \mathbb{E}\langle\|\mathbf{x}\|^2\rangle = n\rho_n$ (again by the Nishimori identity):

$$\left|\frac{df_{n,\epsilon}(t)}{dR_1}\right| = -\frac{df_{n,\epsilon}(t)}{dR_1} = \frac{\rho_n}{2\alpha_n}\left(1 - \frac{\mathbb{E}\|\langle\mathbf{x}\rangle\|^2}{k_n}\right) \leq \frac{\rho_n}{2\alpha_n} . \tag{173}$$

∎

**Proof of Lemma 13**  We define the two functions:

$$\widetilde{F}(R_1) := F_{n,\epsilon}(t) - \frac{\sqrt{R_1}}{m_n}2S\sum_{i=1}^n |\widetilde{Z}_i| \quad , \quad \widetilde{f}(R_1) := \mathbb{E}\widetilde{F}(R_1) = f_{n,\epsilon}(t) - \frac{\sqrt{R_1}}{\alpha_n}2S\,\mathbb{E}|\widetilde{Z}_1| . \tag{174}$$

Because of (167), we see that the second derivative of $\widetilde{F}(R_1)$ is positive so that it is convex. Without the extra term $F_{n,\epsilon}(t)$ is not necessarily convex in $R_1$, although $f_{n,\epsilon}(t)$ is (it can be shown easily). Note that $\widetilde{f}(R_1)$ is convex too. Convexity allows us to use the following standard lemma:

**Lemma 14** (A convexity bound). *Let $G$ and $g$ be two convex functions. Let $\delta > 0$ and define $C_\delta(x) \equiv g'(x+\delta) - g'(x-\delta) \geq 0$. Then:*

$$|G'(x) - g'(x)| \leq \delta^{-1} \sum_{u\in\{x-\delta, x, x+\delta\}} |G(u) - g(u)| + C_\delta(x) .$$

Define $A := \frac{1}{m_n}\sum_{i=1}^n |\widetilde{Z}_i| - \mathbb{E}|\widetilde{Z}_i|$. From (174), we directly obtain:

$$\widetilde{F}(R_1) - \widetilde{f}(R_1) = F_{n,\epsilon}(t) - f_{n,\epsilon}(t) - \sqrt{R_1}2SA . \tag{175}$$

Thanks to (166) and (168) the difference of derivatives (w.r.t. $R_1$) reads:

$$\widetilde{F}'(R_1) - \widetilde{f}'(R_1) = \frac{\rho_n}{\alpha_n}\big(\mathbb{E}\langle\mathcal{L}\rangle - \langle\mathcal{L}\rangle\big) - \frac{\rho_n}{2\alpha_n}\left(\frac{\|\mathbf{X}^*\|^2}{k_n} - 1 + \frac{\mathbf{X}^*\cdot\widetilde{\mathbf{Z}}}{k_n\sqrt{R_1}}\right) - \frac{SA}{\sqrt{R_1}} . \tag{176}$$

Let $\delta \in (0, s_n)$. Define $C_\delta(R_1) := \widetilde{f}'(R_1 + \delta) - \widetilde{f}'(R_1 - \delta) \geq 0$ (this is well-defined because $\delta < s_n \leq R_1$). Combining (175) and (176) with Lemma 14 gives:

$$\frac{\rho_n}{\alpha_n}\big|\langle\mathcal{L}\rangle - \mathbb{E}\langle\mathcal{L}\rangle\big| \leq \delta^{-1} \sum_{u\in\{R_1-\delta, R_1, R_1+\delta\}} \big|\big(F_{n,\epsilon}(t) - f_{n,\epsilon}(t)\big)_{R_1=u}\big| + 2S|A|\sqrt{u}$$

$$+ C_\delta(R_1) + \frac{S|A|}{\sqrt{R_1}} + \frac{\rho_n}{2\alpha_n}\left|\frac{\|\mathbf{X}^*\|^2}{k_n} - 1 + \frac{\mathbf{X}^*\cdot\widetilde{\mathbf{Z}}}{k_n\sqrt{R_1}}\right|$$

$$\leq \delta^{-1} \sum_{u\in\{R_1-\delta, R_1, R_1+\delta\}} \big|\big(F_{n,\epsilon}(t) - f_{n,\epsilon}(t)\big)_{R_1=u}\big|$$

$$+ C_\delta(R_1) + S|A|\left(\frac{1}{\sqrt{R_1}} + \frac{6\sqrt{R_1}}{\delta}\right) + \frac{\rho_n}{2\alpha_n}\left|\frac{\|\mathbf{X}^*\|^2}{k_n} - 1 + \frac{\mathbf{X}^*\cdot\widetilde{\mathbf{Z}}}{k_n\sqrt{R_1}}\right| . \tag{177}$$

The last inequality follows from $\sqrt{R_1 + \delta} + \sqrt{R_1 - \delta} \leq 2\sqrt{R_1}$. Taking the square and then the expectation on both sides of the inequality (177), and making use of $\left(\sum_{i=1}^{6} v_i\right)^2 \leq 6 \sum_{i=1}^{6} v_i^2$ (by convexity) yields:

$$\mathbb{E}\left[\left(\langle \mathcal{L}\rangle - \mathbb{E}\langle\mathcal{L}\rangle\right)^2\right] \leq \frac{6}{\delta^2}\left(\frac{\alpha_n}{\rho_n}\right)^2 \sum_{u \in \{R_1 - \delta, R_1, R_1 + \delta\}} \mathbb{V}\mathrm{ar}\left(F_{n,\epsilon}(t)\big|_{R_1 = u}\right) + 6\left(\frac{\alpha_n C_\delta(R_1)}{\rho_n}\right)^2$$
$$+ 6\left(\frac{\alpha_n}{\rho_n}\right)^2 S^2 \mathbb{E}[A^2]\left(\frac{1}{R_1} + \frac{12}{\delta} + \frac{36R_1}{\delta^2}\right) + \frac{3}{2}\mathbb{V}\mathrm{ar}\left(\frac{\|\mathbf{X}^*\|^2}{k_n} + \frac{\mathbf{X}^* \cdot \widetilde{\mathbf{Z}}}{k_n\sqrt{R_1}}\right). \quad (178)$$

By Proposition 9, under our assumptions, the free entropy $F_{n,\epsilon}(t) = {}^{\ln \mathcal{Z}_{t,\epsilon}}/m_n$ concentrates such that:

$$\mathbb{V}\mathrm{ar}\left(F_{n,\epsilon}(t)\right) \leq \frac{C}{n\alpha_n\rho_n} \quad (179)$$

where $C$ is a polynomial in $\left(S, \left\|\frac{\varphi}{\sqrt{\Delta}}\right\|_\infty, \left\|\frac{\partial_x\varphi}{\sqrt{\Delta}}\right\|_\infty, \left\|\frac{\partial_{xx}\varphi}{\sqrt{\Delta}}\right\|_\infty\right)$ with positive coefficients. Remark that, by independence of the noise variables, we have:

$$\mathbb{E}[A^2] \leq \frac{1 - 2/\pi}{n\alpha_n^2} < \frac{1}{n\alpha_n^2} \ . \quad (180)$$

Also, the last term on the right hand side of (178) satisfies:

$$\mathbb{V}\mathrm{ar}\left(\frac{\|\mathbf{X}^*\|^2}{k_n} + \frac{\mathbf{X}^* \cdot \widetilde{\mathbf{Z}}}{k_n\sqrt{R_1}}\right) = \mathbb{V}\mathrm{ar}\left(\frac{\|\mathbf{X}^*\|^2}{k_n}\right) + \mathbb{V}\mathrm{ar}\left(\frac{\mathbf{X}^* \cdot \widetilde{\mathbf{Z}}}{k_n\sqrt{R_1}}\right) = \frac{n}{k_n^2}\mathbb{V}\mathrm{ar}\left((X_1^*)^2\right) + \frac{n}{k_n^2 R_1}\mathbb{V}\mathrm{ar}\left(X_1^* \widetilde{Z}_1\right)$$
$$\leq \frac{S^4}{n\rho_n} + \frac{1}{n\rho_n R_1} \ . \quad (181)$$

Plugging (179), (180) and (181) back in (178) yields

$$\mathbb{E}\left[\left(\langle \mathcal{L}\rangle - \mathbb{E}\langle\mathcal{L}\rangle\right)^2\right] \leq \frac{18C\alpha_n}{n\rho_n^3\delta^2} + 6\left(\frac{\alpha_n C_\delta(R_1)}{\rho_n}\right)^2 + \frac{6S^2}{n\rho_n^2}\left(\frac{1}{R_1} + \frac{12}{\delta} + \frac{36R_1}{\delta^2}\right) + \frac{3S^4}{2n\rho_n} + \frac{3}{2n\rho_n R_1}$$
$$\leq \frac{18C\alpha_n}{n\rho_n^3\delta^2} + 6\left(\frac{\alpha_n C_\delta(R_1)}{\rho_n}\right)^2 + \frac{294S^2}{n\rho_n^2}\frac{R_1}{\delta^2} + \frac{3}{2n\rho_n}\left(S^4 + \frac{1}{R_1}\right), \quad (182)$$

where the last inequality follows from $R_1^{-1} \leq \delta^{-1} \leq {}^{R_1}/\delta^2$.

The next step is to integrate both sides of (182) over $\mathcal{B}_n := [s_n, 2s_n]^2$. By assumption, the families of functions $(q_\epsilon)_{\epsilon \in \mathcal{B}_n}$ and $(r_\epsilon)_{\epsilon \in \mathcal{B}_n}$ are regular. Therefore, $R^t : (\epsilon_1, \epsilon_2) \mapsto (R_1(t, \epsilon), R_2(t, \epsilon))$ is a $\mathcal{C}^1$-diffeomorphism whose Jacobian determinant $|J_{R^t}|$ satisfies $\forall \epsilon \in \mathcal{B}_n : |J_{R^t}(\epsilon)| \geq 1$. Besides, $R^t(\mathcal{B}_n) \subseteq [s_n, K_n] \times [s_n, 2s_n + 1]$ where $K_n := 2s_n + \frac{\alpha_n}{\rho_n}r_{\max}$. Therefore,

$$\int_{\mathcal{B}_n} d\epsilon \frac{294S^2}{n\rho_n^2}\frac{R_1(t,\epsilon)}{\delta^2} \leq \frac{294S^2}{n\rho_n^2}\int_{\mathcal{B}_n} d\epsilon\frac{K_n}{\delta^2} = \frac{294S^2}{n\rho_n^2}\frac{K_n s_n^2}{\delta^2} \leq 294S^2\left(M_{\rho/\alpha} + r_{\max}\right)\frac{\alpha_n s_n^2}{n\rho_n^3\delta^2} \ , \quad (183)$$

where we use that $K_n = \left({}^{2s_n\rho_n}/\alpha_n + r_{\max}\right)\left({}^{\alpha_n}/\rho_n\right) \leq (M_{\rho/\alpha} + r_{\max})\left({}^{\alpha_n}/\rho_n\right)$ as $s_n \leq {}^{1}/2$ and ${}^{\rho_n}/\alpha_n \leq M_{\rho/\alpha}$. We now upper bound the integral of $\left({}^{\alpha_n C_\delta(R_1)}/\rho_n\right)^2$. Remember that $C_\delta(R_1) := \widetilde{f}'(R_1 + \delta) - \widetilde{f}'(R_1 - \delta) \geq 0$. By the definition (174) of $\widetilde{f}$ and the upper bound (173), we have

$$|\widetilde{f}'(R_1)| \leq \frac{\rho_n}{2\alpha_n} + \frac{S}{\alpha_n\sqrt{R_1}}\mathbb{E}|\widetilde{Z}_1| \leq \frac{\rho_n}{2\alpha_n} + \frac{S}{\alpha_n\sqrt{R_1}} \ , \quad (184)$$

where we use that $\mathbb{E}|\widetilde{Z}_1| \leq 1$. The inequality (184) implies $|C_\delta(R_1)| \leq \left(\rho_n + {}^{2S}/\sqrt{s_n - \delta}\right)/\alpha_n$. Then,

$$\int_{\mathcal{B}_n} d\epsilon\, C_\delta(R_1(t,\epsilon))^2$$
$$\leq \frac{1}{\alpha_n}\left(\rho_n + \frac{2S}{\sqrt{s_n - \delta}}\right)\int_{\mathcal{B}_n} d\epsilon\, C_\delta(R_1(t,\epsilon))$$
$$= \frac{1}{\alpha_n}\left(\rho_n + \frac{2S}{\sqrt{s_n - \delta}}\right)\int_{R^t(\mathcal{B}_n)} \frac{dR_1 dR_2}{|J_{R^t}((R^t)^{-1}(R_1, R_2))|}\, C_\delta(R_1)$$
$$\leq \frac{1}{\alpha_n}\left(\rho_n + \frac{2S}{\sqrt{s_n - \delta}}\right)\int_{s_n}^{2s_n + 1} dR_2 \int_{s_n}^{K_n} dR_1 C_\delta(R_1)$$
$$\leq \frac{1}{\alpha_n}\left(\rho_n + \frac{2S}{\sqrt{s_n - \delta}}\right)\int_{s_n}^{2s_n + 1} dR_2\left(\widetilde{f}(K_n + \delta) - \widetilde{f}(K_n - \delta) + \widetilde{f}(s_n - \delta) - \widetilde{f}(s_n + \delta)\right).$$

By the mean value theorem and the upper bound (173), we have (uniformly in $R_2$)

$$|\widetilde{f}(R_1 - \delta) - \widetilde{f}(R_1 + \delta)| \leq 2\delta\left(\frac{\rho_n}{2\alpha_n} + \frac{S}{\alpha_n\sqrt{R_1 - \delta}}\right) \leq \frac{\delta}{\alpha_n}\left(\rho_n + \frac{2S}{\sqrt{s_n - \delta}}\right).$$

Therefore,

$$\int_{\mathcal{B}_n} d\epsilon \left(\frac{\alpha_n C_\delta(R_1)}{\rho_n}\right)^2 \leq \frac{(1 + s_n)\delta}{\rho_n^2}\left(\rho_n + \frac{2S}{\sqrt{s_n - \delta}}\right)^2 \leq \frac{3\delta}{2\rho_n^2}\left(\frac{1 + 2S}{\sqrt{s_n - \delta}}\right)^2 = \frac{3(1 + 2S)^2\delta}{2\rho_n^2(s_n - \delta)}. \quad (185)$$

For all $\epsilon \in \mathcal{B}_n$, we have $R_1(t, \epsilon) \geq s_n$ so (remember that $\int_{\mathcal{B}_n} d\epsilon = s_n^2$)

$$\int_{\mathcal{B}_n} d\epsilon \frac{3}{2n\rho_n}\left(S^4 + \frac{1}{R_1(t, \epsilon)}\right) \leq \frac{3}{2n\rho_n}\int_{\mathcal{B}_n} d\epsilon\left(S^4 + \frac{1}{s_n}\right) \leq \frac{3s_n}{2n\rho_n}(S^4 s_n + 1) \leq \frac{3s_n}{2n\rho_n}\left(\frac{S^4}{2} + 1\right). \quad (186)$$

Integrating (182) over $\epsilon \in \mathcal{B}_n$ and making use of (183), (185), (186) yields

$$\int_{\mathcal{B}_n} d\epsilon\, \mathbb{E}\left[(\langle\mathcal{L}\rangle - \mathbb{E}\langle\mathcal{L}\rangle)^2\right] \leq \frac{\alpha_n s_n^2}{n\rho_n^3\delta^2}\left(18C + 294S^2(M_{\rho/\alpha} + r_{\max})\right) + \frac{3s_n}{2n\rho_n}\left(\frac{S^4}{2} + 1\right) + \frac{9(1 + 2S)^2}{\rho_n^2\left(\frac{s_n}{\delta} - 1\right)}.$$

We can use $\rho_n/\alpha_n \leq M_{\rho/\alpha}$, $\rho_n \leq 1$, $\delta \leq s_n$, and $s_n \leq 1/2$ – in this order – to show that

$$\frac{s_n}{n\rho_n} = \frac{\rho_n^2\delta^2}{\alpha_n s_n} \cdot \frac{\alpha_n s_n^2}{n\rho_n^3\delta^2} \leq \frac{M_{\rho/\alpha}\rho_n\delta^2}{s_n} \cdot \frac{\alpha_n s_n^2}{n\rho_n^3\delta^2} \leq \frac{M_{\rho/\alpha}\delta^2}{s_n} \cdot \frac{\alpha_n s_n^2}{n\rho_n^3\delta^2} \leq M_{\rho/\alpha}s_n \cdot \frac{\alpha_n s_n^2}{n\rho_n^3\delta^2} \leq \frac{M_{\rho/\alpha}}{2} \cdot \frac{\alpha_n s_n^2}{n\rho_n^3\delta^2}.$$

Thus, we finally obtain

$$\int_{\mathcal{B}_n} d\epsilon\, \mathbb{E}\left[(\langle\mathcal{L}\rangle - \mathbb{E}\langle\mathcal{L}\rangle)^2\right] \leq C_1\frac{\alpha_n s_n^2}{n\rho_n^3\delta^2} + C_2\frac{1}{\rho_n^2\left(\frac{s_n}{\delta} - 1\right)}, \quad (187)$$

where $C_1 := 18C + 294S^2 r_{\max} + (294S^2 + 3S^4/8 + 3/4)M_{\rho/\alpha}$ and $C_2 := 9(1 + 2S)^2$.

The convergence of the ratio $\delta/s_n$ to zero is a necessary condition for the second term on the right-hand side of (187) to vanish. If $\delta/s_n \to 0$ then $\left(\rho_n^2(s_n/\delta - 1)\right)^{-1} = \Theta(\delta/\rho_n^2 s_n)$. In that situation, both terms on the right-hand side of (187) are equivalent, that is, $\alpha_n s_n^2/n\rho_n^3\delta^2 = \Theta(\delta/\rho_n^2 s_n)$, if we choose $\delta \propto (\alpha_n/n\rho_n)^{\frac{1}{3}}s_n$. Note that we can choose $\delta \propto (\alpha_n/n\rho_n)^{\frac{1}{3}}s_n$ and still make sure that $\delta \in (0, s_n)$ because there exists $m_{\rho/\alpha}$ such that $\frac{\rho_n}{\alpha_n} > \frac{m_{\rho/\alpha}}{n}$ for all $n > 0$. Plugging the choice $\delta = \left(\frac{m_{\rho/\alpha}\alpha_n}{n\rho_n}\right)^{\frac{1}{3}}s_n$ back in (187) ends the proof of the lemma,

$$\int_{\mathcal{B}_n} d\epsilon\, \mathbb{E}\left[(\langle\mathcal{L}\rangle - \mathbb{E}\langle\mathcal{L}\rangle)^2\right] \leq \frac{C_1}{m_{\rho/\alpha}}\frac{1}{\rho_n^2\left(\frac{\rho_n n}{\alpha_n m_{\rho/\alpha}}\right)^{\frac{1}{3}}} + C_2\frac{1}{\rho_n^2\left(\frac{\rho_n n}{\alpha_n m_{\rho/\alpha}}\right)^{\frac{1}{3}} - \rho_n^2}$$

$$\leq \left(\frac{C_1}{m_{\rho/\alpha}} + C_2\right)\frac{1}{\rho_n^2\left(\frac{\rho_n n}{\alpha_n m_{\rho/\alpha}}\right)^{\frac{1}{3}} - \rho_n^2}.$$

$\blacksquare$

# E   Proof of Proposition 4

Before proving the proposition, we recall a few definitions for reader's convenience. We suppose that (H1), (H2), (H3) hold and that $\Delta = \mathbb{E}_{X\sim P_0}[X^2] = 1$. For all $n \in \mathbb{N}^*$, we define the interval $\mathcal{B}_n := [s_n, 2s_n]$ where $(s_n)_{n\in\mathbb{N}^*}$ is a sequence that takes its values in $(0, 1/2]$. Let $r_{\max} := -2\,\partial I_{P_{\text{out}}}/\partial q\big|_{q=1,\rho=1}$ a nonnegative real number. We have $X_i^* \overset{\text{iid}}{\sim} P_{0,n}$, $\mathbf{A}_\mu \overset{\text{iid}}{\sim} P_A$ and $\Phi_{\mu i}, V_\mu, W_\mu^*, Z_\mu, \widetilde{Z}_i \overset{\text{iid}}{\sim} \mathcal{N}(0, 1)$ for $i = 1 \ldots n$ and $\mu = 1 \ldots m_n$. For fixed $t \in [0, 1]$ and $R = (R_1, R_2) \in [0, +\infty) \times [0, t + 2s_n]$, consider the observations:

$$\begin{cases} Y_\mu^{(t,R_2)} &= \varphi\big(S_\mu^{(t,R_2)}, \mathbf{A}_\mu\big) + Z_\mu\,, \quad 1 \leq \mu \leq m_n \\ &\sim P_{\text{out}}\Big(\,\cdot\,\Big|S_\mu^{(t,R_2)}\Big) \\ \widetilde{Y}_i^{(t,R_1)} &= \sqrt{R_1}\, X_i^* + \widetilde{Z}_i \quad\quad, \quad 1 \leq i \leq n \end{cases} ;$$

where $S_\mu^{(t,R_2)} = S_\mu^{(t,R_2)}(\mathbf{X}^*, \mathbf{W}^*) := \sqrt{\frac{1-t}{k_n}}(\mathbf{\Phi X}^*)_\mu + \sqrt{R_2}\, V_\mu + \sqrt{t + 2s_n - R_2}\, W_\mu^*$. The joint posterior density of $(\mathbf{X}^*, \mathbf{W}^*)$ given $(\mathbf{Y}^{(t,R_2)}, \widetilde{\mathbf{Y}}^{(t,R_1)}, \mathbf{\Phi}, \mathbf{V})$ is:

$$dP(\mathbf{x}, \mathbf{w}|\mathbf{Y}^{(t,R_2)}, \widetilde{\mathbf{Y}}^{(t,R_1)}, \mathbf{\Phi}, \mathbf{V})$$

$$= \frac{1}{\mathcal{Z}_{t,R}}\prod_{i=1}^{n} dP_{0,n}(x_i)\, e^{-\frac{1}{2}\left(\sqrt{R_1}x_i - \widetilde{Y}_i^{(t,R_1)}\right)^2}\prod_{\mu=1}^{m_n}\frac{dw_\mu}{\sqrt{2\pi}}e^{-\frac{w_\mu^2}{2}}P_{\text{out}}(Y_\mu^{(t,R_2)}|S_\mu^{(t,R_2)}(\mathbf{x}, w_\mu))\,,$$

where $\mathcal{Z}_{t,R}$ is the normalization. The angular brackets $\langle - \rangle_{n,t,R}$ denotes the expectation w.r.t. this posterior. The scalar overlap is the quantity $Q := \frac{1}{k_n} \sum_{i=1}^{n} X_i^* x_i$. We define:

$$F_2^{(n)}(t, R) := \mathbb{E}\langle Q \rangle_{n,t,R} \quad \text{and} \quad F_1^{(n)}(t, R) := -2\frac{\alpha_n}{\rho_n} \frac{\partial I_{P_{\text{out}}}}{\partial q}\bigg|_{q=\mathbb{E}\langle Q \rangle_{n,t,R}, \rho=1} .$$

We now repeat and prove Proposition 4.

**Proposition 4 (extended).** *Suppose that (H1), (H2), (H3) hold and that $\Delta = \mathbb{E}_{X \sim P_0}[X^2] = 1$. For all $\epsilon \in \mathcal{B}_n$, there exists a unique global solution $R(\cdot, \epsilon) : [0, 1] \to [0, +\infty)^2$ to the second-order ODE:*

$$y'(t) = \big(F_1^{(n)}(t, y(t)), F_2^{(n)}(t, y(t))\big) \quad , \quad y(0) = \epsilon .$$

*This solution is continuously differentiable and its derivative $R'(\cdot, \epsilon)$ satisfies:*

$$R'([0,1], \epsilon) \subseteq \left[0, \frac{\alpha_n}{\rho_n} r_{\max}\right] \times [0, 1] .$$

*Besides, for all $t \in [0, 1]$, $R(t, \cdot)$ is a $\mathcal{C}^1$-diffeomorphism from $\mathcal{B}_n$ onto its image whose Jacobian determinant is greater than, or equal to, one:*

$$\forall \epsilon \in \mathcal{B}_n : \det J_{R(t,\cdot)}(\epsilon) \geq 1 ,$$

*where $J_{R(t,\cdot)}$ denotes the Jacobian matrix of $R(t, \cdot)$.*
*Finally, the same statement holds if, for a fixed $r \in [0, r_{\max}]$, we instead consider the second-order ODE:*

$$y'(t) = \left(\frac{\alpha_n}{\rho_n} r, F_2^{(n)}(t, y(t))\right) \quad , \quad y(0) = \epsilon .$$

*Proof.* We only give the proof for the ODE $y' = \big(F_1^{(n)}(t, y), F_2^{(n)}(t, y)\big)$ since the one for the ODE $y' = \big(\alpha_n r / \rho_n, F_2^{(n)}(t, y)\big)$ is simpler and follows the same arguments.

By Jensen's inequality and Nishimori identity (see Lemma 5):

$$\mathbb{E}\langle Q \rangle_{n,t,R} = \frac{\mathbb{E}\|\langle \mathbf{x} \rangle_{n,t,R}\|^2}{k_n} \leq \frac{\mathbb{E}\langle \|\mathbf{x}\|^2 \rangle_{n,t,R}}{k_n} = \frac{\mathbb{E}\|\mathbf{X}^*\|^2}{k_n} = 1 ,$$

i.e., $\mathbb{E}\langle Q \rangle_{n,t,R} \in [0, 1]$. By Lemma 7, the function $q \mapsto I_{P_{\text{out}}}(q, 1)$ is continuously twice differentiable, concave and nonincreasing on $[0, 1]$. Therefore, $q \mapsto -2\partial I_{P_{\text{out}}}/\partial q\big|_{q,\rho=1}$ is nonnegative and nondecreasing on $[0, 1]$, which implies $-2\partial I_{P_{\text{out}}}/\partial q\big|_{q,\rho=1} \in [0, r_{\max}]$. We have thus shown that the function $F : (t, R) \mapsto (F_1^{(n)}(t, R), F_2^{(n)}(t, R))$ is defined on all

$$\mathcal{D}_n := \left\{(t, R_1, R_2) \in [0, 1] \times [0, +\infty)^2 : R_2 \leq t + 2s_n\right\} ,$$

and takes its values in $[0, \alpha_n r_{\max}/\rho_n] \times [0, 1]$.

To invoke Cauchy-Lipschitz theorem, we have to check that $F$ is continuous in $t$ and uniformly Lipschitz continuous in $R$ (meaning the Lipschitz constant is independent of $t$). We can show that $F$ is continuous on $\mathcal{D}_n$ and that, for all $t \in [0, 1]$, $F(t, \cdot)$ is differentiable on $(0, +\infty) \times (0, t + 2s_n)$ thanks to the standard theorems of continuity and differentiation under the integral sign. The domination hypotheses are indeed verified because we assume that (H1), (H2) hold. To check the uniform Lipschitzianity, we show that the Jacobian matrix $J_{F(t,\cdot)}(R)$ of $F(t, \cdot)$ is uniformly bounded in $(t, R)$. For all $(R_1, R_2) \in (0, +\infty) \times (0, t + 2s_n)$, we have:

$$J_{F(t,\cdot)}(R) = \begin{bmatrix} c(t, R) & c(t, R) \\ 1 & 1 \end{bmatrix} \begin{bmatrix} \frac{\partial F_2^{(n)}}{\partial R_1}\big|_{t,R} & 0 \\ 0 & \frac{\partial F_2^{(n)}}{\partial R_2}\big|_{t,R} \end{bmatrix} , \tag{188}$$

with $c(t, R) := -2\frac{\alpha_n}{\rho_n} \frac{\partial^2 I_{P_{\text{out}}}}{\partial q^2}\big|_{q=F_2^{(n)}(t,R), \rho=1}$ and

$$\frac{\partial F_2^{(n)}}{\partial R_1}\bigg|_{t,R} = \frac{1}{k_n} \sum_{i,j=1}^{n} \mathbb{E}\big[\big(\langle x_i x_j \rangle_{n,t,R} - \langle x_i \rangle_{n,t,R} \langle x_j \rangle_{n,t,R}\big)^2\big] ; \tag{189}$$

$$\frac{\partial F_2^{(n)}}{\partial R_2}\bigg|_{t,R} = \frac{1}{k_n} \sum_{\mu=1}^{m_n} \mathbb{E}\Big[\big\|\langle u'_{Y_\mu^{(t,R)}}(s_\mu^{(t,R)})\mathbf{x}\rangle_{n,t,R} - \langle u'_{Y_\mu^{(t,R)}}(s_\mu^{(t,R)})\rangle_{n,t,R}\langle \mathbf{x}\rangle_{n,t,R}\big\|^2\Big] . \tag{190}$$

The function $u'_y(\cdot)$ is the derivative of $u_y : x \mapsto \ln P_{\text{out}}(y|x)$. Both $\partial F_2^{(n)}/\partial R_1$ and $\partial F_2^{(n)}/\partial R_2$ are clearly nonnegative. Using the assumption (H1), we easily obtain from (189) that

$$0 \leq \frac{\partial F_2^{(n)}}{\partial R_1}\bigg|_{t,R} \leq \frac{4S^4 n}{\rho_n} . \tag{191}$$

In the proof of Lemma 7, under the hypothesis (H2) we obtain the upper bound (93) on $|u'_y(x)|$. It yields $\forall x \in \mathbb{R} : \left|u'_{Y_\mu^{(t,R)}}(x)\right| \leq (2\|\varphi\|_\infty + |Z_\mu|)\|\partial_x\varphi\|_\infty$. Then, we easily see from (189) that

$$0 \leq \left.\frac{\partial F_2^{(n)}}{\partial R_2}\right|_{t,R} \leq 8S^2(4\|\varphi\|_\infty^2 + 1)\|\partial_x\varphi\|_\infty^2 \frac{\alpha_n n}{\rho_n} \,. \tag{192}$$

Finally, by Lemma 7, $q \mapsto -\frac{\partial^2 I_{P_{\mathrm{out}}}}{\partial q^2}\big|_{q,\rho=1}$ is nonnegative continuous on the interval $[0,1]$, so it is bounded by a constant $C$ and $c(t,R) \in [0, 2^{C\alpha_n/\rho_n}]$. Combining the later with (188), (191) and (192) shows that $J_{F(t,\cdot)}(R)$ is uniformly bounded in $(t,R) \in \{(t,R_1,R_2) \in [0,1] \times (0,+\infty)^2 : R_2 < t + 2s_n\}$. By the mean-value theorem, this implies that $F$ is uniformly Lipschitz continuous in $R$.

By the Cauchy-Lipschitz theorem, for all $\epsilon \in \mathcal{B}_n$ there exists a unique solution to the initial value problem $y' = F(t,y), y(0) = \epsilon$ that we denote $R(\cdot,\epsilon) : [0,\delta] \to [0,+\infty)^2$. Here $\delta \in [0,1]$ is such that $[0,\delta]$ is the maximal interval of existence of the solution. Because $F$ has its image in $[0, {}^{\alpha_n r_{\max}}/\rho_n] \times [0,1]$, we have that $\forall t \in [0,\delta] : R(t,\epsilon) \in [s_n, 2s_n + t\alpha_n r_{\max}/\rho_n] \times [s_n, 2s_n + t]$, which means that $\delta = 1$ (the solution never leaves the domain of definition of $F$).

Each initial condition $\epsilon \in \mathcal{B}_n$ is tied to a unique solution $R(\cdot,\epsilon)$. This implies that the function $\epsilon \mapsto R(t,\epsilon)$ is injective. Its Jacobian determinant is given by Liouville's formula [14, Chapter V, Corollary 3.1]:

$$\det J_{R(t,\cdot)}(\epsilon) = \exp \int_0^t ds \left(\frac{\partial F_1^{(n)}}{\partial R_1} + \frac{\partial F_2^{(n)}}{\partial R_2}\right)\bigg|_{s,R(s,\epsilon)}$$

$$= \exp \int_0^t ds \left(c\big(s,R(s,\epsilon)\big)\frac{\partial F_2^{(n)}}{\partial R_1}\bigg|_{s,R(s,\epsilon)} + \frac{\partial F_2^{(n)}}{\partial R_2}\bigg|_{s,R(s,\epsilon)}\right).$$

This Jacobian determinant is greater than, or equal to, one since we saw that all of $c(t,R)$, $\partial F_1^{(n)}/\partial R_1$ and $\partial F_2^{(n)}/\partial R_2$ are nonnegative. The fact that the Jacobian determinant is bounded away from 0 uniformly in $\epsilon$ implies by the inverse function theorem that the injective function $\epsilon \mapsto R(t,\epsilon)$ is a $\mathcal{C}^1$-diffeomorphism from $\mathcal{B}_n$ onto its image. □

# F  Proof of Theorem 2 for a general discrete prior with finite support

In the whole appendix we assume that $P_{0,n} := (1 - \rho_n)\delta_0 + \rho_n P_0$ where $P_0$ is a discrete distribution with finite support $\mathrm{supp}(P_0) \subseteq \{\pm v_1, \ldots, \pm v_K\}$ with $0 < v_1 < v_2 < \cdots < v_K$. For all $i$, $P_0(v_i) = p_i^+$, $P_0(-v_i) = p_i^-$ with $p_i^+, p_i^- \geq 0$ and $p_i := p_i^+ + p_i^- > 0$. Of course, $\sum_{i=1}^K p_i = 1$. Note that the second moment of $X \sim P_0$ is $\mathbb{E}[X^2] = \sum_{j=1}^K p_j v_j^2$.

For $\rho_n, \alpha_n > 0$ we denote the variational problem appearing in Theorem 1 by

$$I(\rho_n, \alpha_n) := \inf_{q \in [0, \mathbb{E}X^2]} \sup_{r \geq 0} i_{\mathrm{RS}}(q,r;\alpha_n,\rho_n) \,, \tag{193}$$

where the potential $i_{\mathrm{RS}}$ is defined in (7). Let $X^* \sim P_{0,n} \perp Z \sim \mathcal{N}(0,1)$. We define for all $r \geq 0$:

$$\psi_{P_{0,n}}(r) := \mathbb{E}\left[\ln \int dP_{0,n}(x)e^{-\frac{r}{2}x^2 + rX^*x + \sqrt{r}xZ}\right] \tag{194}$$

$$= \mathbb{E}\left[\ln\left(1 - \rho_n + \rho_n \sum_{i=1}^K e^{-\frac{rv_i^2}{2}}\big(p_i^+ e^{rX^*v_i + \sqrt{r}Zv_i} + p_i^- e^{-rX^*v_i - \sqrt{r}Zv_i}\big)\right)\right].$$

Note that $I_{P_{0,n}}(r) := I(X^*; \sqrt{r}X^* + Z) = \frac{r\rho_n \mathbb{E}[X^2]}{2} - \psi_{P_{0,n}}(r)$ where $X \sim P_0$ so

$$I(\rho_n, \alpha_n) = \inf_{q \in [0, \mathbb{E}X^2]} I_{P_{\mathrm{out}}}(q, \mathbb{E}X^2) + \sup_{r \geq 0}\left\{\frac{rq}{2} - \frac{1}{\alpha_n}\psi_{P_{0,n}}\left(\frac{\alpha_n}{\rho_n}r\right)\right\}. \tag{195}$$

The latter expression for $I(\rho_n, \alpha_n)$ is easier to work with. We point out that $\psi_{P_{0,n}}$ is twice differentiable, nondecreasing, strictly convex and $\frac{\rho_n \mathbb{E}X^2}{2}$-Lipschitz on $[0, +\infty)$ (see Lemma 6) while $I_{P_{\mathrm{out}}}(\cdot, \mathbb{E}X^2)$ is nonincreasing and concave on $[0, \mathbb{E}X^2]$ (see [7, Appendix B.2, Proposition 18]).

Our goal is now to compute the limit of $I(\rho_n, \alpha_n)$ when $\alpha_n := \gamma\rho_n|\ln\rho_n|$ for a fix $\gamma > 0$ and $\rho_n \to 0$. We first look where the supremum over $r$ is reached depending on the value of $q \in [0, \mathbb{E}X^2]$.

**Lemma 15.** *Let $P_{0,n} := (1 - \rho_n)\delta_0 + \rho_n P_0$ where $P_0$ is a discrete distribution with finite support $\mathrm{supp}(P_0) \subseteq \{\pm v_1, \pm v_2, \ldots, \pm v_K\}$ with $0 < v_1 < v_2 < \cdots < v_K$. Let $\alpha_n := \gamma\rho_n|\ln\rho_n|$ for a fix $\gamma > 0$. Define $g_{\rho_n} : r \in (0, +\infty) \mapsto \frac{2}{\rho_n}\psi'_{P_{0,n}}\left(\frac{\alpha_n}{\rho_n}r\right)$ and $\forall \rho_n \in (0, e^{-1}), \forall j \in \{1, \ldots, K\}$:*

$$a_{\rho_n}^{(j)} := g_{\rho_n}\left(\frac{2(1 - |\ln\rho_n|^{-\frac{1}{4}})}{\gamma v_j^2}\right) \quad , \quad b_{\rho_n}^{(j)} := g_{\rho_n}\left(\frac{2(1 + |\ln\rho_n|^{-\frac{1}{4}})}{\gamma v_j^2}\right). \tag{196}$$

Let $X \sim P_0$. For $\rho_n$ small enough we have

$$\rho_n \mathbb{E}[X]^2 < a_{\rho_n}^{(K)} < b_{\rho_n}^{(K)} < a_{\rho_n}^{(K-1)} < b_{\rho_n}^{(K-1)} < \cdots < a_{\rho_n}^{(1)} < b_{\rho_n}^{(1)} < \mathbb{E}[X^2] \,, \tag{197}$$

and for all $j \in \{1, \ldots, K\}$ :

$$\lim_{\rho_n \to 0} a_{\rho_n}^{(j)} = \mathbb{E}[X^2 \mathbf{1}_{\{|X| > v_j\}}] \quad ; \quad \lim_{\rho_n \to 0} b_{\rho_n}^{(j)} = \mathbb{E}[X^2 \mathbf{1}_{\{|X| \geq v_j\}}] \,. \tag{198}$$

Besides, for every $q \in (\rho_n \mathbb{E}[X]^2, \mathbb{E}[X^2])$ there exists a unique $r_n^*(q) \in (0, +\infty)$ such that

$$\frac{r_n^*(q)q}{2} - \frac{1}{\alpha_n} \psi_{P_{0,n}} \left( \frac{\alpha_n}{\rho_n} r_n^*(q) \right) = \sup_{r \geq 0} \frac{rq}{2} - \frac{1}{\alpha_n} \psi_{P_{0,n}} \left( \frac{\alpha_n}{\rho_n} r \right) \,, \tag{199}$$

and $\forall j \in \{1, \ldots, K\}, \forall q \in [a_{\rho_n}^{(j)}, b_{\rho_n}^{(j)}]$:

$$\frac{2(1 - |\ln \rho_n|^{-\frac{1}{4}})}{\gamma v_j^2} \leq r_n^*(q) \leq \frac{2(1 + |\ln \rho_n|^{-\frac{1}{4}})}{\gamma v_j^2} \,. \tag{200}$$

The bounds (200) are tight, namely, $r_n^*(a_{\rho_n}^{(j)}) = \frac{2(1 - |\ln \rho_n|^{-\frac{1}{4}})}{\gamma v_j^2}, r_n^*(b_{\rho_n}^{(j)}) = \frac{2(1 + |\ln \rho_n|^{-\frac{1}{4}})}{\gamma v_j^2}$.

*Proof.* For every $q \in (0, 1)$ we define $f_{\rho_n, q} : r \in [0, +\infty) \mapsto \frac{rq}{2} - \frac{1}{\alpha_n} \psi_{P_{0,n}} \left( \frac{\alpha_n}{\rho_n} r \right)$ whose supremum over $r$ we want to compute. The derivative of $f_{\rho_n, q}$ with respect to $r$ reads

$$f'_{\rho_n, q}(r) = \frac{q}{2} - \frac{1}{\rho_n} \psi'_{P_{0,n}} \left( \frac{\alpha_n}{\rho_n} r \right) \,. \tag{201}$$

The derivative $\psi'_{P_{0,n}}$ is continuously increasing and thus one-to-one from $(0, +\infty)$ onto $(\rho_n^2 \mathbb{E}[X]^2/2, \rho_n \mathbb{E}[X^2]/2)$. Therefore, if $q \in (0, \rho_n \mathbb{E}[X]^2]$ then $f'_{\rho_n, q} \leq 0$ and the supremum of $f_{\rho_n, q}$ is achieved at $r = 0$. On the contrary, if $q \in (\rho_n, \mathbb{E}[X^2])$ then there exists a unique solution $r_n^*(q) \in (0, +\infty)$ to the critical point equation $f'_{\rho_n, q}(r) = 0$. As $f_{\rho_n, q}$ is concave ($\psi_{P_{0,n}}$ is convex) this solution $r_n^*(q)$ is the global maximum of $f_{\rho_n, q}$. We now transform the critical point equation:

$$f_{\rho_n, q}(r) = 0 \Leftrightarrow \frac{2}{\rho_n} \psi'_{P_{0,n}} \left( \frac{\alpha_n}{\rho_n} r \right) = q \Leftrightarrow g_{\rho_n}(r) = q \,, \tag{202}$$

where $g_{\rho_n} : r \mapsto \frac{2}{\rho_n} \psi'_{P_{0,n}} \left( \frac{\alpha_n}{\rho_n} r \right)$ is continuously increasing and one-to-one from $(0, +\infty)$ to $(\rho_n \mathbb{E}X^2, \mathbb{E}X^2)$.

By definition of $a_{\rho_n}^{(j)}$ and $b_{\rho_n}^{(j)}$, $r_n^*(a_{\rho_n}^{(j)}) = 2\left(1 - |\ln \rho_n|^{-\frac{1}{4}}\right)/\gamma v_j^2$ and $r_n^*(b_{\rho_n}^{(j)}) = 2\left(1 + |\ln \rho_n|^{-\frac{1}{4}}\right)/\gamma v_j^2$. Besides, if $q = g_{\rho_n}(r_n^*(q)) \in [a_{\rho_n}^{(j)}, b_{\rho_n}^{(j)}]$ then

$$\frac{2(1 - |\ln \rho_n|^{-\frac{1}{4}})}{\gamma v_j^2} \leq r_n^*(q) \leq \frac{2(1 + |\ln \rho_n|^{-\frac{1}{4}})}{\gamma v_j^2}$$

as $g_{\rho_n}$ is increasing. Because $g_{\rho_n}$ is increasing with $0 < v_1 < \cdots < v_k$, it is clear that we have the ordering (197) provided that $\rho_n$ is close enough to 0.

It remains to prove the limits (198). In order to so, we first rewrite the derivative of $\psi_{P_{0,n}}$. For all $r \geq 0$, we have:

$$\psi'_{P_{0,n}}(r) = \frac{1}{2} \mathbb{E}\left[ X^* \frac{\rho_n \sum_{i=1}^K v_i e^{-\frac{rv_i^2}{2}} \left( p_i^+ e^{rX^* v_i + \sqrt{r}Z v_i} - p_i^- e^{-rX^* v_i - \sqrt{r}Z v_i} \right)}{1 - \rho_n + \rho_n \sum_{i=1}^K e^{-\frac{rv_i^2}{2}} \left( p_i^+ e^{rX^* v_i + \sqrt{r}Z v_i} + p_i^- e^{-rX^* v_i - \sqrt{r}Z v_i} \right)} \right]$$

$$= \frac{\rho_n^2}{2} \sum_{j=1}^K p_j^+ v_j \mathbb{E}\left[ \frac{\sum_{i=1}^K v_i e^{-\frac{rv_i^2}{2}} \left( p_i^+ e^{rv_i v_j + \sqrt{r}Z v_i} - p_i^- e^{-rv_i v_j - \sqrt{r}Z v_i} \right)}{1 - \rho_n + \rho_n \sum_{i=1}^K e^{-\frac{rv_i^2}{2}} \left( p_i^+ e^{rv_i v_j + \sqrt{r}Z v_i} + p_i^- e^{-rv_i v_j - \sqrt{r}Z v_i} \right)} \right]$$

$$+ \frac{\rho_n^2}{2} \sum_{j=1}^K p_j^- v_j \mathbb{E}\left[ \frac{\sum_{i=1}^K v_i e^{-\frac{rv_i^2}{2}} \left( p_i^- e^{rv_i v_j + \sqrt{r}Z v_i} - p_i^+ e^{-rv_i v_j - \sqrt{r}Z v_i} \right)}{1 - \rho_n + \rho_n \sum_{i=1}^K e^{-\frac{rv_i^2}{2}} \left( p_i^- e^{rv_i v_j + \sqrt{r}Z v_i} + p_i^+ e^{-rv_i v_j - \sqrt{r}Z v_i} \right)} \right]$$

$$= \frac{\rho_n}{2} \sum_{j=1}^K \mathbb{E}\left[ \frac{p_j^+ v_j \sum_{i=1}^K v_i e^{-\frac{r(v_i - v_j)^2}{2} + \sqrt{r}Z(v_i - v_j)} \left( p_i^+ - p_i^- e^{-2rv_i v_j - 2\sqrt{r}Z v_i} \right)}{\frac{1 - \rho_n}{\rho_n} e^{-\frac{rv_j^2}{2} - \sqrt{r}Z v_j} + \sum_{i=1}^K e^{-\frac{r(v_i - v_j)^2}{2} + \sqrt{r}Z(v_i - v_j)} \left( p_i^+ + p_i^- e^{-2rv_i v_j - 2\sqrt{r}Z v_i} \right)} \right]$$

$$+ \frac{\rho_n}{2} \sum_{j=1}^K \mathbb{E}\left[ \frac{p_j^- v_j \sum_{i=1}^K v_i e^{-\frac{r(v_i - v_j)^2}{2} + \sqrt{r}Z(v_i - v_j)} \left( p_i^- - p_i^+ e^{-2rv_i v_j - 2\sqrt{r}Z v_i} \right)}{\frac{1 - \rho_n}{\rho_n} e^{-\frac{rv_j^2}{2} - \sqrt{r}Z v_j} + \sum_{i=1}^K e^{-\frac{r(v_i - v_j)^2}{2} + \sqrt{r}Z(v_i - v_j)} \left( p_i^- + p_i^+ e^{-2rv_i v_j - 2\sqrt{r}Z v_i} \right)} \right] \,.$$

The latter expression is shorten to

$$\psi'_{P_{0,n}}(r) = \frac{\rho_n}{2} \sum_{j=1}^{K} p_j^+ v_j \mathbb{E}\big[h(Z, r, v_j; \rho_n, \mathbf{v}, \mathbf{p}^+, \mathbf{p}^-)\big] + p_j^- v_j \mathbb{E}\big[h(Z, r, v_j; \rho_n, \mathbf{v}, \mathbf{p}^-, \mathbf{p}^+)\big] \; ; \qquad (203)$$

where $\mathbf{v} := (v_1, v_2, \dots, v_K)$, $\mathbf{p}^+ := (p_1^+, p_2^+, \dots, p_K^+)$, $\mathbf{p}^- := (p_1^-, p_2^-, \dots, p_K^-)$ and we define $\forall (z, r, u) \in \mathbb{R} \times [0, +\infty) \times (0, +\infty)$:

$$h(z, r, u; \rho_n, \mathbf{v}, \mathbf{p}^+, \mathbf{p}^-)$$

$$:= \frac{\sum_{i=1}^{K} v_i e^{-\frac{r(v_i - u)^2}{2} + \sqrt{r}z(v_i - u)}\big(p_i^+ - p_i^- e^{-2rv_i u - 2\sqrt{r}z v_i}\big)}{\frac{1-\rho_n}{\rho_n} e^{-\frac{ru^2}{2} - \sqrt{r}zu} + \sum_{i=1}^{K} e^{-\frac{r(v_i - u)^2}{2} + \sqrt{r}z(v_i - u)}\big(p_i^+ + p_i^- e^{-2rv_i u - 2\sqrt{r}z v_i}\big)} \; . \qquad (204)$$

Note that $\forall z \in \mathbb{R}$ :

$$h\left(z, \frac{2(1 + |\ln \rho_n|^{-\frac{1}{4}})|\ln \rho_n|}{v_k^2}, v_j; \rho_n, \mathbf{v}, \mathbf{p}^\pm, \mathbf{p}^\mp\right) \xrightarrow[\rho_n \to 0]{} \begin{cases} 0 & \text{if } j < k \; ; \\ v_j & \text{if } j \geq k \; . \end{cases} \qquad (205)$$

$$h\left(z, \frac{2(1 - |\ln \rho_n|^{-\frac{1}{4}})|\ln \rho_n|}{v_k^2}, v_j; \rho_n, \mathbf{v}, \mathbf{p}^\pm, \mathbf{p}^\mp\right) \xrightarrow[\rho_n \to 0]{} \begin{cases} 0 & \text{if } j \leq k \; ; \\ v_j & \text{if } j > k \; . \end{cases} \qquad (206)$$

By the dominated convergence theorem, making use of the identity (203) and the limit (205), we have $\forall k \in \{1, \dots, K\}$ :

$$a_{\rho_n}^{(k)} := g_{\rho_n}\left(\frac{2(1 - |\ln \rho_n|^{-\frac{1}{4}})}{\gamma v_k^2}\right) = \frac{2}{\rho_n} \psi'_{P_{0,n}}\left(\frac{2(1 - |\ln \rho_n|^{-\frac{1}{4}})|\ln \rho_n|}{v_k^2}\right)$$

$$= \sum_{j=1}^{K} p_j^+ v_j \mathbb{E}\left[h\left(z, \frac{2(1 - |\ln \rho_n|^{-\frac{1}{4}})|\ln \rho_n|}{v_k^2}, v_j; \mathbf{v}, \mathbf{p}^+, \mathbf{p}^-\right)\right]$$

$$+ \sum_{j=1}^{K} p_j^- v_j \mathbb{E}\left[h\left(z, \frac{2(1 - |\ln \rho_n|^{-\frac{1}{4}})|\ln \rho_n|}{v_k^2}, v_j; \mathbf{v}, \mathbf{p}^-, \mathbf{p}^+\right)\right]$$

$$\xrightarrow[\rho_n \to 0]{} \sum_{j>k} p_j^+ v_j^2 + \sum_{j>k} p_j^- v_j^2 = \mathbb{E}[X^2 \mathbf{1}_{\{|X| > v_k\}}] \; .$$

Similarly, using this time the limit (206), we have $\forall k \in \{1, \dots, K\}$ :

$$b_{\rho_n}^{(k)} := g_{\rho_n}\left(\frac{2(1 + |\ln \rho_n|^{-\frac{1}{4}})}{\gamma v_k^2}\right) = \frac{2}{\rho_n} \psi'_{P_{0,n}}\left(\frac{2(1 + |\ln \rho_n|^{-\frac{1}{4}})|\ln \rho_n|}{v_k^2}\right)$$

$$= \sum_{j=1}^{K} p_j^+ v_j \mathbb{E}\left[h\left(z, \frac{2(1 + |\ln \rho_n|^{-\frac{1}{4}})|\ln \rho_n|}{v_k^2}, v_j; \mathbf{v}, \mathbf{p}^+, \mathbf{p}^-\right)\right]$$

$$+ \sum_{j=1}^{K} p_j^- v_j \mathbb{E}\left[h\left(z, \frac{2(1 + |\ln \rho_n|^{-\frac{1}{4}})|\ln \rho_n|}{v_k^2}, v_j; \mathbf{v}, \mathbf{p}^-, \mathbf{p}^+\right)\right]$$

$$\xrightarrow[\rho_n \to 0]{} \sum_{j \geq k} p_j^+ v_j^2 + \sum_{j \geq k} p_j^- v_j^2 = \mathbb{E}[X^2 \mathbf{1}_{\{|X| \geq v_k\}}] \; .$$

$\square$

Note that $\lim_{\rho_n \to 0} b_{\rho_n}^{(j)} = \lim_{\rho_n \to 0} a_{\rho_n}^{(j-1)}$. Thus, Lemma 15 essentially states that in the limit $\rho_n \to 0$ the segment $[0, \mathbb{E}[X^2]]$ can be broken into $K$ subsegments $[a_{\rho_n}^{(j)}, b_{\rho_n}^{(j)}]$, and for $q \in [a_{\rho_n}^{(j)}, b_{\rho_n}^{(j)}]$ the point at which the supremum over $r$ is achieved is located in an interval shrinking on $r^* := 2/\gamma v_j^2$. The next step is then to determine what is the limit of $\frac{1}{\alpha_n} \psi_{P_{0,n}}\left(\frac{\alpha_n}{\rho_n} \frac{2}{\gamma v_j^2}\right)$.

**Lemma 16.** *Let $P_{0,n} := (1 - \rho_n)\delta_0 + \rho_n P_0$ where $P_0$ is a discrete distribution with finite support $\mathrm{supp}(P_0) \subseteq \{\pm v_1, \pm v_2, \dots, \pm v_K\}$ with $0 < v_1 < v_2 < \cdots < v_K$. Let $\alpha_n := \gamma \rho_n |\ln \rho_n|$ for a fix $\gamma > 0$. Then, for every $k \in \{1, \dots, K\}$ :*

$$\lim_{\rho_n \to 0} \frac{1}{\alpha_n} \psi_{P_{0,n}}\left(\frac{\alpha_n}{\rho_n} \frac{2(1 \pm |\ln \rho_n|^{-\frac{1}{4}})}{\gamma v_k^2}\right) = \frac{\mathbb{E}[X^2 \mathbf{1}_{\{|X| \geq v_k\}}]}{\gamma v_k^2} - \frac{\mathbb{P}(|X| \geq v_k)}{\gamma} \; . \qquad (207)$$

*Proof.* Fix $k \in \{1, \ldots, K\}$. The function $\psi_{P_0,n}$ is Lipschitz continuous with Lipschitz constant $\frac{\rho_n \mathbb{E}[X^2]}{2}$. Therefore:

$$\left| \frac{1}{\alpha_n} \psi_{P_0,n} \left( \frac{\alpha_n}{\rho_n} \frac{2(1 \pm |\ln \rho_n|^{-\frac{1}{4}})}{\gamma v_k^2} \right) - \frac{1}{\alpha_n} \psi_{P_0,n} \left( \frac{\alpha_n}{\rho_n} \frac{2}{\gamma v_k^2} \right) \right|$$

$$\leq \frac{\rho_n \mathbb{E}[X^2]}{2\alpha_n} \left| \frac{\alpha_n}{\rho_n} \frac{2|\ln \rho_n|^{-\frac{1}{4}}}{\gamma v_k^2} \right| = \frac{\mathbb{E}[X^2]}{\gamma v_k^2} |\ln \rho_n|^{-\frac{1}{4}} .$$

The latter inequality shows that the limits of $\frac{1}{\alpha_n} \psi_{P_0,n} \left( \frac{\alpha_n}{\rho_n} \frac{2(1+|\ln \rho_n|^{-1/4})}{\gamma v_k^2} \right)$ and $\frac{1}{\alpha_n} \psi_{P_0,n} \left( \frac{\alpha_n}{\rho_n} \frac{2(1-|\ln \rho_n|^{-1/4})}{\gamma v_k^2} \right)$ are the same and equal to the limit of $\frac{1}{\alpha_n} \psi_{P_0,n} \left( \frac{\alpha_n}{\rho_n} \frac{2}{\gamma v_k^2} \right)$. To compute the latter we first write $\psi_{P_0,n}(r)$ in a more explicit form. We have for all $r \geq 0$:

$$\psi_{P_0,n}(r) := \mathbb{E}\left[ \ln \int dP_{0,n}(x) e^{-\frac{r}{2}x^2 + rX^*x + \sqrt{r}xZ} \right]$$

$$= \mathbb{E}\left[ \ln \left( 1 - \rho_n + \rho_n \sum_{i=1}^K e^{-\frac{rv_i^2}{2}} \left( p_i^+ e^{rX^*v_i + \sqrt{r}Zv_i} + p_i^- e^{-rX^*v_i - \sqrt{r}Zv_i} \right) \right) \right]$$

$$= (1 - \rho_n)\mathbb{E}\left[ \ln \left( 1 - \rho_n + \rho_n \sum_{i=1}^K e^{-\frac{rv_i^2}{2}} \left( p_i^+ e^{\sqrt{r}Zv_i} + p_i^- e^{-\sqrt{r}Zv_i} \right) \right) \right]$$

$$+ \rho_n \sum_{j=1}^K p_j^+ \mathbb{E}\left[ \ln \left( 1 - \rho_n + \rho_n \sum_{i=1}^K e^{-\frac{rv_i^2}{2}} \left( p_i^+ e^{rv_jv_i + \sqrt{r}Zv_i} + p_i^- e^{-rv_jv_i - \sqrt{r}Zv_i} \right) \right) \right]$$

$$+ \rho_n \sum_{j=1}^K p_j^- \mathbb{E}\left[ \ln \left( 1 - \rho_n + \rho_n \sum_{i=1}^K e^{-\frac{rv_i^2}{2}} \left( p_i^+ e^{-rv_jv_i + \sqrt{r}Zv_i} + p_i^- e^{rv_jv_i - \sqrt{r}Zv_i} \right) \right) \right].$$

By symmetry of $Z \sim \mathcal{N}(0, 1)$ we can replace $Z$ by $-Z$ in the expectations of the last sum. It comes:

$$\psi_{P_0,n}(r) = (1 - \rho_n)\mathbb{E}\left[ \ln \left( 1 - \rho_n + \rho_n \sum_{i=1}^K e^{-\frac{rv_i^2}{2}} \left( p_i^+ e^{\sqrt{r}Zv_i} + p_i^- e^{-\sqrt{r}Zv_i} \right) \right) \right]$$

$$+ \rho_n \sum_{j=1}^K p_j^+ \mathbb{E}\left[ \ln \left( 1 - \rho_n + \rho_n \sum_{i=1}^K e^{-\frac{rv_i^2}{2}} \left( p_i^+ e^{rv_jv_i + \sqrt{r}Zv_i} + p_i^- e^{-rv_jv_i - \sqrt{r}Zv_i} \right) \right) \right]$$

$$+ \rho_n \sum_{j=1}^K p_j^- \mathbb{E}\left[ \ln \left( 1 - \rho_n + \rho_n \sum_{i=1}^K e^{-\frac{rv_i^2}{2}} \left( p_i^- e^{rv_jv_i + \sqrt{r}Zv_i} + p_i^+ e^{-rv_jv_i - \sqrt{r}Zv_i} \right) \right) \right]$$

$$= (1 - \rho_n)\mathbb{E}\left[ \ln \left( 1 - \rho_n + \rho_n \sum_{i=1}^K e^{-\frac{rv_i^2}{2}} \left( p_i^+ e^{\sqrt{r}Zv_i} + p_i^- e^{-\sqrt{r}Zv_i} \right) \right) \right]$$

$$+ \frac{\rho_n r \mathbb{E}[X^2]}{2} + \rho_n \ln \rho_n + \rho_n \sum_{j=1}^K p_j^+ \mathbb{E}\left[ \ln \widetilde{h}(Z, r, v_j; \rho_n, \mathbf{v}, \mathbf{p}^+, \mathbf{p}^-) \right]$$

$$+ \rho_n \sum_{j=1}^K p_j^- \mathbb{E}\left[ \ln \widetilde{h}(Z, r, v_j; \rho_n, \mathbf{v}, \mathbf{p}^-, \mathbf{p}^+) \right] , \tag{208}$$

where $\mathbf{v} := (v_1, v_2, \ldots, v_K)$, $\mathbf{p}^+ := (p_1^+, p_2^+, \ldots, p_K^+)$, $\mathbf{p}^- := (p_1^-, p_2^-, \ldots, p_K^-)$ and we define $\forall (z, r, u) \in \mathbb{R} \times [0, +\infty) \times (0, +\infty)$:

$$\widetilde{h}(z, r, u; \rho_n, \mathbf{v}, \mathbf{p}^\pm, \mathbf{p}^\mp)$$

$$:= \frac{1 - \rho_n}{\rho_n} e^{-\frac{ru^2}{2} - \sqrt{r}zu} + \sum_{i=1}^K e^{-\frac{r(v_i - u)^2}{2} + \sqrt{r}z(v_i - u)} \left( p_i^\pm + p_i^\mp e^{-2rv_iu - 2\sqrt{r}zv_i} \right) . \tag{209}$$

It follows directly from (208) that:

$$\frac{1}{\alpha_n} \psi_{P_0,n} \left( \frac{\alpha_n}{\rho_n} \frac{2}{\gamma v_k^2} \right) = \frac{A_{\rho_n}}{\gamma} + \frac{\mathbb{E}[X^2]}{\gamma v_k^2} - \frac{1}{\gamma} + \frac{1}{\gamma} \sum_{j=1}^K p_j^+ \mathbb{E}\left[ \frac{\ln \widetilde{h}\left(Z, \frac{2|\ln \rho_n|}{v_k^2}, v_j; \rho_n, \mathbf{v}, \mathbf{p}^+, \mathbf{p}^- \right)}{|\ln \rho_n|} \right]$$

$$+ \frac{1}{\gamma} \sum_{j=1}^K p_j^- \mathbb{E}\left[ \frac{\ln \widetilde{h}\left(Z, \frac{2|\ln \rho_n|}{v_k^2}, v_j; \rho_n, \mathbf{v}, \mathbf{p}^-, \mathbf{p}^+ \right)}{|\ln \rho_n|} \right] , \tag{210}$$

where

$$A_{\rho_n} = \frac{1-\rho_n}{\rho_n|\ln\rho_n|}\mathbb{E}\ln\left(1-\rho_n+\rho_n\sum_{i=1}^{K}e^{-\frac{v_i^2}{v_k^2}|\ln\rho_n|}\left(p_i^+e^{\left(\frac{2v_i^2|\ln\rho_n|}{v_k^2}\right)^{\frac{1}{2}}Z}+p_i^-e^{-\left(\frac{2v_i^2|\ln\rho_n|}{v_k^2}\right)^{\frac{1}{2}}Z}\right)\right).$$

Next we show that $A_{\rho_n}$ vanishes when $\rho_n \to 0$. We can use the inequalities $\frac{x}{1+x} \leq \ln(1+x) \leq x$ valid for all $x > -1$ to get the following bounds on $A_{\rho_n}$:

$$A_{\rho_n} \leq \frac{1-\rho_n}{|\ln\rho_n|}\left(\mathbb{E}\left[\sum_{i=1}^{K}e^{-\frac{v_i^2}{v_k^2}|\ln\rho_n|}\left(p_i^+e^{\left(\frac{2v_i^2|\ln\rho_n|}{v_k^2}\right)^{\frac{1}{2}}Z}+p_i^-e^{-\left(\frac{2v_i^2|\ln\rho_n|}{v_k^2}\right)^{\frac{1}{2}}Z}\right)\right]-1\right)$$

$$= \frac{1-\rho_n}{|\ln\rho_n|}\left(\sum_{i=1}^{K}p_ie^{-2\frac{v_i^2}{v_k^2}|\ln\rho_n|}-1\right)\leq -\frac{1-\rho_n}{|\ln\rho_n|}\;;$$

$$A_{\rho_n} \geq \frac{1-\rho_n}{|\ln\rho_n|}\mathbb{E}\left[\frac{\sum_{i=1}^{K}e^{-\frac{v_i^2}{v_k^2}|\ln\rho_n|}\left(p_i^+e^{\left(\frac{2v_i^2|\ln\rho_n|}{v_k^2}\right)^{\frac{1}{2}}Z}+p_i^-e^{-\left(\frac{2v_i^2|\ln\rho_n|}{v_k^2}\right)^{\frac{1}{2}}Z}\right)-1}{1-\rho_n+\rho_n\sum_{i=1}^{K}e^{-\frac{v_i^2}{v_k^2}|\ln\rho_n|}\left(p_i^+e^{\left(\frac{2v_i^2|\ln\rho_n|}{v_k^2}\right)^{\frac{1}{2}}Z}+p_i^-e^{-\left(\frac{2v_i^2|\ln\rho_n|}{v_k^2}\right)^{\frac{1}{2}}Z}\right)}\right]$$

$$\geq -\frac{1}{|\ln\rho_n|}\;.$$

The last inequality follows from $(x-1)/(1-\rho_n+\rho_n x) \geq -1/(1-\rho_n)$ for $x > 0$. Together the upper bound and lower bound imply that $|A_{\rho_n}| \leq 1/|\ln\rho_n| \xrightarrow[\rho_n\to 0]{} 0$. The last step before concluding the proof is to compute the limits of each summand in both sums over $j \in \{1,\ldots,K\}$ in (210). Note that $\forall z \in \mathbb{R}$:

$$\widetilde{h}\left(z, \frac{2|\ln\rho_n|}{v_k^2}, v_j; \rho_n, \mathbf{v}, \mathbf{p}^{\pm}, \mathbf{p}^{\mp}\right) = (1-\rho_n)e^{|\ln\rho_n|\left(1-\frac{v_j^2}{v_k^2}-\sqrt{\frac{2v_j^2}{v_k^2|\ln\rho_n|}}z\right)}$$

$$+\sum_{i=1}^{K}e^{-|\ln\rho_n|\left(\frac{(v_i-v_j)^2}{v_k^2}-\sqrt{\frac{2}{|\ln\rho_n|}}\frac{v_i-v_j}{v_k}z\right)}\left(p_i^{\pm}+p_i^{\mp}e^{-4|\ln\rho_n|\frac{v_i}{v_k}\left(\frac{v_j}{v_k}+\frac{z}{\sqrt{2|\ln\rho_n|}}\right)}\right). \quad (211)$$

From (211) we easily deduce the following pointwise limits for every $z \in \mathbb{R}$ :

$$\frac{\ln\widetilde{h}\left(z, \frac{2|\ln\rho_n|}{v_k^2}, v_j; \rho_n, \mathbf{v}, \mathbf{p}^{\pm}, \mathbf{p}^{\mp}\right)}{|\ln\rho_n|} \xrightarrow[\rho_n\to 0]{} \begin{cases} 1-\frac{v_j^2}{v_k^2} & \text{if } j < k\;; \\ 0 & \text{if } j \geq k\;. \end{cases} \quad (212)$$

By the dominated convergence theorem, making use of the pointwise limits (212), we have:

$$\sum_{j=1}^{K}p_j^+\mathbb{E}\left[\frac{\ln\widetilde{h}\left(Z, \frac{2|\ln\rho_n|}{v_k^2}, v_j; \rho_n, \mathbf{v}, \mathbf{p}^+, \mathbf{p}^-\right)}{|\ln\rho_n|}\right]+p_j^-\mathbb{E}\left[\frac{\ln\widetilde{h}\left(Z, \frac{2|\ln\rho_n|}{v_k^2}, v_j; \rho_n, \mathbf{v}, \mathbf{p}^-, \mathbf{p}^+\right)}{|\ln\rho_n|}\right]$$

$$\xrightarrow[\rho_n\to 0]{} \sum_{j<k}(p_j^+ + p_j^-)\left(1-\frac{v_j^2}{v_k^2}\right) = \mathbb{P}(|X|<v_k) - \frac{\mathbb{E}[X^2\mathbf{1}_{\{|X|<v_k\}}]}{v_k^2}\;. \quad (213)$$

Combining the identity (210), $\lim_{\rho_n\to 0}A_{\rho_n}=0$ and the limit (213) yields:

$$\lim_{\rho_n\to 0}\frac{1}{\alpha_n}\psi_{P_0,n}\left(\frac{\alpha_n}{\rho_n}\frac{2}{\gamma v_k^2}\right) = \frac{\mathbb{E}[X^2]}{\gamma v_k^2} - \frac{1}{\gamma} + \frac{\mathbb{P}(|X|<v_k)}{\gamma} - \frac{\mathbb{E}[X^2\mathbf{1}_{\{|X|<v_k\}}]}{\gamma v_k^2}$$

$$= \frac{\mathbb{E}[X^2\mathbf{1}_{\{|X|\geq v_k\}}]}{\gamma v_k^2} - \frac{\mathbb{P}(|X|\geq v_k)}{\gamma}\;,$$

thus ending the proof of the proposition. $\qquad\square$

We can now use Lemmas 15 and 16 to determine the limits when $\rho_n \to 0$ of the infimum of $\sup_{r\geq 0} i_{\mathrm{RS}}(q,r;\alpha_n,\rho_n)$ over $q$ restrained to different subsegments of $[0,\mathbb{E}X^2]$.

**Proposition 14.** *Let $P_{0,n} := (1-\rho_n)\delta_0 + \rho_n P_0$ where $P_0$ is a discrete distribution with finite support* $\mathrm{supp}(P_0) \subseteq \{\pm v_1, \pm v_2, \ldots, \pm v_K\}$ *with* $0 < v_1 < v_2 < \cdots < v_K$. *Let $\alpha_n := \gamma\rho_n|\ln\rho_n|$ for a fix $\gamma > 0$. Then, $\forall k \in \{1,\ldots,K\}$ :*

$$\lim_{\rho_n\to 0^+}\inf_{q\in[a_{\rho_n}^{(k)},b_{\rho_n}^{(k)}]}\sup_{r\geq 0}i_{\mathrm{RS}}(q,r;\alpha_n,\rho_n) = \min\left\{I_{P_{\mathrm{out}}}(\mathbb{E}[X^2\mathbf{1}_{\{|X|>v_k\}}],\mathbb{E}X^2) + \frac{\mathbb{P}(|X|>v_k)}{\gamma},\right.$$

$$\left.I_{P_{\mathrm{out}}}(\mathbb{E}[X^2\mathbf{1}_{\{|X|\geq v_k\}}],\mathbb{E}X^2) + \frac{\mathbb{P}(|X|\geq v_k)}{\gamma}\right\} \quad (214)$$

*while* $\forall k \in \{2, \dots, K\}$ :

$$\lim_{\rho_n \to 0^+} \inf_{q \in [b_{\rho_n}^{(k)}, a_{\rho_n}^{(k-1)}]} \sup_{r \geq 0} i_{\mathrm{RS}}(q, r; \alpha_n, \rho_n) = I_{P_{\mathrm{out}}}(\mathbb{E}[X^2 \mathbf{1}_{\{|X| \geq v_k\}}], \mathbb{E}X^2) + \frac{\mathbb{P}(|X| \geq v_k)}{\gamma} , \tag{215}$$

*and*

$$\lim_{\rho_n \to 0^+} \inf_{q \in [0, a_{\rho_n}^{(K)}]} \sup_{r \geq 0} i_{\mathrm{RS}}(q, r; \alpha_n, \rho_n) = I_{P_{\mathrm{out}}}(0, \mathbb{E}X^2) , \tag{216}$$

$$\liminf_{\rho_n \to 0^+} \inf_{q \in [b_{\rho_n}^{(1)}, 1]} \sup_{r \geq 0} i_{\mathrm{RS}}(q, r; \alpha_n, \rho_n) \geq \frac{1}{\gamma} . \tag{217}$$

*Proof.* In the whole proof $\rho_n$ is close enough to 0 for the ordering (197) to hold. First we prove (214). Fix $k \in \{1, \dots, K\}$. By Lemma 15, for all $q \in [a_{\rho_n}^{(k)}, b_{\rho_n}^{(k)}]$ we have

$$\sup_{r \geq 0} \frac{rq}{2} - \frac{1}{\alpha_n} \psi_{P_{0,n}}\left(\frac{\alpha_n}{\rho_n} r\right) = \frac{r_n^*(q)q}{2} - \frac{1}{\alpha_n} \psi_{P_{0,n}}\left(\frac{\alpha_n}{\rho_n} r_n^*(q)\right)$$

where $\frac{2(1 - |\ln \rho_n|^{-\frac{1}{4}})}{\gamma v_k^2} \leq r_n^*(q) \leq \frac{2(1 + |\ln \rho_n|^{-\frac{1}{4}})}{\gamma v_k^2}$. This and the fact that $\psi_{P_{0,n}}$ is increasing imply that $\forall q \in [a_{\rho_n}^{(k)}, b_{\rho_n}^{(k)}]$ :

$$I_{P_{\mathrm{out}}}(q, \mathbb{E}[X^2]) + \frac{q}{\gamma v_k^2}\left(1 - |\ln \rho_n|^{-\frac{1}{4}}\right) - \frac{1}{\alpha_n} \psi_{P_{0,n}}\left(\frac{\alpha_n}{\rho_n} \frac{2(1 + |\ln \rho_n|^{-\frac{1}{4}})}{\gamma v_k^2}\right)$$

$$\leq \sup_{r \geq 0} i_{\mathrm{RS}}(q, r; \alpha_n, \rho_n)$$

$$\leq I_{P_{\mathrm{out}}}(q, \mathbb{E}[X^2]) + \frac{q}{\gamma v_k^2}\left(1 + |\ln \rho_n|^{-\frac{1}{4}}\right) - \frac{1}{\alpha_n} \psi_{P_{0,n}}\left(\frac{\alpha_n}{\rho_n} \frac{2(1 - |\ln \rho_n|^{-\frac{1}{4}})}{\gamma v_k^2}\right) . \tag{218}$$

These inequalities are valid for every $q \in [a_{\rho_n}^{(k)}, b_{\rho_n}^{(k)}]$ so the same inequalities will hold if we take the infimum over $q \in [a_{\rho_n}^{(k)}, b_{\rho_n}^{(k)}]$ in (218). Note that $q \mapsto I_{P_{\mathrm{out}}}(q, \mathbb{E}[X^2]) + \frac{q}{\gamma v_k^2}(1 \mp |\ln \rho_n|^{-\frac{1}{4}})$ are concave functions on $[a_{\rho_n}^{(k)}, b_{\rho_n}^{(k)}]$ so the minimum of each function is achieved at either endpoint $a_{\rho_n}^{(k)}$ or $b_{\rho_n}^{(k)}$. It comes:

$$\inf_{q \in [a_{\rho_n}^{(k)}, b_{\rho_n}^{(k)}]} I_{P_{\mathrm{out}}}(q, \mathbb{E}[X^2]) + \frac{q}{\gamma v_k^2}\left(1 \pm |\ln \rho_n|^{-\frac{1}{4}}\right) - \frac{1}{\alpha_n} \psi_{P_{0,n}}\left(\frac{\alpha_n}{\rho_n} \frac{2(1 \mp |\ln \rho_n|^{-\frac{1}{4}})}{\gamma v_k^2}\right)$$

$$= -\frac{1}{\alpha_n} \psi_{P_{0,n}}\left(\frac{\alpha_n}{\rho_n} \frac{2(1 \mp |\ln \rho_n|^{-\frac{1}{4}})}{\gamma v_k^2}\right) + \min_{q \in \{a_{\rho_n}^{(k)}, b_{\rho_n}^{(k)}\}} I_{P_{\mathrm{out}}}(q, \mathbb{E}[X^2]) + \frac{q}{\gamma v_k^2}\left(1 \pm |\ln \rho_n|^{-\frac{1}{4}}\right)$$

$$\xrightarrow[\rho_n \to 0]{} \frac{\mathbb{P}(|X| \geq v_k)}{\gamma} - \frac{\mathbb{E}[X^2 \mathbf{1}_{\{|X| \geq v_k\}}]}{\gamma v_k^2} + \min_{q \in \left\{\substack{\mathbb{E}[X^2 \mathbf{1}_{\{|X| > v_k\}}], \\ \mathbb{E}[X^2 \mathbf{1}_{\{|X| \geq v_k\}}]}\right\}} I_{P_{\mathrm{out}}}(q, \mathbb{E}[X^2]) + \frac{q}{\gamma v_k^2}$$

$$= \min \left\{ I_{P_{\mathrm{out}}}(\mathbb{E}[X^2 \mathbf{1}_{\{|X| > v_k\}}], \mathbb{E}[X^2]) + \frac{\mathbb{P}(|X| > v_k)}{\gamma}, \right.$$

$$\left. I_{P_{\mathrm{out}}}(\mathbb{E}[X^2 \mathbf{1}_{\{|X| \geq v_k\}}], \mathbb{E}[X^2]) + \frac{\mathbb{P}(|X| \geq v_k)}{\gamma} \right\} . \tag{219}$$

The limit when $\rho_n \to 0$ follows from (198) in Lemma 15 and (207) in Lemma 16. Taking the infimum over $q \in [a_{\rho_n}^{(k)}, b_{\rho_n}^{(k)}]$ in (218) and using the fact that the upper and lower bounds have the same limit (219) ends the proof of (214).

We now turn to the proof of the limit (215). Fix $k \in \{2, \dots, K\}$. As the supremum of nondecreasing functions, the function $\widetilde{\psi}_{P_{0,n}} : q \in [0, \mathbb{E}X^2] \mapsto \sup_{r \geq 0} \frac{rq}{2} - \frac{1}{\alpha_n} \psi_{P_{0,n}}\left(\frac{\alpha_n}{\rho_n} r\right)$ is nondecreasing. The fact that $I_{P_{\mathrm{out}}}(\cdot, \mathbb{E}X^2)$ and $\widetilde{\psi}_{P_{0,n}}$ are respectively nonincreasing and nondecreasing imply that:

$$I_{P_{\mathrm{out}}}(a_{\rho_n}^{(k-1)}, \mathbb{E}X^2) + \widetilde{\psi}_{P_{0,n}}(b_{\rho_n}^{(k)})$$

$$\leq \inf_{q \in [b_{\rho_n}^{(k)}, a_{\rho_n}^{(k-1)}]} \sup_{r \geq 0} i_{\mathrm{RS}}(q, r; \alpha_n, \rho_n) \leq I_{P_{\mathrm{out}}}(b_{\rho_n}^{(k)}, \mathbb{E}X^2) + \widetilde{\psi}_{P_{0,n}}(a_{\rho_n}^{(k-1)}) . \tag{220}$$

By Lemma 15, we have

$$\widetilde{\psi}_{P_{0,n}}(b_{\rho_n}^{(k)}) = \frac{r_n^*(b_{\rho_n}^{(k)})b_{\rho_n}^{(k)}}{2} - \frac{1}{\alpha_n}\psi_{P_{0,n}}\left(\frac{\alpha_n}{\rho_n}r_n^*(b_{\rho_n}^{(k)})\right),$$

$$\widetilde{\psi}_{P_{0,n}}(a_{\rho_n}^{(k-1)}) = \frac{r_n^*(a_{\rho_n}^{(k-1)})a_{\rho_n}^{(k-1)}}{2} - \frac{1}{\alpha_n}\psi_{P_{0,n}}\left(\frac{\alpha_n}{\rho_n}r_n^*(a_{\rho_n}^{(k-1)})\right),$$

where $r_n^*(b_{\rho_n}^{(k)}) = 2\left(1+|\ln\rho_n|^{-1/4}\right)/\gamma v_k^2$ and $r_n^*(a_{\rho_n}^{(k-1)}) = 2\left(1-|\ln\rho_n|^{-1/4}\right)/\gamma v_{k-1}^2$. Making use of the limits (198) in Lemma 15 and (207) in Lemma 16 yields:

$$\lim_{\rho_n\to 0^+}\widetilde{\psi}_{P_{0,n}}(b_{\rho_n}^{(k)}) = \frac{\mathbb{E}[X^2\mathbf{1}_{\{|X|\geq v_k\}}]}{\gamma v_k^2} - \frac{\mathbb{E}[X^2\mathbf{1}_{\{|X|\geq v_k\}}]}{\gamma v_k^2} + \frac{\mathbb{P}(|X|\geq v_k)}{\gamma} = \frac{\mathbb{P}(|X|\geq v_k)}{\gamma};$$

$$\lim_{\rho_n\to 0^+}\widetilde{\psi}_{P_{0,n}}(a_{\rho_n}^{(k-1)}) = \frac{\mathbb{E}[X^2\mathbf{1}_{\{|X|>v_{k-1}\}}]}{\gamma v_{k-1}^2} - \frac{\mathbb{E}[X^2\mathbf{1}_{\{|X|\geq v_{k-1}\}}]}{\gamma v_{k-1}^2} + \frac{\mathbb{P}(|X|\geq v_{k-1})}{\gamma} = \frac{\mathbb{P}(|X|\geq v_k)}{\gamma}.$$

Besides, as $\lim_{\rho_n\to 0^+} b_{\rho_n}^{(k)} = \lim_{\rho_n\to 0^+} a_{\rho_n}^{(k-1)} = \mathbb{E}[X^2\mathbf{1}_{\{|X|\geq v_k\}}]$ and $I_{P_{\text{out}}}$ is continuous, we have:

$$\lim_{\rho_n\to 0^+} I_{P_{\text{out}}}(b_{\rho_n}^{(k)},\mathbb{E}X^2) = \lim_{\rho_n\to 0^+} I_{P_{\text{out}}}(a_{\rho_n}^{(k-1)},\mathbb{E}X^2) = I_{P_{\text{out}}}(\mathbb{E}[X^2\mathbf{1}_{\{|X|\geq v_k\}}],\mathbb{E}X^2).$$

Thus, the lower and upper bounds in (220) have the same limit. It ends the proof of (215).

The proof of (216) is similar to the one of (215). We have that

$$I_{P_{\text{out}}}(a_{\rho_n}^{(K)},\mathbb{E}X^2) + \widetilde{\psi}_{P_{0,n}}(0)$$
$$\leq \inf_{q\in[0,a_{\rho_n}^{(K)}]}\sup_{r\geq 0} i_{\text{RS}}(q,r;\alpha_n,\rho_n) \leq I_{P_{\text{out}}}(0,\mathbb{E}X^2) + \widetilde{\psi}_{P_{0,n}}(a_{\rho_n}^{(K)}). \quad (221)$$

Clearly $\widetilde{\psi}_{P_{0,n}}(0) = 0$ while $\lim_{\rho_n\to 0^+} I_{P_{\text{out}}}(a_{\rho_n}^{(K)},\mathbb{E}X^2) = I_{P_{\text{out}}}(0,\mathbb{E}X^2)$ by continuity of $I_{P_{\text{out}}}$ and $\lim_{\rho_n\to 0^+} a_{\rho_n}^{(K)} = 0$. By Lemma 15, $\widetilde{\psi}_{P_{0,n}}(a_{\rho_n}^{(K)}) = r_n^*(a_{\rho_n}^{(K)})a_{\rho_n}^{(K)}/2 - \psi_{P_{0,n}}\left(\frac{\alpha_n}{\rho_n}r_n^*(a_{\rho_n}^{(K)})\right)/\alpha_n$ where $r_n^*(a_{\rho_n}^{(K)}) = 2(1-|\ln\rho_n|^{-1/4})/\gamma v_K^2$. It follows from the limits (198) in Lemma 15 and (207) in Lemma 16 that $\lim_{\rho_n\to 0^+}\widetilde{\psi}_{P_{0,n}}(a_{\rho_n}^{(K)}) = 0$. Thus, the lower and upper bounds in (221) have the same limit. It ends the proof of (216).

It remains to prove (217). The fact that $I_{P_{\text{out}}}(\cdot,\mathbb{E}X^2)$ and $\widetilde{\psi}_{P_{0,n}}$ are respectively nonincreasing and nondecreasing imply that

$$\inf_{q\in[b_{\rho_n}^{(k)},\mathbb{E}X^2]}\sup_{r\geq 0} i_{\text{RS}}(q,r;\alpha_n,\rho_n) \geq I_{P_{\text{out}}}(\mathbb{E}X^2,\mathbb{E}X^2) + \widetilde{\psi}_{P_{0,n}}(b_{\rho_n}^{(1)}) = \widetilde{\psi}_{P_{0,n}}(b_{\rho_n}^{(1)}). \quad (222)$$

Hence, the inequality (217) follows from taking the limit inferior on both sides of (222) and the limit

$$\widetilde{\psi}_{P_{0,n}}(b_{\rho_n}^{(1)}) = \frac{r_n^*(b_{\rho_n}^{(1)})b_{\rho_n}^{(1)}}{2} - \frac{\psi_{P_{0,n}}\left(\frac{\alpha_n}{\rho_n}r_n^*(b_{\rho_n}^{(1)})\right)}{\alpha_n}$$
$$\xrightarrow[\rho_n\to 0^+]{} \frac{\mathbb{E}[X^2\mathbf{1}_{\{|X|\geq v_1\}}]}{\gamma v_1^2} - \frac{\mathbb{E}[X^2\mathbf{1}_{\{|X|\geq v_1\}}]}{\gamma v_1^2} + \frac{\mathbb{P}(|X|\geq v_1)}{\gamma} = \frac{1}{\gamma}.$$

$\square$

**Proposition 15.** *Let $P_{0,n} := (1-\rho_n)\delta_0 + \rho_n P_0$ where $P_0$ is a discrete distribution with finite support* $\operatorname{supp}(P_0) \subseteq \{-v_K,-v_{K-1},\ldots,-v_1,v_1,v_2,\ldots,v_K\}$ *with* $0 < v_1 < \cdots < v_K < v_{K+1} = +\infty$. *Let* $\alpha_n := \gamma\rho_n|\ln\rho_n|$ *for a fix* $\gamma > 0$.

*Then the quantity $I(\rho_n,\alpha_n) := \inf_{q\in[0,\mathbb{E}_{X\sim P_0}X^2]}\sup_{r\geq 0} i_{\text{RS}}(q,r;\alpha_n,\rho_n)$ converges when $\rho_n\to 0^+$ and*

$$\lim_{\rho_n\to 0^+} I(\rho_n,\alpha_n) = \min_{1\leq k\leq K+1}\left\{I_{P_{\text{out}}}(\mathbb{E}[X^2\mathbf{1}_{\{|X|\geq v_k\}}],\mathbb{E}[X^2]) + \frac{\mathbb{P}(|X|\geq v_k)}{\gamma}\right\}. \quad (223)$$

*Proof.* The proof goes in two steps. We first prove a upper bound on the limit superior of $I(\rho_n,\alpha_n)$, and then prove a lower bound on the limit inferior thats turns out to match the limit superior.

**Upper bound on the limit superior**   Note the following trivial upper bound:

$$I(\rho_n, \alpha_n) \leq \min_{1 \leq k \leq K} \left\{ \inf_{q \in [a_{\rho_n}^{(k)}, b_{\rho_n}^{(k)}]} \sup_{r \geq 0} i_{\mathrm{RS}}(q, r; \alpha_n, \rho_n) \right\}. \tag{224}$$

The upper bound on the limit superior of $I(\rho_n, \alpha_n)$ thus directly follows from (224) and Proposition 14 on the limits of the infimums over $q \in [a_{\rho_n}^{(k)}, b_{\rho_n}^{(k)}]$

$$\limsup_{\rho_n \to 0^+} I(\rho_n, \alpha_n) \leq \min_{1 \leq k \leq K} \min \left\{ I_{P_{\mathrm{out}}}\left(\mathbb{E}[X^2 \mathbf{1}_{\{|X| > v_k\}}], \mathbb{E}[X^2]\right) + \frac{\mathbb{P}(|X| > v_k)}{\gamma}, \right.$$

$$\left. I_{P_{\mathrm{out}}}\left(\mathbb{E}[X^2 \mathbf{1}_{\{|X| \geq v_k\}}], \mathbb{E}[X^2]\right) + \frac{\mathbb{P}(|X| \geq v_k)}{\gamma} \right\}$$

$$= \min_{1 \leq k \leq K+1} \left\{ I_{P_{\mathrm{out}}}\left(\mathbb{E}[X^2 \mathbf{1}_{\{|X| \geq v_k\}}], \mathbb{E}[X^2]\right) + \frac{\mathbb{P}(|X| \geq v_k)}{\gamma} \right\}. \tag{225}$$

**Matching lower bound on the limit inferior**   The lower bound on the limit inferior is obtained by studying the infimum on each segment of the following partition:

$$[0, \mathbb{E}X^2] = [0, a_{\rho_n}^{(K)}] \cup \left( \bigcup_{k=1}^{K} [a_{\rho_n}^{(k)}, b_{\rho_n}^{(k)}] \right) \cup \left( \bigcup_{k=2}^{K} [b_{\rho_n}^{(k)}, a_{\rho_n}^{(k-1)}] \right) \cup [b_{\rho_n}^{(1)}, \mathbb{E}X^2]. \tag{226}$$

By Proposition 14, we directly have:

$$\liminf_{\rho_n \to 0^+} \inf_{q \in \bigcup_{k=1}^{K} [a_{\rho_n}^{(k)}, b_{\rho_n}^{(k)}]} \sup_{r \geq 0} i_{\mathrm{RS}}(q, r; \alpha_n, \rho_n)$$

$$= \min_{1 \leq k \leq K+1} \left\{ I_{P_{\mathrm{out}}}\left(\mathbb{E}[X^2 \mathbf{1}_{\{|X| \geq v_k\}}], \mathbb{E}[X^2]\right) + \frac{\mathbb{P}(|X| \geq v_k)}{\gamma} \right\};$$

$$\liminf_{\rho_n \to 0^+} \inf_{q \in \bigcup_{k=2}^{K} [b_{\rho_n}^{(k)}, a_{\rho_n}^{(k-1)}]} \sup_{r \geq 0} i_{\mathrm{RS}}(q, r; \alpha_n, \rho_n)$$

$$= \min_{2 \leq k \leq K} \left\{ I_{P_{\mathrm{out}}}\left(\mathbb{E}[X^2 \mathbf{1}_{\{|X| \geq v_k\}}], \mathbb{E}[X^2]\right) + \frac{\mathbb{P}(|X| \geq v_k)}{\gamma} \right\};$$

$$\liminf_{\rho_n \to 0^+} \inf_{q \in [0, a_{\rho_n}^{(K)}]} \sup_{r \geq 0} i_{\mathrm{RS}}(q, r; \alpha_n, \rho_n)$$

$$= I_{P_{\mathrm{out}}}\left(0, \mathbb{E}[X^2]\right) = I_{P_{\mathrm{out}}}\left(\mathbb{E}[X^2 \mathbf{1}_{\{|X| \geq +\infty\}}], \mathbb{E}[X^2]\right) + \frac{\mathbb{P}(|X| \geq +\infty)}{\gamma};$$

$$\liminf_{\rho_n \to 0^+} \inf_{q \in [b_{\rho_n}^{(1)}, 1]} \sup_{r \geq 0} i_{\mathrm{RS}}(q, r; \alpha_n, \rho_n)$$

$$\geq \frac{1}{\gamma} = I_{P_{\mathrm{out}}}\left(\mathbb{E}[X^2 \mathbf{1}_{\{|X| \geq v_1\}}], \mathbb{E}[X^2]\right) + \frac{\mathbb{P}(|X| \geq v_1)}{\gamma}.$$

Following the partition (226), the limit inferior of $\inf_{q \in [0, \mathbb{E}X^2]} \sup_{r \geq 0} i_{\mathrm{RS}}(q, r; \alpha_n, \rho_n)$ is equal to the minimum of the above four limits inferior. It comes:

$$\liminf_{\rho_n \to 0^+} I(\rho_n, \alpha_n) \geq \min_{1 \leq k \leq K+1} \left\{ I_{P_{\mathrm{out}}}\left(\mathbb{E}[X^2 \mathbf{1}_{\{|X| \geq v_k\}}], \mathbb{E}[X^2]\right) + \frac{\mathbb{P}(|X| \geq v_k)}{\gamma} \right\}. \tag{227}$$

We see that the lower bound (227) on the limit inferior matches the upper bound (225) on the limit superior, thus ending the proof.   □

**Proof of Theorem 2**   Combining Theorem 1 together with Proposition 15 ends the proof of Theorem 2.