[Reviews · NeurIPS 2020]

Review 1

Summary and Contributions: This paper extends recent work on the "all-or-nothing" phenomenon observed in binary compressive sensing (i.e., a sparse linear model) to a class of generalized linear models. Roughly speaking, the all-or-nothing phenomenon demonstrates that the mean-square error undergoes a rapid phase transition from its maximum value to zero after crossing a sampling threshold.

Strengths: Extending the all-or-nothing analysis to a broad family of generalized linear models is a challenging problem. This paper makes progress towards this goal by 1) characterizing the mutual information of the generalized model via a "single-letter" variational formula 2) applying this bound to develop a non-rigorous analysis of the all-or-nothing phenomenon for generalized linear models via the replica method 3) demonstrating agreement between these non-rigorous predictions and the performance of the GAMP algorithm. These are all novel contributions and important steps towards a full understanding of the generalized linear models case. (The authors now claim a rigorous bound.)

Weaknesses: Edit after author feedback: I believe that the work is much stronger now that the authors have established rigorous MMSE bounds. I have increased my overall score accordingly. The main weakness of the paper is that the connection to the all-or-nothing phenomenon is non-rigorous, whereas prior work for linear models establishes rigorous bounds. Given this limitation, the paper would have been stronger had it increased the focus on the agreement between the non-rigorous prediction and state-of-the-art algorithms. While the mutual information bound is interesting on its own, it would be more interesting with an explicit connection to the all-or-nothing phenomenon.

Correctness: The claims seems correct.

Clarity: The paper is relatively well-written, although some sections, especially the introduction, could be streamlined to make the key points more clearly. I would have preferred if the paper were more explicit and direct in the introduction about the fact that the all-or-nothing analysis is non-rigorous. (This suggestion is now moot given that the authors have established a rigorous connection.)

Relation to Prior Work: The paper makes extensive references to prior work, but I think it would be presented more clearly if it placed itself in the category of statistical physics bounds on phase transitions in sparse models, rather than in the category of the all-or-nothing phenomenon, which is established rigorously. (Again, one can ignore this comment given the new rigorous bounds.)

Reproducibility: Yes

Additional Feedback:


Review 2

Summary and Contributions: This paper studies the fundamental limits of a sparse estimation problem that can be viewed as estimation in a generalized linear estimation model or learning the parameters a prediction rule for a one layer neural network (i.e., the perceptron). Previous work has rigorously characterized interesting phase transitions for quality of recovery (with optimal methods) in settings where "sparsity" means that a fixed *fraction* of components is nonzero. The contribution of the present paper is to extend these results to sub-linear sparsity where the fraction of nonzeros converges to zero as the problem dimensions diverge. In this regime, very recent work focusing on the *linear* model has demonstrate an "all-or-nothing" phenomenon where the MSE converges to a step function as a function the sample complexity. The results of the paper depend on some nontrivial technical advances. Most significant is a refinement of the adaptive interpolation method that allows for different problem dimensions to diverge at different rates. As a theory paper, this is a nice and timely contribution to active research area showcasing the value of exact asymptotic theory. From the perspective of machine learning, this paper would benefit by provided some more discussion about how the results relate more usual learning problem.

Strengths: The paper builds upon sophisticated and very recent tools to provide interesting and exact analysis of a (stylized but still relevant) inference problem. This a solid contribution for the fundamental problem of learning the parameters of the perception, which should be of interest to the theoretical side of ML

Weaknesses: There is a lot of notation set up needed to understand and interpret the results. Some simple examples upfront may help reach a broader audience. It would be nice if analysis of all-or-nothing behavior from the potential (Section 3 int he Supplement) were rigorous instead of heuristic.

Correctness: The proof it quite long but I am fairly confident that the results are sound.

Clarity: It is written well enough given the space constraints. (More examples and discussion would definitely help though.)

Relation to Prior Work: I think the authors do a decent job in this regard.

Reproducibility: Yes

Additional Feedback: I suggest the authors look at the at Section 3.1 as well as the proof in the online version of citation [33], which deals with the linear model. I suspect the same ideas could be applied with minor modification to establish the all-or-nothing behavior from the potential function rigorously. In any event, the illustrations of the normalized potential (Figure 2 in [33]) may also be helpful. EDIT: I have read the authors response and appreciate that they have made the result rigorous. My score remains the same.


Review 3

Summary and Contributions: The paper considers the generalized linear model, in the high-dimensional sparse setting. i.e. a sublinear number of features are important (or even informative) for predicting the outcomes or observations. In this setting the authors prove the existence of an 'all-or-nothing phenomenon' (see GZ, RXZ) wherein there exists a function m_*(n) (sublinear in the ambient dimension n) such that - if the # samples m > (1+eps) m_*, the error in estimating the signal vanishes; and - if m < (1-eps) m_*, the error converges to the 'trivial' error of estimating from the prior. Classical statistical or learning theory typically demonstrates a 'graceful' decay of error as the number of samples grows large. The current work is in a line of work demonstrating a 'sharp cutoff' phenomenon instead. This was initiated by GZ, RXZ in linear regression setting. This is closely related to notions like channel capacity in information theory, but is different in the sublinear scaling, which can cause technical complications. It is analogous to the 'cutoff phenomenon' in mixing of markov chains (see reference D) The authors carry this out using an extension of statistical physics techniques to the 'sublinear' regime. The steps are 1. To compare (or interpolate) the mutual information in the current problem with that of simpler, 'scalar' observation problem with a calibrated signal-noise-ratio. Via an adaptive interpolation method (BM), the authors then prove upper and lower bounds that (asymptotically) coincide for an appropriately calibrated scalar problem 2. Demonstrate that the mutual information goes from 'trivial' to 'full' as m crosses m_* 3. Link the mutual information with the estimation and generalization errors using the I-MMSE identity. Reference key GZ: Gamarnik , Zadik 'High dimensional regression with ...' RXZ: Reeves, Xu, Zadik 'The all-or-nothing ...' DAM: Deshpande, Abbe, Montanari, 'Asymptotic mutual information ...' D: Diaconis 'The cutoff phenomenon in finite Markov Chains' BM: Barbier Macris 'The adaptive interpolation ....'

Strengths: The papers main strengths are: 1. The problem considered is fundamental, and the phenomenon demonstrated is interesting in its own right, as well for algorithm design questions in statistical estimation and learning. The distinct nature of the phenomenon to relatively 'standard' arguments in statistics and learning is of significant interest to the NeurIPS community, and I would welcome more of it. 2. In terms of techniques, the proof methods have been used signficantly in the past few years (see BM and later papers) but the application to the sublinear data regime is also nice. I think the paper would make a valuable addition to the NeurIPS program.

Weaknesses: I don't think the paper has signficant weaknesses. The model setting is admittedly quite limited, but given the relatively sparse literature on the cutoff phenomenon in learning, I do not think this is a strong complaint. Some suggestions to improve: - I would recommend the authors to distinguish the all-or-nothing or cutoff phenomenon from usual statistical bounds that the machine learning and NeurIPS community is familiar with. - Mention that 'teacher student' setting is another phrasing of 'well-specified' models in statistics, 'realizable setting' in learning theory, and 'proper learning' in computer science. - Regularizing noise: if this is needed for the proof, preferably write it as such. Presumably a limiting argument works for Delta vanishing, e.g. one can always add small noise to the observvations. - It would also be nice to have a rigorous proof for the MMSE portion. Since this is not done in the current paper, but potentially accessible to present techniques, the authors should identify Eq (10) as a 'Claim'.

Correctness: The paper's results seem correct to me, while I did not verify the proofs in full detail. The authors also provide code to reproduce the synthetic experiments.

Clarity: The paper is well written and easy to understand for the general reader. Some small suggestions regarding writing are in the Other comments portion of the review.

Relation to Prior Work: The related work is correctly referenced and introduced. The i-mmse connection was first introduced in DAM, so I recommend to include the reference.

Reproducibility: Yes

Additional Feedback: - L.62 'more precisions' is weird phrasing - L. 69 intuitively explain the scaling of alpha_n - L.81 This neural-network interpretation is somewhat forced. It is okay to care about models/algs that are not neural nets. - L.98 n^{-1/9} is odd scaling... is this a proof artifact? If so rephrase the line to indicate this. - L.131 'absent of' is odd - Table 1 suggests for zero noise there is not an 'all or nothing' phenomenon for linear models since gamma_c = 0 - Figure 1: what are the lighter colored lines, presumably potential value at q=1? EDIT POST RESPONSE: After reading the author response, I am maintaining my score on the paper.


Review 4

Summary and Contributions: In this paper, the authors study the estimation error of generalized linear models (GLM) in cases which the amount of samples is sublinear with respect to the total number of features of the signal. They motivate this problem by first providing an estimation interpretation in which the goal is to find the lowest reconstruction error as number of samples grow. They also provide a learning interpretation which consists of a teacher-student scenario in which the student wants to learn the weights of the teacher’s perceptron with general activation function. To this end, the authors provide a variational formula for the asymptotic mutual information per sample as system size goes to infinity. This Theorem is the main contribution of the paper. To highlight the importance of the provided Theorem, they use analytical arguments to show the existence of the all-or-nothing phenomenon for the estimation error of a GLM and the generalization error of a perceptron neural network with a general activation function. To illustrate this phenomenon, the authors plot the obtained MMSE for Bernoulli and Bernoulli-Rademacher priors and various activation functions. It is shown that in all different configurations, the MMSE jumps from a value close to 1 to a value close to 0.

Strengths: The paper is well written and results are non-trivial and interesting. They consider the GLM which can be seen as an extension of others works that consider the linear model. They compare their theoretical result with prior work done on the linear model.

Weaknesses: While the linear model is motivated by compressed sensing, it would be interesting if authors could tie the GLM model with other applications. For example, where is the teacher-student network interpretation used in practical applications? Figures 2 and 3 do not have sufficient captions. What is the small plot inside the big one? Why do some of the dotted lines divide into two lines? The authors addressed this in the rebuttal.

Correctness: I did not check the proofs. The empirical methodology looks correct.

Clarity: The paper is well written. It is sometimes hard to follow the math, but I think the authors have made an effort to make it flow and move proofs to the end of the paper or supplementary material.

Relation to Prior Work: The authors mention prior work and highlight distinctions with their work throughout the paper.

Reproducibility: Yes

Additional Feedback:

[Author Response · NeurIPS 2020]

We thank the reviewers for their thorough comments and suggestions. Let us first address the main weakness of the
submission: all the reviewers point out that the discussion on the asymptotic MMSE and the all-or-nothing phenomenon
is nonrigorous. We are now able to resolve this point. The main difficulty in order to make the discussion on the MMSE
rigorous is to compute the limit of the variational formula in Theorem 1 when $\rho_n$ vanishes. Since the submission
deadline we have proved that the latter limit is given by the minimum of a finite set of values whenever $P_0$ is a discrete
distribution with a finite support. We can combine this limit with Theorem 1 to obtain the result:

**Theorem 2.** *Suppose that $\Delta > 0$ and that $P_{0,n} := (1 - \rho_n)\delta_0 + \rho_n P_0$ where $P_0$ is a discrete distribution with fi-*
*nite support* $\mathrm{supp}(P_0) \subseteq \{-v_K, -v_{K-1}, \ldots, -v_1, v_1, v_2, \ldots, v_K\}$ *where* $0 < v_1 < v_2 < \cdots < v_K < v_{K+1} := +\infty$.
*Further assume that the hypotheses (H2), (H3) of Theorem 1 hold. Let* $\rho_n = \Theta(n^{-\lambda})$ *with* $\lambda \in (0, {}^1\!/{}_9)$ *and*
$\alpha_n = \gamma\rho_n|\ln\rho_n|$ *with* $\gamma > 0$. *Then (in what follows* $X \sim P_0$):

$$\lim_{n\to+\infty} \frac{I(\boldsymbol{X}^*; \boldsymbol{Y}|\boldsymbol{\Phi})}{m_n} = \min_{1\le k\le K+1}\left\{ I_{P_{\mathrm{out}}}\big(\mathbb{E}[X^2\mathbb{1}_{\{|X|\ge v_k\}}], \mathbb{E}[X^2]\big) + \frac{\mathbb{P}(|X|\ge v_k)}{\gamma} \right\}. \tag{1}$$

Once we have Theorem 2 we can obtain the asymptotic MMSE with a little more work. First we have to introduce a
modified inference problem where in addition to the observations $\boldsymbol{Y}$ we are given $\widetilde{\boldsymbol{Y}}^{(\tau)} = \sqrt{\alpha_n\tau/\rho_n}\,\boldsymbol{X}^* + \widetilde{\boldsymbol{Z}}$. When $\tau$
is close enough to 0, i.e., $\tau < {}^2\!/\gamma v_K^2$, the analysis yielding Theorem 2 can be adapted to obtain the limit

$$\lim_{n\to+\infty} \frac{I(\boldsymbol{X}^*; \boldsymbol{Y}, \widetilde{\boldsymbol{Y}}^{(\tau)}|\boldsymbol{\Phi})}{m_n} = \min_{1\le k\le K+1}\left\{ I_{P_{\mathrm{out}}}\big(\mathbb{E}[X^2\mathbb{1}_{\{|X|\ge v_k\}}], \mathbb{E}[X^2]\big) + \frac{\mathbb{P}(|X|\ge v_k)}{\gamma} + \frac{\tau\mathbb{E}[X^2\mathbb{1}_{\{|X|<v_k\}}]}{2} \right\}.$$

We can then apply the I-MMSE identity[1] to obtain the asymptotic MMSE:

**Theorem 3.** *Suppose that $\Delta > 0$ and that $P_{0,n} := (1 - \rho_n)\delta_0 + \rho_n P_0$ where $P_0$ is a discrete distribution with fi-*
*nite support* $\mathrm{supp}(P_0) \subseteq \{-v_K, -v_{K-1}, \ldots, -v_1, v_1, v_2, \ldots, v_K\}$ *where* $0 < v_1 < v_2 < \cdots < v_K < v_{K+1} := +\infty$.
*Further assume that the hypotheses (H2), (H3) of Theorem 1 hold. Let* $\rho_n = \Theta(n^{-\lambda})$ *with* $\lambda \in (0, {}^1\!/{}_9)$ *and*
$\alpha_n = \gamma\rho_n|\ln\rho_n|$ *with* $\gamma > 0$.

*If the minimization problem on the right-hand side of* (1) *has a unique solution* $k^* \in \{1, \ldots, K+1\}$ *then*

$$\lim_{n\to+\infty} \frac{\mathbb{E}\|\boldsymbol{X}^* - \mathbb{E}[\boldsymbol{X}^*|\boldsymbol{Y}, \boldsymbol{\Phi}]\|^2}{k_n} = \mathbb{E}\big[X^2\mathbb{1}_{\{|X|<v_{k^*}\}}\big]. \tag{2}$$

When $P_{0,n}$ is a Bernoulli or Bernoulli-Rademacher distribution we have $K = 1$ and Theorem 3 corresponds to the
all-or-nothing phenomenon. If $K > 1$ the phenomenology is different and the asymptotic MMSE exhibits up to $K$
sharp phase transitions.

If our submission is accepted we shall use the additional content page to state both Theorems 2 and 3 in Section 2.2.
We shall refer the reader to the supplementary material for the proofs. We shall slightly rewrite the introduction and
Section 3 to take into account that the discussion on the asymptotic MMSE is now rigorous and not only heuristic.

Let us now address the main other points raised by the reviewers.

• About the figures. The horizontal lines in Figure 1 indeed correspond to the value of the potential at $q = 1$. We plot
these lines to highlight how the minimum of the potential shifts from being close to $q = 0$ to being close to $q = 1$ at
the all-or-nothing phase transition. We will make this clear in the caption. The inset plots in Figures 2 and 3 are zooms
on tight intervals around the all-or-nothing phase transitions. We propose to get rid of these inset plots and instead to
pick a tighter interval for the x-axis of the main plots (in the submission $\alpha_n/\alpha_c(\rho_n) \in [0.1, 10]$). In Figure 2 and 3 the
dotted lines correspond to the performance of GAMP algorithms as predicted by the state evolution equations (see the
paragraph "Further remarks" in Section 3). The departure of a dotted line from the thicker line having the same color
puts in evidence the algorithmic-to-statical gap mentioned in the paragraph "Further remarks". When this gap is
small the dotted line is not visible on the figures as it matches the MMSE curve. Our captions will be corrected.

• The scaling $\rho_n = \Theta(n^{-\lambda})$ with $\lambda < {}^1\!/{}_9$ is indeed a proof artifact. To be more precise, we have to lower bound the
vanishing rate of $\rho_n$ by $n^{-1/9}$ if we want that the upper bound of Lemma 10 in the supplementary material vanishes.

• The value $\gamma_c = 0$ for noiseless linear models in Table 1 indeed shows that there is no all-or-nothing phenomenon
in this case, and is in agreement with the existing literature. E.g., Reeves, Xu, and Zadik[2] locate the transition at a
sample size $n^*$ that vanishes when the noise variance approaches zero.

Finally, we take note of the additional reference as well as the rephrasings and grammatical corrections suggested by
the reviewers in order to improve our manuscript.

## Footnotes

[1] The derivative of ${}^{I(\boldsymbol{X}^*; \boldsymbol{Y}, \widetilde{\boldsymbol{Y}}^{(\tau)}|\boldsymbol{\Phi})}\!/{}_{m_n}$ with respect to $\tau$ at $\tau = 0$ is equal to half the MMSE of the original problem.

[2] Galen Reeves, Jiaming Xu, and Ilias Zadik. "The all-or-nothing phenomenon in sparse linear regression". In: *Proceedings of the Thirty-Second Conference on Learning Theory* (Phoenix, USA). Vol. 99. PMLR, 2019, pp. 2652–2663.


[Meta-Review · NeurIPS 2020]

This paper studied the generalized linear model under sublinear sparsity. Theorem 1 establishes reduction of the asymptotic mutual information to a low-dimensional variational problem. Under a certain assumption on the signals, the all-or-nothing phenomenon, previously reported for the linear model under sublinear sparsity, is also shown to be observed in the generalized linear model. The main weakness, as most of the reviewers pointed out, was in a non-rigorous step in the analysis of the all-or-nothing phenomenon, but the authors stated in their response that they have succeeded in providing a rigorous proof on the basis of the suggestion by Reviewer #2 in his review. Now the whole results have been derived rigorously, and all the review scores are well above the acceptance threshold, so that I am glad to recommend acceptance of this paper.